# The Five Ws of Multi-Agent Communication: Who Talks to Whom, When, What, and Why
# A Survey from MARL to Emergent Language and LLMs

**Jingdi Chen**[*]                                                    *jingdic@arizona.edu*
*University of Arizona*
*Tucson, AZ 85719*

**Hanqing Yang**                                                 *hanqing3@andrew.cmu.edu*
*Carnegie Mellon University*
*Pittsburgh, PA 15213*

**Zongjun Liu**                                                      *msnliu@arizona.edu*
*University of Arizona*
*Tucson, AZ 85719*

**Carlee Joe-Wong**                                            *cjoewong@andrew.cmu.edu*
*Carnegie Mellon University*
*Pittsburgh, PA 15213*

**Reviewed on OpenReview:** *https://openreview.net/forum?id=LGsedOQQVq*

## Abstract

Multi-agent sequential decision-making underpins many real-world systems, from autonomous vehicles and robotics to collaborative AI assistants. In dynamic and partially observable environments, effective communication is essential for reducing uncertainty and enabling coordination. Although research on multi-agent communication (MA-Comm) spans diverse paradigms, we organize this survey explicitly around the *Five Ws of communication*: who communicates with whom, what is communicated, when communication occurs, and why communication is beneficial. This lens provides a coherent structure for synthesizing diverse approaches and exposing shared design principles across paradigms. Within *Multi-Agent Reinforcement Learning (MARL)*, early work relied on hand-designed or implicit communication protocols, followed by trainable, end-to-end mechanisms optimized for reward and control. While effective, these approaches often yield task-specific and weakly interpretable communication, motivating research on *Emergent Language (EL)*, where agents develop more structured or symbolic protocols through interaction. EL methods, however, still face challenges in grounding, generalization, and scalability, which have driven recent interest in *large language models (LLMs)* as a means to leverage natural language priors for reasoning, planning, and coordination in open-ended multi-agent settings. This progression motivates our survey: we analyze how communication paradigms evolve in response to the limitations of earlier approaches and how MARL, EL, and LLM-based systems address complementary aspects of multi-agent communication. This paper provides a unified survey of MA-Comm across MARL, EL, and LLM-based multi-agent systems. Organized around the *Five Ws*, we examine how different paradigms motivate, structure, and operationalize communication, reveal cross-paradigm trade-offs, and identify open challenges in communication, coordination, and learning. By offering systematic comparisons and design-oriented insights, this survey helps the community extract effective communication design

---

[*]This work was done during Jingdi Chen's postdoctoral research at Carnegie Mellon University.

patterns and supports the development of hybrid systems that combine learning, language, and control to meet diverse task, scalability, and interpretability requirements.

## Contents

# 1  Introduction

Multi-agent decision-making plays a critical role in a wide range of real-world applications, including robotics Gu et al. (2016); Zhang et al. (2015), such as navigation Candido & Hutchinson (2011) and manipulation Pajarinen & Kyrki (2017); autonomous systems Talpaert et al. (2019), including autonomous driving Wang et al. (2019a); and planning under uncertainty Wang et al. (2019b); Cheng et al. (2018). In these settings, multiple agents must act independently while interacting with both the environment and one

another. Depending on the task, agents may cooperate to achieve shared objectives, compete over limited resources, or engage in mixed-motive interactions that involve both collaboration and competition. A central challenge in these systems is uncertainty, which can arise from multiple sources: limited or partial observability of the global state or of other agents' intentions, as well as inherent stochasticity in environment dynamics. Effective multi-agent systems must therefore address problems of coordination, decentralized control, and decision-making under uncertainty, often in real time and with limited communication.

To address these challenges, reinforcement learning (RL) has been widely explored as a framework for training agents to make sequential decisions through trial and error. Deep Reinforcement Learning (DRL), in particular, has been extensively studied in both simulated Mnih et al. (2013); Lillicrap et al. (2015) and real-world Gu et al. (2016); Zhang et al. (2015); Qureshi et al. (2017); Meng et al. (2019) scenarios. While many DRL algorithms assume fully observable environments modeled as Markov Decision Processes (MDPs) Lample & Chaplot (2016); Schulman et al. (2015); Pipattanasomporn et al. (2009); Li et al. (2002), real-world multi-agent systems often involve Partially Observable Markov Decision Processes (POMDPs), where each agent has only a limited view of the system state. In such cases, inter-agent communication can help mitigate partial observability by allowing agents to exchange information and build a more complete representation of the environment Chen et al. (2024b). As a result, an important research direction in MARL focuses on designing effective communication strategies to improve decision-making in complex environments.

Effective communication is a core capability in multi-agent systems, particularly in scenarios requiring coordination, negotiation, or competition among autonomous agents. In MARL, agents must make decisions based on local observations while interacting with others in partially observable and often stochastic environments. Communication enables agents to exchange task-relevant information, such as observations, goals, intentions, or strategies, which helps them align actions, resolve uncertainty, and coordinate more effectively. While this is critical in fully cooperative settings like multi-robot collaboration Li et al. (2002) or smart grid control Pipattanasomporn et al. (2009), communication also plays a nuanced but important role in competitive and mixed-motive scenarios. For example, in adversarial games like StarCraft Samvelyan et al. (2019) or Dota Berner et al. (2019), agents may engage in strategic signaling, misinformation, or negotiation to manipulate opponents or form temporary alliances. In such cases, communication becomes not just a tool for cooperation, but also a mechanism for influencing, deceiving, or adapting to others' behaviors in strategic interactions.

To move beyond manually designed protocols, recent research has focused on **learned communication**, where agents develop communication strategies through interaction. A widely used approach involves enabling gradient flow between agents during training, allowing them to optimize both message content and timing based on task performance. This line of work offers a scalable and flexible alternative to handcrafted rules, enabling agents to discover efficient and adaptive messaging schemes tailored to their goals and environmental dynamics, whether cooperative, competitive, or somewhere in between.

## 1.1 Evolution of Multi-Agent Communication Learning

**Communication in MARL** Research on learning multi-agent communication has evolved significantly, mirroring broader advances in artificial intelligence and multi-agent systems. **Initial efforts** in this area focused on enabling communication in **MARL** to enhance coordination under partial observability. Early studies addressed foundational questions such as *who should communicate* and *what information should be shared* Paulos et al. (2019); Lowe et al. (2017); Foerster et al. (2016a); Sukhbaatar et al. (2016); Jiang & Lu (2018); Das et al. (2018); Rangwala & Williams (2020). Many of these early approaches employed *Centralized Training with Decentralized Execution*, where a centralized coordinator facilitates communication during training while agents act independently during execution. For example, *CommNet* Sukhbaatar et al. (2016) and *IC3Net* Singh et al. (2018) aggregate hidden states across agents to produce shared representations. Later methods introduced more flexible mechanisms, such as attention-based message routing in *TarMAC* Das et al. (2018) and graph-structured message propagation in *DICG* Li et al. (2020), allowing agents to dynamically select communication partners. While these models often rely on centralized components during training, they still support decentralized execution, preserving scalability and autonomy in deployment. While these neural architectures enabled learned communication, they often **lacked interpretability**, as deep neural networks generate complex messages that are difficult to analyze Brown et al. (2020); LeCun et al. (2015).

Moreover, most approaches assumed agents could exchange **continuous, real-valued messages**, ignoring the **practical constraints** of real-world communication networks, which typically rely on discrete and bandwidth-limited transmissions Foerster et al. (2016a); Lowe et al. (2017); Mordatch & Abbeel (2018); Freed et al. (2020b;a). These **limitations** motivated research into **Emergent Discrete Communication**, where agents develop structured, interpretable protocols for exchanging information.

While **MARL** has made substantial strides in learning communication protocols within RL frameworks, alternative approaches for enabling agent cooperation have increasingly drawn attention. One such direction is **Emergent Language (EL)**, which studies how agents can develop structured communication protocols through repeated interaction. This line of work provides insights into how meaningful communication can **emerge** organically in cooperative settings, without requiring pre-defined languages or protocols Lazaridou & Baroni (2020); Li et al. (2022). This shift was motivated by several limitations of traditional MARL-based communication: (1) Although end-to-end learned communication can be highly effective for task optimization, it often produces continuous, opaque message representations that are difficult to interpret, verify, or reuse across tasks and agent populations. (2) Moreover, MARL communication protocols are typically tightly coupled to specific environments and reward structures, making them brittle under changes in agent composition, task semantics, or deployment conditions. These challenges motivated researchers to study communication itself as a learning outcome in **EL** research, rather than merely an internal signal optimized for reward in **MARL** settings.

**Communication in EL**   Learned communication in EL has been a promising avenue for improving interpretability and structure in multi-agent systems. To move beyond opaque continuous message spaces, early research explored how agents could develop discrete communication protocols through interaction—using one-hot messages Chaabouni et al. (2019); Kottur et al. (2017); Havrylov & Titov (2017); Lazaridou et al. (2016); Lee et al. (2017), binary signals Foerster et al. (2016a), or other constrained message formats Eccles et al. (2019). These efforts made agent communication more interpretable to humans, but often revealed coordination challenges, such as zero-shot failures where independently trained agents could not understand each other's learned protocols Hu et al. (2020b). To address this, researchers have begun to design environments and training methods that encourage more robust, generalizable, and human-aligned communication Bullard et al. (2021; 2020). While many of these efforts originate in the context of MARL, their implications extend to multi-agent decision-making more broadly, including settings beyond traditional reinforcement learning. For example, some works have aimed to align emergent communication with natural language Lee et al. (2019); Lowe et al. (2020), while others integrate pre-trained language models to introduce linguistic priors and promote structured, interpretable coordination Lazaridou et al. (2020); Tucker et al. (2021). These developments reflect **a growing trend** toward bridging learned communication protocols with human language to support not only efficient coordination in MARL, but also broader **agent collaboration and human-AI interaction**.

More recently, **Large Language Models (LLMs)** have emerged as a natural next step in this trajectory. By leveraging strong language understanding, reasoning capabilities, and extensive world knowledge acquired from large-scale pretraining, LLMs provide a new foundation for multi-agent communication. **Unlike traditional MARL or EL agents**, LLM-based systems can communicate directly in natural language, infer shared goals, and adapt to novel tasks and teammates with minimal additional training Yang et al. (2025). This capability makes LLMs particularly attractive for settings that require flexible coordination, zero-shot generalization, and interaction with humans.

**Communication in LLMs**   The rise of LLMs has introduced a new paradigm for multi-agent communication, expanding beyond traditional MARL frameworks. Unlike communication in MARL, where agents develop protocols from scratch through reinforcement learning, LLM-based agents leverage pre-trained linguistic knowledge to engage in structured, natural language-based exchanges and decision-making Du et al. (2023); Liang et al. (2023); Wang et al. (2023c). This is distinct from EL research, where the focus is on how agents develop communication protocols in task-driven settings without any pre-existing linguistic knowledge. Recent work has proposed a range of communication architectures for LLM-based multi-agent systems, including direct messaging, chain-of-thought interactions, hierarchical structures, and graph-based exchanges Zhang et al. (2023c); Du et al. (2023); Qian et al. (2023); Hong et al. (2023); Holt et al.; Wu

et al. (2023b); Jiang et al. (2023); Chan et al. (2023); Qian et al. (2024b); Zhuge et al. (2024b). These frameworks enable agents to **collaborate, debate, plan, and reason** across diverse environments while offering scalability and adaptability across tasks. A key distinction is that EL research investigates how communication *emerges from scratch* through interaction and learning, whereas LLM-based communication builds on pre-existing natural language capabilities. Despite this difference, EL offers valuable insights for LLM-based systems, particularly in understanding how structured, compositional, and task-relevant communication can arise from coordination pressures. Bridging these two lines of work suggests a promising path toward hybrid communication frameworks that combine learned behaviors, emergent structure, and the expressiveness of pre-trained language models. LLMs have become attractive for multi-agent systems in part because they address several limitations that persist in EL approaches. Although EL improves interpretability relative to purely latent MARL communication, it typically **requires training from scratch**, careful environment design, and extensive interaction to stabilize meaningful protocols. In contrast, LLMs provide agents with immediate access to rich linguistic structure, commonsense reasoning, and broad world knowledge, enabling zero-shot or few-shot coordination without explicit protocol learning. As a result, the role of communication shifts from a task-specific signaling mechanism to a reusable, semantically grounded interface that can support flexible coordination across tasks and agents.

**EL** and **LLM-based agents** highlight a broader shift in how multi-agent communication is conceptualized, extending beyond the assumptions of conventional **MARL** frameworks. Rather than supplanting earlier paradigms, each line of work emerged in response to concrete limitations in expressiveness, interpretability, scalability, or adaptability encountered at increasing levels of task and interaction complexity. Viewed together, these developments reveal communication not as a fixed design choice, but as an evolving mechanism that co-adapts with agent capabilities, environments, and coordination demands. This evolution motivates **a unified treatment of multi-agent communication** that explicitly traces how communication mechanisms transform as systems move from closed, task-specific settings toward open-ended, human-facing, and deployment-oriented scenarios.

## 1.2 The Need for a Dedicated Survey

Despite substantial progress, most existing work on multi-agent communication is discussed as a subsection within broader MARL surveys Wong et al. (2023); Gronauer & Diepold (2022); Oroojlooy & Hajinezhad (2023); Nguyen et al. (2020); Hernandez-Leal et al. (2019); Zaïem & Bennequin (2019). Given the increasing complexity and diversity of approaches, from MARL-based communication and EL to LLM-driven coordination, there is a growing need for a *dedicated and structured review of MA-Comm*. This survey aims to formalize the field, highlight key research challenges, and provide a roadmap for future research at the intersection of reinforcement learning, natural language processing, and multi-agent cooperation. The main contributions of this paper are as follows:

- **A Unified Survey of Multi-Agent Communication Across Paradigms:** We present the first survey that systematically unifies **MARL-based communication**, **EL**, and **LLM-powered multi-agent systems**, moving beyond traditional MARL-centric views to capture the full evolution of communication in multi-agent decision-making.

- **A Five Ws–Driven Analytical Framework:** We organize the literature using the **Five Ws of communication**, **who** communicates with **whom**, **what** is communicated, **when** communication occurs, **why** communication is needed, and **how** it is motivated and operationalized, providing a consistent lens for comparison across paradigms.

- **Cross-Paradigm Synthesis and Bridging Analysis:** Beyond cataloging methods, we introduce dedicated **bridging sections** that explain how limitations in MARL-Comm motivated EL, and how gaps in MARL and EL led to LLM-based and hybrid LLM–MARL systems, clarifying complementary roles rather than treating paradigms in isolation.

- **Foundational, Formal, and Game-Theoretic Perspectives:** We ground modern communication methods in classical views of communication as action, add concise mathematical formalizations of communication structures, and connect mixed-motive and competitive settings to Nash and Bayesian Nash equilibrium concepts, strengthening theoretical clarity without excessive formalism.

- **Open Challenges and Future Directions:** We identify key open problems spanning **grounding, interpretability, generalization, efficiency, and theoretical guarantees**, and outline future research directions for algorithm design, benchmarking, and human-centric multi-agent communication in increasingly open-ended and safety-critical settings.

**Survey Structure and Organization** In the order of MARL with Communication, EL, and LLM-based Multi-Agent Systems, we structure the paper as follows. Section 2 reviews the foundations of agent communication, showing how modern MARL, EL, and LLM-based approaches extend classical theories of communication as goal-directed, belief-shaping action. Section 3 reviews existing surveys on MARL, emergent communication, and LLM-based agents, identifying the need for a unified perspective on multi-agent communication (MA-Comm). Section 4 outlines the methodological framework and inclusion/exclusion criteria guiding the selection of papers in this survey. Sections 5, 6, 7 present the three core paradigms of multi-agent communication, **MARL with learned communication protocols**, **EL**, **LLM-powered multi-agent communication**. Each section is organized explicitly around the **Five Ws of communication**, **who communicates with whom**, **what is communicated**, **when communication occurs**, **why communication is needed**, **how communication is motivated and operationalized**, providing a unified analytical lens across paradigms. We first introduce essential background, then categorize representative methods according to these dimensions, highlighting how different communication motivations (**why**) shape concrete design choices (**how**). To further strengthen cross-paradigm coherence, we add dedicated **bridging subsections** at the end of each major section, together with a comprehensive **Bridge section** that integrates MARL-, EL-, LLM-based communication, clarifying their relationships, limitations, and complementary roles. Finally, Section 8 synthesizes insights across the survey, discussing key challenges, limitations, and open problems in multi-agent communication, extending beyond implementation-level issues to epistemic perspectives and outlining future research directions on theory, algorithms, benchmarking, and human-centric communication across MARL, EL, and LLM-based systems.

## 2 Foundations of Agent Communication

### 2.1 Foundations of Communication as Action

The study of agent communication builds on a long-standing intellectual tradition that predates modern multi-agent systems, reinforcement learning, and large language models Russell et al. (1995). Early philosophy of language framed communication not as passive information transfer, but as a form of *action*. Speech act theory formalized this view by arguing that utterances perform goal-directed acts such as requesting, committing, or commanding Searle (1969); Austin (1975). Related work by Wittgenstein and Grice further emphasized that meaning depends on context, intention, and shared conventions rather than syntax alone Grice (1957); Arrington (2016). Together, these ideas established a **foundation for viewing communication as a decision-dependent behavior** chosen to influence others' beliefs, intentions, and actions.

This perspective strongly influenced **early artificial intelligence research** on situated language use, where communication was embedded within perception, reasoning, and action rather than treated as an isolated symbolic process Hobbs et al. (1993). In parallel, formal models of language structure—such as context-free grammars and their computational extensions—provided tools for representing and processing linguistic structure Chomsky (1956); Pereira & Warren (1980); Colmerauer (2005), while formal semantics connected language to logic and reasoning through explicit meaning representations Tarski (1956); Montague (1970); Montague & Thomason (1978); Woods (1978). These frameworks enabled principled reasoning about communication but relied heavily on hand-crafted rules and domain-specific representations, limiting scalability in dynamic and multi-agent environments.

Modern learning-based approaches **revisit these foundational ideas under new computational regimes**. Rather than prescribing communication semantics a priori, MARL and EL research allow communication protocols to arise through interaction and optimization Sukhbaatar et al. (2016); Lazaridou et al. (2018). Recent surveys on **foundation agents** further contextualize this evolution by framing communication as one component of modular, brain-inspired agent architectures that integrate memory, world

modeling, goals, reward processing, and social interaction Liu et al. (2025b). From this perspective, contemporary LLM-based and multi-agent systems do not abandon classical views of communication as action, but extend them by coupling language, learning, and coordination within scalable, adaptive, and collaborative agent societies. Many persistent challenges, such as grounding, interpretability, belief alignment, and coordination under uncertainty, can thus be seen as modern instantiations of long-standing foundational questions about how agents act, reason, and communicate to achieve collective objectives.

## 2.2 From Speech Acts to Learned Message Actions

Classical speech act theory views communication as a form of action chosen to achieve specific social or pragmatic effects, such as requesting, committing, or influencing beliefs Austin (1975); Searle (1969). This perspective maps naturally onto modern MARL, where messages are treated as *decision variables* selected by agents to influence future system trajectories Bernstein et al. (2002); Sukhbaatar et al. (2016). In MARL-Comm, sending a message can be understood as executing a learned speech act whose value is defined by its contribution to long-term reward, coordination stability, or belief alignment Singh et al. (2018); Das et al. (2018). Unlike classical speech acts, which rely on shared linguistic conventions and intentional reasoning Grice (1957), MARL message actions derive their semantics implicitly through interaction and optimization. This shift replaces explicit pragmatic rules with reward-driven learning, but preserves the core insight that communication is not passive information exchange; it is an action taken under uncertainty to shape the behavior of others Russell et al. (1995). This connection clarifies why issues such as grounding, interpretability, and generalization arise in learned communication: without explicit semantic constraints, speech-act-like behaviors must be rediscovered through experience rather than assumed a priori Kottur et al. (2017); Lazaridou et al. (2018).

Table 1: Foundational assumptions about communication across classical AI, MARL-based systems, and LLM-based multi-agent systems.

| Aspect | Classical Agent Communication | MARL-Based Communication | LLM-Based Communication |
| --- | --- | --- | --- |
| View of communication | Symbolic action governed by explicit rules (speech acts, logic) | Learned action optimized for reward | Language generation conditioned on prompts and context |
| Semantics | Explicit, hand-defined, logic-based | Implicit, emerges from interaction | Implicit, inherited from pre-training |
| Grounding | Assumed via symbolic world models | Grounded through environmental interaction | Weakly grounded; relies on textual abstractions |
| Interpretability | High (human-readable rules and meanings) | Low to moderate (latent or discrete learned signals) | High at surface level, uncertain at control level |
| Adaptation | Manual rule design | Policy learning via reinforcement signals | Prompting, fine-tuning, or hybrid RL integration |
| Primary limitation | Poor scalability and brittleness | Task specificity and data inefficiency | Lack of control guarantees and grounding |

Table 1 contrasts foundational assumptions about communication across classical AI, MARL-based systems, and LLM-based multi-agent systems. Classical approaches treat communication as a symbolic action with explicitly defined semantics and grounding, offering strong interpretability but limited scalability and adaptability Russell et al. (1995); Austin (1975); Searle (1969). MARL reframes communication as a learned action optimized directly for reward, enabling agents to adapt communication through interaction, but often at the cost of interpretability and generalization beyond co-trained teams Sukhbaatar et al. (2016); Singh et al. (2018); Kottur et al. (2017); Lazaridou et al. (2018). LLM-based systems inherit rich linguistic structure from large-scale pretraining, yielding fluent and human-readable communication, yet they lack strong grounding in environment dynamics and formal control guarantees Park et al. (2023); Mahowald et al. (2024). This comparison highlights that modern hybrid LLM–MARL systems occupy a middle ground, seeking to balance

expressiveness, grounding, and control by combining learning-based communication with language priors (FAIR); Slumbers et al. (2023); Estornell et al. (2025).

## 3 Related Work: Surveying the Three Dimensions of Multi-Agent Communication

Prior efforts to survey communication in multi-agent systems have evolved along several lines. In this work, we focus on three prominent and interrelated dimensions: communication in MARL, emergent language systems, and LLM-driven communication frameworks. While not exhaustive, these dimensions capture a wide spectrum of recent research exploring how agents exchange information to support coordination, learning, and reasoning. Existing MARL surveys often address communication only indirectly, emphasizing policy learning while overlooking communication-specific challenges. Surveys on emergent communication focus on language development and interpretability but are typically siloed from broader multi-agent learning discussions. More recent reviews on LLM-based agents emphasize architectural workflows and coordination capabilities, yet often remain disconnected from prior frameworks. Our survey aims to bridge these perspectives through a unifying framework (Fig. 1) that supports comparative analysis and highlights shared trends, challenges, and open questions across these three influential threads of multi-agent communication research.

### 3.1 Surveys on Communication in MARL

Previous surveys on MARL have primarily provided broad overviews of the field, only occasionally addressing communication as a secondary or supporting topic. Hernandez-Leal et al. Hernandez-Leal et al. (2019) reviewed various MARL methodologies, emphasizing algorithmic frameworks and their critiques, with limited exploration of agent communication strategies. Similarly, Nguyen et al. Nguyen et al. (2020) presented a comprehensive analysis of MARL, highlighting challenges and general solutions but allocating minimal attention to the specifics of communication mechanisms.

Gronauer and Diepold Gronauer & Diepold (2022) conducted an extensive survey focusing predominantly on deep reinforcement learning techniques and their multi-agent extensions. They discussed communication briefly, primarily in the context of agent coordination tasks, without providing an in-depth or structured exploration. Wong et al. Wong et al. (2023) also outlined various challenges and future directions in multi-agent systems, briefly touching upon communication but primarily maintaining a focus on general reinforcement learning issues rather than systematically reviewing communication techniques.

Recent surveys like that of Oroojlooy and Hajinezhad Oroojlooy & Hajinezhad (2023) continue this trend, examining cooperation mechanisms extensively while only superficially addressing communication methodologies, and Beikmohammadi Beikmohammadi (2024) presents a systematic review of 16 MARL communication studies from 2016–2021, focusing on their methodological rigor and identifying gaps such as a lack of standardized environments and consideration of real-world communication constraints. Even the works specifically addressing communication, such as Zaiem et al. Zaïem & Bennequin (2019) and Zhu et al. Zhu et al. (2024) have limitations, either narrowly focusing on MARL-specific methods or not incorporating recent advancements involving emergent languages and large language models (LLMs).

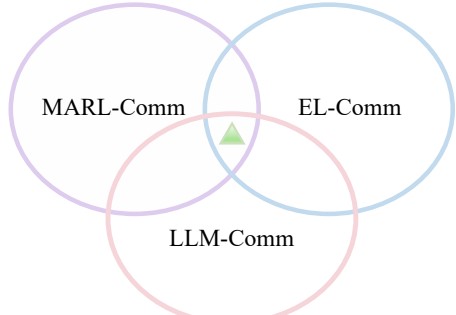

Figure 1: Three core dimensions of multi-agent communication surveyed in this work: MARL-based, emergent language, and LLM-driven communication.

In contrast, our work uniquely addresses the significant gap in the literature by providing the first dedicated, structured survey explicitly focusing on Communication. Our contributions differ significantly from prior surveys in several ways. First, we establish a novel three-dimensional review framework that integrates MARL-based communication approaches, emergent language paradigms, and recent advancements in LLM-driven agent communication. Secondly, we systematically categorize and comprehensively analyze key works across these dimensions, highlighting essential trends, similarities, and differences. Finally, we explicitly outline open

challenges and suggest clear research directions to enhance the efficiency, interpretability, scalability, and applicability of MA-Comm in real-world multi-agent systems.

## 3.2 Surveys on Emergent Communication

Research on emergent language in multi-agent systems has gained traction as a promising pathway toward interpretable and grounded communication. While Boldt and Mortensen Boldt & Mortensen (2024) provide a broad survey covering emergent communication across machine learning, natural language processing, and cognitive science, their review spans beyond the scope of decision-making. In contrast, our focus centers specifically on how emergent language facilitates coordination and policy learning in sequential decision-making settings. They emphasize emergent communication's potential for explainability and human-agent interaction, while also categorizing practical and theoretical use cases. Peters et al. Peters et al. (2024) present a large-scale review of 181 papers, proposing a taxonomy for discrete emergent language. They systematically analyze evaluation metrics and identify challenges in language comparability, compositionality, and semantic alignment. Their contribution is methodological, aiming to standardize how emergent language is assessed. Wolff et al. Wolff et al. (2024) address the limitations of traditional reference games and introduce more realistic, situated environments for emergent language learning. Their work highlights the emergence of bidirectional and sparse communication strategies that arise naturally under coordination pressure in complex MARL settings. Other focused surveys and domain-specific efforts include Chafii et al.Chafii et al. (2023), which explores EC-MARL in future 6G wireless networks, and Gupta et al.Gupta et al. (2020), who analyze emergent communication grounded in graph-based multi-agent topologies. While valuable, these works are contextually bounded and do not offer a general-purpose, integrative perspective.

Prior surveys on emergent language have primarily focused on the linguistic and structural properties of communication protocols, such as syntax, semantics, compositionally, and their resemblance to human language. These studies are often motivated by cognitive science and language understanding, aiming to investigate how structured language can emerge in artificial agents through interaction, without necessarily grounding that communication in task-driven decision-making. In contrast, Boldt and Mortensen's recent survey Boldt & Mortensen (2024) is among the few that explicitly address emergent communication in the context of multi-agent coordination, i.e., how structured language can emerge through agent interactions when multiple agents interact and communicate with each other. This highlights its role not just as a linguistic phenomenon but as a mechanism for enabling effective joint decision-making.

The inclusion of EL in this survey is motivated by the pioneering work of Mordatch & Abbeel (2018), which first established a connection between emergent communication and MARL methods through MAD-DPG Lowe et al. (2017). This work demonstrated how grounded, compositional language can emerge among agents as a mechanism to complete tasks and achieve goals in multi-agent environments. The symbolic and discrete nature of the communication protocols developed in this setting subsequently inspired a range of MARL communication methods, such as Havrylov & Titov (2017); Chen et al. (2024b), which adopted discrete messaging to reduce communication overhead and enhance interpretability. Summarizing EL research in the context of multi-agent communication thus offers an interpretable framework for agent collaboration and promotes the development of human-understandable communication protocols, contributing toward the broader goal of human-centered AI.

Therefore, our survey complements and extends these efforts by incorporating EL as one of the three central dimensions of multi-agent communication. Unlike prior reviews that emphasize emergent communication in isolation, we provide a unified framework that links MARL communication, emergent language, and LLM-driven communication strategies. This broader integration facilitates comparisons, highlights cross-cutting challenges, and sets the stage for designing generalizable and human-aligned communication protocols in future multi-agent systems.

## 3.3 Surveys on LLM-Based Communication

Recent surveys have extensively explored the integration of LLMs into multi-agent systems (MAS), particularly focusing on enhancing inter-agent communication, coordination, and collaboration. Wang et al. Wang et al. (2024a) provide a comprehensive review of LLM-based autonomous agents, categorizing core capabili-

Table 2: Comparison of Surveys Across MA-Comm Dimensions: our survey uniquely bridges three domains, MARL-Comm, EL, and LLM-based MAS, offering a unified framework that highlights cross-cutting trends, systematic methodologies, and open challenges for building scalable, interpretable, and human-centric multi-agent communication systems.

| Survey | Focus Area | Communication Focused | LLM Focused | Emergent Language | Systematic Framework |
|---|---|---|---|---|---|
| Hernandez-Leal et al. (2019) | MARL | ✗ | ✗ | ✗ | ✗ |
| Nguyen et al. (2020) | MARL | ✗ | ✗ | ✗ | ✗ |
| Gronauer & Diepold (2022) | MARL | ▲ | ✗ | ✗ | ✗ |
| Wong et al. (2023) | MARL | ▲ | ✗ | ✗ | ✗ |
| Oroojlooy & Hajinezhad (2023) | MARL | ▲ | ✗ | ✗ | ✗ |
| Beikmohammadi (2024) | MARL-Comm | ✓ | ✗ | ✗ | ✓ |
| Zaïem & Bennequin (2019) | MARL-Comm | ✓ | ✗ | ✗ | ✓ |
| Zhu et al. (2024) | MARL-Comm | ✓ | ✗ | ✗ | ✓ |
| Boldt & Mortensen (2024) | Emergent | ▲ | ✗ | ✓ | ✓ |
| Peters et al. (2024) | Emergent | ✓ | ✗ | ✓ | ✓ |
| Wolff et al. (2024) | Emergent | ✓ | ✗ | ✓ | ✓ |
| Chafii et al. (2023) | Emergent (Wireless) | ▲ | ✗ | ✓ | ✗ |
| Gupta et al. (2020) | Emergent (Graphs) | ▲ | ✗ | ✓ | ✓ |
| Wang et al. (2024a) | LLM | ▲ | ✓ | ✗ | ✓ |
| Li et al. (2024e) | LLM | ✓ | ✓ | ✗ | ✓ |
| Guo et al. (2024b) | LLM | ▲ | ✓ | ✗ | ✓ |
| Yi et al. (2024) | LLM (Dialogue) | ▲ | ✓ | ✗ | ✓ |
| Guan et al. (2025) | LLM (Dialogue Eval.) | ▲ | ✓ | ✗ | ✓ |
| Hu et al. (2024b) | LLM (Games) | ▲ | ✓ | ✗ | ✓ |
| Tran et al. (2025a) | LLM (Collaboration) | ✓ | ✓ | ✗ | ✓ |
| Sun et al. (2025b) | LLM (Coordination) | ▲ | ✓ | ✗ | ✓ |
| Sun et al. (2024a) | LLM (MARL) | ▲ | ✓ | ✗ | ✓ |
| Mosquera et al. (2024) | LLM (Evaluation) | ✓ | ✓ | ✗ | ✗ |
| Cemri et al. (2025b) | LLM (Evaluation) | ✓ | ✓ | ✗ | ✗ |
| Liu et al. (2025a) | LLM (Future) | ▲ | ✓ | ✗ | ✓ |
| Aratchige & Ilmini (2025) | LLM (Technological Aspects) | ✓ | ✓ | ✗ | ✓ |
| **Our Survey** | **All Above** | ✓ | ✓ | ✓ | ✓ |

ties such as communication, planning, memory, and tool use. Li et al. Li et al. (2024e) offer a workflow-centric perspective, outlining key infrastructure components including profile modeling, self-action reasoning, perception, interaction, and learning in LLM-based MAS.

Guo et al. Guo et al. (2024b) survey the progress and challenges of large language model-based multi-agents, covering aspects such as reasoning, cooperation, and agent modularity. In the context of dialogue systems, Yi et al. Yi et al. (2024) and Guan et al. Guan et al. (2025) analyze recent advances and evaluation strategies for LLM-based multi-turn conversations, highlighting communication bottlenecks and coordination failures. Hu et al. Hu et al. (2024b) focuses on LLM-based agents in game environments, discussing planning, interaction dynamics, and decision-making complexity. While these surveys highlight the growing role of LLMs in multi-agent decision-making, they largely treat communication as a secondary concern rather than a primary focus, leaving open the need for a more targeted synthesis of communication strategies in LLM-powered multi-agent systems.

Several surveys have examined collaboration and coordination among LLM agents more directly. Tran et al. Tran et al. (2025a) review mechanisms for multi-agent collaboration using LLMs, while Sun et al. Sun et al. (2025b) survey coordination strategies across diverse application domains. Sun et al. Sun et al. (2024a) specifically explore how LLMs are being integrated with MARL, summarizing current progress and proposing future research directions. Aratchige and Ilmini Aratchige & Ilmini (2025) further examine the technological challenges in building effective LLM-based multi-agent systems, addressing issues like communication reliability, shared memory, and uncertainty handling.

Complementary to these broad surveys, several evaluation studies have investigated the fundamental limitations of LLM-based multi-agent cooperation. Mosquera et al. Mosquera et al. (2024) empirically assess cooperative behaviors in LLM-augmented agents using the Melting Pot benchmark, while Cemri et al. Cemri et al. (2025b) provide an in-depth analysis of why multi-agent LLM systems often fail, identifying critical weaknesses in alignment, communication consistency, and robustness. Moreover, Liu et al. Liu et al. (2025a)

present a broader vision of 'foundation agents', exploring brain-inspired architectures, evolutionary collaboration mechanisms, and the pursuit of safe, scalable multi-agent intelligence.

In contrast to existing surveys that primarily focus on architectural capabilities, evaluation benchmarks, or emerging challenges in LLM-based multi-agent systems, our work frames LLM-enabled communication as one of three central paradigms within a broader taxonomy of MA-Comm. By connecting LLM-driven interaction with traditional MARL communication protocols and EL systems, we provide an integrative perspective that spans diverse research communities. While MA-Comm addresses a broad range of coordination and decision-making objectives beyond human involvement, the incorporation of LLMs introduces new opportunities for more natural, interpretable, and potentially human-aligned communication. This broader lens helps uncover shared challenges and promising directions for advancing communication strategies across agent types and application domains.

Table 2 provides a structured comparison of major surveys across the key dimensions of MA-Comm. While earlier works focus narrowly on either MARL communication, emergent language, or LLM-based systems, none comprehensively address all three paradigms together. In contrast, our survey uniquely bridges these domains, offering a unified framework that highlights cross-cutting trends, systematic methodologies, and open challenges for building scalable, interpretable, and human-centric multi-agent communication systems.

## 4    Methodology and Scope of the Survey

This survey follows a systematic literature review methodology to analyze research on multi-agent communication across three closely related but traditionally separate domains: communication in multi-agent reinforcement learning (MARL-Comm), emergent language (MA-EL), and large language model–based multi-agent communication (LLM-Comm). Our goal is not to exhaustively enumerate every paper in these areas, but to provide a principled, representative synthesis that highlights core modeling assumptions, methodological trends, and open challenges across domains.

### 4.1    Search Strategy and Temporal Scope

We conducted a structured literature search covering works published before 2025. To capture foundational context, a small number of earlier seminal works (e.g., Dec-POMDPs, classical team decision theory, and communication-constrained control) were included where necessary to ground later developments. Primary searches were performed using Google Scholar with combinations of keywords such as *multi-agent communication*, *MARL communication*, *emergent language*, *signaling games*, *multi-agent LLM*, and *language-based coordination*. In addition, we manually examined proceedings of major AI and machine learning venues (NeurIPS, ICML, ICLR, AAAI, AAMAS) and relevant workshops. Backward and forward citation tracking was used to identify influential works not surfaced by keyword search alone. For each search term, the stopping criterion was the absence of new relevant papers on a full results page.

### 4.2    Inclusion and Exclusion Criteria

Papers were **included** in this survey based on the following criteria: (1) the work explicitly treats **communication between agents** as a core component of the problem formulation or solution, rather than as a peripheral implementation detail; (2) the setting involves **multiple decision-making agents**, including cooperative, competitive, or mixed-motive scenarios; (3) the paper presents a **methodological contribution**, empirical evaluation, or formal framework related to communication learning, representation, or usage; and (4) the work is published in a peer-reviewed journal, conference, workshop, thesis, or book chapter, or is a **technically substantive arXiv preprint** included selectively to reflect recent advances in fast-moving subfields (e.g., LLM-based multi-agent communication) that are not yet fully represented in archival venues.

Conversely, papers were **excluded** if (1) communication was mentioned only superficially without being modeled or analyzed; (2) the work focused solely on single-agent learning or fully centralized control without decentralized execution; (3) the paper was purely conceptual or a position piece without technical substance; or (4) the work was available only as a non-peer-reviewed preprint and lacked sufficient technical depth,

empirical validation, or methodological clarity to warrant inclusion, despite the fast-paced nature of emerging areas such as LLM-based multi-agent communication.

These inclusion and exclusion criteria were applied consistently across all three domains considered in this survey (MARL-Comm, MA-EL, and LLM-Comm), with domain-specific judgment used only to account for differences in terminology, experimental conventions, and evaluation practices.

### 4.3 Coverage Across Domains and Selection for Detailed Discussion

The initial search yielded several hundred candidate papers across the three domains. After applying the inclusion and exclusion criteria, we retained a curated corpus of representative works spanning MARL-Comm, emergent language, and LLM-based multi-agent systems. Within this corpus, a subset of papers is discussed in depth in the main text and tables. These papers were selected based on one or more of the following factors: (1) the introduction of a widely adopted communication architecture, framework, or benchmark; (2) an explicit or implicit investigation of **who communicates with whom**, **what**, **when**, **why**, and **how** agents communicate, especially works that analyze how the motivation for communication (**why**) directly informs the design of communication mechanisms (**how**); such works are discussed and compared in depth in the main text; (3) demonstrated influence on subsequent research, as evidenced by citation patterns or follow-up works; and (4) the ability to illustrate contrasts or connections across domains (e.g., between MARL-Comm and LLM-Comm). The remaining works are cited or summarized more briefly to provide breadth and contextual coverage without redundancy. This balance reflects an intentional trade-off between comprehensive coverage and analytical depth, ensuring that core ideas are examined rigorously while situating them within the broader literature.

### 4.4 Limitations of Scope

We note that the rapid pace of research—particularly in LLM-based multi-agent systems—means that new results continue to emerge beyond the temporal scope of any fixed survey. Accordingly, this survey reflects the state of the literature up to August 2024 and prioritizes works that clarify fundamental modeling assumptions, communication mechanisms, and design trade-offs, rather than attempting exhaustive coverage of all contemporaneous results. The methodological criteria described above are intended to make these selection decisions transparent, principled, and reproducible. In total, this survey considers approximately 400 works across MARL-Comm, MA-EL, and LLM-Comm, with approximately 100–130 representative papers examined in depth through detailed taxonomies, comparative tables, and focused discussion.

## 5 Communication in Multi-Agent Reinforcement Learning

Communication in **Multi-Agent Reinforcement Learning (MARL-Comm)** plays a central role in enabling agents to coordinate, share information, and act collectively in dynamic and partially observable environments. As agents operate under decentralized execution, effective communication must address fundamental challenges such as limited observability, bandwidth constraints, and non-stationarity induced by learning teammates. Early MARL systems relied on hand-designed protocols, but recent advances have enabled agents to *learn when, what, and how to communicate* end-to-end as part of the decision-making process. This shift has positioned communication as a first-class control mechanism, tightly coupled with policy learning and long-horizon coordination.

**Review Scope and Structure** This section begins with background and foundational concepts in MARL-Comm (Section 5.1), followed by an overview of related work and key research directions (Section 5.2). We then organize the MARL-Comm literature using an updated Five Ws perspective. Specifically, we examine **who communicates with whom and what is shared** through message relevance, recipient selection, and information encoding (Section 5.3); **when communication occurs** under timing, frequency, and resource constraints (Section 5.4); and **why communication is beneficial and how it is operationalized** via training objectives, protocol design, and control shaping (Section 5.5). We conclude with a discussion that synthesizes insights across the Five Ws (Section 5.6) and a bridge section that motivates the transition from MARL-Comm to emergent language (Section 5.7). Figures 2 and 3 summarize the resulting taxonomy and

end-to-end MARL communication pipeline. Throughout the section, we highlight representative algorithms, evaluation settings, and design trade-offs, grounding all discussions within the Markov Decision Process framework Puterman (2014) to provide a unified and extensible foundation for communication-centric MARL research.

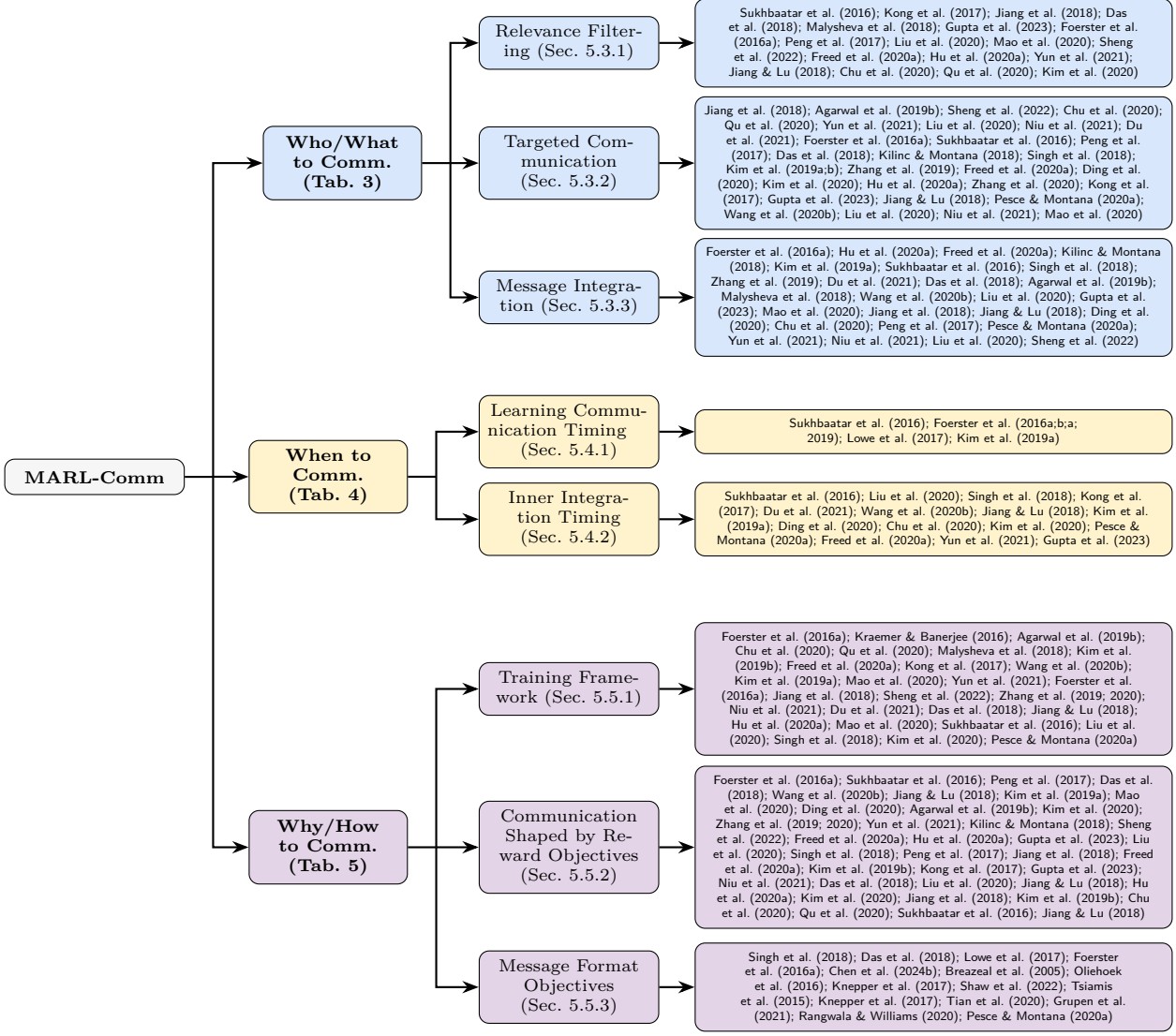

Figure 2: MARL-Comm Agents Taxonomy

## 5.1 Background of MARL-Comm

Reinforcement learning trains models to make sequential decisions in uncertain environments by maximizing cumulative rewards through trial and error Sutton & Barto (2018). While it is traditionally designed for single agents, many reinforcement learning applications involve multiple agents, making MARL a more appropriate framework Busoniu et al. (2008). MARL studies decision-making among agents in shared environments, with applications in cyber-physical systems Adler & Blue (2002); Wang et al. (2016), communication networks Cortes et al. (2004); Choi et al. (2009), and social science Castelfranchi (2001); Leibo et al. (2017). MARL problems can be fully cooperative, fully competitive, or mixed, depending on the agents' objectives. Multi-agent communication is a fundamental aspect of MARL, enabling agents to share information, coordinate strategies, and achieve collective objectives efficiently. In MARL, agents operate in

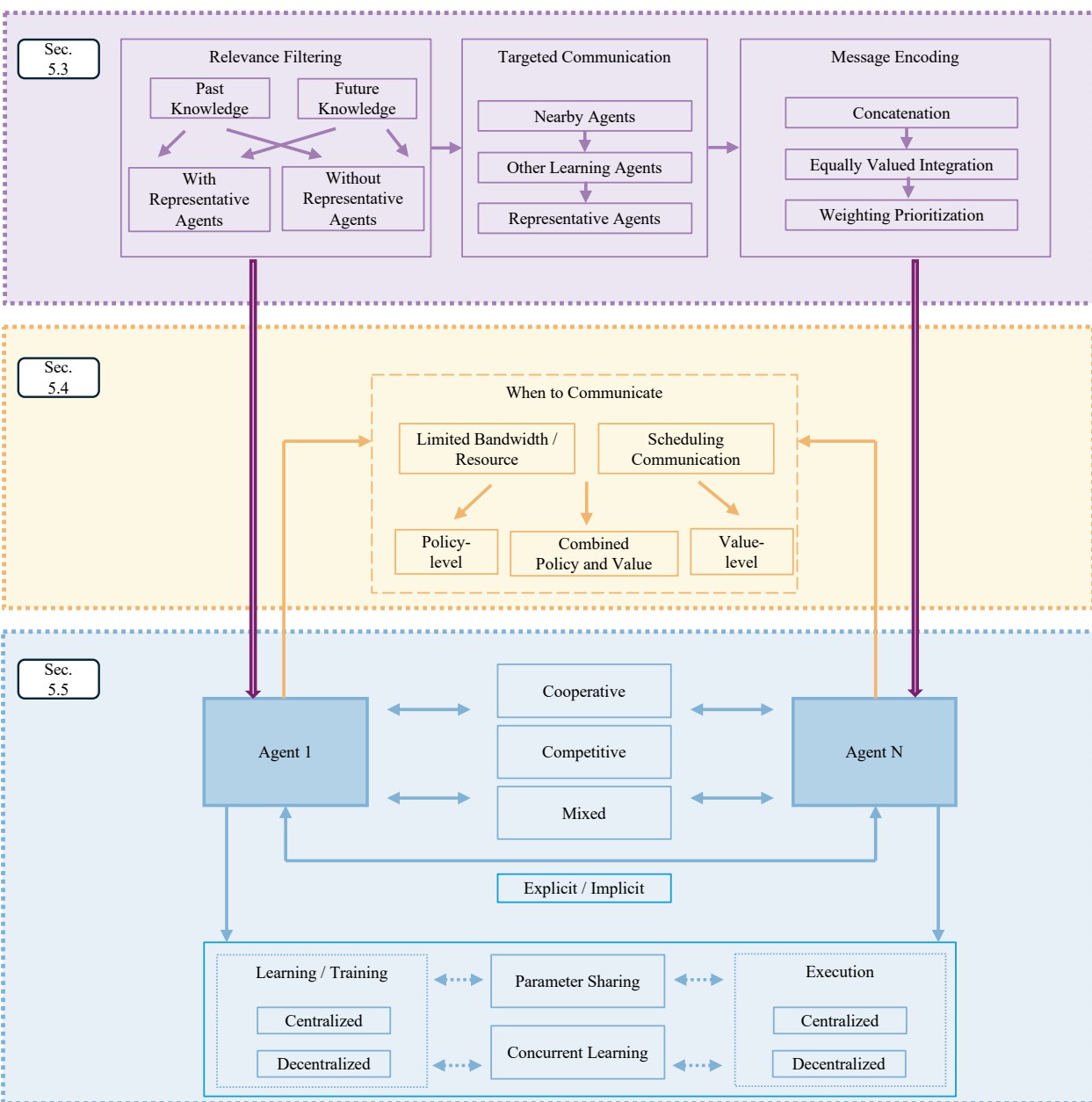

Figure 3: Organization of the key dimensions of communication design in Sec. 5: what and whom to communicate with (Sec. 5.3), when to communicate under resource constraints (Sec. 5.4), why communicate and how communication is shaped by interaction scenarios and learning architectures (Sec. 5.5). It highlights how various communication mechanisms are conditioned on task structure, bandwidth limitations, agent relationships, and training/execution paradigms.

a shared environment, and their joint actions influence the state transitions and rewards. Communication allows agents to overcome partial observability, align their policies, and avoid conflicts or redundant actions.

**Formal Framework**   Decentralized Partially Observable Markov Decision Processes (Dec-POMDPs) Bernstein et al. (2002) are a multi- agent extension of a partially observable Markov decision process, which models cooperative MARL, where agents lack complete information about the environment and have only local observations. Communication is crucial for strategy coordination in this scenario. A Dec-POMDP with communication is often modeled as a tuple Chen et al. (2024b) $D = \langle S, A, P, \Omega, O, I, n, R, \gamma, f \rangle$, where $I = \{1, 2, \ldots, n\}$ is a set of $n$ agents, $S$ is the joint *state* space and $A = A_1 \times A_2 \times \cdots \times A_n$ is the joint *action* space. Here $\boldsymbol{a} = (a_1, a_2, \ldots, a_n) \in A$ denotes the joint action of all agents and $A_i$ is the individual action space of agent $i$. The joint action influences the environment, which transitions to a new state and provides a global reward $r$. $P(\boldsymbol{s}'|\boldsymbol{s}, \boldsymbol{a}) : S \times A \times S \to [0, 1]$ is the *state transition function*. $\Omega$ is the *observation* space. $O(\boldsymbol{s}, i) : S \times I \to \Omega$ is a function that maps from the joint state space to distributions of observations for each agent $i$. $R(\boldsymbol{s}, \boldsymbol{a}) : S \times A \to \mathbb{R}$ is the *reward function* in terms of state $s$ and joint action $\boldsymbol{a}$, and $\gamma$ is the discount factor. Communication can be modeled as an additional action space, where agents send and receive messages $m_i$ to share information. At each time step, agent $i$ generates a message $m_i = f_i(o_i, h_i)$ based on its local observation $o_i \in O$ and internal state $h_i \in H$, where $f_i : O \times H \to M$ is a message-generation function that maps observations and internal states to a message space $M$, often implemented as a neural network. Each agent $i$ receives messages $\{m_j\}_{j \in \mathcal{N}_i}$ from a (possibly dynamic) subset of other agents $\mathcal{N}_i \subseteq \{1, \ldots, N\} \setminus \{i\}$, and aggregates them into a context vector $c_i = g_i(\{m_j\}_{j \in \mathcal{N}_i})$, where $g_i$ is an aggregation function such as averaging, concatenation, or attention. The agent's policy $\pi_i$ uses the aggregated context $c_i$ along with its local observation $o_i$ to select an action $a_i \sim \pi_i(o_i, c_i)$. This framework highlights the dual role of communication: (1) enabling agents to share local observations and (2) facilitating coordination by aligning their policies based on shared information.

Based on this background, we provide a review of communication-based MARL algorithms in the following subsections, with a particular focus on works whose primary novelty lies in the design of communication strategies. One category of such algorithms aims to enhance cooperative multi-agent learning by improving the efficiency and relevance of the communication process. These works either introduce mechanisms such as gating and selective message filtering to reduce redundant communication or leverage advanced techniques like attention mechanisms and graph-based communication protocols to ensure meaningful message exchange. Another category focuses on extended MARL setups, such as hierarchical or task-specific communication, utilizing the idea that communication messages can serve as embeddings or representations of tasks or roles within the multi-agent system.

## 5.2   Overview and Taxonomy of MARL-Comm

Communication plays a crucial role in MARL by enabling agents to share information, coordinate effectively, and tackle complex tasks in partially observable environments. Early foundational works, such as CommNet Sukhbaatar et al. (2016), DIAL, and RIAL Foerster et al. (2016a), pioneered communication learning for cooperative agents. CommNet uses a shared neural network to process local observations and aggregates messages via mean pooling for decentralized decision-making. While effective in small-scale settings, it struggles with scalability and communication delays. DIAL and RIAL introduced more explicit communication protocols, where agents exchange discrete or continuous messages and jointly learn both their policies and communication strategies through reinforcement learning. These approaches, while innovative, require fully connected communication networks, leading to high bandwidth demands as the number of agents increases.

To address the scalability and efficiency challenges, subsequent research has focused on adaptive and selective communication mechanisms. For instance, IC3Net Singh et al. (2018) employs gating mechanisms to allow agents to dynamically decide whether to communicate based on their current states, reducing redundant messages. TarMAC Das et al. (2018) introduces multi-headed attention to improve the relevance of communicated information, enabling selective sharing among agents. Similarly, ATOC Jiang & Lu (2018) forms dynamic communication groups based on agent proximity and employs bi-directional LSTMs to aggregate messages within groups. More advanced methods, such as SARNet Rangwala & Williams (2020), incorporate structured reasoning and attention mechanisms to assess the relevance of received messages and past memories, enabling more nuanced decision-making. These methods collectively explore key questions such

as *when*, *what*, and *with whom* agents should communicate, offering significant performance improvements in MARL tasks.

Recent work also explores communication constraints, such as limited bandwidth and noisy channels. For example, SchedNet Kim et al. (2019a) selects only a subset of agents to communicate, optimizing within bandwidth limits, while IMAC Wang et al. (2020b) regularizes message entropy to reduce communication overhead. These approaches emphasize the trade-offs between communication efficiency and performance, paving the way for practical MARL systems in real-world scenarios. To further advance MARL-Comm, new methods must address challenges such as scalability, interpretability, and performance guarantees under realistic constraints Chen et al. (2024b). Future work should explore integrating sparse, discrete communication protocols with theoretical guarantees, such as regret minimization, to enhance robustness and applicability in large-scale environments Chen et al. (2024b).

Based on the foundational advancements in MARL-Comm, we provide a review of communication-based MARL algorithms in the following subsections, with a particular focus on works whose primary novelty lies in designing efficient and scalable communication strategies. One category of such algorithms seeks to enhance MARL-Comm by introducing mechanisms like attention or gating to optimize when, what, and with whom agents should communicate. These approaches aim to reduce redundant communication, improve message relevance, and address practical constraints such as limited bandwidth or noisy channels.

**Key Challenges** Despite its potential, multi-agent communication in MARL faces several core challenges: **(1). Scalability in Large State and Agent Spaces:** In large-scale environments, each agent typically observes only a small portion of the global state. Agents often have limited local observations, making it difficult to infer the global state or the intentions of other agents. Communication is essential for enabling coordination under these conditions, but exchanging high-dimensional or unfiltered observations among many agents can be inefficient and computationally prohibitive Cao et al. (2013). The challenge lies in designing scalable protocols that selectively share information while preserving task-relevant signals. **(2). Communication Overhead:** As the number of agents increases, communication complexity grows rapidly, leading to significant overhead. Techniques such as message filtering, attention mechanisms, and hierarchical structures aim to mitigate this. For instance, CommNet Sukhbaatar et al. (2016) uses a shared centralized architecture to enable continuous communication, but it does not scale well with the number of agents. MADDPG Lowe et al. (2017) introduces a centralized critic and decentralized actors, where agents generate communication as part of their actions. However, its policy network often overfits to specific agent configurations, limiting generalizability in large-scale scenarios. **(3). Non-Stationarity:** In MARL, each agent operates in a non-stationary environment because other agents are concurrently learning and updating their policies. This dynamic context makes it difficult to learn stable communication strategies, as the meaning and utility of messages may shift over time. **Credit Assignment:** In cooperative settings, determining the contribution of each agent to the collective reward is non-trivial, yet essential for effective learning. Without proper credit assignment, agents cannot accurately update their policies based on individual impact, which can lead to suboptimal or unstable learning dynamics. Communication can help agents better coordinate their actions, but it also introduces additional dependencies between agents' behaviors and messages, making it more complex to attribute success or failure to specific contributions. **(5). Interpretability:** Learned communication protocols in MARL are often embedded in latent continuous spaces, making them hard to interpret. Unlike hand-crafted protocols with explicit semantics, learned messages lack transparency, making it difficult to analyze their meaning, influence on decision-making, or alignment with human reasoning. **(6). Lack of Theoretical Guarantees:** Many MARL-Comm methods rely on heuristic mechanisms for learning continuous or discrete communication Foerster et al. (2016a); Singh et al. (2018); Jiang & Lu (2018); Das et al. (2018); Rangwala & Williams (2020); Havrylov & Titov (2017); Mordatch & Abbeel (2018). While some approaches model discrete, human-like messages, they often lack formal guarantees such as policy convergence or regret minimization, limiting their theoretical robustness and practical reliability.

**Key Methodologies** Several approaches have been proposed to address these challenges, which can be broadly categorized as: **(1). Centralized Communication**, such as CommNet (Sukhbaatar et al. (2016)), uses a shared neural network to aggregate messages from all agents and compute a global context. The message generation and aggregation steps are formalized as: $m_i = f_i(o_i, h_i); \quad c_i = \frac{1}{N-1} \sum_{j \neq i} m_j$, where $c_i$ is the

averaged message from all other agents. While this approach simplifies coordination, it introduces a single point of failure and may not scale well to large systems. **(2). Decentralized Communication**, such as Tar-MAC (Das et al. (2018)), enables agents to communicate directly with each other using attention mechanisms. The message aggregation step is formalized as: $c_i = \sum_{j \neq i} \alpha_{ij} m_j, \quad \alpha_{ij} = \text{softmax}(u_i^T W u_j)$, where $\alpha_{ij}$ is the attention weight between agents $i$ and $j$, $u_i$ and $u_j$ are learned embeddings, and $W$ is a learned weight matrix. This approach is more scalable but requires robust protocols to handle message delays and redundancies. **(3). Learning to Communicate**, such as IC3Net (Singh et al. (2018)), focuses on end-to-end learning of communication protocols. The policy update step is formalized as: $a_i \sim \pi_i(o_i, c_i), \quad c_i = g_i(\{m_j\}_{j \neq i})$, where $g_i$ is a learned aggregation function. This approach allows agents to develop communication strategies that are tailored to the specific task and environment. **(4). Emergent Communication**, such as in Mordatch & Abbeel (2018); Lazaridou et al. (2016), studies how agents can develop their emergent communication protocols through interaction, without explicit supervision. The message generation step is formalized as: $m_i = f_i(o_i, h_i), \quad f_i$ is learned through reinforcement learning. This approach is often studied in the context of cooperative games or language learning tasks. We will use a separate section to introduce this line of work in Sec. 6. **(5). LLM-Guided Communication:** Approaches such as Li et al. leverage Large Language Models (LLMs) to guide agent communication toward more interpretable, natural language-like structures. The message generation process is formalized as $m_i = f_i(o_i, h_i, \text{LLM})$, where $f_i$ is a function conditioned not only on the agent's observation $o_i$ and internal state $h_i$, but also informed by a pre-trained LLM. This integration allows agents to generate structured, human-understandable messages while preserving or even enhancing task performance. LLM-guided communication has shown promise in tasks like collaborative planning and ad-hoc teamwork, leading to improved generalization and faster communication emergence. We provide a detailed discussion of this research direction in Sec. 7. **(6). Return Gap Minimization**, such as in Chen et al. (2024b), quantifies the gap between the optimal expected average return of an ideal policy with full observability, i.e., $\pi^* = [\pi_i^*(a_i|o_1, \ldots, o_n), \forall i]$ and the optimal expected average return of a communication-enabled, partially-observable policy, i.e., $\pi = [\pi_i(a_i|o_i, \boldsymbol{m_{-i}}), \forall i]$. This result enables such works to recast multi-agent communication into a novel online clustering problem over the local observations at each agent, with messages as cluster labels and the upper bound on the return gap as clustering loss.

### 5.3 Who, Whom and What to Communicate: Information and Recipient Selection

One of the main challenges in MARL-Comm is determining what information should be shared among which agents in order to maximize coordination while minimizing redundant or unnecessary communication. Previous surveys have often treated **what** to communicate and **who** or **whom** to communicate with as separate topics, even though the two are tightly coupled in both problem formulation and algorithmic design. In practice, selecting message content and selecting recipients are usually handled jointly Das et al. (2018); Rangwala & Williams (2020), and many methods learn these choices together. This section therefore presents a unified discussion in which both **information selection and recipient selection** are viewed as part of the question of 'what to communicate'. We identify three key challenges in jointly determining *what* to communicate and *to whom*: (1). **Relevance Filtering:** Determining which information is essential for coordination and selecting the recipients who will benefit from receiving it. (2). **Avoiding Redundancy:** Ensuring messages are sent only to agents who cannot infer the information independently, thereby reducing unnecessary or duplicated communication. (3). **Compression and Encoding:** Constructing compact message representations tailored to the intended recipients while meeting bandwidth and scalability constraints.

### 5.3.1 Relevance Filtering of Messages and Recipients

A central component of determining *what* to communicate is identifying *which agents* should receive each piece of information. Relevance filtering therefore involves jointly selecting message content and appropriate recipients, ensuring that communication is both informative and directed toward agents for whom the information is actionable. After communication links are established by a communication policy, agents must determine which information should be shared and which recipients will benefit from receiving it. Under partial observability, local observations become especially valuable, and agents often rely on historical experience, intended actions, or anticipated future plans to generate informative messages tailored to specific recipients.

**Encoding Past Knowledge**   Past observations, actions, and internal states provide a rich source of information for relevance filtering. These signals are frequently encoded into compact messages that are selectively routed to relevant recipients. Models from the Recurrent Neural Networks (RNN) family (e.g., Long Short-Term Memory (LSTM), Gated Recurrent Unit (GRU)) are especially effective for this purpose, as they capture temporal structure and selectively preserve information that may be useful for downstream decision-making. Representative works encode either historical observations Sukhbaatar et al. (2016); Kong et al. (2017); Jiang et al. (2018); Das et al. (2018); Malysheva et al. (2018); Kilinc & Montana (2018); Singh et al. (2018); Pesce & Montana (2020a); Zhang et al. (2019); Wang et al. (2020b); Liu et al. (2020); Mao et al. (2020); Sheng et al. (2022); Freed et al. (2020a); Hu et al. (2020a); Zhang et al. (2020); Gupta et al. (2023); Niu et al. (2021); Du et al. (2021); Yun et al. (2021) or action–observation histories Foerster et al. (2016a); Peng et al. (2017), depending on how message relevance is defined for different recipients. Nevertheless, if there is a representative agent, messages will be generated and transformed from agents to the representative agent, and then from the representative agent to agents. Therefore, we differentiate recent works on using existing knowledge as messages into the following two cases.

- **Through Representative Agents:** When using a representative agent, communicated messages are generated in two stages. First, local observations can be encoded Sukhbaatar et al. (2016); Kong et al. (2017); Jiang et al. (2018); Das et al. (2018); Malysheva et al. (2018) or directly sent Das et al. (2018); Malysheva et al. (2018) to the representative agent. Then, the representative agent aggregates these local (encoded) observations and produces either a single global message for all agents Gupta et al. (2023) or individualized messages tailored to each agent Sukhbaatar et al. (2016); Kong et al. (2017); Jiang et al. (2018); Das et al. (2018); Malysheva et al. (2018). This centralized mechanism removes the burden of message integration from individual agents. Because the representative agent is responsible for determining both the structure and the routing of the aggregated information, the design intrinsically links *what* information is communicated with *whom* it is intended for. Systems such as CommNet Sukhbaatar et al. (2016) and MS-MARL-GCM Kong et al. (2017) generate a shared global message that captures information relevant to all teammates, while architectures like HAMMER Gupta et al. (2023) or TarMAC Das et al. (2018) enable agent-specific messages, reflecting differentiated informational needs across recipients. In these settings, the representative module operationalizes recipient selection by shaping the content and distribution of messages simultaneously.

- **Without Representative Agents:** Without a representative agent, messages are exchanged directly among agents. DIAL and RIAL Foerster et al. (2016a) encode past observations and actions into messages. BiCNet Peng et al. (2017) shares both local and global observations among agents. Other approaches directly communicate raw observations Liu et al. (2020), or rely on feed-forward networks Liu et al. (2020); Mao et al. (2020); Sheng et al. (2022); Freed et al. (2020a), MLPs Freed et al. (2020a); Hu et al. (2020a), autoencoders Freed et al. (2020a), CNNs Liu et al. (2020); Mao et al. (2020), RNNs Sukhbaatar et al. (2016); Kong et al. (2017); Jiang et al. (2018); Das et al. (2018); Malysheva et al. (2018), or GNNs Liu et al. (2020); Freed et al. (2020a) to encode local observations for communication. In decentralized settings, selecting message content is inseparable from determining which agents require that information: broadcasting, neighborhood-based routing, or attention-guided message passing all implicitly define *whom* the information is meant for. Methods such as DIAL Foerster et al. (2016a), BiCNet Peng et al. (2017), or GNN-based communication Liu et al. (2020); Freed et al. (2020a) thus jointly learn the relevance of information and the appropriate recipients, reflecting a direct coupling between *what* is transmitted and *who* benefits from it.

  In addition, some methods communicate more specialized information. For example, in GAXNet Yun et al. (2021), agents exchange attention weights to coordinate the combination of hidden states from neighboring agents, providing fine-grained recipient-aware message content.

**Encoding Future Knowledge**   We use 'Future Knowledge' to refer to either intended actions Jiang & Lu (2018), a policy fingerprint (i.e., current action probabilities in a particular state) Chu et al. (2020); Qu et al. (2020), or future plans Kim et al. (2020), which can be generated by simulating a model of environment dynamics Kim et al. (2020). As intentions and plans are state-related, recent works usually

encode intentions together with local observations to generate more relevant messages. Because future-oriented signals are meaningful only to agents whose decisions depend on anticipating the sender's behavior, this category naturally couples *what* is communicated with *whom* it is communicated to. In intention-based methods such as ATOC Jiang & Lu (2018) or fingerprint-sharing approaches Chu et al. (2020); Qu et al. (2020), the shared information is selectively transmitted to teammates whose policies must coordinate tightly with the sender. Planning-based communication Kim et al. (2020) similarly directs predictive rollouts or abstracted plans to agents whose future actions are interdependent. Across these approaches, future knowledge functions as a targeted communication mechanism in which message content is shaped by the specific recipients who benefit most from predictive information.

### 5.3.2 Redundancy Reduction via Targeted Communication

A key feature of human communication is the ability to engage in targeted interactions. Instead of broadcasting messages to all agents, as explored in previous work CommNet Sukhbaatar et al. (2016), RIAL and DIAL Foerster et al. (2016a), and IC3Net Singh et al. (2018), directing specific messages to intended recipients can enhance collaboration in complex environments. For instance, in a team of search-and-rescue robots, a message like 'smoke is coming from the kitchen' is relevant to a firefighter but irrelevant to a bomb defuser. Targeted communication therefore directly reflects the "whom" dimension, specifying which agents are expected to receive and act on a given piece of information. In this section, we focused on how agents learn **whom to communicate** to while performing cooperative tasks in partially observable environments. The communicatee type defines which agents are potential recipients of messages in a MARL-Comm system. In the literature, communicatee types can be categorized based on whether agents communicate directly with one another or through a representative agent.

**Communicating with Nearby Agents** In many MARL systems, communication is only allowed between nearby agents. Nearby agents are defined in various ways, such as agents that are observable Yun et al. (2021), agents within a certain distance Jiang et al. (2018); Agarwal et al. (2019b); Sheng et al. (2022), or agents that are neighbors on a graph Chu et al. (2020); Qu et al. (2020). Neighboring agents can also emerge during learning instead of being predefined, as in GA-Comm Liu et al. (2020), MAGIC Niu et al. (2021), and FlowComm Du et al. (2021), which explicitly learn a graph structure among agents. In these works, "whom" is determined either by spatial proximity or by a learned interaction graph, binding message relevance to a specific recipient set. However, in GA-Comm and MAGIC, a central unit (e.g., GNN) learns the graph structure and coordinates messages, so the agents do not communicate directly and are instead managed through a representative agent. Dynamic Graph Networks (DGN) Jiang et al. (2018) restrict communication to the three closest neighbors based on a predefined distance metric. This locality-based communication is particularly important in dynamic environments, where agents' positions and states change frequently, requiring timely and context-specific message exchanges to maintain effective coordination and avoid communication bottlenecks. Similarly, the Agent-Entity Graph Agarwal et al. (2019b) also uses distance to measure proximity, allowing communication between agents as long as they are close to each other. MAGNet-SA-GS-MG Malysheva et al. (2018) uses attention mechanisms to optimize communication by selecting the most relevant messages, with a pretrained graph limiting communication to neighboring agents. Learning Structured Communication (LSC) Sheng et al. (2022) enables agents within a cluster radius to decide whether to become a leader agent, with all non-leader agents communicating with the leader in that cluster. This structured communication protocol improves both communication efficiency and interpretability. NeurComm Chu et al. (2020) and Intention Propagation (IP) Qu et al. (2020) restrict communication to neighbors on a preset graph structure during learning, facilitating coordination by sharing decision-making policies and intentions (i.e., action probabilities) with neighboring agents. FlowComm Du et al. (2021) dynamically manages communication using message-passing techniques in a flow-based system, optimizing the routing of information based on environmental flow, while also learning a graph structure to identify neighboring agents. Lastly, GAXNet Yun et al. (2021) uses attention mechanisms to combine hidden states from observable neighboring agents, allowing for selective message passing that helps agents focus on critical information while filtering out irrelevant data. Across these methods, both the message content and the chosen recipients are co-determined by spatial, structural, or learned adjacency relations.

**Communicating with Other Learning Agents**   When nearby agents are not explicitly defined (e.g., via proximity-based graphs), communication typically occurs among all learning agents or through mechanisms that selectively determine recipients. Several methods support communication in this scenario by either broadcasting messages or learning whom to communicate with. DIAL (Differentiable Inter-Agent Learning) Foerster et al. (2016a) enables differentiable communication between agents, allowing them to optimize message content jointly via backpropagation. RIAL (Reinforced Inter-Agent Learning) Foerster et al. (2016a) instead uses reinforcement learning to select messages that maximize collective rewards, assuming a fixed set of recipients. CommNet Sukhbaatar et al. (2016) facilitates communication by averaging hidden states across all agents, implicitly assuming full communication among teammates. BiCNet (Bidirectionally Coordinated Network) Peng et al. (2017) applies RNNs to support recurrent communication of local and global observations, again relying on shared message exchange among all agents. TarMAC (Targeted Multi-Agent Communication) Das et al. (2018) introduces learned attention weights that determine *whom* to send messages to at each step, enabling targeted communication and reducing unnecessary message passing. MADDPG-M (Multi-Agent DDPG with Messages) Kilinc & Montana (2018) extends the MADDPG framework with continuous message exchange but does not explicitly control recipient selection, assuming messages are accessible to all agents. IC3Net (Implicit Communication in Collective Intelligence) Singh et al. (2018) enhances communication efficiency by allowing agents to learn a gating mechanism that determines *when* to communicate, reducing bandwidth usage. SchedNet Kim et al. (2019a) implements a scheduler that selects which agents can communicate in each timestep, effectively learning both *when* and *who* communicates under constrained bandwidth. DCC-MD Kim et al. (2019b) scales message exchange by dynamically dropping messages based on importance, which indirectly controls the recipient set. VBC (Value-Based Communication) Zhang et al. (2019) uses value function information to prioritize which state information to share, helping agents coordinate by highlighting key states without broadcasting all messages. Diff Discrete Freed et al. (2020a) and I2C (Intentional Information Communication) Ding et al. (2020) reduce communication load by encouraging sparse or goal-relevant communication. I2C, in particular, guides agents to share only information aligned with their current intentions, which can implicitly filter recipients. Information Sharing (IS) Kim et al. (2020) focuses on transmitting internal state representations to synchronize behavior in cooperative tasks. ETCNet (Event-Triggered Communication Network) Hu et al. (2020a) triggers communication only upon detecting significant environmental changes, thus reducing redundant message broadcasts. Finally, Variable-length Coding Freed et al. (2020a) and TMC (Temporal Message Compression) Zhang et al. (2020) compress historical information into concise messages, improving efficiency but still assuming full communication unless paired with selective mechanisms. In all these settings, redundancy reduction hinges on selecting both *what* to send and *to whom* it should be directed, linking content relevance to intended recipients.

**Communicating through Representative Agents**   In systems where agents communicate via a representative agent, communication between agents is mediated by an intermediary or a representative agent. This intermediary coordinates, processes, and distributes messages to and from individual agents, enhancing communication efficiency and often simplifying the coordination process. Several approaches have been proposed for this type of communication, as summarized here. MS-MARL-GCM Kong et al. (2017) introduces a master agent who gathers local observations and hidden states from agents within the environment. The master agent then sends a common message back to each agent, thereby reducing the need for direct communication between individual agents. This centralized communication model improves coordination in partially observable environments. Similarly, HAMMER Gupta et al. (2023) employs a central representative agent that collects local observations from agents in the multi-agent system (MAS) and distributes both a global and individualized message to each agent. This ensures efficient communication by aggregating and distributing relevant information to agents, facilitating better decision-making. ATOC Jiang & Lu (2018) uses an LSTM to link nearby agents who choose to join a communication group. The representative agent in this system sends coordinated messages to the group members, streamlining communication and improving the ability of agents to act cohesively based on shared information. MD-MADDPG Pesce & Montana (2020a) maintains a shared memory among agents, allowing them to selectively store local observations in the memory. The representative agent then loads and relays this information, facilitating communication while reducing message overload. This method ensures that only relevant information is communicated to the necessary agents. IMAC Wang et al. (2020b) defines a scheduler that aggregates encoded information from all

agents and sends individualized messages back to each agent. This centralized scheduling of communication helps agents focus on their individual tasks while still benefiting from global knowledge. In GA-Comm Liu et al. (2020) and MAGIC Niu et al. (2021), a global message processor is used to aggregate information from agents and then distribute messages based on agent-specific weights. This enables more customized communication, where each agent receives a message tailored to its unique context within the system. Finally, Gated-ACML Mao et al. (2020) takes a slightly different approach, allowing agents to decide whether or not to communicate through a message coordinator. This introduces an additional level of decision-making, where agents evaluate the necessity of communication based on the relevance of the information at hand. These methods illustrate the utility of using representative agents to enhance communication in multi-agent systems by centralizing, filtering, and distributing messages in ways that improve overall system efficiency and agent coordination. Because the representative agent explicitly determines recipients, targeted communication becomes part of the central design choice: the intermediary decides *whom* each synthesized message is intended for, embedding the "whom" dimension into the communication architecture itself.

### 5.3.3 Compression and Encoding via Message Weighting

Recent studies in MARL-Comm often handle multiple incoming messages collectively. Message Integration refers to the method by which received messages are combined before being passed into the agents' internal models. In cases where a representative agent is used, the representative agent typically consolidates and coordinates messages, providing a unified single message to each agent, this eliminates the need for individual agents to manage the combination of messages. However, when no representative agent is involved, each agent must independently determine how to merge multiple incoming messages. Since the communicated messages reflect the sender's personal interpretation of the learning process or the environment, some messages may be prioritized over others. These integration mechanisms primarily shape **what information** is ultimately preserved, compressed, or emphasized, although in some architectures, e.g., attention-based weighting, the prioritization of messages can indirectly influence which agents' information becomes most influential during coordination.

**Message Concatenation** In this approach, incoming messages are concatenated into a single input vector without applying explicit preferences or weighting. This ensures that no information is discarded, but it increases the input dimensionality, which may become problematic as the number of agents grows. Methods such as DIAL Foerster et al. (2016a), RIAL Foerster et al. (2016a), ETCNet Hu et al. (2020a), Variable-length Coding Freed et al. (2020a), MADDPG-M Kilinc & Montana (2018), and Diff Discrete Freed et al. (2020a) adopt this approach. For instance, DIAL and ETCNet concatenate scalar message values, preserving simplicity while retaining essential information. SchedNet Kim et al. (2019a) uses shorter message vectors to reduce communication overhead before concatenation, allowing agents to operate more efficiently within limited bandwidth settings. Since all messages are treated equally without prioritization, this method is particularly suitable for small-scale systems, where preserving the entirety of transmitted data is more beneficial than selectively filtering or compressing messages. Because concatenation retains the full set of incoming messages, it directly shapes **what** information enters the agent's policy without making any distinction about **whom** the preserved content originated from; thus, concatenation reflects a content-preservation choice rather than a recipient-selection mechanism.

**Equally Valued Message Integration** When agents are assumed to contribute equally, their messages are integrated using permutation-invariant operations such as summation or averaging, rather than concatenation. While concatenation also treats messages uniformly in the sense that each is included, it preserves ordering and increases dimensionality, whereas averaging or summing compresses the messages into a fixed-size representation without assuming any order. Techniques such as CommNet Sukhbaatar et al. (2016), IC3Net Singh et al. (2018), and VBC Zhang et al. (2019) employ this form of integration. In CommNet, all incoming messages are averaged to produce a single vector, ensuring equal influence from each sender. IC3Net uses a similar averaging mechanism over messages from neighboring agents. FlowComm Du et al. (2021) instead sums the messages, maintaining equal contribution while enabling gradient-based message refinement. This approach is especially effective in cooperative tasks where all agents are assumed to have similarly relevant information, allowing simple yet effective message integration without the need for attention or learned weighting. Because every incoming message is weighted identically, this integration strategy

reflects a specific choice of **what** information the agent retains, namely, an aggregated summary of all inputs, while implicitly treating all senders as equally relevant, without introducing any mechanism to differentiate **who** or **whom** the information came from.

**Weighted Message Prioritization** In this category, agents assign different levels of importance to received messages based on their relevance or utility to the current task. Some approaches rely on handcrafted rules to selectively prune messages deemed less useful. For example, DCC-MD Kim et al. (2019b) and TMC Zhang et al. (2020) implement manual thresholds on message magnitude or relevance scores to discard low-salience messages, thereby reducing communication overhead. More commonly, modern approaches utilize attention mechanisms to learn these weights dynamically. Attention-based models such as TarMAC Das et al. (2018), Agent-Entity Graph Agarwal et al. (2019b), and MAGNet-SA-GS-MG Malysheva et al. (2018) compute relevance scores for each incoming message and combine them using a weighted sum. This enables the receiving agent to focus on the most informative messages while suppressing noise from less critical sources. By assigning different weights to messages, this approach specifies **what** information should be emphasized and what should be ignored. It also introduces an implicit notion of **whom** the agent attends to, because messages from more relevant senders receive higher importance and have a stronger effect on the final representation.

**The Use of Neural Networks** Neural networks are frequently employed to automatically learn the importance of messages, enabling agents to extract and prioritize useful information without hand-crafted rules. In IMAC Wang et al. (2020b), GA-Comm Liu et al. (2020), and HAMMER Gupta et al. (2023), simple multilayer perceptrons (MLPs) are used to integrate messages while implicitly learning which ones are most valuable. Gated-ACML Mao et al. (2020) takes a similar approach but introduces a gating mechanism to regulate communication links with a central representative agent. For more complex settings, convolutional neural networks (CNNs) are used to capture spatial dependencies among agents, as in DGN Jiang et al. (2018), which is particularly beneficial in dynamic environments where agent positions change frequently and updated spatial context is essential. Recurrent neural networks (RNNs), including Long Short-Term Memory (LSTM) variants, are widely applied for sequential decision-making tasks in communication-centric coordination. These are used in ATOC Jiang & Lu (2018), I2C Ding et al. (2020), NeurComm Chu et al. (2020), BiCNet Peng et al. (2017), MD-MADDPG Pesce & Montana (2020a), and GAXNet Yun et al. (2021), where they capture temporal dependencies across communication rounds, allowing agents to make informed decisions based on evolving message histories. Lastly, graph neural networks (GNNs) have gained popularity in works such as MAGIC Niu et al. (2021), GA-Comm Liu et al. (2020), and LSC Sheng et al. (2022). These approaches utilize a learned graph structure over agents to model interactions and assign relevance to neighboring messages, often combining topological information with attention mechanisms to improve communication efficiency and task performance. These models determine **what** information should be extracted from messages, because they learn to highlight useful patterns and suppress noise. They also influence **whom** the agent effectively listens to, since learned graph edges, attention scores, or recurrent dependencies determine which senders have a stronger impact on the final communication representation.

Table 3: Taxonomy of Approaches to 'What to Communicate' (Sec. 5.3) in MARL-Comm. The taxonomy also illustrates how message content is closely linked to the intended recipients, since relevance filtering, targeted communication, and weighting mechanisms depend on whom each message is for.

| Category | Approach | Representative Works |
|---|---|---|
| **Relevance Filtering and Recipient Matching (Sec. 5.3.1)** | *Encoding Past Knowledge* (information selection tied to fixed or learned recipients) (with/without representative agents) | • With representative agents: Sukhbaatar et al. (2016); Kong et al. (2017); Jiang et al. (2018); Das et al. (2018); Malysheva et al. (2018); Gupta et al. (2023)
• Without representative agents: Foerster et al. (2016a); Peng et al. (2017); Liu et al. (2020); Mao et al. (2020); Sheng et al. (2022); Freed et al. (2020a); Hu et al. (2020a); Yun et al. (2021) |

**Table 3 – continued from previous page**

| Category | Approach | Representative Works |
|---|---|---|
| | *Encoding Future Knowledge* (shared with specific downstream recipients) (intended actions, fingerprints, plans) | Jiang & Lu (2018); Chu et al. (2020); Qu et al. (2020); Kim et al. (2020) |
| **Targeted Communication and Recipient Selection (Sec. 5.3.2)** | *Nearby Agents* (proximity-based recipient selection) (static/dynamic proximity) | • Static: Jiang et al. (2018); Agarwal et al. (2019b); Sheng et al. (2022); Chu et al. (2020); Qu et al. (2020); Yun et al. (2021)
• Dynamic: Liu et al. (2020); Niu et al. (2021); Du et al. (2021) |
| | *Other Learning Agents* (learned recipient sets) (learned communicatee sets) | Foerster et al. (2016a); Sukhbaatar et al. (2016); Peng et al. (2017); Das et al. (2018); Kilinc & Montana (2018); Singh et al. (2018); Kim et al. (2019a;b); Zhang et al. (2019); Freed et al. (2020a); Ding et al. (2020); Kim et al. (2020); Hu et al. (2020a); Zhang et al. (2020) |
| | *Representative Agents* (centralized routing of recipients) (centralized message processing) | Kong et al. (2017); Gupta et al. (2023); Jiang & Lu (2018); Pesce & Montana (2020a); Wang et al. (2020b); Liu et al. (2020); Niu et al. (2021); Mao et al. (2020) |
| **Message Integration and Recipient-Aware Weighting (Sec. 5.3.3)** | *Concatenation* (scalar/short vectors) (no recipient differentiation) | Foerster et al. (2016a); Hu et al. (2020a); Freed et al. (2020a); Kilinc & Montana (2018); Kim et al. (2019a) |
| | *Equal Weighting* (averaging/summing) (treating all senders as equally relevant) | Sukhbaatar et al. (2016); Singh et al. (2018); Zhang et al. (2019); Du et al. (2021) |
| | *Weighted Prioritization* (manual or attention-based) (recipient-dependent message importance) | • Manual pruning: Kim et al. (2019b); Zhang et al. (2020)
• Attention: Das et al. (2018); Agarwal et al. (2019b); Malysheva et al. (2018) |
| | *Neural Integration Models* (MLPs, CNNs, RNNs, GNNs) (learning sender–recipient relevance patterns) | • MLPs: Wang et al. (2020b); Liu et al. (2020); Gupta et al. (2023); Mao et al. (2020)
• CNNs: Jiang et al. (2018)
• RNNs/LSTMs: Jiang & Lu (2018); Ding et al. (2020); Chu et al. (2020); Peng et al. (2017); Pesce & Montana (2020a); Yun et al. (2021)
• GNNs: Niu et al. (2021); Liu et al. (2020); Sheng et al. (2022) |

Table 3 presents a taxonomy of representative approaches to the problem of 'what to communicate' in MARL-Comm. It organizes key strategies into three overarching categories: relevance filtering, targeted communication, and message integration. Within each category, we highlight how agents address information selection and compression by focusing on different communication design choices. For relevance filtering,

methods are grouped by whether they encode past knowledge (e.g., histories or observations) or future knowledge (e.g., intentions or plans), and whether communication occurs via representative agents or in a peer-to-peer manner. Targeted communication is further classified by proximity-based strategies (static vs. dynamic neighborhoods), interaction with other learning agents, and centralized communication through representative agents. Finally, message integration methods are categorized by how agents combine received messages: concatenation, equal weighting, weighted prioritization (manual or attention-based), and neural architectures such as MLPs, CNNs, RNNs, and GNNs. Across all categories, the choice of **what** information to communicate is closely tied to **who** that information is sent to, since message relevance, filtering, and weighting depend on the intended recipients. This connection illustrates how communication content and communication partners must be considered together when designing scalable and effective MARL-Comm systems. This structured overview reveals trends in how communication is selectively managed to balance informativeness, efficiency, and scalability in multi-agent systems.

### 5.4 When to Communicate: Timing and Adaptation under Constraints

A fundamental challenge in MARL-Comm is determining *when* agents should exchange information. While continuous communication keeps agents fully informed, it can be costly, inefficient, or even counterproductive in decentralized environments with limited bandwidth or computational resources. Thus, effective communication timing is critical for balancing the benefits of coordination with practical constraints. Instead of communicating at every time step, agents must learn to identify key moments when communication provides meaningful advantage, enabling better decision-making and collaboration under resource constraints. Beyond general bandwidth constraints, sharing a communication medium introduces additional complexity, particularly when multiple agents must avoid message collisions. While protocols like Wi-Fi and LTE manage this at the packet level through well-established scheduling mechanisms, such low-level access control is typically handled separately from decision-making in MARL systems. Instead, the key challenge in MARL-Comm is higher-level: determining *when* to communicate so that information is shared efficiently and meaningfully without overwhelming the channel. This involves two main concerns: (1) selecting concise, task-relevant messages, and (2) coordinating communication timing to minimize redundancy and interference Kim et al. (2019a). Rather than optimizing communication protocols directly, recent MARL approaches address these issues by integrating message selection and scheduling into agents' learning policies.

#### 5.4.1 Learning When to Communicate under Constraints

Certain constraints, such as discrete decisions about whether or when to communicate, can make communication timing non-differentiable, posing challenges for gradient-based learning algorithms. This non-differentiability hinders end-to-end training of communication protocols. One common workaround is *centralized training with decentralized execution* (CTDE), where agents are trained in a centralized manner with full access to global information and differentiable approximations of discrete choices (e.g., via soft gating or policy gradients). This allows agents to learn effective communication strategies during training, even if message transmission is limited or constrained during execution. At test time, agents deploy these learned strategies under stricter, often non-differentiable, conditions. This section reviews methods that address communication constraints by learning *when* and *how* to communicate, including approaches based on CTDE, differentiable gating mechanisms, public belief modeling, and distributed communication scheduling.

**Learning to communicate through backpropagation** Sukhbaatar et al. Sukhbaatar et al. (2016) introduced a 'Communication Network' (CommNet) that processes partial observations from multiple agents to determine their actions. Each layer in this multi-layered network has several cells, where each cell takes as input its own previous hidden state $h_m^{i-1}$ and aggregated messages $c_m^i$ from other agents' hidden states. The updated hidden state is computed as $h_m^{i+1} = \sigma(C^i c_m^i + H^i h_m^i)$, where $\sigma$ is a non-linear activation function. The final layer outputs the agent's action, and the entire network is trained using backpropagation based on a reward signal. This approach outperformed baseline models where communication within the network occurred through discrete symbols. Subsequent research shifted toward *centralized training with decentralized execution*, as demonstrated by Deep Recurrent Q-Networks (DRQN) Hausknecht & Stone (2015). Unlike independent Deep Q-Networks (DQN) van Hasselt et al. (2015), DRQN does not assume full observability and is better suited for partially observable multi-agent environments. By maintaining internal memory

through recurrent units, DRQN helps mitigate issues of non-stationarity arising from concurrent policy updates across agents, making it more robust in decentralized multi-agent settings.

Foerster et al. Foerster et al. (2016b) extended this by proposing the two following centralized communication schemes where agents share network weights and learn to communicate effectively in partially observable environments with shared rewards.

- **Deep Distributed Recurrent Q-Network Foerster et al. (2016b):** The Deep Distributed Recurrent Q-Network (DDRQN) extends Deep Recurrent Q-Networks (DRQN) by adapting them to multi-agent settings through modifications to the independent Q-learning paradigm. Specifically, DDRQN introduces three key modifications to standard independent Q-learning: (1) experience replay is disabled, as concurrently learning agents invalidate each other's past experiences; (2) each agent's previous action is appended to the current input to capture temporal dependencies; and (3) a single Q-network is shared across all agents, rather than maintaining separate networks. This weight sharing improves learning efficiency and reduces computational overhead.

  Despite using a shared network, each agent receives individualized inputs—including its own observation, previous action, and agent identifier—allowing the shared Q-network to specialize behavior for each agent. The Q-function learned by the Recurrent Neural Network (RNN) is defined as:

$$Q(o_t^m, h_{t-1}^m, m, a_{t-1}^m, a_t^m; \theta), \tag{1}$$

  where $o_t^m$ and $a_t^m$ are the observation and current action of agent $m$ at time $t$, $a_{t-1}^m$ is its previous action, $h_{t-1}^m$ is the hidden state, and $\theta$ denotes the shared network parameters.

  Training proceeds in two stages: (1) agents interact with the environment using an $\epsilon$-greedy policy based on the current Q-function, and (2) the shared network is updated using the Bellman equation, optimizing temporal-difference error over episodes.

- **Differentiable Inter-Agent Learning (DIAL) Foerster et al. (2016a):** DIAL introduces a framework where, at each time step, agent $m$ outputs not only an environment action $a_t^m$ but also a communication action $c_t^m$. During centralized training, communication is unrestricted and differentiable, enabling gradient flow across agents; during decentralized execution, messages are constrained to low-bandwidth discrete channels. Unlike traditional Q-networks that only estimate value functions, DIAL's Q-network additionally outputs real-valued messages $c_t^m$, which are sent at time $t+1$ to other agents' Q-networks (all agents share weights).

  Two types of gradients flow through the network: (1) the reward gradient from the agent's own Q-learning loss and (2) an error gradient from message recipients, which propagates back through the communication channel. This allows the sending agent to directly adjust its messages to minimize downstream DQN loss, accelerating the emergence of effective communication.

  Experimental results show that both DIAL and DDRQN eventually converge to similar performance—100% of Oracle rewards for $n = 3$ agents, and 90% for $n = 4$. However, DIAL converges substantially faster, requiring only 20,000 episodes compared to DDRQN's 500,000. For $n = 4$, DIAL maintains reliable convergence, while DDRQN may fail to converge altogether. The performance gains are attributed to DIAL's differentiable message passing, which enables more direct and efficient coordination learning during training.

**Learning to communicate through parameter sharing**  Besides the differentiable inter-agent learning (DIAL) proposed in Foerster et al. (2016a), the authors of this paper also proposed another communication approach based on centralized learning decentralized execution using deep Q-learning with a recurrent network to address partial observability, i.e., reinforced inter-agent learning (RIAL). RIAL has two variants, using either independent or centralized Q-learning. In independent Q-learning, each agent learns its own network, treating others as part of the environment. Another variant trains a single shared network for all agents. During execution, agents act independently based on their unique observations. RIAL combines DRQN with independent Q-learning to learn both environmental and communication actions. Each agent's Q-network, $Q^a$, takes as input the hidden state $h_t^a$, observation $o_t^a$, and the messages $m_{t-1}^a$ received from

other agents at the previous time step. To manage the potentially large action space, RIAL factorizes $Q^a$ into two separate output heads: $Q^a_u$ for environmental actions $u^a_t \in U$ and $Q^a_m$ for communication actions $m^a_t \in M$, where $U$ and $M$ denote the respective action spaces. An $\epsilon$-greedy policy is used to select both types of actions, requiring the Q-network to produce $|U| + |M|$ outputs per time step. Both $Q_u$ and $Q_m$ are trained using DQN, with two key modifications: disabling experience replay to address non-stationarity, and feeding the actions $u_t$ and $m_t$ as inputs at the next time step to account for partial observability. Even though the agents learn independently, the decentralized execution is identical to the training phase.

**Learning to Communicate through Public Belief**   The DIAL method Foerster et al. (2016a) assumes the presence of an idealized communication channel where agents can freely exchange information without affecting the environment. However, in many practical scenarios, explicit communication is unavailable or constrained. To address this, Foerster et al. Foerster et al. (2019) introduced the concept of *public belief*, which captures a shared probabilistic estimate over latent state features, based on common observations. At time step $t$, a latent feature $f_t$ is inferred, with its publicly observable part denoted as $f^{pub}_t$. The public belief is updated as:

$$\mathcal{B}_t = P(f_t | f^{pub}_1, f^{pub}_2, \dots, f^{pub}_t).$$

Building on this, they proposed the **Bayesian Action Decoder (BAD)**, a method that uses public belief to enable coordination without direct communication. Each agent selects actions based on its private observation and a shared virtual policy $\hat{\pi}_t$, computed from the public belief and known policies of all agents. This virtual policy maps private observations to likely actions, enabling agents to infer others' private states from their observable actions. Specifically, after observing an action $a^m_t$, an agent updates its belief about another's private state $f^m_t$ using:

$$P(f^m_t | a^m_t, \mathcal{B}_t, f^{pub}_t, \hat{\pi}_t) \propto \mathbf{1}(\hat{\pi}_t(f^m_t) = a^m_t) P(f^m_t | \mathcal{B}_t, f^{pub}_t).$$

This approach was evaluated on the cooperative card game *Hanabi*, where players cannot see their own cards and must rely on hints and inference. BAD achieved state-of-the-art performance in two-player settings, with agents learning implicit conventions (e.g., using a hint like "red" to signal which card to play) that facilitated non-verbal coordination. The success of BAD demonstrates how shared beliefs can support effective communication even in the absence of explicit channels.

**Scheduling Communication in a Distributed Manner**   A common approach to enhancing coordination in multi-agent systems is by enabling distributed communication, where agents exchange information to act cohesively without relying on a centralized controller. This is often achieved through the CTDE paradigm Lowe et al. (2017), which is also adopted by many of the previously discussed methods in this section. **SchedNet** Kim et al. (2019a) exemplifies this approach by using reinforcement learning to learn a scheduling policy that determines which agents should communicate at each time step. This dynamic scheduling is particularly important in practical scenarios where communication occurs over a shared medium—such as a wireless frequency channel—introducing two key constraints: limited bandwidth and contention for medium access. SchedNet addresses these constraints by jointly learning (1) a message encoder, (2) a policy for scheduling which agents can broadcast, and (3) an action selector that uses both local observations and received messages to make decisions.

### 5.4.2   Timing of Message Integration

Message integration plays a critical role in determining not only *how*, but also *when* communicated information affects an agent's behavior during both learning and execution. By message integration, we refer to the process of incorporating received communication into an agent's internal computation, such as in its policy or value function, during a training round or decision cycle. In most MARL-Comm methods, communication is treated as additional input and incorporated into either the policy network, the value network, or both. The specific timing of this integration—whether messages influence immediate action selection, delayed value updates, or both—affects how quickly agents adapt to new information and coordinate their behavior. Poorly timed integration may delay responses to critical cues or cause misaligned coordination. To examine these effects, we categorize recent approaches into three strategies: policy-level integration, value-level integration, and joint policy-and-value integration Zhu et al. (2022).

**Policy-Level Communication Messages Integration**   Incorporating communication messages into the policy model directly influences *when* agents adjust their decisions based on received information. By conditioning their next action on the latest messages from other agents, policies can dynamically adapt to evolving environments and coordination needs in real time. Typically, messages are concatenated with an agent's local observations or hidden states and fed into the policy network. This integration allows the agent to immediately update its behavior at every decision step based on the most recent communicated information, rather than relying solely on its own partial observations. In **policy gradient methods**, such as REINFORCE Sukhbaatar et al. (2016); Liu et al. (2020); Singh et al. (2018); Kong et al. (2017), the timing of communication affects the entire trajectory distribution: agents collect rewards over episodes while continuously adjusting their policies based on ongoing message exchanges. Communication influences the probability of action sequences during learning, making timely message integration critical for effective gradient estimation and coordination. **Actor-critic approaches** Du et al. (2021); Wang et al. (2020b); Jiang & Lu (2018); Kim et al. (2019a); Ding et al. (2020); Chu et al. (2020); Kim et al. (2020); Pesce & Montana (2020a); Freed et al. (2020a); Yun et al. (2021); Gupta et al. (2023) further emphasize timing by using critics that evaluate future returns conditioned on both local observations and communication messages. Here, agents can adapt their action choices at each step based on newly received messages, optimizing not only for immediate rewards but also for longer-term cooperation based on shared information. Thus, in policy-level integration, communication timing determines *when* new information influences the agent's decision-making pipeline, enabling agents to react to dynamic changes in their teammates' states and strategies throughout the learning and execution process.

**Value-Level Communication Messages Integration**   In value-level integration, communication messages are incorporated into the value function (or action-value function), directly influencing how agents evaluate the long-term utility of their actions. By conditioning value estimates on both local observations and received messages, agents can better assess expected returns under partial observability and dynamic conditions. Many works following DQN-style frameworks adopt this strategy Foerster et al. (2016a); Jiang et al. (2018); Zhang et al. (2019); Agarwal et al. (2019b); Sheng et al. (2022); Kim et al. (2019b). Typically, messages are concatenated with local observations or embedded as additional features for the value network. This allows incoming communication to immediately affect an agent's Q-value estimates, which in turn guide future action selection via $\epsilon$-greedy or similar policies. Unlike policy-level integration, where messages directly shape the action distribution, value-level integration influences behavior indirectly, through changes in the estimated returns of different actions. In both cases, the timing of message reception matters. However, in value-level integration, updates often occur after reward feedback is received, introducing a temporal delay in how communicated information affects future decisions. This lag can influence when agents benefit from new information, particularly in environments with fast-changing dynamics. Incorporating communication into the value function allows agents to reason about the long-term consequences of actions informed by peer input, thereby enhancing coordination in sequential decision-making tasks.

**Combined Policy and Value Communication Messages Integration**   In combined integration approaches, communication messages are incorporated into both the policy and value models, enabling agents to adjust not only their immediate action selection but also their evaluation of future outcomes based on received information. This dual integration enhances the agent's ability to dynamically adapt its decisions *when* new messages are received, influencing both short-term reactions and long-term planning. This approach is often based on actor-critic methods, where messages are taken as additional inputs to both the actor and the critic. Some works feed the messages separately into the policy and value networks Agarwal et al. (2019b); Peng et al. (2017), allowing independent processing of communication information in action generation and value estimation. Others first combine messages with local observations to form a shared internal representation, which is then jointly used by both the actor and critic models Niu et al. (2021); Das et al. (2018); Liu et al. (2020); Freed et al. (2020a). By integrating communication into multiple stages of decision-making, agents can react promptly to new information at the policy level while simultaneously refining their value predictions. The timing of communication thus has a compounded effect: when messages arrive at critical decision points, they can immediately influence an agent's behavior, either by directly altering action probabilities (in policy-level integration) or by updating value estimates that drive action

selection (in value-level integration, e.g., via Q-learning). At the same time, these updates can shift agents' expectations of long-term rewards, enhancing coordination over both immediate and future timescales.

| Category | Approach | Representative Works |
|---|---|---|
| **Learning Communication Timing (Sec. 5.4.1)** | Backpropagation-based Communication | Sukhbaatar et al. (2016); Foerster et al. (2016a;b) |
| | Parameter Sharing (RIAL/DRQN) | Foerster et al. (2016a) |
| | Public Belief Modeling | Foerster et al. (2016a; 2019) |
| | Distributed Scheduling (MAC) | Lowe et al. (2017); Kim et al. (2019a) |
| **Inner Integration Timing (Sec. 5.4.2)** | Policy-Level Integration | • REINFORCE Sukhbaatar et al. (2016); Liu et al. (2020); Singh et al. (2018); Kong et al. (2017)
• Actor-critic approaches Du et al. (2021); Wang et al. (2020b); Jiang & Lu (2018); Kim et al. (2019a); Ding et al. (2020); Chu et al. (2020); Kim et al. (2020); Pesce & Montana (2020a); Freed et al. (2020a); Yun et al. (2021); Gupta et al. (2023) |
| | Value-Level Integration | • DQN-like frameworks Foerster et al. (2016a); Jiang et al. (2018); Zhang et al. (2019); Agarwal et al. (2019b); Sheng et al. (2022); Kim et al. (2019b) |
| | Combined Policy-Value Integration | • Feeding the messages separately into the policy and value networks Agarwal et al. (2019b); Peng et al. (2017)
• Combining messages with local observations to form a shared internal representation, which is then jointly used by both the actor and critic models Niu et al. (2021); Das et al. (2018); Liu et al. (2020); Freed et al. (2020a) |

Table 4: Taxonomy of Approaches to 'When to Communicate' in MARL-Comm

Table 4 summarizes key approaches to determining when agents should communicate in multi-agent reinforcement learning systems. The first part of the table highlights learning-based strategies for adapting communication under bandwidth or resource constraints, including backpropagation-enabled frameworks like CommNet and DIAL, centralized training with shared parameters, public belief modeling for environments without explicit channels, and distributed scheduling using reinforcement learning. The second part focuses on the timing of message integration within an agent's learning model, either at the policy level (directly influencing action selection), at the value level (shaping return estimates) or both. Together, these approaches aim to balance timely and efficient communication with coordination demands, especially in dynamic, partially observable, or bandwidth-constrained settings.

### 5.5 Why and How to Communicate: Purpose-Driven Protocol Design

After identifying what to communicate, who should communicate, whom to communicate to, and when communication should occur, the remaining question is how agents should structure and exchange information. The choice of protocol is not only a matter of implementation. It reveals the underlying purpose of communication and the specific advantages agents seek to obtain. In practice, how agents communicate reflects why they communicate in the first place. For example, agents may communicate to reduce uncertainty, to coordinate joint actions, or to stabilize learning under partial observability, and each of these goals motivates

a different protocol design. For this reason, we discuss how and why together in this subsection and treat protocol design as a purpose-driven process that links communication structure with communication intent. Effective protocols must support coordination, preserve interpretability, and remain scalable as the number of agents increases. They must also determine whether information exchange should rely on centralized support or be achieved through fully decentralized interactions, and whether messages should use explicit human-interpretable symbols or implicit latent representations optimized for learning.

### 5.5.1 Training Frameworks for Communication Learning

The training framework in an MARL-Comm system defines how agents use the collected experiences, including observations, actions, rewards, and messages, to improve performance. It also implicitly determines **why** communication is needed and **how** communication mechanisms should be designed, since different frameworks create different informational needs and coordination requirements. Different frameworks specify how this information is processed and shared during learning, shaping both the motivation for communication and the mechanisms through which agents exchange information.

In **decentralized learning**, each agent is trained independently using only its respective experience. Because no centralized information is available, communication is motivated by the need to reduce uncertainty caused by partial observability and non-stationarity. Agents communicate in order to stabilize learning and anticipate the behaviors of others. This approach allows agents to learn their individual policies based solely on local observations and rewards. However, decentralized learning faces significant challenges in multi-agent settings, primarily due to the non-stationary environment introduced by constantly adapting agents. As agents evolve, the environment each agent perceives is continuously changing, complicating the learning process.

In **centralized learning**, all agents are trained jointly using the combined experience of every agent. Here, communication during execution is not needed to supply missing global context, because global information is already used to construct joint policies. Instead, the reason for communication shifts toward enabling the central learner to shape coordinated local behaviors. This approach transforms the multi-agent system into a stationary environment where global information can be utilized to guide the training process. However, centralized learning suffers from the curse of dimensionality, as the joint policy space grows exponentially with the number of agents, making it computationally expensive and difficult to search.

To address the drawbacks of purely decentralized or centralized learning, the **Centralized Training with Decentralized Execution (CTDE)** paradigm Foerster et al. (2016a); Kraemer & Banerjee (2016) has become the dominant training framework for MARL-Comm. In CTDE, communication serves two distinct purposes. During training, communication is shaped by centralized information that guides the learning of local policies. During execution, communication helps agents reconstruct or approximate the centralized information that is no longer available. This dual purpose leads to diverse communication mechanisms, including learned message encoders, attention-based routing, and critic-informed message usage. In CTDE, agents learn their local policies with the benefit of centralized information during training, such as access to global observations or the actions of other agents. However, during execution, agents operate autonomously, using only their local information. This approach balances the benefits of centralized training with the flexibility of decentralized execution, allowing agents to learn more robust policies while avoiding the challenges of joint policy search during execution.

In addition, **Parameter Sharing** schemes Foerster et al. (2016a) have emerged as a solution to further reduce the complexity of multi-agent systems. Here, the motivation for communication is different: because all agents share the same model, communication becomes a means for a shared policy to differentiate agent roles or contextual behaviors. This motivation influences the design of communication mechanisms that help distinguish agents that otherwise follow identical network parameters. In these methods, agents share a common set of parameters (such as neural network weights), effectively training a single model that can control all agents. While this reduces the computational overhead of training multiple policies, it may limit the diversity of agent behaviors, as all agents are governed by the same model.

Lastly, **Concurrent Learning** addresses situations where agents must simultaneously learn to interact with each other while managing their own policies. Methods like MD-MADDPG Pesce & Montana (2020a) and

IS Kim et al. (2020) allow agents to train in tandem, coordinating their learning processes while maintaining individuality in decision-making. Here, communication is motivated by the need for agents to track each other's evolving behaviors without a shared replay buffer or central controller, which leads to mechanisms that propagate up-to-date observations, actions, or intentions among agents.

Before examining each training framework in detail, it is important to clarify how they jointly determine both **why** agents communicate and **how** communication mechanisms are ultimately designed. The training setup shapes what information is available, what information becomes unavailable at execution, and what gaps agents must fill through communication. Centralized learning reduces the need for communication during training but creates a pressure for communication at execution time to approximate the global context that is no longer accessible. Decentralized learning creates the opposite pressure: agents begin with only local views, so communication becomes necessary to stabilize learning and reduce uncertainty. CTDE blends these motivations by using centralized information to train policies while requiring agents to reconstruct or approximate that information during execution. Across these frameworks, the underlying reason **why** communication is needed varies, and this difference naturally produces different communication mechanisms. When communication is required to overcome partial observability, agents exchange local signals. When it is required to propagate global structure learned during centralized training, communication becomes critic-shaped or implicit. When it is required to diversify behaviors under shared parameters, communication emphasizes identity and role differentiation. In the following content, we examine each training framework in depth, compare their assumptions and communication implications, and highlight how their distinct motivations lead to different choices for how agents communicate.

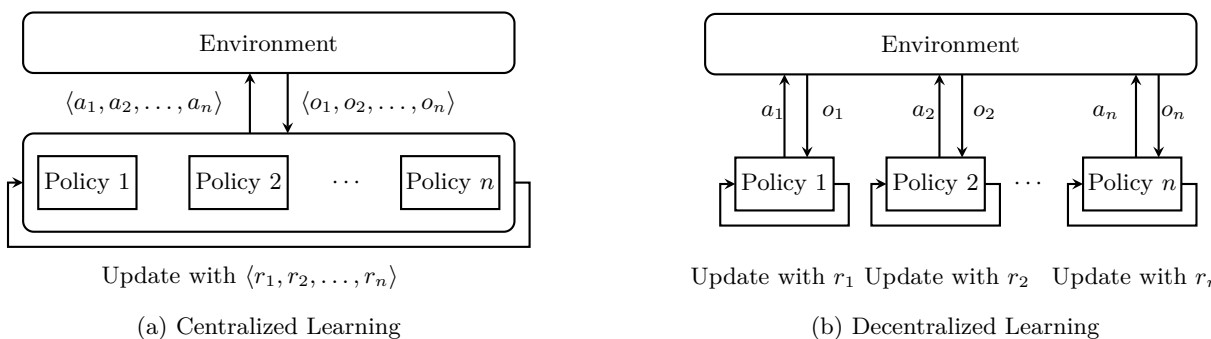

(a) Centralized Learning

(b) Decentralized Learning

Figure 4: Centralized and Decentralized Learning. In centralized learning (a), policies are jointly optimized with shared knowledge and gradient flow. In decentralized learning (b), each agent operates and learns independently using local observations and rewards.

**Centralized Learning** In the centralized learning paradigm, the experiences of all agents, including their observations, actions, rewards, and messages, are aggregated by a central unit that learns to optimize the collective behavior of the system. As illustrated in Fig. 4a, this centralized controller leverages a global view of the environment to coordinate agents and compute joint policies or value functions. By having access to system-wide information, the learning process can exploit dependencies and correlations that may be inaccessible in decentralized settings. This global access also shapes **why communication** is needed at execution. Because the central learner integrates information from all agents during training, individual agents do not learn to infer missing global context on their own. Once the central unit is removed at execution, communication becomes a mechanism for agents to reconstruct the global structure they relied on during training. In this way, centralized learning implicitly motivates communication as a tool to approximate training-time information using decentralized signals.

Historically, early centralized-learning communication works such as CommNet Sukhbaatar et al. (2016) provided a single continuous communication channel trained through centralized backpropagation. This addressed **why** to communicate by assuming that all agents benefit from sharing a unified global feature space, and it addressed **how** to communicate by averaging hidden states across agents. BiCNet Peng et al. (2017) improved this by introducing a recurrent structure across agents, motivated by the need to propagate richer dependencies through the centralized model. The communication mechanism therefore

became sequential and contextual, reflecting the centralized model's need to capture ordered interactions among agents.

DIAL Foerster et al. (2016a) strengthened the link between **why** and **how** by making the communication channel differentiable. Because centralized learning used gradient flow to shape the content of messages, DIAL justified why communication is needed as a way to convey representations that directly support joint optimization. Its communication mechanism then became real-valued messages shaped by gradient signals. IC3Net Singh et al. (2018) added a gating module to learn when communication is helpful. This responded to the observation that centralized training revealed mixed incentives in some tasks, which motivated the need for selective communication as part of the how.

The next generation of centralized-learning works used attention. TarMAC Das et al. (2018) showed that centralized gradients could train attention weights to choose informative senders. This created a strong connection between why and how: the reason to communicate became the need to capture targeted relevance under global supervision, and the communication architecture became a learned attention mechanism. Similarly, graph-based methods such as ATOC Jiang et al. (2018) used centralized feedback to discover which neighborhoods of agents should interact. This motivated communication as a way to share local group-specific context, while the communication protocol became graph attention guided by centralized training. MAGIC Niu et al. (2021) continued this trend by learning dynamic graph structures under centralized gradients, again tying the motivation for communication to the need to recover global information using local graph-based message passing. Finally, representative-agent models such as HAMMER Gupta et al. (2023) used a central module to aggregate information and broadcast processed guidance, making communication necessary to distribute the centrally computed strategy. The communication mechanism therefore became a centralized-to-decentralized broadcast path.

In practice, centralized learning is typically confined to the training phase. Most recent works assume that, during execution, agents operate independently without access to a central controller. This decoupling, commonly referred to as **CTDE**, reflects the fact that maintaining a centralized controller at runtime is often infeasible in real-world deployments, particularly in dynamic, distributed, or bandwidth-constrained environments. As such, centralized learning serves as a practical way to facilitate coordination during training while preserving scalability and autonomy during execution.

Because centralized learning does not require agents to exchange explicit messages during training, the communication that emerges afterward is often implicit, critic-shaped, or encoded in latent vectors rather than explicit symbolic messages. This explains why centralized-learning methods frequently produce communication protocols that rely on hidden representations, shared critics, or centralized guidance rather than explicit message structures. The distinction between **why communication is needed** and **how it is implemented** becomes clearer: the training process reveals what global information agents must reconstruct, and the communication architecture becomes the tool that allows them to recover this information without a central controller.

**Decentralized Learning** As illustrated in Fig. 4b, decentralized learning frameworks enable each agent to collect experience and optimize its policy independently, using only locally available information. Agents learn from their own observation-action-reward-message trajectories, without reliance on a centralized controller or shared global knowledge. This paradigm is suitable for scenarios where communication is constrained, delayed, or unreliable, and where agents must operate under partial observability or localized perspectives. In decentralized settings, the reason **why** communication emerges is directly tied to agents lacking access to the information that other agents observe. Agents must therefore create message exchanges that compensate for missing state information. The structure of these messages reflects **how** decentralized systems attempt to share only what is necessary to support coordination.

Several recent approaches adopt this decentralized formulation, including MAGNet-SA-GS-MG Malysheva et al. (2018), Agent-Entity Graph Agarwal et al. (2019b), DCC-MD Kim et al. (2019b), NeurComm Chu et al. (2020), Intention Propagation (IP) Qu et al. (2020), and Diff Discrete Freed et al. (2020a). These works share a common challenge: each agent must infer missing information from peers without a centralized learner, which clarifies **why** explicit communication becomes critical and **how** different solutions attempt to encode that needed information.

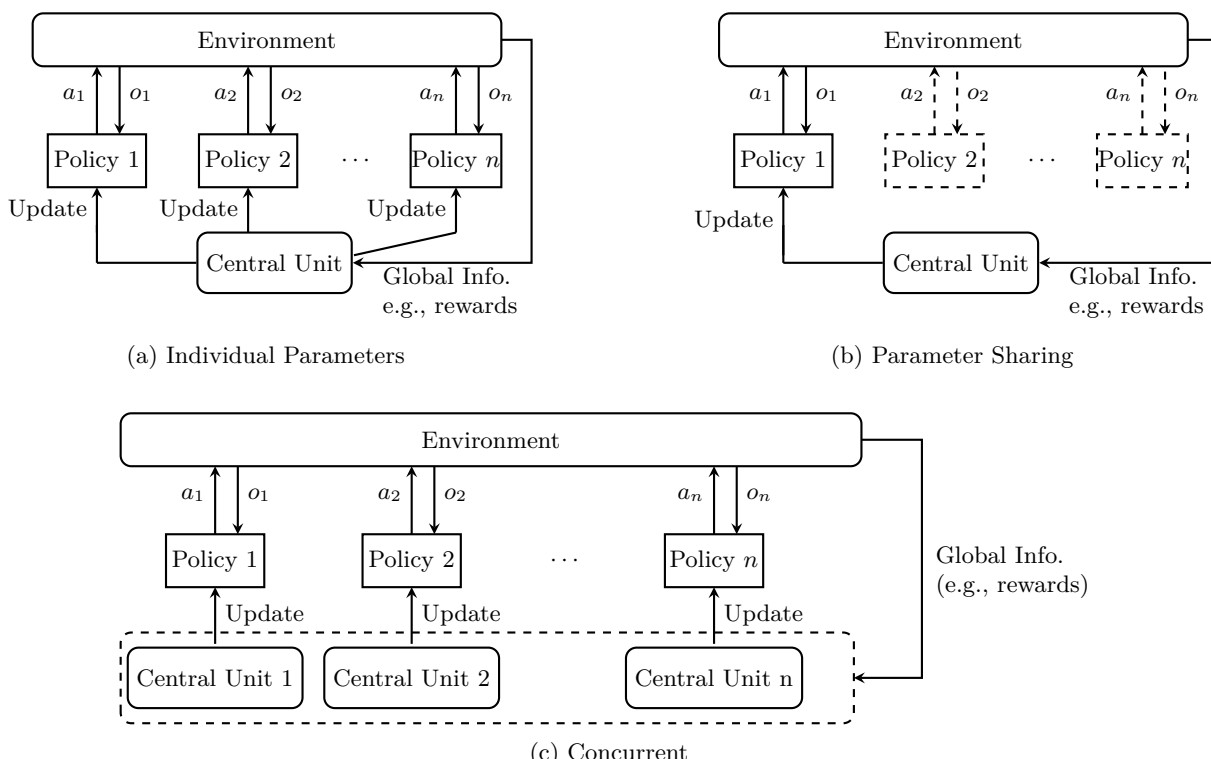

Figure 5: Three types of CTDE Scheme.

MAGNet-SA-GS-MG Malysheva et al. (2018) employs a modular attention mechanism to choose relevant senders and receivers. The challenge lies in determining which information should be exchanged and which should be ignored, clarifying **why** communication must be selective rather than uniform. The architecture specifies **how** this selectivity is implemented through multi-scale attention over spatially varying contexts. This improves over earlier work by enabling dynamic sender-receiver assignment rather than fixed communication channels.

Agent-Entity Graph Agarwal et al. (2019b) introduces a dynamic graph representation to handle interactions based on proximity. The challenge motivating communication is that agents cannot jointly track entity relationships from partial views. The method clarifies **why** messages are needed: agents must share relational cues that reveal structured dependencies. It also specifies **how** this is achieved: through graph-based message passing between agent-entity nodes. Compared to earlier fully connected communication schemes, this work improves scalability by allowing messages only along relevant edges rather than across all agent pairs.

DCC-MD Kim et al. (2019b) addresses the difficulty of scaling communication in large multi-agent populations. The challenge here is bandwidth, which explains **why** only a subset of messages should be transmitted. The method defines **how** pruning is performed through relevance-driven message dropping. Compared with MAGNet-SA-GS-MG, DCC-MD focuses more directly on communication efficiency and scalability rather than relational or spatial structure.

NeurComm Chu et al. (2020) learns decentralized communication policies directly through neural encoders. Here the challenge is that agents cannot predict future behavior of their teammates, which explains **why** communication must encode policy-relevant latent information. The method explains **how** agents exchange compact neural messages that approximate their internal policy state. Relative to Agent-Entity Graph, NeurComm improves expressiveness by learning latent intention cues rather than relying only on structural proximity.

Intention Propagation (IP) Qu et al. (2020) targets the difficulty of anticipating others' immediate actions. This makes clear **why** communication is used: agents need a way to reduce uncertainty about teammates'

next moves. The method defines **how** this is done by passing explicit intended actions across local neighborhoods. Compared with NeurComm, which uses latent embeddings, IP improves transparency and interpretability by exchanging explicit one-step action predictions.

Diff Discrete Freed et al. (2020a) aims to produce compact, discrete communication protocols that improve interpretability. The challenge that motivates communication is the need for discrete coordination signals that are robust under noise. This clarifies **why** soft continuous messages may be insufficient. The method shows **how** to generate discrete messages through a differentiable Gumbel-softmax relaxation. This approach improves over DCC-MD by producing structured, interpretable messages rather than only pruning continuous ones.

Together, these decentralized methods demonstrate that the challenges agents face under partial observability shape both **why** communication is needed and **how** communication is designed. As the field evolves chronologically, methods progress from structural message passing (Agent-Entity Graph) to latent policy-aware messaging (NeurComm), explicit intention sharing (IP), selective attention mechanisms (MAGNet-SA-GS-MG), scalable pruning techniques (DCC-MD), and finally discrete, interpretable protocols (Diff Discrete). Each step refines prior approaches by addressing limitations in expressiveness, scalability, or interpretability.

**Centralized Training with Decentralized Execution (CTDE)**  In the CTDE framework (Fig. 5), the experiences from all agents are made available for optimizing their individual policies. This design addresses a core challenge of multi-agent learning: agents must coordinate despite receiving only partial observations during execution. CTDE clarifies **why** communication becomes necessary at execution. During training, agents benefit from global information through the centralized critic, but during execution this global structure disappears. Agents therefore must exchange messages to reconstruct or approximate the information they relied on during training. By using gradients computed from the combined experiences of all agents, CTDE guides each agent's local policy while still allowing decentralized execution in the environment. The key benefit is that CTDE exploits rich global information during training without requiring centralized control at runtime.

One critical component of CTDE is *parameter sharing* Foerster et al. (2016a), which improves data efficiency by using a single set of parameters (such as a shared Q-function or shared policy) across agents instead of maintaining separate models. Even with shared parameters, agents can still learn diverse behaviors because they encounter different observations and roles in the environment. Parameter sharing also informs **how** communication is structured. Since agents operate under identical models, communication often focuses on exchanging role-identifying information or contextual cues that help them break symmetry during execution.

Based on these mechanisms, recent CTDE-based works can be grouped into three main approaches:

**(1). Individual Policy Parameters.** In this category, each agent maintains its own policy parameters while a central unit aggregates experiences from all agents to compute global guidance signals, as shown in Fig. 5a. This family of methods typically uses policy gradient algorithms such as REINFORCE Kong et al. (2017) or actor-critic variants Wang et al. (2020b); Kim et al. (2019a); Mao et al. (2020); Yun et al. (2021). The challenge motivating communication here is that agents must act under partial observability even though training relies on access to full system information. This explains **why** agents later need to communicate: they must approximate the influence of centralized gradients using messages. The training setup also shapes **how** communication emerges, often in the form of critic-shaped messages or latent signals that mimic the centralized feedback agents received during learning. Relative to decentralized learning approaches, these methods improve stability by grounding each local update in global training data, which reduces the non-stationarity caused by concurrently learning policies.

**(2). Parameter Sharing.** Here, a single set of parameters is shared across all agent policies or critic networks, as illustrated in Fig. 5b. Parameter sharing improves data efficiency and can accelerate training. When DQN-like methods are used, agents may learn a unified Q-function from all experiences Foerster et al. (2016a); Jiang et al. (2018); Sheng et al. (2022), or a second global Q-function may be added to complement local updates Zhang et al. (2019; 2020). Actor-critic methods instead train one shared actor using all observation-action samples while relying on a centralized critic to provide gradients Niu et al. (2021); Du et al. (2021); Das et al. (2018); Jiang & Lu (2018); Hu et al. (2020a); Mao et al. (2020). In some cases,

REINFORCE-style updates are still used, relying solely on episode-level rewards Sukhbaatar et al. (2016); Liu et al. (2020); Singh et al. (2018). Parameter sharing highlights **why** communication becomes essential: if all agents are governed by the same network, they must exchange contextual information to differentiate their behavior. The approach also influences **how** communication is designed, often producing identity-indicating messages or local-disambiguation signals that help agents specialize. Compared with individual-parameter methods, parameter-sharing techniques improve sample efficiency and reduce computational cost, but require more deliberate communication to avoid homogenized behaviors.

**(3). Concurrent Learning.** In scenarios where storing all experiences in a centralized buffer is impractical, each agent maintains its own buffer while assuming access to the actions and observations of other agents. These decentralized backups are used to train a centralized critic per agent, as shown in Fig. 5c. Methods such as Kim et al. (2020); Pesce & Montana (2020a) adopt this variant. The challenge arises when agents must learn concurrently but still use global information during training. This explains **why** communication is needed: agents require a way to maintain coordination despite updating policies at different times. It also shapes **how** communication is exchanged, often through implicit synchronization signals or messages that refine the critic's predictive accuracy. Relative to parameter-sharing approaches, concurrent CTDE offers greater behavioral diversity and avoids the restrictions of a shared policy class, while still maintaining the advantages of centralized critic guidance.

CTDE balances centralized training and decentralized execution, becoming a standard framework for multi-agent reinforcement learning. This flexibility is particularly valuable in environments where agents must coordinate effectively while preserving individual autonomy during execution. Across all CTDE variants, the underlying training architecture clarifies **why** communication is required and **how** communication protocols are shaped by the structure of the centralized information available during learning.

### 5.5.2 Communication Objectives and Control Shaping

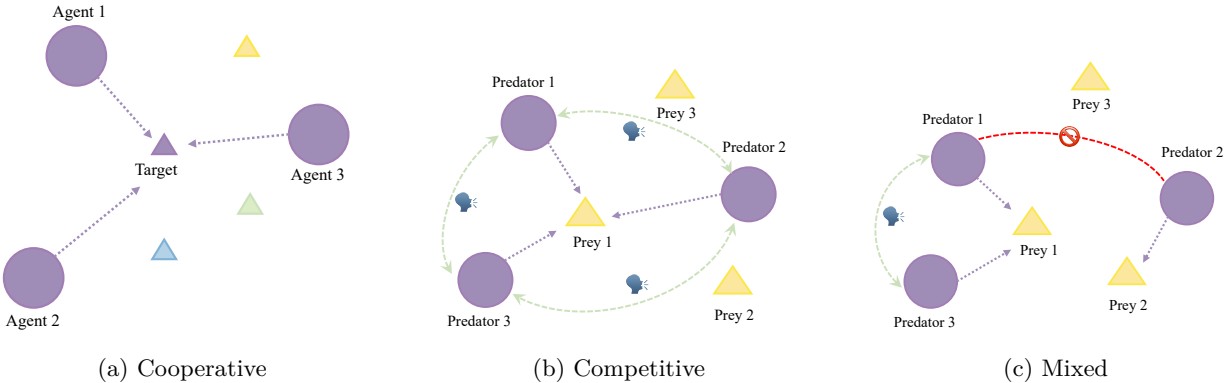

(a) Cooperative  (b) Competitive  (c) Mixed

Figure 6: Communication shaped by reward objectives: cooperative (a), competitive (b), and mixed (c).

The design of reward functions plays a central role in shaping how agents communicate and coordinate in multi-agent systems. By tailoring these reward signals, agents can be incentivized to pursue specific inter-action patterns and group dynamics. Depending on how these rewards are structured, agent interactions tend to fall into one of three primary categories: *cooperative*, *competitive*, or a combination of both, referred to as *mixed* or *cooperative-competitive* scenarios Ning & Xie (2024); Busoniu et al. (2008); Matignon et al. (2012). Each paradigm requires distinct communication strategies to support the underlying objective. For instance, fully cooperative tasks encourage agents to share informative messages that maximize joint utility, while competitive tasks may discourage communication or foster deception. A number of recent approaches validate their adaptability by testing across multiple behavioral regimes Liu et al. (2020); Das et al. (2018); Singh et al. (2018); Jiang et al. (2018); Sheng et al. (2022). These differences across reward structures clarify **why** communication is encouraged or suppressed in each setting and **how** agents adapt their messaging patterns to suit the underlying incentive landscape. In **cooperative settings** (Fig. 6a), agents are encouraged to work together to maximize a shared objective. A common approach is to assign the same reward to all

agents, i.e., $r_1 = r_2 = \cdots = r_N$, so that collective performance is prioritized over individual gain. This equal reward distribution incentivizes agents to support one another and avoid failures that could harm the group's overall outcome, thereby fostering stronger coordination. Under this incentive structure, communication becomes useful because agents benefit jointly from shared information, and coordination-oriented message passing naturally emerges as the most effective mechanism for improving collective outcomes. **Competitive settings** (Fig. 6b), by contrast, emphasize individual success. Each agent seeks to maximize its own reward, potentially at the expense of others. In fully competitive tasks—often modeled as zero-sum games—the total reward is fixed, such that $\sum_{i=1}^{N} r_i = 0$. Here, one agent's gain directly results from another's loss, and the objective becomes outperforming opponents rather than collaborating. This creates a very different motivation for communication: sharing information may disadvantage the sender, so agents often reduce, gate, or strategically time their messages. The resulting communication protocols therefore reflect adversarial incentives rather than cooperative ones. **Mixed settings** (Fig. 6c), also known as general-sum games, incorporate both cooperative and competitive dynamics. In these environments, agents have partially aligned or independent goals, and rewards are neither fully shared nor strictly opposed. Rather than involving multiple tasks, agents in mixed settings may find themselves cooperating in some parts of a task while competing in others. For example, in autonomous driving, vehicles may cooperate to avoid collisions and maintain smooth traffic flow (shared safety objective), but compete for limited road resources such as merging lanes or parking spots. Similarly, in multi-player video games or robot soccer, teams may exhibit internal cooperation while competing against opposing teams. These scenarios reflect many realistic multi-agent applications and require more nuanced communication strategies to manage the trade-offs between collaboration and competition. Because incentives shift over time, agents must learn **why** communication is beneficial in some contexts but risky in others, and **how** to adjust message content, timing, or selectivity accordingly. This leads to more complex, context-aware communication patterns than those found in purely cooperative or competitive environments. In the following content, we review representative methods from each category in depth and analyze how their communication strategies arise from the underlying reward structures.

**Cooperative Scenarios**   In cooperative scenarios, the central challenge is that agents must align their behaviors to maximize a shared objective, often under partial observability or limited local information. This structure creates a clear reason for communication: agents benefit when they share information that reduces uncertainty and improves group coordination, which explains **why** communication is encouraged in these environments. A common formulation assigns a shared team reward, where all agents receive the same signal regardless of individual contributions Foerster et al. (2016a); Sukhbaatar et al. (2016); Peng et al. (2017); Das et al. (2018); Wang et al. (2020b); Jiang & Lu (2018); Kim et al. (2019a); Mao et al. (2020); Ding et al. (2020); Agarwal et al. (2019b); Kim et al. (2020); Zhang et al. (2019; 2020); Yun et al. (2021); Kilinc & Montana (2018); Sheng et al. (2022); Freed et al. (2020a); Hu et al. (2020a); Gupta et al. (2023). This equal reward structure naturally promotes cooperation because agents succeed only when the team succeeds.

An alternative design uses individualized rewards that still depend on the performance of others Liu et al. (2020); Singh et al. (2018); Peng et al. (2017); Jiang et al. (2018); Freed et al. (2020a); Kim et al. (2019b); Kong et al. (2017); Gupta et al. (2023). This does not force identical incentives, but agents remain motivated to support teammates because their own returns are tied to overall group performance. Many works strengthen this coupling by adding penalty terms, such as collision costs or inefficiency penalties Niu et al. (2021); Das et al. (2018); Liu et al. (2020); Jiang & Lu (2018); Hu et al. (2020a); Kim et al. (2020); Jiang et al. (2018); Kim et al. (2019b), while others use neighborhood-based reward sharing to promote local cooperation among nearby agents Chu et al. (2020); Qu et al. (2020). These reward designs highlight **why** communication becomes useful: agents need access to each other's observations, planned actions, or intentions to avoid penalties and improve joint outcomes.

Early cooperative communication frameworks approached this need with simple differentiable messaging. DIAL Foerster et al. (2016a) and CommNet Sukhbaatar et al. (2016) introduced end-to-end trainable communication protocols that allowed agents to exchange continuous messages. These methods were among the first to demonstrate **how** communication can directly improve collective performance by enabling shared

latent representations. Their simplicity made them widely applicable, but message exchange was dense and often unnecessary.

Later works improved communication efficiency by making message exchange selective. ATOC Jiang & Lu (2018) introduced an attention mechanism that lets agents decide when communication is useful. This innovation addressed the challenge of unnecessary communication and provided a more scalable solution. IC3Net Singh et al. (2018) expanded on this by giving each agent a communication gating function that learns to open or close communication channels based on individual returns, which offered a principled way to adapt messaging to diverse cooperative tasks.

Subsequent methods developed more structured communication to improve coordination in complex environments. GA-Comm Liu et al. (2020) aligned individual incentives with team outcomes through reward shaping, which reduced conflicts among agents and encouraged more meaningful messages. TarMAC Das et al. (2018) introduced targeted communication using multi-head attention, allowing agents to direct different parts of their messages to different recipients. MAGIC Niu et al. (2021) built on these ideas with modular communication components and dynamic attention to handle dense multi-agent interactions more effectively.

Across these developments, the trend is clear. Early methods provided continuous messaging without structure. Later approaches introduced selective, targeted, and modular mechanisms that better match the cooperative incentive structure. These improvements reflect **how** communication strategies evolve as the underlying cooperative challenge becomes more complex and as agents learn when and what to share to maximize joint performance.

**Competitive Scenarios**   Competitive MARL presents a distinct challenge because agents pursue conflicting objectives. Each agent aims to maximize its own reward while possibly obstructing the progress of others. This creates instability in learning, as policies must adapt to dynamic opponents that continually change their behaviors. These adversarial incentives explain **why** communication becomes delicate. Messages can help an agent anticipate threats or opportunities, but they can also reveal private intentions that opponents might exploit. As a result, communication must be selective and strategically controlled to avoid harming individual performance.

Most benchmarks for competitive MARL, such as the StarCraft Multi-Agent Challenge (SMAC) Zhang et al. (2019; 2020); Samvelyan et al. (2019), primarily evaluate coordination within a single team under adversarial pressure. They do not model communication across opposing teams, since sharing intentions with adversaries is rarely beneficial. This means that explicit competitive communication remains understudied, even though adversarial dynamics highlight the importance of selective information sharing.

IC3Net Singh et al. (2018) provides one of the first concrete examples of **how** controlled communication can emerge in competitive settings. It introduces a learnable gating mechanism that allows each agent to decide when communication is advantageous. This mechanism helps agents avoid revealing unnecessary information, while still exchanging useful signals in moments that directly improve their own utility. IC3Net demonstrates that even self-interested agents can benefit from conditional communication, such as sharing information just before seizing a goal or evading pursuit. Compared with earlier cooperative-oriented methods, IC3Net improves robustness by allowing agents to suppress communication when it creates vulnerability.

MADDPG Lowe et al. (2017) addresses another fundamental challenge in competitive tasks: unstable learning due to interdependent strategies. It adopts a CTDE paradigm with a centralized critic that observes the global state and all agent actions during training. This helps agents learn stable policies in environments where opponents' decisions strongly influence the optimal response. Although MADDPG does not impose an explicit communication channel, its training structure clarifies **why** communication would later become useful. Agents learn policies that depend on global information during training, and message passing at execution time provides a decentralized method for reconstructing that missing context.

Differentiable Inter-Agent Learning (DIAL) Foerster et al. (2016a) offers a complementary perspective by enabling end-to-end learning of communication through differentiable channels. In competitive settings, this capability can yield private or encoded communication strategies that improve coordination within a team without leaking exploitable information. This reflects a different notion of **how** communication can operate

in adversarial environments: not merely sharing raw information, but transmitting abstract or encrypted signals shaped by competitive pressure.

Further insights come from the work of Deka and Sycara Deka & Sycara (2021), who explore the emergence of diverse roles in competitive teams. Their graph neural network architecture and decentralized training process encourage agents to specialize, leading to heterogeneous strategies that counter varied opponent tactics. Specialization offers another path toward improved communication, because agents can tailor messages to complement distinct roles rather than sharing uniform signals.

These works collectively illustrate that communication in competitive MARL follows different principles from cooperative settings. It must be selective, strategically motivated, and often implicit. IC3Net provides conditional sharing, MADDPG stabilizes adversarial learning and motivates communication during execution, DIAL explores optimized latent channels, and role-specialization methods highlight structured coordination under competition. Although competitive communication is still in its early stages, these foundational studies reveal **how** adversarial pressures reshape communication needs and offer promising directions for future research, including deception-resistance, privacy maintenance, and selective signaling.

**Mixed Scenarios** Mixed scenarios involve agents that pursue individual reward functions that may be partially aligned, entirely independent, or conflicting. These environments blend cooperative and competitive dynamics, requiring agents to adapt their strategies and communication behaviors based on context. This hybrid structure explains **why** communication becomes more nuanced than in purely cooperative or competitive tasks. Agents must decide whether sharing information will produce mutual benefit or expose vulnerabilities, and they must recognize how communication affects both short-term and long-term incentives. Mixed settings can also be viewed as a generalization of competitive scenarios because they relax the zero-sum constraint, allowing cooperation and adversarial behavior to emerge simultaneously. The central challenge is determining when communication is beneficial, what information should be shared, and how agents balance self-interest with opportunities for collaboration.

One of the foundational works for mixed settings is MADDPG Lowe et al. (2017), which introduces a CTDE paradigm. By allowing each agent to condition its policy on other agents' actions during training via a centralized critic while preserving decentralized execution, MADDPG provides a robust framework for learning in cooperative-competitive environments. This structure clarifies **how** communication may later be used at execution time: agents rely on global information during training, and message passing becomes a way to approximate that structure once the centralized critic is removed.

RIAL and DIAL Foerster et al. (2016a) extend this perspective by supporting both private and shared communication channels, enabling agents to negotiate implicitly and adopt dynamic roles. Although initially designed for cooperative tasks, their communication flexibility helps explain **how** agents adapt signals to mixed-reward conditions, selectively sharing or withholding information based on changing incentives.

Liu et al. Liu et al. explore the emergence of team-oriented behaviors in competitive multi-agent soccer through decentralized population-based training and reward shaping. Their results illustrate that agents can progress from self-interested strategies toward cooperative patterns without explicit communication, highlighting a different mechanism for coordination that complements message-based approaches.

DGN Jiang et al. (2018) considers environments where agents choose between collecting food for positive rewards or attacking others for greater gain. Its graph-based communication mechanism enables agents to shift from competitive to cooperative behavior, demonstrating an improvement over earlier architectures by exploiting relational structure and proximity-based interactions. IC3Net Singh et al. (2018) contributes another dimension by allowing agents to learn when to communicate, which is particularly useful in mixed tasks where signaling must be conditional on the reward context.

TarMAC Das et al. (2018) proposes targeted communication and evaluates it in Predator-Prey settings, showing that addressing specific receivers improves performance when cooperation and competition must coexist. I2C Ding et al. (2020) builds upon this by introducing incentive-based messaging that accounts for communication cost, encouraging agents to exchange information only when the expected benefit outweighs the overhead. MAGIC Niu et al. (2021) further expands scalability by combining modular communication

with dynamic attention, helping agents focus information flow under shifting cooperative and competitive pressures.

Ryu et al. Ryu et al. (2021) incorporate cooperative and competitive biases into training to improve adaptability, offering a principled way to shape communication tendencies. Santos et al. Santos et al. (2022) investigate dynamic communication availability at execution time, showing how agents can adjust to fluctuations in bandwidth or connectivity.

Together, these studies show that communication in mixed settings depends on reward design, environmental structure, and temporal context. They highlight **how** agents learn to determine when collaboration is beneficial, when independence is necessary, and how to navigate the uncertainty that arises when cooperation and competition coexist.

### 5.5.3 Why and How Communication Formats Matter

In multi agent reinforcement learning, communication formats determine how agents share information and resolve coordination challenges under partial observability. This makes it essential to understand **why agents need to communicate** and **how the chosen format operationalizes that need**. The central difficulty comes from decentralized execution, where agents hold only partial and often inconsistent local views of the environment. As a result, communication formats arise as different solutions to this problem. Explicit communication compensates for missing information through direct message exchange, while implicit communication compensates by allowing agents to interpret or infer hidden intentions from observed behavior. In the following, we examine both formats and show how each method addresses a specific challenge, why communication is needed in that scenario, and how the architecture implements its solution. Cooperative multi agent reinforcement learning has proven effective for collaborative decision-making in diverse domains Canese et al. (2021); Oroojlooy & Hajinezhad (2023). Multi agent environments introduce additional challenges such as partial observability and non stationarity. To address these issues, many approaches adopt the centralized training with decentralized execution paradigm, where agents are trained with access to global information but must operate based only on local observations during execution Rashid et al. (2018); Wang et al. (2020a); Yu et al. (2022a). This information gap explains **why communication becomes necessary**: agents lose access to global state at execution time and must reconstruct missing information through interaction. The **how** depends on whether the system uses explicit messages or behavioral cues.

**Explicit Communication Protocols** Explicit communication directly addresses the challenge of partial observability by enabling agents to transmit structured messages. The **why** is to supply missing information that cannot be inferred locally, and the **how** is realized through differentiable message passing, attention mechanisms, or selective routing modules. Each method chooses its architecture according to the severity of partial observability and the coordination demands of the task. Explicit communication allows agents to exchange observations, latent features, or intentions. Although this enhances coordination, it increases bandwidth cost and may weaken decentralization. Below, we review representative methods and explain their challenge, **why to communicate**, **how to communicate**, and how each improves over earlier work.

- **DIAL** Foerster et al. (2016a). The challenge is learning meaningful messages that support coordination. The **why** is that agents require information beyond their local view. The **how** is through differentiable communication channels that allow gradients to shape message content. This improves over earlier discrete protocols by enabling end to end learning of compact and task specific messages.

- **CommNet** Sukhbaatar et al. (2016). The challenge is scalability when many agents communicate simultaneously. The **why** is that agents must share aggregated context to align behavior. The **how** is through averaging hidden states into a shared representation. This improves efficiency but sacrifices expressiveness, motivating later advances.

- **ATOC** Jiang & Lu (2018). The challenge is eliminating unnecessary communication that causes noise or congestion. The **why** is that communication should occur only when coordination difficulty is high. The **how** is through a learned attention gate that identifies beneficial communication moments. This improves over CommNet by introducing selectivity and reducing redundant messaging.

- **TarMAC** Das et al. (2018). The challenge is routing messages to the correct recipients in heterogeneous teams. The **why** is that agents often need information from specific teammates rather than broadcast messages. The **how** is through multi head target addressing using attention. This improves over ATOC by making communication directional and role aware.

- **IC3Net** Singh et al. (2018). The challenge is handling environments where communication may be harmful or risky. The **why** is that agents should share information only when it improves their own utility. The **how** is through a learnable communication gate that decides whether to speak or stay silent. This improves over TarMAC by enabling context dependent suppression of communication, which is valuable in competitive or noisy environments.

The evolution from CommNet to ATOC to TarMAC to IC3Net shows increasing refinement in both **why communication occurs** and **how messages are structured and routed**. Early works use simple broadcast communication, while later works carefully regulate message timing, direction, and necessity.

**Implicit Communication Protocols**  Implicit communication begins with a different challenge. In many environments, explicit message passing is limited by bandwidth, connectivity, or strategic concerns. The **why** for communication is therefore to preserve coordination under communication constraints. The **how** is to encode information in observable actions or learned latent beliefs, allowing agents to infer intentions from behavior. Implicit communication avoids explicit message passing. Instead, agents learn to interpret movements, actions, or representation shaped cues as informative signals Oliehoek et al. (2016). This reduces bandwidth cost but requires stronger inference capabilities. Below, we summarize main methods with their challenge, **why**, **how**, and improvement.

- **Action based implicit communication**. Knepper et al. Knepper et al. (2017), Shaw et al. Shaw et al. (2022), and Tsiamis et al. Tsiamis et al. (2015). The challenge is coordinating when explicit messaging is unavailable. The **why** is the need to reveal intent through observable behavior. The **how** is choosing actions that encode information for teammates to interpret. These works improve over purely reactive behaviors by formalizing intention revealing actions.

- **Representation based implicit communication**. Tian et al. Tian et al. (2020) introduce belief modules and auxiliary rewards to help agents infer hidden information. The challenge is reconstructing others' states without explicit signals. The **why** is the demand for coordination in settings without communication channels. The **how** is learning internal beliefs that summarize inferred information. This improves over action based approaches by providing more expressive internal models.

  Grupen et al. Grupen et al. (2021) show that spatial cues can act as implicit signals in fully decentralized systems. Their improvement is removing reliance on centralized components.

  Li et al. Li et al. (2023a) propose TACO, which begins with explicit communication but gradually removes this information as agents learn to infer missing context. The challenge is transitioning from explicit to implicit communication. The **why** is to reduce reliance on global information while maintaining coordination quality. The **how** is by training agents to approximate global information through local inference. This improves scalability and robustness in decentralized settings.

Together, these methods progress from simple behavioral cues to increasingly sophisticated latent inference, showing how communication can be embedded implicitly without dedicated channels.

**Comparison and Discussion**  Explicit and implicit communication reflect different answers to the question of **why communication is needed**. Explicit communication compensates for severe partial observability by exchanging structured messages. Implicit communication compensates for communication limits or strategic risks by encoding signals in behavior or learned beliefs. They also reflect different answers to **how communication should occur**. Explicit communication emphasizes message content, routing, and compression. Implicit communication emphasizes behavioral expressiveness and inference. Across both categories, improvements move toward greater selectivity, robustness, and decentralization.

Table 5: Summary of Approaches to 'How to Communicate?' in MARL-Comm

| Category | Sub-Category | Representative Approaches |
|---|---|---|
| **Training Framework (Sec. 5.5.1)** | Centralized Learning | Foerster et al. (2016a); Kraemer & Banerjee (2016) |
| | Decentralized Learning | Agent-Entity Graph Agarwal et al. (2019b), NeurComm Chu et al. (2020), Intention Propagation (IP) Qu et al. (2020), MAGNet-SA-GS-MG Malysheva et al. (2018), DCC-MD Kim et al. (2019b), Diff Discrete Freed et al. (2020a). |
| | Centralized Training with Decentralized Execution (CTDE) | • Individual Policy Parameters: REINFORCE Kong et al. (2017), Actor-critic-based Wang et al. (2020b); Kim et al. (2019a); Mao et al. (2020); Yun et al. (2021)
• Parameter Sharing: local Q-function Foerster et al. (2016a); Jiang et al. (2018); Sheng et al. (2022), global Q-function Zhang et al. (2019; 2020), shared actor receiving gradient guidance from a centralized critic Niu et al. (2021); Du et al. (2021); Das et al. (2018); Jiang & Lu (2018); Hu et al. (2020a); Mao et al. (2020), rewards sampled over entire episodes via REINFORCE Sukhbaatar et al. (2016); Liu et al. (2020); Singh et al. (2018)
• Concurrent Learning: centralized critic to help guide the optimization of its local policy Kim et al. (2020); Pesce & Montana (2020a) |

Table 5 – continued from previous page

| Category | Approach | Representative Works |
|---|---|---|
| **Communication Shaped by Reward Objectives (Sec. 5.5.2)** | Cooperative Scenarios. | • Shared team reward, all agents receive the same signal regardless of individual contributions Foerster et al. (2016a); Sukhbaatar et al. (2016); Peng et al. (2017); Das et al. (2018); Wang et al. (2020b); Jiang & Lu (2018); Kim et al. (2019a); Mao et al. (2020); Ding et al. (2020); Agarwal et al. (2019b); Kim et al. (2020); Zhang et al. (2019; 2020); Yun et al. (2021); Kilinc & Montana (2018); Sheng et al. (2022); Freed et al. (2020a); Hu et al. (2020a); Gupta et al. (2023)
• Individualized rewards that are still dependent on the performance of other agents Liu et al. (2020); Singh et al. (2018); Peng et al. (2017); Jiang et al. (2018); Freed et al. (2020a); Kim et al. (2019b); Kong et al. (2017); Gupta et al. (2023)
• Incorporating penalties Niu et al. (2021); Das et al. (2018); Liu et al. (2020); Jiang & Lu (2018); Hu et al. (2020a); Kim et al. (2020); Jiang et al. (2018); Kim et al. (2019b)
• Neighborhood-based reward sharing to strengthen localized cooperation Chu et al. (2020); Qu et al. (2020); Sukhbaatar et al. (2016); Jiang & Lu (2018). |
| | Competitive Scenarios | • Learnable gating mechanism enabling agents to selectively communicate: IC3Net Singh et al. (2018)
• Centralized critics during training: MADDPG Lowe et al. (2017)
• End-to-end learning of communication messages through differentiable channels: DIAL Foerster et al. (2016a)
• Leveraging graph neural networks and decentralized training Deka & Sycara (2021) |

**Table 5 – continued from previous page**

| Category | Approach | Representative Works |
|---|---|---|
| | Mixed Scenarios | • Centralized training with decentralized execution paradigm: MADDPG Lowe et al. (2017)
• Learn both shared and private communication channels, supporting implicit negotiation and role emergence in partially aligned settings: RIAL and DIAL Foerster et al. (2016a)
• Decentralized population-based training and reward shaping Liu et al.; Jiang et al. (2018)
• Gating mechanism for communication: IC3Net Singh et al. (2018)
• Targeted communication architecture evaluated on mixed scenarios: TarMAC Das et al. (2018); Ding et al. (2020)
• Incorporate cooperative and competitive biases into agent training Ryu et al. (2021)
• Dynamic communication availability at execution time Santos et al. (2022) |
| **Message Format Objectives (Sec. 5.5.3)** | Explicit communication | Singh et al. (2018); Das et al. (2018); Lowe et al. (2017); Foerster et al. (2016a); Chen et al. (2024b); Breazeal et al. (2005) |
| | Implicit communication | Oliehoek et al. (2016); Knepper et al. (2017); Shaw et al. (2022); Tsiamis et al. (2015); Knepper et al. (2017); Tian et al. (2020); Grupen et al. (2021) |

Table 5 provides a structured summary of the key approaches addressing the question of how to communicate in MARL-Comm. The taxonomy is organized into four major categories: training frameworks, reward-shaped communication strategies, communication message formats, and protocol designs. Each category is further subdivided into specific approaches, such as centralized learning, decentralized learning, and the increasingly popular centralized training with decentralized execution. Representative works are provided for each sub-category, highlighting influential algorithms and methods from the literature. The table captures both explicit and implicit communication protocols, as well as variations tailored for cooperative, competitive, and mixed-motive scenarios. This summary offers a comprehensive reference for understanding how communication mechanisms are operationalized in MARL systems, balancing coordination, scalability, and interpretability.

## 5.6 Discussions

To provide a unified perspective on MARL-Comm, this subsection synthesizes communication methods using a structured lens: *what* information is exchanged, *who* speaks, *whom* it is directed to, *when* communication occurs, and finally *how* and *why* communication is carried out. While Table 6 retains its original three columns for clarity, our discussion expands these categories into a more complete framework that reflects recent conceptual developments in MARL-Comm.

Table 6 illustrates the diversity of communication design choices. Early methods emphasize simple message content and uniform broadcasting, while recent works introduce selective roles, structured recipients, and more expressive encoding mechanisms. Below, we reinterpret these trends using the order of what, when, and how, with each part incorporating the related dimensions of who, whom, and why.

Table 6: Representative MARL-Comm methods categorized by what they communicate, whom they communicate with, and how messages are encoded or aggregated.

| Method | What to Comm. | Whom to Comm. | How to Comm. |
|---|---|---|---|
| DIAL Foerster et al. (2016a) | Past Knowledge (w/o rep) | Other Learning Agents | Concatenation |
| RIAL Foerster et al. (2016a) | Past Knowledge (w/o rep) | Other Learning Agents | Concatenation |
| CommNet Sukhbaatar et al. (2016) | Past Knowledge (rep) | Other Learning Agents | Equal Weighting |
| IC3Net Singh et al. (2018) | Past Knowledge (rep) | Other Learning Agents | Equal Weighting |
| TarMAC Das et al. (2018) | Past Knowledge (rep) | Other Learning Agents | Weighted |
| MADDPG-M Kilinc & Montana (2018) | Past Knowledge (w/o rep) | Other Learning Agents | Concatenation |
| ETCNet Hu et al. (2020a) | Past Knowledge (w/o rep) | Other Learning Agents | Concatenation |
| HAMMER Gupta et al. (2023) | Past Knowledge (rep) | Representative Agent | Weighted |
| GA-Comm Liu et al. (2020) | Past Knowledge (rep) | Representative Agent | Weighted |
| MAGIC Niu et al. (2021) | Past Knowledge (rep) | Representative Agent | Weighted |
| ATOC Jiang & Lu (2018) | Future Knowledge | Representative Agent | RNN-based |
| FlowComm Du et al. (2021) | Past Knowledge (w/o rep) | Nearby Agents | Equal Weighting |
| GAXNet Yun et al. (2021) | Past Knowledge (w/o rep) | Nearby Agents | RNN and Attention |
| DGN Jiang et al. (2018) | Past Knowledge (w/o rep) | Nearby Agents | CNN |
| I2C Ding et al. (2020) | Future Knowledge | Other Learning Agents | RNN |
| MD-MADDPG Pesce & Montana (2020a) | Past Knowledge (rep) | Representative Agent | RNN |
| Gated-ACML Mao et al. (2020) | Past Knowledge (rep) | Representative Agent | Gated NN |
| MAGNet-SA-GS-MG Malysheva et al. (2018) | Past Knowledge (rep) | Nearby Agents | Attention-based |
| NeurComm Chu et al. (2020) | Future Knowledge | Nearby Agents | RNN |
| Agent-Entity Graph Agarwal et al. (2019b) | Past Knowledge (w/o rep) | Nearby Agents | Attention-based |

### 5.6.1 Who Communicates with Whom and What Is Shared

The first design decision concerns the nature of the information being exchanged. Methods transmit local observations, latent encodings, predicted intentions, or learned value-related features. What agents choose to broadcast is tightly connected to who is responsible for speaking and whom the message is intended for.

In early systems such as DIAL Foerster et al. (2016a) and RIAL Foerster et al. (2016a), all agents share low-level hidden states, reflecting a simple assumption where every agent serves as both sender and receiver. CommNet Sukhbaatar et al. (2016) introduces representation-based sharing, enabling agents to reason over a pooled latent feature but still maintaining a fully inclusive broadcasting pattern. As the field evolves, newer models refine both the content and the routing structure. For instance, TarMAC Das et al. (2018) assigns messages to specific teammates, guided by role or spatial relevance. MAGIC Niu et al. (2021) and GA-Comm Liu et al. (2020) reduce redundancy by designating representative agents that compress group-level information. IC3Net Singh et al. (2018) and Gated-ACML Mao et al. (2020) further introduce learned speaking decisions, where only agents that expect their information to be useful choose to communicate. This progression reflects a shift toward more deliberate and purpose-driven messaging.

### 5.6.2 When Communication Is Necessary

Timing is increasingly treated as an independent design dimension. In many classical architectures, communication is continuous and unconditional. However, always-on communication is rarely efficient and can overwhelm learning dynamics. Recent methods identify the need to transmit information only when it becomes valuable.

Event-driven communication, as seen in ETCNet Hu et al. (2020a), activates messaging only when significant environmental changes occur. Predictive communication in ATOC Jiang & Lu (2018) triggers information exchange before agents enter cooperative interactions. Gating mechanisms in IC3Net Singh et al. (2018) and other selective frameworks learn to suppress communication in uninformative moments, reducing bandwidth and improving robustness. These advances show that the temporal aspect of communication is not merely an optimization detail but a fundamental part of multi-agent coordination.

### 5.6.3 Why Communication Helps and How It Is Used

The underlying reason for communicating determines the structure of the communication protocol. When the purpose is to reduce partial observability, methods favor explicit message passing with expressive encoders and learnable aggregation functions. When the goal is to align distributed representations or coordinate long-horizon planning, models adopt attention-based or role-aware mechanisms that shape the influence of messages. When communication must be conservative or strategically safe, selective gating or representative-based designs become preferable.

Concatenation, equal weighting, and simple pooling dominate early algorithms such as DIAL Foerster et al. (2016a), RIAL Foerster et al. (2016a), and MADDPG-M Kilinc & Montana (2018), providing scalable but coarse message integration. Attention-based methods such as TarMAC Das et al. (2018), MAGIC Niu et al. (2021), GAXNet Yun et al. (2021), and MAGNet-SA-GS-MG Malysheva et al. (2018) introduce finer-grained routing that reflects task relevance. Neural architectures such as RNNs used in I2C Ding et al. (2020) and NeurComm Chu et al. (2020), CNNs used in DGN Jiang et al. (2018), and gated networks as in Gated-ACML Mao et al. (2020) offer richer temporal or spatial structure. Across all these approaches, the choice of how to encode and aggregate messages is closely connected to why communication is beneficial in the first place.

Overall, framing MARL-Comm through what, when, and how clarifies the role of each design element. Future work is likely to explore more adaptive message content, richer temporal scheduling, and tighter alignment between communication purpose and communication protocol.

### 5.7 Bridge: From MARL-Comm to Emergent Language

Table 7: From MARL-based communication to emergent language: limitations in MARL-Comm that motivated EL research.

| Aspect | MARL-Based Communication | Motivation Toward Emergent Language |
|---|---|---|
| **Message form** | • Continuous or latent vectors;
• Entangled with policy networks. | • Discrete symbols or tokens;
• Explicit communicative units. |
| **Interpretability** | • High task performance;
• Low human interpretability. | • Focus on semantic structure;
• Human-readable protocols. |
| **Generalization** | • Jointly trained agents;
• Weak zero-shot transfer. | • Partner generalization;
• Population-level consistency. |
| **Semantic grounding** | • Implicit, reward-driven semantics. | • Explicit symbol–state alignment. |
| **Core limitation** | • Opaque, non-transferable signals. | • Study why and how language emerges. |

Although MARL-Comm has achieved strong empirical performance in cooperative control, much of this progress relies on learning task-specific signaling protocols that are tightly coupled to the environment, reward design, and training population. Consequently, the learned messages are often difficult to interpret, hard to verify, and challenging to transfer to new tasks or unseen teammates. Moreover, many MARL-Comm methods assume continuous or high-bandwidth communication channels, which can mask the semantic structure of what is being communicated and abstract away practical constraints.

These limitations motivated research on emergent language, which shifts the focus from optimizing task return to understanding how discrete, structured communication protocols arise through interaction. In particular, emergent language studies aim to characterize when communication becomes necessary, what compositional structure can emerge, and how protocol stability and generalization behave across different games, environments, and agent populations. This perspective complements MARL-Comm by emphasizing interpretability and generalization as first-class objectives, rather than byproducts of task optimization. Table 7 highlights how limitations in MARL-based communication, particularly **interpretability** and **generalization**, motivated the emergence of discrete and semantically grounded communication research.

## 6 Emergent Language in Multi-agent Systems

Deep learning advances in natural language processing and multi-agent reinforcement learning have enabled a new paradigm for studying how communication systems can arise through interaction. This line of research, commonly referred to as **Emergent Language (EL)** or emergent communication, uses learning-based agents to simulate the formation of human-like signaling behaviors without predefined semantics Boldt & Mortensen (2024). Large-scale self-play systems, such as AlphaZero Silver et al. (2017) and OpenAI's hide-and-seek agents Baker et al., have demonstrated that complex coordination strategies can emerge purely from environmental dynamics and reward feedback. Building on these insights, early EL studies extended self-play to discrete communication settings, showing how symbolic protocols can arise from task-driven interaction Foerster et al. (2016a); Lazaridou et al. (2016); Havrylov & Titov (2017); Mordatch & Abbeel (2018). While modeling language emergence offers a compelling lens for understanding communication, EL research has also become increasingly relevant for designing scalable, interpretable coordination mechanisms in artificial multi-agent systems.

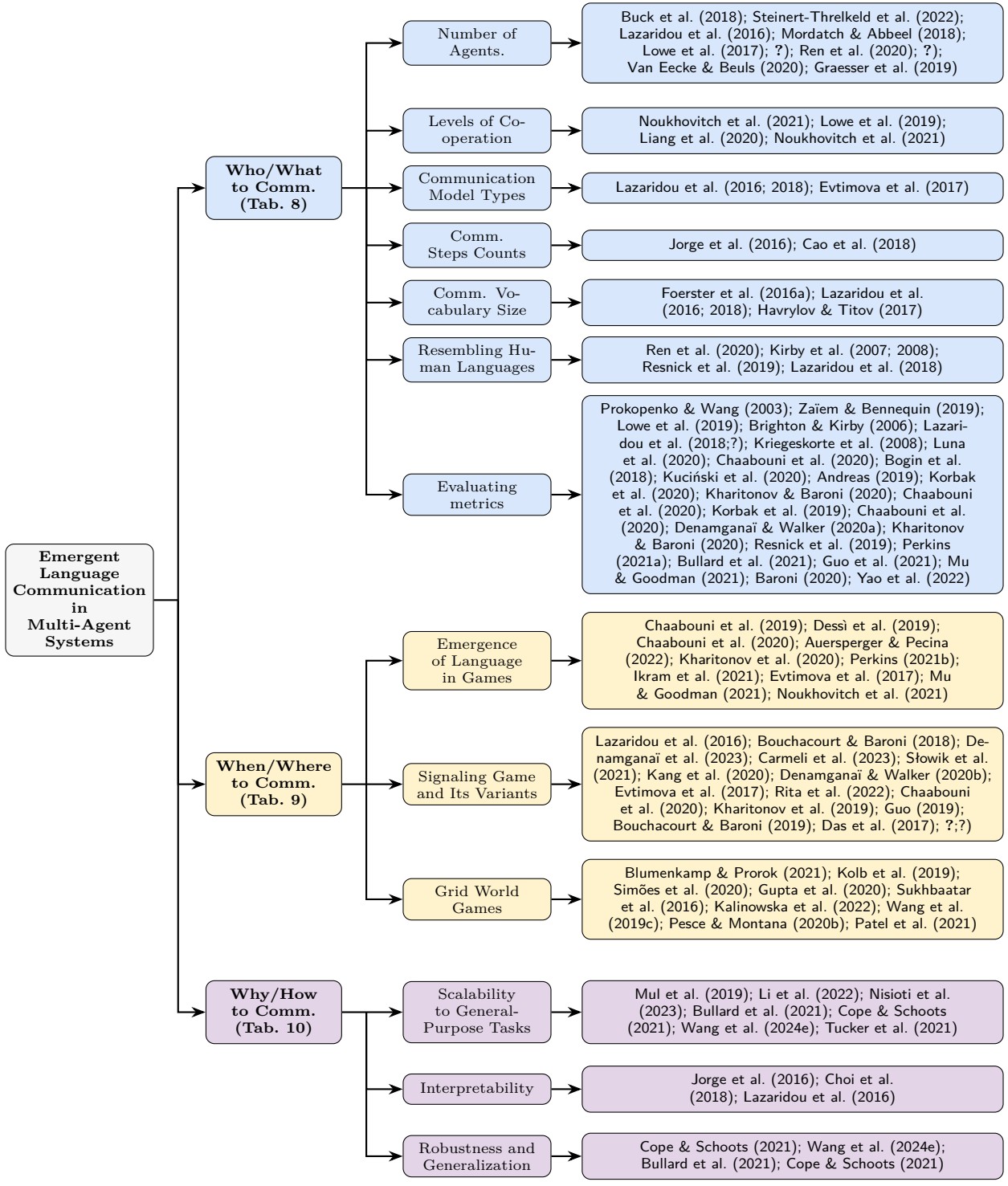

Figure 7: EL-Comm Agents Taxonomy

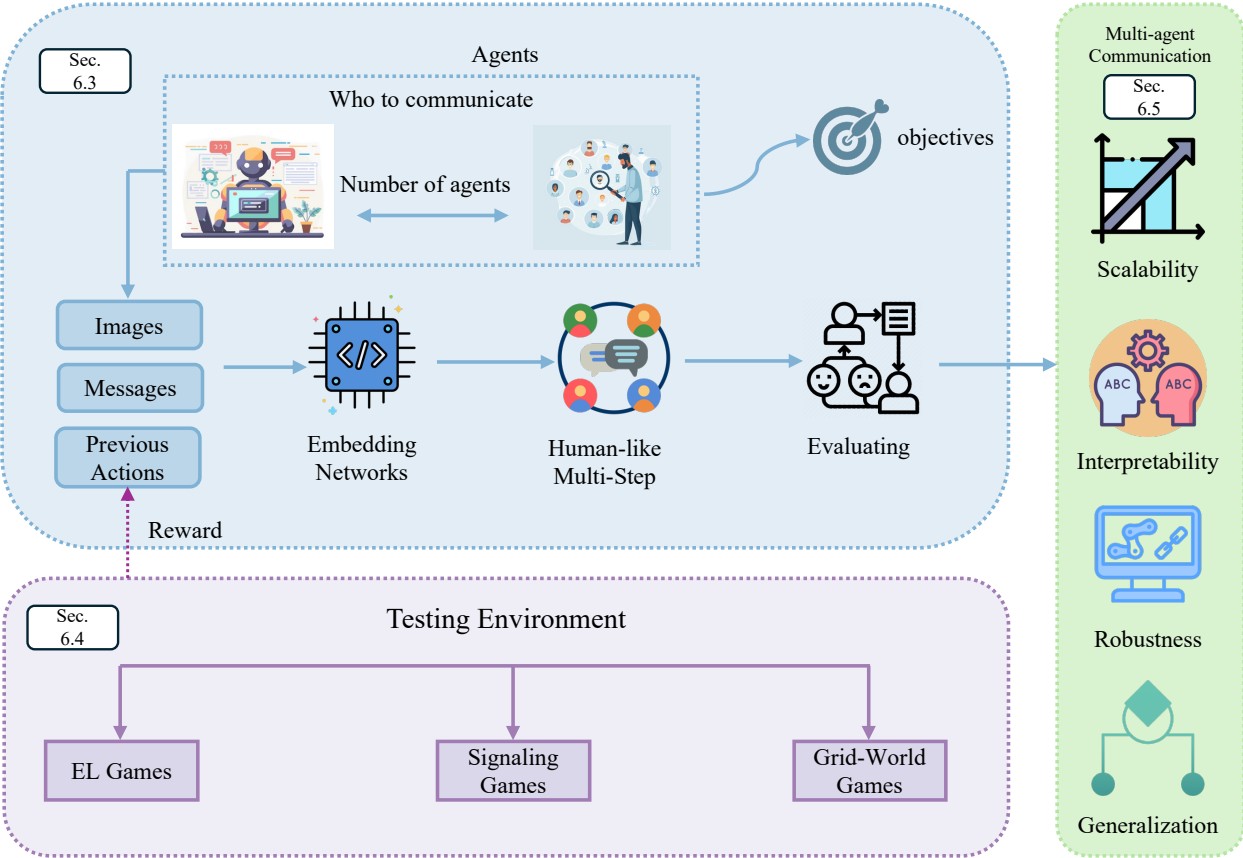

Figure 8: Illustrating the organization of Sec. 6 under the **Five Ws** lens. We analyze **who communicates with whom** and **what is communicated** via population roles, recipient selection, and message encoding/metrics (Sec. 6.3), including multi-step dialogue and compositionality). Next, we examine **when** and **where** communication occurs through standardized toolkits, one-shot versus iterative protocols, and grounded or embodied environments (Sec. 6.4). Finally, we discuss **why** communication emerges and **how** it is learned and shaped, covering motivations, interpretability/structured protocols, and robustness/generalization (Sec. 6.5).

**Review Scope and Structure**   Emergent Language Communication (EL-Comm) investigates how agents develop communication protocols *from interaction alone*, without explicit supervision or predefined meaning. This section begins with foundational background and core concepts (Section 6.1), followed by an overview and taxonomy of the EL literature (Section 6.2). We then organize prior work using the Five Ws perspective: **who communicates with whom and what is shared** (Section 6.3), **when and where communication occurs** through interaction structure and environment design (Section 6.4), and **why and how communication emerges** via learning objectives, protocol structure, and generalization properties (Section 6.5). Section 6.6 synthesizes these dimensions, and Section 6.7 connects EL-Comm to LLM-based communication, highlighting limitations that motivate language-grounded and hybrid approaches. Figures 7 and 8 summarize the taxonomy and end-to-end EL pipeline, guiding readers through key design choices, environments, and evaluation criteria.

## 6.1   Background and Foundations of Emergent Language

Emergent language has become a rapidly advancing area of research within artificial intelligence, particularly in the realm of multi-agent reinforcement learning. Historically, the study of language emergence focused on understanding the origins of human language, with limited attention to its applicability for artificial agents. However, recent advancements in reinforcement learning aim to enable agents to develop communication protocols with capabilities comparable to or even exceeding human language. These efforts extend beyond the statistical approaches commonly used in natural language processing, prompting critical questions about the conditions for language emergence and how to effectively evaluate its success. To provide context for the taxonomy and analysis, this subsection outlines the fundamental concepts of communication and linguistics and offers background knowledge of emergent language research Peters et al. (2024).

Communication fundamentally involves the exchange of signals that convey information, whether intentional, such as speech, or unintentional, like bodily reactions. According to Watzlawik's 'Interactional View' Watzlawick et al. (2011), all behavior communicates, making communication universal and essential. It occurs through various channels and can be broadly categorized into intrapersonal, interpersonal, group, public, and mass communication, depending on its purpose and audience Peters et al. (2024). In emergent language research, two forms of communication are commonly studied: interpersonal and group communication. Interpersonal communication involves interactions between entities that influence each other, with each entity perceiving its own environment, often overlapping through a shared communication channel. Noise may affect the process by distorting perception or communication. Group communication extends this to multiple entities, focusing on collective goals or tasks, making it more formal than interpersonal communication, which often has a social character. Most population-based EL research focuses on group communication, while intrapersonal, public, and mass communication remain largely unexplored.

Natural language is one of humanity's greatest achievements, serving as the foundation for all forms of communication. It enables us to convey highly complex information through a discrete and humanly manageable amount of utterances. Much AI research focuses on developing natural language models for tasks like translation and text generation Wolf et al. (2020); Brown et al. (2020); Lauriola et al. (2022); Khurana et al. (2023). However, many in the AI community argue that current statistical models trained on static datasets lack true language understanding for effective human cooperation Mordatch & Abbeel (2018); Choi et al. (2018); Lazaridou et al. (2021); Merrill et al. (2021). Emerging EL research in AI aims to enable agents to communicate like humans, enhancing cooperation, performance, and generalization, and fostering meaningful interaction between humans and AI Mordatch & Abbeel (2018). While explicit forms of EC are widely studied, some survey papers excluded implicit communication Peters et al. (2024), such as spatial positioning in multi-agent systems Grupen et al. (2022).

Emergent Language (EL) is a form of communication that develops naturally among artificial agents through interaction, without explicit programming. It arises from the agents' need to cooperate and solve tasks in their environment Brandizzi (2023). EL involves agents creating, adapting, and refining linguistic structures to exchange information effectively Lipowska & Lipowski (2022). Research focuses on understanding how elements like syntax Ueda et al. (2022), semantics Colas et al. (2020); Qiu et al. (2022), and pragmatics Lowe et al. (2019) emerge and enhance agent performance and cooperation. A natural language-like communication system would make artificial agents more accessible, comprehensible, and powerful Iocchi et al. (2022);

Noukhovitch et al. (2021); Lazaridou & Baroni (2020). Initially focused on the origins of language Steels (1997), EL research now emphasizes enabling agents to harness communication mechanisms akin to natural language Lazaridou & Baroni (2020). Modern EL in computer science involves self-learned, reusable, teachable, and interpretable communication protocols Nowak & Krakauer (1999); Lazaridou et al. (2016); Li & Bowling (2019). The ultimate goal is seamless and extensible communication between machines and humans Lazaridou & Baroni (2020); Yao et al..

In this section, we review papers that proposed agent-based computer models in which communication emerges as a general paradigm for coordinating behavior in multi-agent systems Boldt & Mortensen (2024). These models feature agents with communication channels that generate discrete message symbols, aiming to mimic human-like language. The structure and content of the messages are not predefined but instead emerge naturally during training, shaped by the agents' interactions and the characteristics of their environments. Such approaches have been explored within a variety of agentic frameworks, often leveraging deep learning methods and neural networks optimized through gradient descent.

## 6.2 Overview and Taxonomy of Emergent Language

Communication arises from conventions and rules developed through the need or benefit of coordination. Lewis Lewis (2008) formalized such scenarios as 'coordination problems' and introduced a simple signaling game, where a speaker describes an object and a listener identifies it among options. This game significantly influenced emergent language (EL) research in computer science. Early studies focused on specific aspects of emergent communication (EC) through hand-crafted simulations Wagner et al. (2003); Steels (1997); Nowak & Krakauer (1999); Kirby (2002); Cangelosi & Parisi (2012); Christiansen & Kirby (2003); Batali (1998); Oliphant & Batali (1997); Steels (1995); Skyrms (2002); Smith et al. (2003), often relying on supervised learning and non-situated settings, which limited their exploration of complex linguistic features Wagner et al. (2003). Recently, EL research has gained momentum Foerster et al. (2016a); Lazaridou et al. (2016); Havrylov & Titov (2017); Bouchacourt & Baroni (2018); Cao et al. (2018); Mordatch & Abbeel (2018); Das et al. (2017) leveraging multi-agent reinforcement learning (MARL) approaches Agarwal et al. (2019a); Blumenkamp & Prorok (2021); Brandizzi et al. (2022); Iocchi et al. (2022); Chaabouni et al. (2022); Gupta et al. (2021); Lo & Sengupta; Lowe et al. (2019); Vanneste et al. (2022b); Yu et al. (2022b) to study more intricate features of language emergence.

A key goal of EL research in MARL is to enable agents to autonomously develop communication systems that support both agent-to-agent and agent-to-human interaction in a natural language (NL)-like manner Wagner et al. (2003); Bouchacourt & Baroni (2018); Iocchi et al. (2022); Lo & Sengupta; Bogin et al. (2018); Noukhovitch et al. (2021). Reinforcement learning (RL) methods are particularly promising for this purpose. They not only create agents capable of flexible, practical communication Lazaridou & Baroni (2020) but also offer insights into the evolution of NL itself Galke et al.. Unlike NLP models, which rely on static datasets to mimic NL Steinert-Threlkeld et al. (2022); Bender & Koller (2020), EL focuses on agents designing and learning their own communication systems through active, goal-driven experiences Lemon (2022). This experiential approach contrasts with the shallow understanding of language seen in large language models (LLMs) Manning & Schutze (1999); Qiu et al. (2020); Wolf et al. (2020). In EL, agents develop communication while solving tasks, mirroring natural processes Lewis (2008), leading to a deeper grasp of their environment Bogin et al. (2018). Advances in EL pave the way for innovative multi-agent systems and more human-centric AI applications Lazaridou & Baroni (2020).

Various research areas and questions have emerged in EL. Recent studies explore diverse settings, including semi-cooperative environments Noukhovitch et al. (2021); Liang et al. (2020), adversaries Blumenkamp & Prorok (2021); Yu et al. (2022b), noise in messages Cope & Schoots (2020), and social structures Dubova et al.; Fitzgerald & Tagliabue (2022). Grounding EL has been approached through representation learning Lin et al. (2021), combining supervised learning with self-play Lowe et al. (2020), or using EL agents for fine-tuning NL models Yao et al.. Other work examines the emergence of NL-like characteristics, such as internal and external pressures Luna et al. (2020); Kalinowska et al. (2022); Dagan et al. (2021); Iocchi et al. (2022); Rita et al. (2022), compositionality Resnick et al. (2020), generalization Chaabouni et al. (2022), expressivity Guo et al., and the connection between compositionality and generalization Chaabouni et al. (2020). Unlike NLP models like LLMs, which imitate language statistically without addressing the

functional purpose of communication Mordatch & Abbeel (2018), EL treats language as a tool for achieving meaningful outcomes Brandizzi et al. (2022). Agents must learn EL through necessity or benefits Luna et al. (2020), in settings that encourage communication, such as cooperative scenarios Noukhovitch et al. (2021). However, EL faces significant challenges. Encouraging communication can lead to task-specific gibberish rather than NL-like language Mu & Goodman (2021), making proper incentives critical. Additionally, measuring successful communication and assessing language properties like syntax, semantics, and pragmatics are essential for guiding and evaluating EL development Lowe et al. (2019); Chaabouni et al. (2020). The following sections delve into these challenges and related approaches.

Existing survey publications on EL research address similar topics but differ in scope. Peters et al. Peters et al. (2024) categorized these surveys into three main directions, and we expand on this framework by including more recent works and providing broader descriptions. Surveys focusing on learning settings explore the design of environments and learning processes, as seen in works Van Eecke & Beuls (2020); Lipowska & Lipowski (2022); Denamganaï & Walker (2020b). Those emphasizing methods review learning and evaluation techniques, including studies Lowe et al. (2019); Lemon (2022); Korbak et al. (2020); LaCroix (2019); Mihai & Hare (2021); Galke & Raviv (2024); Vanneste et al. (2022a). Lastly, surveys offering general overviews provide broad insights into the EL field, as demonstrated in Iocchi et al. (2022); Lazaridou & Baroni (2020); Galke et al.; Hernandez-Leal et al. (2019); Fernando et al. (2020); Brandizzi (2023); Boldt & Mortensen (2024); Peters et al. (2024).

There are also several surveys papers that reviews the EL research. Some surveys focus on learning problems, environments, and language learning design. van Eecke and Beuls Van Eecke & Beuls (2020) reviewed the language game paradigm, categorizing experiments and highlighting properties like symmetric agent roles and autonomous behavior. Our survey extends beyond the language game paradigm, providing more details about RL-based communication protocols. Similarly, Lipowska and Lipowski Lipowska & Lipowski (2022) explored EL in MARL, emphasizing naming games, protolanguage explainability, and sociocultural factors like migration and teachability. While these are included in our analysis, we integrate them into a broader context. Denamganaï and Walker Denamganaï & Walker (2020b) reviewed referential games, introducing the ReferentialGym framework and metrics like positive signaling and listening Lowe et al. (2019). Our survey includes referential games but covers a wider range of metrics and approaches.

Some surveys focused on learning and evaluation methods. Korbak et al. Korbak et al. (2020) discussed compositionality metrics, introducing the tree reconstruction error to address gaps in analyzing semantic aspects. LaCroix LaCroix (2019) critiqued the focus on compositionality in EL, suggesting reflexivity as a better direction, though metrics for this remain undeveloped. Lemon Lemon (2022) reviewed language grounding, emphasizing the need for better data collection, but did not propose metrics. Lowe et al. Lowe et al. (2019) highlighted language utility over semantics, proposing metrics like positive signaling and listening, and Mihai and Hare Mihai & Hare (2021) emphasized exploring semantic factors rather than low-level hashes but did not introduce new metrics. Galke and Raviv Galke & Raviv (2024) identified pressures and biases aligning neural EL with human NL, focusing on NL phenomena rather than training biases. Vanneste et al. Vanneste et al. (2022a) reviewed discretization methods for MARL-based EL, recommending DRU, straight-through DRU, and Gumbel-Softmax for general use, though these are not central to our survey.

There are also more existing surveys providing general overviews or focus on unique aspects of EL research that do not fit into the settings or methods categories. Hernandez-Leal et al. Hernandez-Leal et al. (2019) offered a broad review of multi-agent reinforcement learning (MARL), categorizing work into emergent behavior, communication, cooperation, and agent modeling. While their survey is valuable for understanding MARL's historical context, it lacks detailed analysis of EL-specific metrics, which we address in this work. Brandizzi and Iocchi Iocchi et al. (2022) emphasized human-in-the-loop interactions in EC research, highlighting the underrepresentation of human-AI communication. They explored interaction types but did not provide a comprehensive review of EL papers, as we do. Moulin-Frier and Oudeyer Moulin-Frier & Oudeyer (2020) connected MARL to linguistic theories, identifying challenges like decentralized learning and intrinsic motivation but omitted detailed metric discussions. Galke et al. Galke et al. reviewed RL-based EC approaches, focusing on the gap between emergent and human language, particularly in compositionality. Fernando et al. Fernando et al. (2020) proposed a novel drawing-based communication system, which, while interesting, falls outside our survey's scope. Suglia et al. Suglia et al. (2024) reviewed visually grounded lan-

guage games, categorizing tasks and models but primarily focusing on multimodal grounding rather than EL itself. Zhu et al. Zhu et al. (2024) structured EC works along nine dimensions but focused more on learning tasks than EL emergence. Lazaridou and Baroni Lazaridou & Baroni (2020) provided a summary of the EL field but did not emphasize metrics or taxonomy, which are central to our work. Similarly, Brandizzi Brandizzi (2023) categorized EC research and identified open challenges but lacked a structured taxonomy or a systematic literature approach, which we include with more publications. Boldt et al. Boldt & Mortensen (2024) offer a review of EL research, however, it mainly focused specifically on the goals and applications of EL research.

### 6.3 Who, Whom and What to Communicate: Preliminary Considerations

In emergent communication systems, the foundational questions of **who communicates with whom** and **what** information is exchanged play a decisive role in determining whether a structured protocol can arise at all. Unlike MARL-Comm where these dimensions are often shaped by explicit architectural design, emergent language settings require these components to be discovered through interaction. Thus, separating the analysis along the axes of '**who/whom**' and '**what**' is essential for understanding how communication emerges from minimal assumptions.

We group **who/whom and what** together in this subsection since these dimensions are tightly coupled in emergent communication: the identity and number of interacting agents directly constrain the informational content that must be transmitted, and conversely, the nature of the information structure shapes which agents must communicate and how frequently. Studying them in isolation would obscure these dependencies, whereas a unified treatment allows us to characterize how agent roles, task structure, and message representations co-evolve.

To provide a principled organization of this space, we examine the following dimensions that collectively address the core **who/whom/what** challenges: (1) specifying the communicating parties (who/whom), (2) determining how social incentives shape when and why communication is beneficial (why + whom), (3) identifying the representations used to encode messages (what), (4) structuring the interaction protocol through which information is exchanged (what + whom), (5) regulating the expressive capacity of the communication channel (what), and (6) shaping the form and linguistic structure of emergent messages (what). (7) evaluating the structural and functional properties of the resulting messages through formal metrics that assess informativeness, compositionality, and generalization (what). Each of these dimensions reflects a distinct foundational challenge that any emergent communication system must resolve.

We therefore structure this subsection around these perspectives, as each cluster of works addresses a different component of the broader 'who/whom/what' problem. In the following paragraphs, we summarize representative studies within each category and highlight how they refine our understanding of how agents decide who communicates with whom and what information is transmitted.

#### 6.3.1 Population Size and Communicative Roles

A central challenge in emergent communication is determining **which agents participate in communication** and **how their roles structure the emergence of a shared protocol**. Unlike MARL-Comm, where communication roles are often tied to task-driven coordination needs, research in EL has historically treated 'who communicates with whom' as a structural design choice. This choice directly constrains what messages can be produced, how interaction unfolds, and the expressive capacity of the resulting communication system. Solutions in the EL literature typically instantiate one of three communication configurations: single-agent, dual-agent, or population settings Peters et al. (2024), each progressively increasing the complexity of the who/whom structure and the flexibility available for language-like behavior to emerge.

Early work adopts **single-agent settings**, which are comparatively rare but useful when the goal is to model *human–machine interfaces.* Buck et al. Buck et al. (2018) use a single agent to learn interpretable representations that facilitate human–AI interaction, while Steinert et al. Steinert-Threlkeld et al. (2022) focus on model fine-tuning where the emergent messages serve as internal scaffolding rather than inter-agent communication. These works address the **who/whom** challenge by effectively removing multi-agent

interaction, showing that emergent symbol structures can arise even without a conversational partner, but at the expense of neglecting genuine dialogue dynamics.

The field then evolved toward **dual-agent communication**, which became the dominant paradigm for studying emergent protocols. Lazaridou et al. Lazaridou et al. (2016) introduced the canonical speaker–listener referential game, establishing clear communicative roles: one agent selects a message and the other interprets it. This structured who/whom configuration enabled rigorous study of how symbolic communication emerges from cooperative pressures. Subsequent work Mordatch & Abbeel (2018) deepened this analysis, examining how variations in input distributions and training dynamics influence emergent utterances. The same speaker–listener paradigm was later adopted in grounded settings such as Lowe et al. (2017), where messages come from a fixed K-dimensional one-hot vocabulary. While this restricts what can be communicated, it isolates how role asymmetry shapes the form of emergent messages. Across these dual-agent systems, the improvement over single-agent designs lies in introducing role specialization, which allows researchers to disentangle message production from message interpretation and thus study the structural properties of emergent language.

More recent works explore **population-based communication**, where multiple agents interact simultaneously. These settings address a new challenge—how protocols scale and stabilize when many agents must coordinate or compete. Agarwal et al. **?** show that populations enable *community regularization*, which stabilizes emergent languages and prevents degenerate conventions from dominating. Ren et al. Ren et al. (2020) demonstrate that multi-agent populations naturally support *language evolution* through iterated learning, allowing compositional structure to emerge over generations of agents. Additional studies Graesser et al. (2019); Van Eecke & Beuls (2020) examine how larger populations provide more opportunities to shape the emergent protocol through selective partner interaction, variation in roles, and exposure to heterogeneous communication partners. Population settings therefore improve over earlier paradigms by revealing how social structure—not just pairwise coordination—affects which protocols are selected, transmitted, and stabilized.

Together, these works define the 'who/whom' landscape by showing how the number of interacting agents, the asymmetry or symmetry of their roles, and the social structure of the population constrain the interaction dynamics and shape the space of possible emergent messages.

### 6.3.2 Cooperation Level and Recipient Selection

Another challenge in emergent communication research is understanding how social incentives that are encoded through the reward structure shape both **who** communicates with **whom** and what information must be exchanged. Because communication is costly, ambiguous, or strategically manipulable, different cooperative structures impose distinct pressures on the emergence, stability, and semantic content of messages. Thus, **the degree of cooperation** in the environment is not merely a contextual detail but a primary determinant of when communication becomes necessary and which communication strategies are viable.

To examine this challenge, prior work has analyzed emergent communication under three primary incentive structures: **fully cooperative**, **semi-cooperative**, and **fully competitive** environments. Each structure provides a different answer to the '**who/whom**' and '**what**' dimensions, as the reward topology dictates which agents depend on one another and what types of information are beneficial to share.

In early work, **fully cooperative** settings dominate the literature because agents share a global reward and therefore rely on communication to jointly reduce uncertainty and coordinate successful task completion. Studies such as Noukhovitch et al. Noukhovitch et al. (2021) show that when all agents receive identical rewards and have no individual incentives, communication naturally emerges as a tool for aligning beliefs and actions. Here, the 'who communicates with whom' dimension is simple: every agent's payoff is intertwined while the 'what' dimension centers on transmitting task-relevant referential or relational features needed for group success.

**Semi-cooperative** environments, examined in works such as Lowe et al. Lowe et al. (2019) and Liang et al. Liang et al. (2020), introduce both shared and individual rewards. This incentive structure complicates **who** should communicate with **whom**: agents must decide whether sharing information benefits the group, themselves, or potentially empowers competitors. The challenge therefore shifts toward balancing cooperative

information exchange with self-interested decision-making. These studies demonstrate that communication content becomes more selective and strategic, which means the agents often share partial or compressed signals that advance both personal and collective utility.

**Fully competitive** environments are comparatively rare in emergent communication research. Noukhovitch et al. Noukhovitch et al. (2021) present one of the only systematic examinations. Here, the incentive structure makes the **who/whom** dimension adversarial: agents may choose to conceal information or even communicate deceptively, as sharing accurate messages directly benefits opponents. As a result, meaningful communication often fails to emerge unless some cooperative substructure exists. These results highlight the limits of emergent communication under purely competitive incentives and highlight that '**what**' is communicated becomes strategically suppressed or intentionally misleading.

Taken together, these works show that the levels of cooperation fundamentally determine the communication topology (**who speaks to whom**) and the semantic structure of messages (**what must be conveyed**). Cooperative environments encourage broad, informative communication; semi-cooperative environments induce selective and self-interested signaling; and competitive environments inhibit communication altogether or promote deceptive strategies.

### 6.3.3 Architectural Mechanisms for Encoding What is Communicated

Representational choices strongly influence how emergent communication develops: before agents can form a protocol, they must first determine **what** information from raw perceptual input should be encoded and transmitted. Existing EL systems approach this problem by designing neural architectures that extract, compress, and structure evidence for communication. We group these works together because they address the same underlying question of how architectural choices determine the content of messages, their design decisions also collectively define the dominant strategies for specifying '**what**' is communicated and how this interacts with the roles of speaker and listener.

**CNN-based Communication Architectures**   One early line of work focuses on improving the perceptual encoding available to the speaker, thereby refining **what** information enters the communication channel. In the referential games of Lazaridou et al. Lazaridou et al. (2016), Convolutional Neural Networks are used to transform raw images into higher-quality visual features. This architectural change strengthens the communicative '**what**' by ensuring that messages are grounded in more discriminative perceptual representations, enabling the listener to resolve the target more reliably.

**Attention-based Communication Architectures**   Subsequent work by Evtimova et al. Evtimova et al. (2017) incorporates an attention mechanism into the message-generation process, allowing the encoding network to emphasize salient object regions. Attention not only clarifies **what** information should be highlighted in the message but also implicitly addresses **whom** the speaker is informing, since the mechanism prioritizes features most useful for the listener's decision-making process. The resulting messages generalize better to unseen objects precisely because they reflect feature relevance rather than uniform encoding.

**LSTM-based Communication Architectures**   Lazaridou et al. Lazaridou et al. (2018) further extend the representational pipeline by introducing an LSTM-based recurrent decoder, drawing on DRQN principles Hausknecht & Stone (2015). Unlike earlier one-shot symbol generation, the recurrent policy maintains a temporal memory of previously produced tokens and intermediate states. This enables messages to form multi-token sequences, enriching the expressive space available for communication and allowing '**what**' is communicated to unfold over time. Recurrent encoding thus supports more structured and compositional message formation, a property important for later EL work.

Across these studies, the architectural template is consistent: a CNN encoder produces a hidden state $h_{LR}$; a recurrent decoder generates a message $m = g(h_{LR}, \theta_g)$; and a listener-side LSTM encodes this message into $z = h(m, \theta_h)$ before comparing it with candidate images to select the correct target. **Speaker and listener** parameters are optimized jointly without weight sharing, reflecting their asymmetric contributions to the communication pipeline.

Collectively, these architectures clarify the '**what**' dimension of emergent communication by showing how perceptual encoders, attention mechanisms, and recurrent policies influence the semantic content of messages and determine how effectively information is conveyed between agents.

### 6.3.4 Multi-Step Interaction and Dialog Structure

A well-established model of language emergence was outlined above, but the literature has since expanded to include several variants that incorporate more interactive, human-like communication patterns. One line of work, exemplified by Jorge et al. (2016) and Cao et al. (2018), introduces multi-step exchanges that more closely resemble dialog.

In Jorge et al. (2016), the task remains referential, identifying target images (celebrities), yet both agents participate in message generation. The guessing agent begins the interaction by posing questions derived from the candidate images. The answering agent, who knows the target, responds with binary signals such as 'yes' or 'no', similar to the mechanics of the Guess Who game. The most effective performance is achieved with two rounds of questioning and answering. The authors report that the questioning agent tends to focus on distinct visual attributes across the two questions, thereby accumulating information in a structured manner.

Cao et al. Cao et al. (2018) also examine multi-step communication but shift from referential games to negotiation tasks in which agents bargain over items with different utilities. Their framework introduces two complementary communication channels: a symbolic channel for exchanging abstract information and a proposal channel that communicates trade offers that may be accepted or rejected by the partner. Under cooperative reward structures, the symbolic channel helps agents reveal preference information, enabling more efficient negotiation. Their findings show that this additional communicative structure helps agents converge toward a Nash equilibrium and reduces variability in joint optimality, improving the robustness of the learned bargaining strategies.

Taken together, these studies demonstrate how multi-step dialogs broaden the space of 'what' can be communicated by allowing agents to iteratively refine exchanged information, and they clarify 'whom' each message is directed to at different stages of the interaction protocol.

### 6.3.5 Vocabulary Size and Message Structure

A persistent challenge in emergent communication is determining how much expressive bandwidth agents need to convey task-relevant information. Vocabulary size directly constrains **what agents can communicate**: too small a symbol set limits expressiveness and forces overly compressed encoding strategies, whereas larger vocabularies or sequence-based messages allow richer semantic distinctions and the emergence of structured communication. We group these works together because they examine how expanding or restricting the channel capacity shapes the semantic space of messages, thereby defining the limits of **what** information agents can reliably exchange.

**Simple 1-Bit Messages**   The vocabulary size significantly impacts the effectiveness of communication between agents. Foerster et al. Foerster et al. (2016a) initially experimented with simple **1-bit messages** in scenarios requiring only minimal information exchange, such as solving the switch riddle or determining numerical parity. These settings illustrate how extremely limited vocabularies restrict agents to transmitting single binary distinctions, sufficient only for tasks with low informational complexity.

**Single-symbol Messages with a Larger Vocabulary Size**   As tasks became more complex, researchers investigated whether agents required **larger symbol sets** to convey more nuanced distinctions. Lazaridou et al. Lazaridou et al. (2016) explored single-symbol messages with **vocabulary sizes ranging from 0 to 100**, showing that increasing the cardinality of the message space allowed agents to encode more fine-grained referential information. This line of work demonstrated that vocabulary size acts as a design lever governing what types of attributes or relational distinctions agents can represent.

**NLP-introduced Message Sequences**   Subsequent research inspired by **natural language introduced recurrent architectures** capable of generating symbol *sequences* rather than isolated tokens Havrylov &

Titov (2017); Lazaridou et al. (2018). In these models, the vocabulary remains fixed, but the combinatorial power of sequences vastly expands the potential message space. It is important to note that these symbols initially lack inherent meaning; their association with perceptual or task-specific features emerges entirely through training. One of the key advantages of sequence-based communication is that it enables compositional structure to appear. For example, a speaker may produce a description such as 'red box' by combining learned representations for 'red' and 'box,' even if that specific attribute combination was not seen during training.

Taken together, these works show how vocabulary size and message structure determine the expressive boundary of what agents can communicate. By regulating the symbol budget or enabling combinatorial sequences, researchers shape the granularity, compositionality, and generalizability of emergent languages, thereby addressing the fundamental question of what information the communication channel can support.

### 6.3.6 Message Structure and Compositionality Toward Human-Like Language

A persistent challenge in emergent communication is understanding what structural properties languages acquire when agents learn to communicate from scratch. Earlier work in EL primarily examined whether agents could transmit task-relevant signals at all, emphasizing referential accuracy or cooperative success. Yet the resulting protocols often remained brittle, uninterpretable, and highly specialized to their training environments. These limitations motivated a shift toward investigating whether emergent languages can develop **human-like structural properties** that support generalization, interpretability, and systematic reuse of concepts. This line of research moves beyond the mechanics of **who** communicates with **whom** and **what** information is conveyed, and instead asks **what form the communication system itself ultimately takes** and whether it resembles linguistic systems observed in humans.

Human language exhibits low-level regularities, such as compositional semantics, morphology, and syntax, as well as higher-level pragmatic behaviors shaped by social interaction and cultural transmission. These properties allow humans to generalize to unseen contexts, compose meanings productively, and coordinate effectively with diverse partners. Accordingly, researchers explore whether emergent protocols can develop analogous linguistic structure and identify the environmental, architectural, or learning pressures under which such structure arises. By examining these questions, the field aims not only to obtain deeper theoretical understanding but also to address long-standing shortcomings of earlier EL systems—including lack of compositionality, limited transfer, and the opacity of learned messages—by grounding emergent language design in principles that characterize natural communication systems.

Emergent communication research therefore seeks to understand how and why certain linguistic traits arise, ranging from compositional semantics and syntactic regularities to pragmatic behaviors influenced by social context. The goal is not to recreate human language verbatim, but rather to study how different task settings, agent architectures, and learning dynamics generate pressures that lead to more structured and interpretable emergent protocols. This perspective treats language emergence as a complex adaptive process shaped by interactions among agents rather than explicit engineering of linguistic rules.

Despite the ambition of this research direction, no existing study attempts a full reconstruction of human language. Most works focus on a single linguistic property at a time, which risks overlooking the deep interdependencies among linguistic phenomena. For example, Ren et al. Ren et al. (2020) show that iterated learning—imperfect transmission of language across generations—induces compositionality, consistent with longstanding results from cognitive science Kirby et al. (2007; 2008). However, many studies of compositionality rely on fixed-population settings with no generational transmission, raising questions about whether observed compositional structure truly arises from factors such as representational capacity Resnick et al. (2019) or perceptual features Lazaridou et al. (2018), or whether important cultural pressures are being omitted.

This fragmentation highlights a broader challenge: human-like linguistic structure likely emerges only when multiple pressures are jointly present, e.g., expressivity, learnability, cultural transmission, environmental diversity, and social incentives. Research that isolates only one such factor may therefore produce languages with limited resemblance to natural communication systems, even if they succeed on specific tasks.

Taken together, these studies address the highest-level component of the '**what**' dimension: not merely what information agents choose to convey, but **what structural form** the communication system itself acquires. They collectively demonstrate that human-like linguistic properties emerge only under specific combinations of environmental pressures and learning dynamics, highlighting the need for more integrated and comprehensive approaches to studying emergent language.

### 6.3.7 Metrics for Evaluating What Is Communicated

Evaluating emergent communication requires more than verifying whether agents succeed on a task. Early studies relied heavily on task accuracy (e.g., correct image selection) as the primary indicator of successful communication, but such measures offer limited insight into whether the learned messages are interpretable, structured, or capable of generalizing beyond the training environment. This limitation prompted a shift toward formal evaluation metrics that capture **what properties the emergent messages themselves exhibit**. By moving beyond task reward, these metrics operationalize the '**what**' dimension of the Five-Ws framework and provide a principled basis for diagnosing weaknesses in communication protocols, comparing competing approaches, and assessing whether an emergent language possesses meaningful linguistic structure rather than task-specific heuristics.

**Entropy** Choice accuracy remains a widely used baseline, but metrics such as purity have limited ability to assess interpretability. For example, forcing agents to describe broad categories (e.g., 'dog') instead of fine-grained features (e.g., color or shape) can inflate success rates while obscuring the actual content of communication. This limitation has motivated the development of metrics that assess communication independently of downstream task reward. Prokopenko and Wang Prokopenko & Wang (2003), for instance, analyze public belief entropy, showing that it decreases as communication conventions stabilize. Similarly, Zaiem et al. Zaïem & Bennequin (2019) quantify how much uncertainty a message reduces, following suggestions by Lowe et al. Lowe et al. (2019) to measure informativeness rather than task reward alone.

Beyond information-theoretic metrics, researchers have developed increasingly fine-grained tools to measure structural properties of emergent languages. Whereas concrete properties like vocabulary size are easily quantified, abstract ones, such as **compositionality** and **generalizability**, require careful formalization, and their evaluation has become central to the field.

**Compositionality** Compositionality refers to the principle that complex utterances derive meaning from the combination of simpler components (e.g., 'red car' = car that is red), distinguishing structured communication from holistic signaling. The most common metric is topographic similarity Brighton & Kirby (2006); Lazaridou et al. (2018), which measures correlation between distances in the referent feature space and message space. Lazaridou et al. Lazaridou et al. (2018) apply Spearman's rank correlation ($\rho$) using cosine distances for referents and Levenshtein distances for message strings. More sophisticated metrics include representation similarity analysis Kriegeskorte et al. (2008); Luna et al. (2020), which compares internal agent representations with referent features. Disentanglement-based metrics evaluate whether message components correspond to independent attributes—via positional and bag-of-words disentanglement Chaabouni et al. (2020), context independence Bogin et al. (2018), or conflict count Kuciński et al. (2020). Tree reconstruction error Andreas (2019) examines how well learned messages match a compositional semantic grammar. Meta-analyses by Korbak et al. Korbak et al. (2020) show that many metrics capture only basic forms of compositionality, while Kharitonov and Baroni Kharitonov & Baroni (2020) and Chaabouni et al. Chaabouni et al. (2020) challenge whether such metrics reliably predict generalization to novel objects.

**Generalizability** Generalization measures how well agents communicate in conditions not seen during training. In emergent communication, a common benchmark is whether agents can describe objects composed of novel attribute combinations Korbak et al. (2019); Chaabouni et al. (2020); Denamganaï & Walker (2020a); Kharitonov & Baroni (2020); Resnick et al. (2019); Perkins (2021a). Other forms of generalization include robustness to new communication partners Bullard et al. (2021), adaptation to new environments Guo et al. (2021); Mu & Goodman (2021), and separation of syntax from semantics Baroni (2020). Yao et al. Yao et al. (2022) propose a data-driven metric that compares an emergent language's machine translation performance

to human language, under the intuition that emergent protocols closer to natural language better support transfer and broad applicability.

Overall, evaluation research specifies what properties an emergent communication system must possess to be considered successful. Through metrics that assess informativeness, compositional structure, and generalization, this line of work clarifies the operational meaning of the 'what' dimension and provides rigorous criteria for analyzing and comparing emergent languages.

### 6.3.8 Discussion: Who–Whom–What in Emergent Language

Table 8: Taxonomy of the Who/Whom and What Components in Emergent Communication: Population, Incentives, Architectures, Protocols, Expressive Capacity, Structure, and Evaluation Metrics (Sec. 6.3)

| Category | Sub-Category | Representative Approaches |
|---|---|---|
| **Population Size and Communicative Role Structure (who/whom) (Sec. 6.3.1)** | Single agent | • Human-machine interfaces Buck et al. (2018);
• Model fine-tuning Steinert-Threlkeld et al. (2022). |
| | Dual-agent settings /speaker and listener | • Each with distinct roles, emergent utterances come from a fixed set and the communication symbols are in the K-dimensional one-hot encoding sample Lazaridou et al. (2016); Mordatch & Abbeel (2018); Lowe et al. (2017);
• Larger groups of agents: regularization **?**, language evolution Ren et al. (2020), influencing the emergence process **?**Van Eecke & Beuls (2020); Graesser et al. (2019). |
| **Social Incentive Structures and Their Influence on Communication (who/whom + what) (Sec. 6.3.2)** | Fully cooperative settings | Agents share rewards entirely and lack individual incentives and rely on a shared language to coordinate, and communication does not develop if dominance can be achieved without it Noukhovitch et al. (2021). |
| | Semi-cooperative setups | Both shared and individual rewards, introducing the challenge of balancing collective and personal objectives Lowe et al. (2019); Liang et al. (2020). |
| | Fully competitive settings | Agents compete for rewards without shared objectives, often leads to deceptive language, making meaningful communication unlikely without cooperative elements Noukhovitch et al. (2021). |
| **Architectural Mechanisms for Encoding Communicative Content (what) (Sec. 6.3.3)** | CNN-based Visual Feature Extraction | Lazaridou et al. (2016) used CNNs to process image inputs, yielding more informative features that improved referential task performance. |
| | Recurrent Policies for Temporal Encoding | Lazaridou et al. (2018) introduced RNN-based decoders to maintain temporal memory and enhance the generation of longer communication sequences. |

**Table 8 – continued from previous page**

| Category | Approach | Representative Works |
|---|---|---|
| | Attention-Enhanced Communication | Evtimova et al. (2017) added attention mechanisms to guide symbol generation, improving generalization by focusing on salient features in visual inputs. |
| | Jointly Optimized Sender–Receiver Architectures | These works train speaker and listener networks end-to-end using only final task success as the learning signal, without weight sharing between agents. |
| **Multi-Step Dialog Protocols and Interaction Structure (who/whom + what) (Sec. 6.3.4)** | Identifying target task Jorge et al. (2016) | <ul><li>Task: Identifying target images (e.g., celebrities).</li><li>Agent Roles: Asymmetric — guesser initiates questions, answerer replies.</li><li>Communication: Binary responses ('yes'/'no') across two interaction rounds.</li><li>Architecture: MLPs for perception, 2-layer GRUs for state/message updates, DRU for discretization.</li><li>Key Findings: Later questions refine earlier guesses; multi-step dialog improves target accuracy and dialogue consistency.</li></ul> |
| | Agents bargain over items that hold different utilities for each party Cao et al. (2018) | <ul><li>Task: Negotiation over items with differing utilities.</li><li>Agent Roles: Symmetric — both agents negotiate cooperatively.</li><li>Communication: Two channels — symbolic (linguistic) and action-based (trade proposals).</li><li>Architecture: RL agents with value networks and discrete messaging.</li><li>Key Findings: Linguistic messages clarify preferences; leads to Nash equilibrium and reduces variance in joint rewards.</li></ul> |
| **Expressive Capacity of the Channel: Vocabulary and Message Structure (what) (Sec. 6.3.5)** | Simple 1-bit messages | Foerster et al. Foerster et al. (2016a) initially experimented with simple 1-bit messages in scenarios that required minimal information |
| | Single-symbol messages | Lazaridou et al. Lazaridou et al. (2016) explored the use of single-symbol messages with vocabulary sizes ranging from 0 to 100. |
| | Message sequences | Research Lazaridou et al. (2018); Havrylov & Titov (2017) are inspired by natural language, which proposed the adoption of LSTM-based encoder-decoder architectures to create message sequences. |
| **Linguistic Structure and Human-Like Properties of Emergent Messages (what) (Sec. 6.3.6)** | Imperfect language transmission across generations drives compositionality | Ren et al. Ren et al. (2020) demonstrate that iterated learning—imperfect transmission of language across generations—drives compositionality, as supported by earlier studies Kirby et al. (2007; 2008). |

**Table 8 – continued from previous page**

| Category | Approach | Representative Works |
|---|---|---|
| | Fixed-population settings | Many studies fix population size, overlooking iterated learning as a sufficient driver of compositionality, thereby questioning claims that attribute it to model capacity Resnick et al. (2019) or perception Lazaridou et al. (2018). |
| **Evaluation Metrics for Assessing Communicative Properties (what) (Sec. 6.3.7)** | Public belief entropy | Prokopenko and Wang Prokopenko & Wang (2003) analyzed public belief entropy, which decreases as communication conventions form |
| | Using entropy to quantify how much a message reduces uncertainty | Zaiem et al.Zaïem & Bennequin (2019) used entropy to measure how much a message reduces uncertainty, a metric Lowe et al.Lowe et al. (2019) also suggested for evaluating communication protocols beyond task performance |
| | Compositionality | • *Topographic similarity*: measures the correlation between distances in the referent feature space and message space Brighton & Kirby (2006); Lazaridou et al. (2018);
• *Spearman's rank correlation coefficient ($\rho$)*: with cosine similarity for feature space distances and Levenshtein distance for message space distances: Lazaridou et al. (2018);
• *Representation similarity analysis*: Kriegeskorte et al. (2008); Luna et al. (2020);
• *Disentanglement-based Metrics:* Evaluate whether individual message components represent distinct attributes, including:
  – Positional and bag-of-words disentanglement Chaabouni et al. (2020)
  – Context independence Bogin et al. (2018)
  – Conflict count Kuciński et al. (2020)
• *Tree Reconstruction Error:* Measures how well a compositional semantics model reconstructs the structure of the language produced by agents Andreas (2019).
• *Meta-Analysis of Metrics:* Korbak et al. Korbak et al. (2020) found that existing metrics capture basic compositionality but struggle to detect more complex forms.
• *Limitations of Compositionality Metrics:* Kharitonov and Baroni Kharitonov & Baroni (2020) and Chaabouni et al. Chaabouni et al. (2020) questioned whether current metrics reliably assess generalization to novel objects. |

| Category | Approach | Representative Works |
|---|---|---|
| | Generalizability | • *Unseen Attributes:* Agents generalize to novel object attribute combinations Korbak et al. (2019); Chaabouni et al. (2020); Denamganaï & Walker (2020a); Kharitonov & Baroni (2020); Resnick et al. (2019); Perkins (2021a). |
| | | • *New Partners and Environments:* Agents adapt to unfamiliar partners Bullard et al. (2021) and settings Guo et al. (2021); Mu & Goodman (2021). |
| | | • *Syntax–Semantics Separation:* Generalization across linguistic structure Baroni (2020). |
| | | • *MT-based Human Alignment:* Translation performance used to assess similarity to human language Yao et al. (2022). |

Overall, Table 8 synthesizes the **seven foundational perspectives** introduced in Sec. 6.3, providing an integrated view of how emergent communication systems address the coupled questions of **who communicates with whom** and **what is communicated**. The table mirrors the conceptual structure of the subsection: it begins with (1) population size and role structure, which determine the basic communication topology; followed by (2) cooperation levels, which shape the incentives that govern when and why communication is beneficial; then (3) architectural mechanisms that specify **what** perceptual or latent features enter the channel; (4) multi-step dialog protocols that clarify **whom** each message targets at different interaction stages and refine **what** information is exchanged across rounds; (5) expressive capacity constraints, such as vocabulary size and sequence structure, which bound the richness of **what** can be communicated; (6) human-like linguistic structure, which examines **what form** the overall communication system acquires; and finally (7) evaluation metrics, which operationalize the **what** dimension by defining how to measure informativeness, compositionality, and generalization. Together, these perspectives illustrate that emergent communication does not arise from any single design choice. Rather, it is shaped jointly by social structure, representational capacity, interaction protocols, expressive constraints, and the criteria used to evaluate the resulting language. By situating representative works within this unified taxonomy, Table 8 clarifies how each research thread contributes to resolving a different facet of the broader **who/whom/what** challenge and how these threads collectively advance our understanding of structured communication in multi-agent systems.

## 6.4 When and Where to Communicate: Experimental Environments

In addition to clarifying who communicates with whom and what is communicated (Sec. 6.3), emergent language research must also specify **when** agents are allowed to exchange messages and **where** these interactions are embedded within an environment. In practice, the temporal schedule of interaction (e.g., one-shot vs. multi-round exchanges, episodic vs. continual tasks) and the spatial or task context in which communication is grounded (e.g., images, text, graphs, or navigation worlds) together determine the opportunities agents have to develop and refine a protocol. **We therefore treat 'when' and 'where' as tightly coupled:** the design of an experimental environment simultaneously fixes **when** along an episode messages can be sent and **where** in the observation–action loop communication influences behavior.

From this perspective, standardized experimental environments serve not only to simplify implementation, but also to make explicit the assumptions about when and where communication occurs. Well-designed environments allow researchers to vary the timing of messages, the sequence length of interactions, and the grounding context, while keeping other factors fixed. This supports reproducibility, reduces implementation bugs, and enables more reliable cross-paper comparisons. At the same time, emergent communication is highly sensitive to such design choices, so care is needed to avoid introducing systematic biases through environment defaults. The following subsubsections review three prominent families of environments and games, emphasizing how each family encodes assumptions about the temporal and spatial structure of communication.

### 6.4.1 Standardized Toolkits for Controlling Interaction Timing and Topology

A first challenge is to provide reusable infrastructure in which researchers can flexibly specify at which points in an episode messages are exchanged and which observation spaces they refer to. General-purpose toolkits answer this challenge by abstracting common emergent communication patterns into configurable modules, thereby standardizing both **when** and **where** communication is inserted into the learning loop.

Emergence of Language in Games (EGG) Kharitonov et al. (2019) is the most widely used framework for deep learning-based emergent communication. It provides a straightforward Python interface for common emergent communication games, agent architectures, and metrics, making it easy to define at which time step a sender observes a stimulus, when a message is produced, and when a receiver acts on it. Studies that implement new games with EGG, such as Chaabouni et al. (2019); Dessì et al. (2019); Chaabouni et al. (2020); Auersperger & Pecina (2022); Kharitonov et al. (2020), expand the range of temporal and environmental configurations supported by the toolkit and enrich the library of metrics for probing resulting protocols.

Other tools target specific aspects of emergent communication experiments while still providing controlled settings for when and where language is grounded. TexRel Perkins (2021b) offers a synthetic dataset in which compositional language can describe constructed images, effectively fixing the visual '**where**' while allowing researchers to vary the message schedule. HexaJungle Ikram et al. (2021) introduces a suite of grounded environments for emergent communication, exposing richer spatial structure and longer interaction horizons. For experiments beyond standard platforms, many researchers develop custom implementations of games and agents Evtimova et al. (2017); Mu & Goodman (2021); Noukhovitch et al. (2021) on top of general-purpose deep learning libraries such as PyTorch Paszke et al. (2019), explicitly encoding when messages are sent relative to environment transitions.

Across these efforts, standardized frameworks primarily address the 'when/where' question at the infrastructure level: they provide common hooks for inserting communication into different stages of the perception–action cycle and consistent abstractions for grounding messages in visual, textual, or other modalities. This enables systematic exploration of how varying the timing and context of communication affects the emergence and stability of learned protocols.

### 6.4.2 One-Shot and Iterative Communication

A second challenge is to design interaction games that expose specific temporal structures for communication. Signaling games and their variants address this challenge by defining how many messages can be sent, in what order, and against which backdrop of stimuli. In these games, '**when**' is operationalized through the number of rounds and their sequencing, while '**where**' is determined by the modality of the inputs (e.g., images, text, graphs) that messages are meant to describe.

The classical signaling game generally involves two agents, a sender and a receiver Lazaridou et al. (2016). The receiver must identify a target sample from a set that may include distractors using only the sender's message. The sample set can contain images Lazaridou et al. (2016); Bouchacourt & Baroni (2018); Denamganaï et al. (2023), object feature vectors (texts) Carmeli et al. (2023), or graphs Słowik et al. (2021). Here, communication typically occurs in a single round: the sender observes the target (and sometimes distractors) and then produces one message, fixing both **when** the communication happens (once per episode, before the receiver's choice) and **where** it is grounded (in the representational space of the stimuli). Key design

decisions include whether the sender sees only the correct sample or also distractors, potentially different from those given to the receiver Lazaridou et al. (2016), and how many distractors or near-duplicates the receiver must choose among Kang et al. (2020). Although a referential Gym implementation of the signaling game Denamganaï & Walker (2020b) exists, it has seen limited adoption in practice Evtimova et al. (2017).

A closely related variant is the *reconstruction game*. Instead of selecting from a collection, the receiver must reconstruct a sample from the sender's message. The objective is to reproduce the original sample shown to the sender as accurately as possible Rita et al. (2022); Chaabouni et al. (2020). This setup resembles an autoencoder in which the latent bottleneck mimics or supports language, as implemented in EGG Kharitonov et al. (2019). Compared to referential games, reconstruction games change both when and where communication acts: messages are produced once but evaluated via reconstruction quality rather than discrete choice, and the '**where**' is the continuous feature space of the input rather than a discrete index set Guo (2019).

The question–answer game extends these designs by introducing iterative, bilateral communication over multiple rounds Bouchacourt & Baroni (2019). Unlike original signaling and reconstruction games, this framework allows the receiver to ask follow-up or clarifying questions Das et al. (2017); **?**, so that the timing of messages becomes dynamic: agents alternately decide when to query and when to reply. This setting has sparked interest in the symmetry of emergent language Das et al. (2017), even though it remains less widely adopted. By enabling agents to adapt their messages over several turns, question–answer games provide a richer answer to the '**when**' dimension and allow researchers to study how protocols evolve within an extended dialog.

Overall, signaling games and their variants operationalize 'when to communicate' through the number and ordering of interaction rounds and specify '**where**' communication is grounded via the choice of referents or reconstruction targets. They thereby expose how different temporal schedules—from one-shot to multi-round dialogs—affect the emergence, refinement, and symmetry of learned communication protocols.

### 6.4.3 Grounded and Embodied Environments

A third challenge is to understand communication in temporally extended, spatially structured worlds where agents must act and coordinate over many steps. Grid-world and continuous environments tackle this challenge by embedding communication within navigation, control, or manipulation tasks. In these settings, '**when**' refers to which time steps within an episode agents choose (or are allowed) to communicate, while '**where**' refers to the physical or simulated environment in which their messages are grounded.

Grid-world games simulate simplified 2D environments to model scenarios such as warehouse path planning Blumenkamp & Prorok (2021), object movement Kolb et al. (2019), traffic management Simões et al. (2020); Gupta et al. (2020), or maze navigation Sukhbaatar et al. (2016); Kalinowska et al. (2022). These games offer substantial design flexibility: agents may communicate with teammates while moving through the grid or act as external supervisors that issue instructions. Designers can decide whether messages are sent at every time step, only when certain events occur, or at sparse decision points, thereby explicitly shaping when information flows. The complexity of the environment, the observability of local neighborhoods, and the placement of communication channels together determine where messages are grounded and which spatial relations they must encode. Despite their popularity, grid-world implementations vary greatly, making this a highly diverse category.

Continuous-world games extend these ideas to richer physics and observation spaces, introducing greater complexity into the learning process Wang et al. (2019c); Pesce & Montana (2020b). In emergent language approaches, such environments often support multi-task learning in which agents must both interact with the environment and develop communication policies. Continuous 2D or 3D settings Patel et al. (2021) increase realism: agents must decide when to exchange information while navigating, manipulating objects, or coordinating with others under partial observability and noisy dynamics. This increased temporal and spatial complexity raises the bar for effective 'when/where' design but also enhances the potential for deploying emergent communication in real-world scenarios.

Taken together, grid-world and continuous environments ground emergent communication in embodied interaction. They reveal how the timing of messages over long horizons and the spatial structure of the world

jointly influence what kinds of protocols agents can discover, highlighting that decisions about when and where to communicate are inseparable from the tasks in which communication is embedded.

### 6.4.4 Discussion: When and Where Communication Occurs

| Category | Description | Representative Works |
|---|---|---|
| **Standardized Toolkits for Controlling When and Where Agents Interact (when + where)** (Sec. 6.4.1) | General frameworks and toolkits designed for building, training, and evaluating agents in emergent communication tasks. These platforms standardize interfaces, simplify development, and encourage reproducibility. | • EGG: symbolic and neural agent toolkit Kharitonov et al. (2019); 
• Studies that implement new games with EGG, such as Chaabouni et al. (2019); Dessì et al. (2019); Chaabouni et al. (2020); Auersperger & Pecina (2022); Kharitonov et al. (2020), expand its accessible range of games and metrics; 
• TexRel: compositional image-language benchmark Perkins (2021b); 
• HexaJungle: diverse grounded communication tasks Ikram et al. (2021); 
• Custom codebases using PyTorch Evtimova et al. (2017); Mu & Goodman (2021); Noukhovitch et al. (2021). |
| **One-Shot and Iterative Signaling Games: Scheduling When Messages Are Exchanged (when)** (Sec. 6.4.2) | Sender-receiver interaction games where the goal is either to identify a target, reconstruct a message, or iteratively refine communication. These variants differ in message structure, agent roles, and level of interactivity. | • Signaling Game: sender encodes target, receiver selects Lazaridou et al. (2016); Bouchacourt & Baroni (2018); Denamganaï et al. (2023); Carmeli et al. (2023); Słowik et al. (2021); Kang et al. (2020); 
• Signaling game in referential Gym Denamganaï & Walker (2020b); Evtimova et al. (2017) 
• Reconstruction Game: receiver reconstructs input Rita et al. (2022); Chaabouni et al. (2020); 
• Question–Answer Game: multi-round clarification Bouchacourt & Baroni (2019); Das et al. (2017); **?**. |
| **Temporally Extended Embodied Environments: Where Communication Is Grounded in Space and Time (where + when)** (Sec. 6.4.3) | 2D spatial environments for grounded interaction, object manipulation, or path planning. These games range from discrete to continuous settings, often used to study embodiment and communication under navigation or cooperation tasks. | • Pathfinding and object movement Kolb et al. (2019); Sukhbaatar et al. (2016); 
• Traffic coordination and network control Simões et al. (2020); Gupta et al. (2020); 
• Complex planning in mazes Blumenkamp & Prorok (2021); 
• Continuous 2D/3D worlds for realism Wang et al. (2019c); Pesce & Montana (2020b); Patel et al. (2021). |

Table 9: Summary of When and Where (Experimental Environment Settings) for Emergent Communication (Sec. 6.4)

Table 9 synthesizes the three major perspectives introduced in Sec. 6.4, each addressing complementary aspects of **when** agents exchange messages and **where** communication is grounded. Together, these perspectives highlight that experimental environments are not merely platforms for implementing emergent

communication systems; they actively *shape* the temporal structure of communication, the spatial grounding of observations, and the coordination pressures that determine whether communication emerges at all.

The first category, **standardized toolkits**, provides researchers with environments that make the notion of '**when**' explicit through configurable interaction loops and message schedules, while also specifying '**where**' communication is anchored: whether in symbolic referential settings, image-based benchmarks, or custom grounded tasks. These frameworks lower barriers to experimentation and ensure reproducibility, but they also impose default assumptions about temporal sequencing and perceptual context that influence the emergence of language.

The second category, **signaling games and their variants**, directly operationalizes the **when** dimension through carefully designed communication protocols. One-shot signaling games define a single, fixed moment for communication, whereas multi-round question–answer variants introduce iterative temporal structure, enabling agents to refine and negotiate meaning over time. These settings specify '**where**' communication takes place through symbolic inputs, images, graphs, or feature vectors, but their defining characteristic is their explicit control over **communication timing**, which allows researchers to probe how emergent languages depend on interaction order and available rounds of exchange.

The third category, **spatial and continuous environments**, integrates communication into embodied tasks, making '**where**' a central component of the learning problem. Here, agents operate in environments where observations, goals, and opportunities for communication unfold over time and space. These settings naturally expose more complex **when**-to-communicate decisions: agents must determine not only *what* information is useful, but *when* during a trajectory it becomes relevant and *where* within the environment it must be grounded. As task complexity increases—from discrete grid-worlds to rich continuous domains, so do the temporal and spatial dependencies that drive the emergence of structured communication.

Viewed together, the three categories illustrate that '**when**' and '**where**' cannot be treated independently. Temporal scheduling constrains which observations are available and thus where communication must be grounded; conversely, the spatial structure of the environment determines when communication becomes necessary for coordination. By organizing representative environments into this unified taxonomy, Table 9 clarifies how experimental design choices implicitly prescribe distinct pressures on emergent communication and, in doing so, shape the languages learned by multi-agent systems.

## 6.5 Why and How to Communicate: Motivations and Mechanisms

Having examined who communicates with whom and what is communicated in Sec. 6.3, as well as experimental settings in Sec. 6.4, the next foundational perspective concerns **why communication is needed** in multi-agent systems and **how communication should be implemented** to serve those needs. These two dimensions are tightly connected: the reasons agents must communicate directly influence the design of the communication mechanisms, while the available mechanisms constrain what motivations can be satisfied. Treating these dimensions jointly clarifies how communication arises not as an optional feature but as a task-driven, efficiency-driven, and robustness-driven behavior.

Traditional multi-agent systems rely either on handcrafted communication protocols or on learned latent-vector protocols Boldt & Mortensen (2024). Handcrafted protocols specify rigid rules for how to communicate but require significant expert effort and do not scale to open-ended, dynamic environments. Automatically learned continuous communication relaxes these constraints but often produces uninterpretable, brittle representations.

Emergent communication provides an alternative: agents learn **why** to communicate based on task incentives and **how** to communicate based on the structural pressures in their environments. This subsection examines three perspectives that together explain why communication is necessary and how effective communication protocols can be constructed: (1) communication for scalability to general-purpose tasks, (2) communication for interpretability and structured meaning, and (3) communication for robustness and generalization.

### 6.5.1 Motivations for Emergent Communication

One fundamental reason agents communicate is to enable coordination in environments too complex for isolated reasoning. Communication becomes necessary when tasks exhibit partial observability, dynamic uncertainty, or heterogeneous roles. In such cases, agents must exchange information to align beliefs, negotiate goals, or distribute sub-tasks. Thus, the **why** rests on the inadequacy of individual observation and planning.

To address these challenges, emergent communication research investigates **how** communication protocols should be designed to scale across diverse situations. Open-world or open-ended environments such as Minecraft, Starcraft, or Dota 2 contain rich contextual variation, forcing agents to learn communication strategies that adapt flexibly across different states and task phases. Hierarchical or structured communication protocols in referential and navigation tasks Mul et al. (2019); Li et al. (2022); Nisioti et al. (2023) illustrate early attempts to develop scalable mechanisms.

In signaling-based environments Bullard et al. (2021); Cope & Schoots (2021); Wang et al. (2024e); Tucker et al. (2021), adaptability arises from enforcing communication across diverse partners or contexts. Techniques such as symbol remapping or channel randomization reveal how communication mechanisms can be structured to avoid overfitting to a single partner or environment.

Thus, communication improves scalability (why) by providing a flexible means of exchanging situationally relevant information, and scalable mechanisms (how) must support general-purpose, context-dependent adaptation.

### 6.5.2 Interpretability and Structured Protocols

A second motivation for communication is the need for **interpretable coordination**. When agents must collaborate, monitor each other, or provide explanations to humans, messages must carry structured, meaningful content. Communication becomes necessary not only to exchange information but also to make that information understandable and verifiable.

The mechanisms for achieving interpretability revolve around **how** to constrain communication. Discrete symbol vocabularies, bounded message lengths, and structured composition promote clarity and transparency, reducing the opacity associated with continuous-vector protocols. These mechanisms enable researchers to detect when a message encodes properties such as attributes, preferences, or task intentions.

Empirical studies illustrate the interpretability benefits of symbolic communication: Jorge et al. Jorge et al. (2016) analyzed agent messages to identify semantic alignment with features such as hair color. Choi et al. Choi et al. (2018) demonstrated that dynamic environments challenge stable meaning formation, highlighting the need for structured protocols. Purity analyses Lazaridou et al. (2016) show that symbolic messages can sometimes align with human-like categorical distinctions.

In summary, interpretability motivates communication (why) by promoting transparent coordination, and structured symbolic protocols provide mechanisms (how) for agents to produce messages that are analyzable, compositional, and meaningful.

### 6.5.3 Robustness and Generalization

A third motivation for communication arises from the need for **robust and generalizable coordination**. Real-world multi-agent systems encounter uncertainty, noise, new partners, and shifting environments. Agents must communicate in ways that remain reliable under such variability. Communication is valuable because it allows agents to share stabilizing information, correct misperceptions, and maintain coordination even in novel conditions.

To support robustness, communication mechanisms must prevent overfitting and encourage generalizable message structures. Techniques explored in the literature include: Channel noise injection, symbol permutations, or message scrambling Cope & Schoots (2021), which force agents to rely on stable relational structure rather than arbitrary encodings; Population cycling and bottleneck architectures Wang et al. (2024e), which prevent coordination from becoming agent-specific; Protocols that enable zero-shot communication with unseen partners Bullard et al. (2021).

Thus, robustness explains why communication is essential in unpredictable environments, and mechanisms such as noise tolerance, population diversity, and structured bottlenecks explain how systems can maintain communicative effectiveness across contexts.

| Challenge Category | Approach | Representative Works |
|---|---|---|
| **Why Communication Improves Scalability and How Protocols Adapt (Sec. 6.5.1) (why + how)** | Use of open-world, task-rich environments to elicit adaptive and scalable communication strategies | • Hierarchical protocol learning in referential games Mul et al. (2019)
• Grounding communication via autoencoders in navigation Li et al. (2022)
• Autotelic reinforcement learning for adaptive goal generation Nisioti et al. (2023)
• Dual-channel communication in negotiation settings Tucker et al. (2021) |
| | Techniques to enforce protocol generality across diverse partners or contexts | • Symbol remapping, channel randomization Cope & Schoots (2021)
• Quasi-equivalence training across agent populations Bullard et al. (2021) |
| **Why Interpretability Matters and How Structured Protocols Support Meaning (Sec. 6.5.2) (why + how)** | Design protocols with discrete symbols and bounded vocabulary size to mimic human language | • Grounding symbols in interpretable features (e.g., hair color) Jorge et al. (2016)
• Symbol purity and linguistic structure analysis Lazaridou et al. (2016)
• Zero-shot compositional language assessment Choi et al. (2018) |
| **Why Robustness Is Essential and How Communication Mechanisms Promote Generalization (Sec. 6.5.3) (why + how)** | Introduce agent cycling, bottlenecks, and symbol-level constraints to avoid co-adaptation and foster generalization | • Population cycling and bottleneck architectures Wang et al. (2024e)
• Zero-shot communication with unseen agents Bullard et al. (2021)
• Channel noise injection and message scrambling Cope & Schoots (2021) |

Table 10: Summary of Why Communication Is Needed and How It Is Achieved in Emergent Multi-Agent Systems (Sec. 6.5)

### 6.5.4 Discussion: Why and How Communication Emerges

Overall, Table 10 synthesizes the three foundational perspectives developed in Sec. 6.5, each addressing a complementary aspect of why multi-agent systems require communication and how communication mechanisms can be designed to meet these requirements. The subsection begins with (1) the scalability perspective, which explains why communication becomes essential in open-ended or task-diverse settings and highlights how agents develop adaptive protocols that generalize across partners, tasks, and environments. It then examines (2) the interpretability perspective, which clarifies why human-like structure is desirable for transparency and verification, and how discrete, compositional protocols support meaningful communication. Finally, (3) the robustness perspective addresses why communication must withstand perturbations, agent turnover, and environmental noise, and how design interventions such as population cycling, bottleneck structures, or noise injection promote generalization rather than co-adaptation.

Together, these perspectives demonstrate that emergent multi-agent communication is shaped by the alignment between communicative motivation (why agents need to exchange information) and communicative mechanism (how this exchange is implemented). Scalability motivates flexible protocol formation; interpretability motivates symbolic and structured message spaces; and robustness motivates architectures and training regimes that remain effective beyond the training distribution. By organizing the literature according to these why and how dimensions, Sec. 6.5 highlights how different research threads contribute to constructing multi-agent systems capable of reliable, interpretable, and generalizable communication.

## 6.6 Discussions

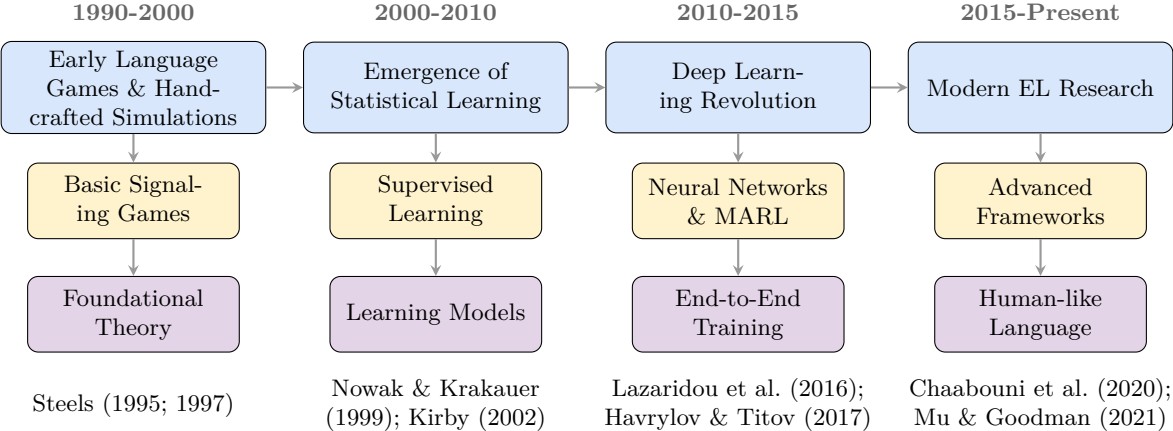

Figure 9: Evolution of Emergent Language Research: Timeline showing major milestones, approaches, and achievements

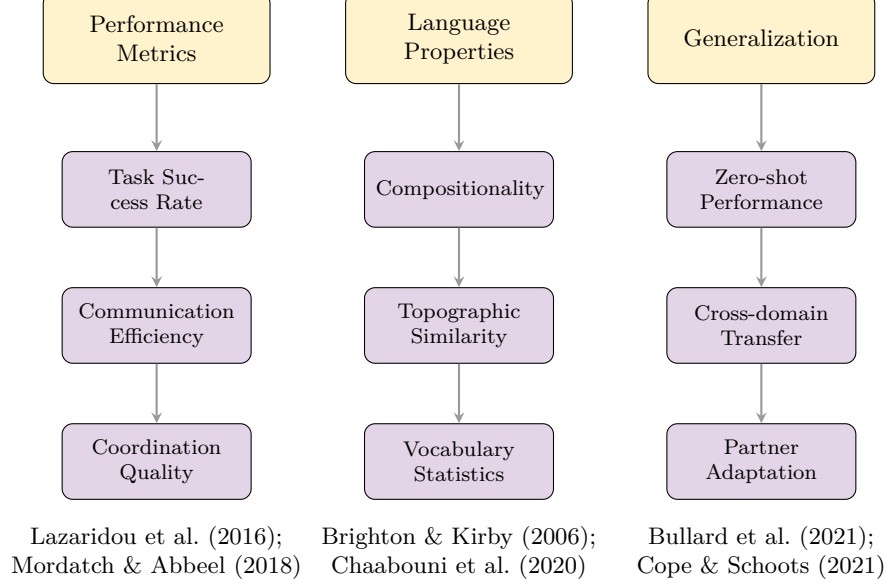

Figure 10: Taxonomy of Evaluation Metrics in Emergent Language Research

### 6.6.1 Who Communicates with Whom and What Is Shared

The first set of considerations in emergent communication concerns who participates in communication, whom messages are directed toward, and what information agents are capable of exchanging. As detailed in Sec.,6.3, population size, role structure, cooperation level, and message design jointly determine the topology and informational bandwidth of the communication process. These assumptions specify the communicative space in which protocols can emerge.

Fig.,9 illustrates how research has gradually expanded these foundational assumptions. Early signaling-game studies focused primarily on dyadic, role-specific communication and symbolic vocabularies. Subsequent neural approaches introduced richer architectural mechanisms (CNNs, RNNs, attention) that broadened what can be communicated. Multi-step protocols and expressive vocabulary choices further shaped whom messages target and how information is structured across a conversation.

Evaluation practices (Fig.,10) matured alongside these developments. Metrics such as compositionality, topographic similarity, and vocabulary statistics emerged as tools for quantifying the structure and semantic content of messages, offering insights into what agents communicate and whether it resembles human-like abstractions. These foundational who–whom–what dimensions form the basis on which more dynamic aspects of communication are built.

### 6.6.2 When and Where Agents Communicate

The second dimension concerns the timing and situational context of communication, addressing when communication is useful and where in an environment or task structure communicative bottlenecks emerge. Sec.,6.4 shows that emergent language is highly sensitive to the structure of the experimental setting. Communication does not arise uniformly; it appears in moments where coordinated behavior depends on information that cannot be inferred from local observations or passive cues.

Referential, reconstruction, question–answer, grid-world, and continuous environments each create different temporal and spatial opportunities for communication. Multi-round dialog games enable clarification over time, while continuous control tasks require communication during navigation or manipulation. Grid-world tasks depend on spatial grounding, constraining where information must be shared for success. These design choices influence not only the availability of communication but also its necessity and strategic value.

The historical development shown in Fig.,9 reflects this progression from static symbolic settings toward dynamic, embodied environments where timing, location, and partial observability determine when communication becomes advantageous. Effective evaluation therefore requires metrics that capture performance across varied environmental contexts and task structures, as organized in Fig.,10.

### 6.6.3 Why Communication Emerges and How It Is Learned

The third dimension captures the functional drivers behind emergent communication: why communication is beneficial in a task and how agents implement communicative strategies that lead to improved coordination, generalization, or robustness, as discussed in Sec.,6.5. Communication emerges when independent decision-making is insufficient to solve open-ended or high-uncertainty tasks, when agents face partner variability or noise, or when concepts must be abstracted and shared to achieve stable coordination.

The how dimension encompasses architectural and algorithmic mechanisms that support communication. These include discrete vocabularies, message bottlenecks, attention modules, agent cycling, and multi-round negotiation schemes. Such mechanisms influence the emergence of interpretability, enabling messages to develop structure similar to natural language. They also contribute to scalability and robustness; for example, population cycling or noise injection prevents co-adaptation and strengthens generalization across agents and environments.

As summarized in Fig.,10, evaluation metrics for these functional properties have expanded beyond task success to include compositionality, partner adaptation, zero-shot generalization, and cross-domain transfer. These metrics reflect the need to assess not only whether agents communicate but also whether the resulting protocols provide benefits that continuous or handcrafted systems cannot. Identifying benchmarks where emergent communication clearly outperforms such baselines remains an important open direction.

### 6.7 Bridge: From MARL-Comm and Emergent Language to LLM-Based Communication

While emergent language research addressed key limitations of MARL-based communication—most notably interpretability and discrete structure—it introduced new challenges related to scalability, generalization, and semantic richness. Emergent communication protocols are typically learned from scratch through task-driven interaction, which makes them sensitive to training environments, agent populations, and reward structures. As a result, many emergent languages struggle to generalize across tasks, adapt to unseen partners, or support open-ended reasoning beyond narrowly defined coordination problems.

At the same time, MARL-based communication continues to face challenges in interpretability, verification, and integration with human-centric decision-making pipelines. Continuous or latent message spaces remain difficult to audit or align with human semantics, limiting their applicability in safety-critical or human-

Table 11: From MARL-Comm and emergent language to LLM-based communication: limitations that motivate LLM-Comm.

| Aspect | MARL-Comm Limitations | Emergent Language Limitations | Motivation for LLM-Comm |
|---|---|---|---|
| **Semantic structure** | • Latent or continuous signals;
• Semantics hard to interpret. | • Discrete symbols;
• Limited expressiveness. | • Rich, compositional language;
• Human-aligned semantics. |
| **Generalization** | • Jointly trained agents;
• Poor transfer across tasks. | • Population-specific languages;
• Zero-shot coordination failures. | • Strong cross-task generalization;
• Pretrained linguistic priors. |
| **Scalability** | • Communication cost grows with agents. | • Learning language from scratch is costly. | • Reusable language representations;
• Scales across agent roles. |
| **Human alignment** | • Hard to interpret or audit. | • Limited grounding in natural language. | • Natural interface for humans;
• Supports explanation and oversight. |
| **Core gap** | • Task-optimized but opaque (MARL);
• Interpretable but brittle and narrow (EL). | | • Open-ended, reusable communication substrate. |

in-the-loop settings. Together, these limitations reveal a gap between task-optimized communication and semantically rich, reusable language.

LLM-based multi-agent communication emerges as a response to this gap. By leveraging large-scale pre-training on natural language, LLMs provide agents with strong semantic priors, compositional structure, and the ability to generalize across tasks and domains without task-specific retraining. Unlike emergent language systems, LLM-based agents do not need to rediscover linguistic structure through interaction, and unlike MARL-based communication, their messages are inherently interpretable and human-aligned. These properties motivate the growing adoption of LLMs as communication backbones for coordination, planning, and reasoning in multi-agent systems. Table 11 summarizes how limitations in MARL-Comm and EL-Comm jointly motivate the shift toward LLM-based multi-agent communication.

# 7 LLM-Powered Multi-Agent Communication

Large language models (LLMs) demonstrate strong performance across domains due to training on large, diverse corpora, which yields emergent commonsense reasoning and robust human language understanding and generation capabilities Liang et al. (2022). Because human language is a general and flexible medium for representing knowledge and intent, it provides a natural interface for reasoning, coordination, and information exchange. Building on these capabilities, LLMs are increasingly deployed as autonomous agents that plan, reason, and act, and more recently as teams of agents in multi-agent systems (MAS) that communicate via natural language. Language-based communication enables agents to coordinate and collaborate in an interpretable, task-agnostic manner without hand-crafted protocols, supporting flexible interaction patterns and broad generalization Guo et al. (2024a); Liu et al. (2025b). Despite challenges such as hallucination and limited grounding Ahn et al. (2022), multimodal LLMs further enhance robustness by integrating language with sensory inputs Durante et al. (2024). Compared to traditional RL agents with fixed protocols, LLM-based agents offer a general communication framework, and hybrid LLM–RL approaches seek to combine

linguistic reasoning with environment-grounded learning Sun et al. (2024a); Wu et al. (2022); Feng et al. (2024).

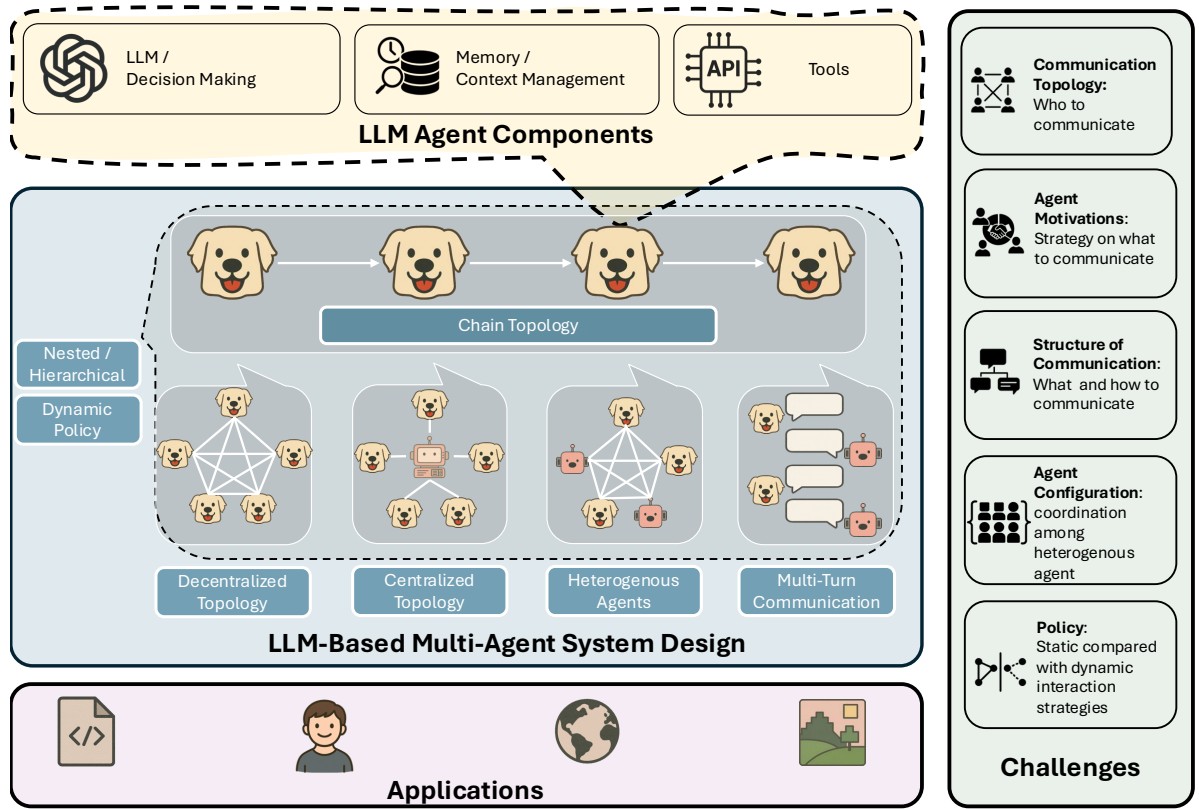

Figure 11: LLM-Based Multi-Agent System Design and Challenges. In an LLM-based multi-agent system, each agent outputs decisions that can be enhanced through memory, context management, and the use of domain-specific tools. Agents communicate through different topologies and can operate in either static or dynamic configurations, with applications spanning diverse domains. Building functional multi-agent systems remains challenging due to numerous design considerations, and several key challenges are summarized.

**Review Scope and Structure** As interest in LLM-powered multi-agent systems grows, communication has become a central organizing theme. Natural language provides a unified interface for agent–environment interaction, agent–agent coordination, and human–agent collaboration. This section reviews LLM-powered multi-agent communication by first introducing the foundations of LLMs and LLM-based agents (Section 7.1), then systematically organizing prior work around who communicates with whom and what is shared (Section 7.2), when and where communication occurs (Section 7.3), and why and how communication is structured (Section 7.4). We further discuss these dimensions in Section 7.5 and conclude by bridging LLM-based communication with multi-agent reinforcement learning, highlighting emergent communication patterns in LLM–MARL hybrid systems and their limitations in decentralized settings (Section 7.6). Fig. 11 provides a high-level system view, illustrating how LLM agent components, communication topologies, agent configurations, and policies jointly shape multi-agent system design and challenges, and Fig. 12 presents a detailed taxonomy that categorizes existing work across communication topology, motivational setting, agent heterogeneity, and communication structure. We conclude by bridging LLM-based communication with multi-agent reinforcement learning, highlighting emergent patterns in LLM–MARL hybrid systems and their limitations in decentralized settings.

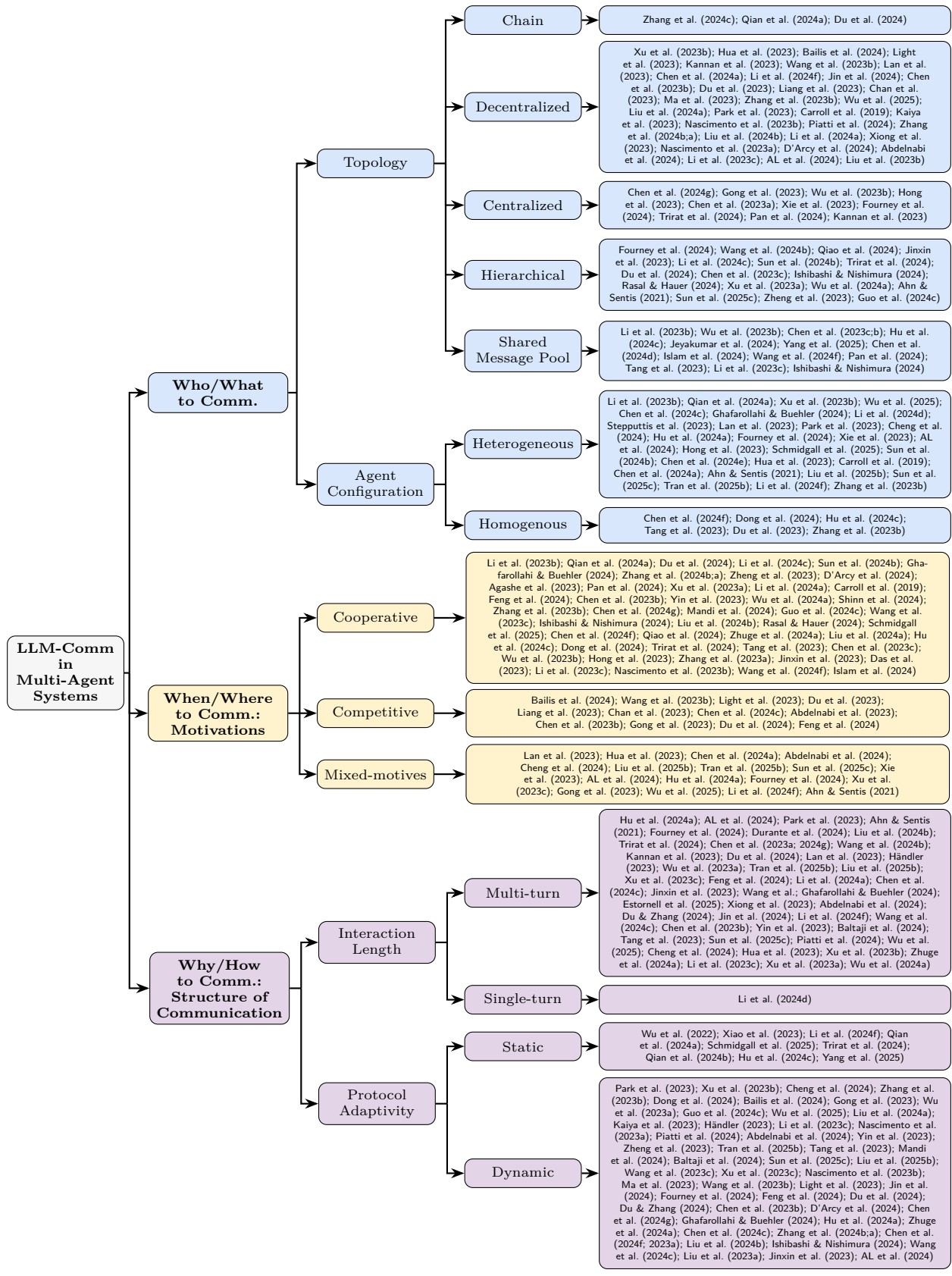

Figure 12: LLM Agent Communication Taxonomy

### 7.1 Background and Foundations of LLMs and LLM Agents

### 7.1.1 Large Language Models (LLMs)

LLMs have emerged as transformative tools for agent-based systems, offering advanced capabilities such as reasoning, planning, and natural language understanding. These capabilities arise not from explicit programming but from the emergent properties of training on large-scale, diverse corpora Liang et al. (2022). When deployed as decision-making agents, LLMs demonstrate robust adaptability in dynamic, multi-agent environments by enabling flexible coordination and communication beyond rule-based constraints: (1). **Emergent Behaviors.** LLMs display emergent abilities—such as strategic planning, abstract reasoning, and social simulation—that were not explicitly encoded during training. These behaviors empower agents to engage in high-level tasks including negotiation, cooperative problem-solving, and context-aware decision-making. In multi-agent systems, such behaviors are particularly valuable for fostering robust coordination strategies in unpredictable settings. (2). **In-Context, Few-shot, and Zero-shot Learning.** LLMs are capable of adapting to new tasks via in-context learning, interpreting task instructions and examples directly from the prompt without parameter updates. This flexibility extends to few-shot and zero-shot settings, where agents can perform novel tasks using minimal or no prior training data. Such capabilities significantly reduce development overhead and enable agents to operate in open-ended, evolving environments with task-agnostic generalization. (3). **Natural Language Communication.** The ability to understand and generate human language makes LLMs ideal for facilitating interpretable communication within multi-agent systems and between agents and humans. Agents can articulate goals, strategies, and rationales using natural language, improving both inter-agent alignment and human oversight. This is critical for collaboration, transparency, and trust in mixed-agent teams. (4). **General-Purpose Reasoning Across Environments.** Importantly, LLMs are not pre-trained to optimize behavior for any specific environment (e.g., a particular game or graphical interface). Instead, they serve as general-purpose cognitive engines that can be fine-tuned or scaffolded to suit domain-specific requirements. This property supports reusability across diverse agent applications—from GUI-based assistants to strategic game agents—while maintaining task flexibility. In summary, the emergent reasoning, flexible learning paradigms, and language-driven communication of LLMs establish them as foundational components for next-generation multi-agent systems. By leveraging these attributes, agents can dynamically adapt to new objectives, collaborate through interpretable dialogue, and function effectively in complex, uncertain environments.

### 7.1.2 LLM Agent

An LLM-based agent is a modular system that integrates reasoning, memory, and task execution around a central language model Durante et al. (2024); Wang et al. (2024a). As illustrated in Fig. 13, these agents are structured into four core components, each contributing to their ability to act coherently and adaptively within dynamic environments: (1). *Profile.* This component encodes the agent's identity, personality traits, and social attributes. Profiles influence interaction style and decision preferences, allowing agents to simulate diverse roles or personas. Profiles can be handcrafted, learned, or aligned with external datasets. (2). *Memory.* Memory systems enable agents to persist and retrieve historical information, such as past interactions or task outcomes. Implemented as embeddings, text logs, or databases, this component supports long-term coherence and task continuity through read/write operations and reflective reasoning. (3). *Planning.* The planning module supports decomposition of goals into actionable steps and the formation of strategies to reach those goals. Plans may be iteratively refined based on agent self-evaluation, feedback from other agents, or environmental input, enabling responsiveness to dynamic task contexts. (4). *Action.* This component operationalizes plans into concrete actions. Actions may involve environmental manipulation, communication, tool usage, or internal reasoning. By integrating outputs from memory and planning, this module allows the agent to execute tasks effectively and adapt behavior in real time. Together, these components form a cohesive architecture for LLM-based agents capable of multi-turn reasoning, human interaction, and decentralized coordination. This modular design facilitates extensibility and customization for a broad range of multi-agent settings.

### 7.2 Who, Whom, and What to Communicate

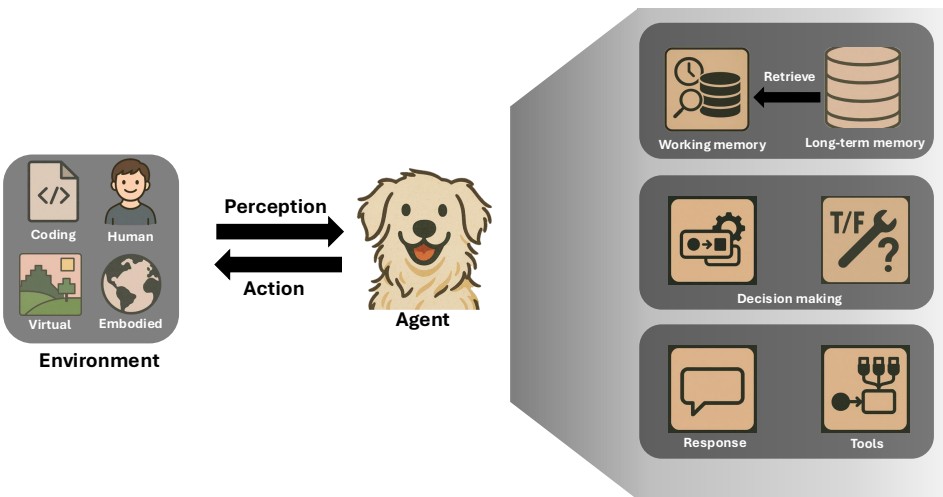

Figure 13: General Architecture of an LLM-Based Agent. The figure depicts how an agent interacts with coding, human, virtual, and embodied environments through perception and action loops. Within the agent, customized modules can be employed to handle memory, decision making, response generation, and tool use, enabling flexible and adaptable performance across domains.

Communication in multi-LLM agent systems introduces challenges that differ substantially from those in traditional MARL or emergent language settings. Unlike gradient-trained policies, LLM agents **communicate through open-ended natural language**, causing issues such as ambiguity, verbosity, hallucination, uncontrolled information flow, and difficulty coordinating roles among heterogeneous agents. These problems often arise because the communication structure is under-specified: it is unclear **who** has the authority or opportunity to speak, **whom** each agent should address or respond to, and **what** information ought to be exchanged for effective coordination. By explicitly analyzing **who**, **whom**, and **what** in communication design, we can systematically constrain interaction patterns, reduce uncertainty, and promote coherent multi-agent behavior. Concretely, these dimensions help address several core **challenges**: **(1)** uncontrolled or redundant message generation, **(2)** misalignment between speakers and intended recipients, **(3)** information overload or omission due to unclear message content, **(4)** degraded coordination in large teams with overlapping responsibilities, and **(5)** failure to maintain shared context or stable interaction protocols. The following subsections examine how **communication topology**, **agent-role configurations**, **message structure** and **communication framework** each operationalize the **who-whom–what** axes, and how these design choices help mitigate the fundamental challenges outlined above.

### 7.2.1 Communication Topology and Interaction Structure

Communication topology is the primary structural mechanism through which LLM-based multi-agent systems define **who** is allowed or expected to speak, **whom** they address, and consequently **what** information is most relevant for exchange. Unlike MARL, where communication edges often reflect spatial adjacency or bandwidth constraints, and unlike emergent language settings where agent roles are fixed by the game structure, LLM-MAS topologies are logical and design-driven. They impose organizational scaffolding that prevents free-form natural language communication from becoming chaotic, redundant, or misaligned with task demands. Below, we synthesize five major topology classes (Figures 14–18) and analyze how each governs **who→whom** interaction and influences **what** gets communicated.

**Chain and Pipeline Topologies** (Fig. 14) describes a linear communication pattern where each agent communicates exclusively with its immediate neighbors. This structure is particularly suited for pipelines or sequential workflows, ensuring messages flow in an organized manner. In sequential communication, agents operate in a step-by-step chain, where each one handles a part of the task and passes the result

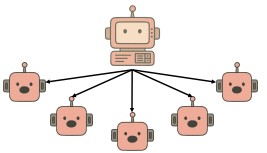

Figure 14: Chain

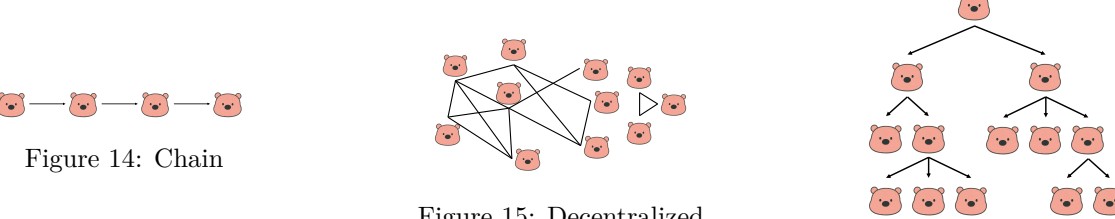

Figure 15: Decentralized

Figure 16: Hierarchical

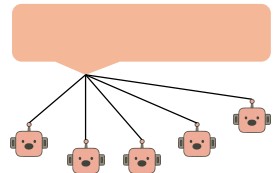

Figure 17: Centralized

Figure 18: Shared Message Pool

to the next Zhang et al. (2024c). In this fixed pipeline, the **who** an agent may listen to is its immediate predecessor, and the **whom** it may speak to is its immediate successor. This rigid pairing shapes **what** is communicated: agents must pass only the intermediate artifact or reasoning needed by the next stage, and cannot rely on global context or cross-checking from non-adjacent agents. This setup breaks down long or complex problems into smaller parts that are easier to process. Some systems follow this pattern for multi-step workflows such as software development, with different agents handling design, coding, testing, and review in a clear sequence Qian et al. (2024a); Du et al. (2024). In these systems, the chain explicitly resolves the **who talks to whom** question by binding each stage to a single upstream provider and a single downstream consumer, which reduces communication redundancy but also restricts **what** information can be transmitted, typically structured specifications, code drafts, or test results formatted for the next specialized agent. Compared with decentralized or shared-memory approaches, these chain-based workflows improve modularity and control but limit opportunities for global coordination or corrective feedback, making them more susceptible to cascading errors when early-stage outputs are flawed.

**Decentralized topology**   (Fig. 15) Decentralized topologies permit agents to communicate directly with multiple peers without centralized coordination, forming fully or partially connected interaction graphs. Unlike chain or hierarchical designs where **who** talks to **whom** is largely fixed by execution order or role assignment, decentralized systems allow communication partners to be selected dynamically based on local decisions, strategic incentives, or environmental context. This flexibility enables richer and more adaptive interaction patterns, but it also shifts the burden of coordination, consistency, and filtering onto the agents themselves.

A first class of decentralized systems arises in **social and strategic game-based environments**, such as Werewolf Bailis et al. (2024), Avalon Wang et al. (2023b), and related role-playing settings Xu et al. (2023b); Hua et al. (2023); Light et al. (2023); Kannan et al. (2023); Lan et al. (2023); Chen et al. (2024a). In these settings, the **who talks to whom** pattern is shaped jointly by explicit game mechanics, such as turn-taking, voting rounds, and phase transitions, as formalized in Avalon- and Werewolf-style benchmarks Light et al. (2023); Bailis et al. (2024); Wang et al. (2023b), and by implicit social roles or personas (e.g., villagers, adversaries, or agents with special abilities) that condition agents' communicative incentives Xu et al. (2023b); Hua et al. (2023); Lan et al. (2023); Chen et al. (2024a). Within these settings, agents actively choose **whom** to address when accusing, defending, or probing others, a decision shown to be central to strategic influence and belief shaping Xu et al. (2023b); Kannan et al. (2023). This partner selection directly determines **what** is communicated: agents generate persuasive arguments, deceptive narratives, and explicit belief statements to sway group decisions or obscure private information Hua et al. (2023); Bailis et al. (2024); Chen et al. (2024a). Across these works, a consistent trend emerges: decentralized interaction enables intention-driven and socially expressive communication, but it also introduces higher ambiguity and vulnerability to misinfor-

mation, as conflicting narratives persist without a central mechanism for reconciliation or verification Light et al. (2023); Wang et al. (2023b); Lan et al. (2023).

A second category consists of **debate-style and critique-based frameworks** Li et al. (2024f); Jin et al. (2024); Chen et al. (2023b); Du et al. (2023); Liang et al. (2023); Chan et al. (2023), which typically adopt fully connected communication graphs in which any agent may address any other. In systems such as LLM-Debate and related variants Du et al. (2023); Liang et al. (2023), **who** speaks to **whom** is governed by explicit debate protocols (e.g., challenger versus defender), while judge-based or reconciliation-oriented designs further structure interactions through evaluator roles Li et al. (2024f); Chen et al. (2023b). In these settings, **what** is communicated is explicitly argumentative, including critiques, counterexamples, evidential claims, and meta-reasoning steps Jin et al. (2024); Chan et al. (2023). Empirical evaluations consistently report improvements in reasoning quality and error detection relative to single-agent baselines, demonstrating the benefit of decentralized peer critique for complex decision-making Du et al. (2023); Liang et al. (2023); Chan et al. (2023). At the same time, these works expose a key limitation of decentralized debate designs: without carefully designed turn-taking rules, aggregation mechanisms, or stopping criteria, communication can become repetitive or diffuse Li et al. (2024f); Chen et al. (2023b), and it becomes difficult to attribute performance gains to specific **who**–**whom** interactions or individual argumentative contributions.

In contrast, **embodied and task-grounded decentralized systems** Ma et al. (2023); Zhang et al. (2023b); Wu et al. (2025); Kannan et al. (2023); Liu et al. (2024a) impose implicit structure on **who** talks to **whom** through physical proximity, complementary roles, or local task dependencies. For example, agents coordinate primarily with spatially nearby peers or role-complementary partners in embodied planning and navigation tasks Ma et al. (2023); Zhang et al. (2023b), while multi-robot and tool-using systems often restrict communication links based on functional dependencies such as navigation–manipulation or perception–action coupling Kannan et al. (2023); Liu et al. (2024a). This partner-selection mechanism induces **sparse but functionally meaningful communication graphs**, reducing redundancy while preserving task relevance. As a consequence, **what** is communicated in these systems is tightly grounded in the environment and task dynamics, typically consisting of spatial descriptions, action proposals, affordance information, or local state updates needed for immediate coordination Ma et al. (2023); Wu et al. (2025). Compared with social-deduction or debate-based decentralized settings, these systems deliberately trade expressive richness and open-ended dialogue for clearer grounding and tighter coupling between messages and environment dynamics Zhang et al. (2023b); Liu et al. (2024a). Empirically, this design yields more predictable and stable behavior, but evaluations are often limited to smaller numbers of agents and narrower task distributions, reflecting a scalability–grounding trade-off inherent to task-grounded decentralized communication.

Finally, **large-scale agent societies and general-purpose decentralized frameworks** Park et al. (2023); Carroll et al. (2019); Kaiya et al. (2023); Nascimento et al. (2023b); Piatti et al. (2024); Zhang et al. (2024b;a); Liu et al. (2024b); Li et al. (2024a); Xiong et al. (2023); Nascimento et al. (2023a); D'Arcy et al. (2024); Abdelnabi et al. (2024); Li et al. (2023c); AL et al. (2024); Liu et al. (2023b) examine communication at population scale, where **who** talks to **whom** is no longer fixed by task structure but emerges from learned utilities, social ties, or cultural and role-based signals. For instance, agent-based social simulations and generative societies allow agents to initiate communication based on perceived social relationships, memory, or long-term preferences Park et al. (2023); Nascimento et al. (2023b;a). Other systems explicitly model interaction incentives or utility-driven partner selection, leading agents to contact peers that maximize expected payoff or coordination benefit Carroll et al. (2019); Piatti et al. (2024); Abdelnabi et al. (2024). In these settings, **what** is communicated extends beyond immediate observations or actions to include recommendations, shared norms, role expectations, and high-level strategies that regulate collective behavior Li et al. (2024a); Zhang et al. (2024b;a). Several works demonstrate that such decentralized exchanges can give rise to emergent division of labor, social conventions, and adaptive group-level organization without centralized control Kaiya et al. (2023); Liu et al. (2024b); Li et al. (2023c); AL et al. (2024). However, empirical analyses also reveal limitations unique to population-scale decentralization: communication becomes increasingly noisy or redundant as the number of agents grows, causal attribution of individual interactions becomes difficult, and system-level evaluation requires aggregate or statistical metrics rather than trajectory-level analysis Xiong et al. (2023); D'Arcy et al. (2024).

Overall, decentralized topologies consistently enable flexible, context-dependent **who**–**whom** selection and richer **what** content than rigid pipelines or hierarchies. At the same time, the literature reveals a common design tension: greater expressive power and adaptability come at the cost of increased ambiguity, coordination overhead, and stabilization difficulty, motivating the use of filtering, aggregation, or hybrid control mechanisms in practical systems.

**Hierarchical topology**   (Fig. 16) structures communication into layers, assigning distinct roles to agents: higher-level agents handle planning, abstraction, and task allocation, while lower-level agents are specialized for executing subtasks or domain-specific actions. This design enables modularity and scalable coordination by decomposing complex tasks. However, effective use relies on strong role specialization, coherent coordination across layers, and a shared understanding of task structure and progression.

Across hierarchical systems, the **who** and **whom** of communication are intentionally predetermined: supervisors decide **who** speaks, **whom** instructions target, and the structure of these interactions strongly constrains **what** is communicated. In multi-layer pipelines such as Magentic-One Fourney et al. (2024), MegaAgent Wang et al. (2024b), AutoAct Qiao et al. (2024), and CGMI Jinxin et al. (2023), higher-level agents issue abstract plans or task decompositions, while lower-level agents respond with execution details or local updates. These works explicitly encode a downward flow of **what**: plans, constraints, and goals; and an upward flow of **what**: progress reports, clarifications, or exceptions. Compared with decentralized systems where **who talks to whom** emerges dynamically, these approaches eliminate ambiguity about communication partners, reducing coordination overhead and yielding more reliable task adherence.

Role-based hierarchical simulations further reinforce how fixed **who talks to whom** mappings shape **what** is exchanged. In hospital simulations Li et al. (2024c), legal-service workflows Sun et al. (2024b), and AutoML pipelines Trirat et al. (2024); Du et al. (2024), agents are embedded in organizational structures mirroring real-world institutions. Here, **who** communicates is dictated by job role, and **whom** they address is defined by institutional hierarchy, e.g., doctors report diagnoses to coordinators, analysts return model results to pipeline managers. Consequently, **what** becomes highly structured: domain-specific evidence, predictions, summaries, or structured updates. These systems improve reliability and interpretability compared to decentralized topologies, which generate more diverse but less predictable messages.

General-purpose hierarchical environments such as AgentVerse Chen et al. (2023c) and related agent orchestration frameworks Ishibashi & Nishimura (2024); Rasal & Hauer (2024); Xu et al. (2023a) extend hierarchical communication beyond fixed pipelines toward **adaptive teaming and organizational alignment**. These systems emphasize explicit role assignment and supervisory coordination, showing that hierarchy is particularly effective when tasks require structured decomposition, global oversight, or consistent execution standards across agents Chen et al. (2023c); Zheng et al. (2023). Several works further demonstrate how hierarchical control supports reliability and scalability in complex workflows. For example, nested or supervisor–worker designs Ahn & Sentis (2021); Wu et al. (2024a) and multi-agent orchestration frameworks Sun et al. (2025c); Guo et al. (2024c) highlight that fixing **who talks to whom** simplifies coordination and stabilizes **what** is communicated—typically structured plans, constraints, or summaries—yielding predictable system behavior across long horizons. At the same time, these studies consistently reveal a key limitation of hierarchical communication. Because interaction patterns are rigidly defined, the space of **what** agents can express is correspondingly constrained, limiting lateral information sharing, peer-to-peer critique, and creative recombination of ideas Xu et al. (2023a); Rasal & Hauer (2024). As a result, hierarchical topologies are less effective in scenarios that benefit from spontaneous collaboration or exploratory reasoning, where decentralized or shared-message-pool designs allow richer and more diverse communication strategies. Taken together, this line of work clarifies a fundamental trade-off: hierarchical systems favor predictability, control, and interpretability at the cost of expressive flexibility. Compared with decentralized designs, they often achieve higher reliability and task adherence, but lower diversity in communication behaviors and reduced capacity for emergent, peer-driven reasoning.

**Centralized topology**   (Fig. 17) involves a single central agent coordinating communication among all agents. This centralized coordinator manages information flow, assigns tasks, and synchronizes actions, simplifying coordination and promoting consistency, particularly beneficial in large-scale deployments.

Centralized systems make the **who** and **whom** dimensions explicit: the central planner is always the **who** that initiates communication, and all subordinate agents are the **whom** it addresses. This fixed mapping strongly constrains **what** is communicated. For instance, in Scalable Multi-Robot Collaboration Chen et al. (2024g), the central controller allocates roles and synchronizes multi-robot teams, meaning the **what** passed downstream consists of structured task assignments and coordination directives, while the **what** returned upstream is limited to progress reports or sensor summaries. Compared with decentralized schemes where agents independently determine partners and message types, this reduces message diversity but substantially increases predictability and task adherence.

A similar pattern appears in MindAgent Gong et al. (2023), where a central manager orchestrates gaming agents by distributing observations and ensuring coherent actions. Because the **who talks to whom** mapping is fixed, agents rarely need to negotiate or infer partners; instead, **what** they communicate is shaped by managerial demands: local states, strategy suggestions, or execution feedback. These systems highlight both the strength and limitation of centralized topologies: despite efficiency and coherence, adaptability drops sharply if the central agent misinterprets context or becomes overloaded.

Centralized coordination is also widely used in general-purpose LLM-MAS frameworks including AutoGen, MetaGPT, AutoAgents, and OpenAgents Wu et al. (2023b); Hong et al. (2023); Chen et al. (2023a); Xie et al. (2023). In these systems, the central planner decomposes high-level goals into subtasks, selects specialized agents, and aggregates outputs. Here, the **who** is the planner generating decomposition instructions; the **whom** are the skill-specific agents; and the **what** becomes a structured interchange of plans, execution traces, and tool outputs. Because all communication flows through the planner, these frameworks gain strong global oversight but sacrifice robustness under noisy or ambiguous downstream messages.

Similarly, workflow-automation and system-integration platforms such as Magentic-One, AutoML-Agent, and AgentCoord Fourney et al. (2024); Trirat et al. (2024); Pan et al. (2024) embed a centralized controller to schedule subtasks, manage dependencies, and enforce execution order. Compared with hierarchical topologies, where information flows along multiple levels, these systems collapse hierarchy into a single locus of authority, simplifying **who talks to whom** but increasing pressure on the central agent to correctly interpret and route **what**. This concentration improves coordination efficiency but also narrows the expressive bandwidth of messages since agents tailor **what** they report to the controller's expectations.

In embodied and strategic game environments Kannan et al. (2023); Gong et al. (2023), centralized oversight ensures consistent high-level goals and prevents divergence among agents operating in complex spatial or adversarial settings. Yet this reliability comes at the expense of flexibility: unlike decentralized topologies where agents dynamically choose **whom** to address and decide **what** to share based on local needs, centralized systems impose rigid communication funnels that may bottleneck under high task complexity.

Overall, centralized topologies contrast sharply with chain, hierarchical, and decentralized structures: they minimize uncertainty in **who talks to whom** mappings and thus constrain **what** to well-defined task primitives, enabling consistency and fault detection at the cost of adaptability and emergent coordination. Their effectiveness therefore hinges on the competence and reliability of the central agent, which becomes both the enabling factor and the primary point of failure.

**Shared message pool topology**  (Fig. 18) employs a global broadcast channel or shared repository where agents asynchronously publish messages and retrieve relevant information. This topology provides flexibility and loose coupling, but may face challenges such as message overload or latency in high traffic scenarios.

Unlike chain or hierarchical topologies, where **who** talks to **whom** is specified by fixed edges, the shared message pool removes explicit addressing entirely. Instead, agents broadcast to a common memory and retrieve whatever they judge to be relevant. This indirection strongly shapes **what** is communicated, since messages must be self contained, interpretable, and useful to recipients that may not be known beforehand.

Dialogue based collaborative systems such as CAMEL Li et al. (2023b), AutoGen Wu et al. (2023b), Agent-Verse Chen et al. (2023c), and Reconcile Chen et al. (2023b) illustrate this pattern clearly. Because agents cannot target specific partners, the **who** is any agent contributing to the shared buffer and the **whom** is every agent capable of reading from it. As a result, **what** tends to include rationales, intermediate reasoning traces, and shared task context that can support the work of diverse collaborators. These systems improve

robustness compared with chain topologies, since any agent can join or leave without breaking the communication path. However, they suffer from ambiguity about which messages are relevant to which agent, which creates a need for retrieval heuristics and memory pruning.

Graph based coordination systems Hu et al. (2024c); Jeyakumar et al. (2024); Yang et al. (2025) use a structured shared memory such as a knowledge graph. Here, the **who** is an updating agent and the **whom** is the global state. Consequently, **what** agents communicate takes the form of structured updates, such as node proposals or constraint assertions. This improves scalability and creates a more interpretable global representation, but requires all agents to align on schemas and update rules, which reduces the flexibility of free form natural language messages.

Task-oriented collaboration frameworks such as CommAI Chen et al. (2024d), MapCoder Islam et al. (2024), MACRec Wang et al. (2024f), AgentCoord Pan et al. (2024), and MedAgents Tang et al. (2023) rely on shared repositories where agents publish plans, intermediate results, or tool outputs. Since no explicit **whom** is specified, **what** is communicated must be anticipatory and modular so that heterogeneous agents can reuse information. Compared with decentralized debates, these systems reduce message conflict and support higher levels of parallelism, but can suffer from coordination bottlenecks if too many agents produce unfiltered content.

Emergent organization approaches such as Theory of Mind agents and self organizing societies Li et al. (2023c); Ishibashi & Nishimura (2024) use the message pool to mediate adaptive roles and norms. Because the **who** and **whom** relations are fluid, **what** includes meta level information such as inferred intentions or social expectations. This enables rich emergent behaviors but also increases the risk of noise and instability.

Relative to decentralized topologies, where **who** selects **whom**, the shared pool simplifies coordination by allowing fully implicit partner selection but removes partner specificity. Relative to hierarchical structures, where **what** is shaped by manager worker relations, the shared pool encourages more symmetric communication but provides less control over message relevance. The effectiveness of this topology relies on relevance filtering, memory organization, and the ability of agents to prioritize among many concurrent messages.

### 7.2.2 Agent Configurations and Communication Capabilities

LLM-based multi-agent systems can operate with either heterogeneous or homogeneous agents. These configurations determine how the system specifies **who** communicates with **whom** and what types of information are exchanged. Figures 19 and 20 illustrate these two settings.

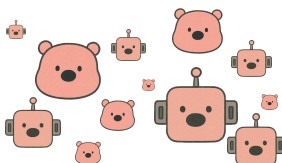

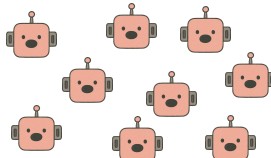

Figure 19: Heterogeneous     Figure 20: Homogeneous

**Heterogeneous agents**   In heterogeneous systems, agents possess different roles, skills, knowledge sources, or identities. This diversity allows the system designer to specify **who** interacts with **whom** based on functional dependencies and also shapes **what** is communicated, since messages must be tailored to the needs of particular recipients. Many role based multi agent systems rely on this structure. Works such as CAMEL Li et al. (2023b), ChatDev Qian et al. (2024a), social reasoning agents Xu et al. (2023b); Hua et al. (2023); Lan et al. (2023); Park et al. (2023); Chen et al. (2024c); Li et al. (2024d); Ghafarollahi & Buehler (2024); Cheng et al. (2024); Chen et al. (2024e), embodied collaborators Wu et al. (2025); Stepputtis et al. (2023); Hu et al. (2024a); Cheng et al. (2024), and multi stage task teams Fourney et al. (2024); Xie et al. (2023); Hong et al. (2023); Sun et al. (2024b); AL et al. (2024); Schmidgall et al. (2025) construct heterogeneity through explicit role prompts. This determines **who** initiates communication, **whom** they direct it to, and **what** content is appropriate for that role. For example, in ChatDev Qian et al. (2024a) the designer agent sends specifications to the coder agent, who returns code to the tester agent. The **who talks**

**to whom** mapping is therefore rigid and the **what** is specialized to each stage. CAMEL Li et al. (2023b) similarly enforces explicit bilateral roles that shape the exchange of rationales and plans.

Some works introduce stronger heterogeneity through tool calling or environment access Chen et al. (2024e); Schmidgall et al. (2025); Hua et al. (2023). Here, the **who talks to whom** structure emerges from capability boundaries. Agents direct messages to those uniquely able to perform actions such as web browsing or simulation control. Because tools require precise inputs, the **what** communicated becomes more structured and often constrained to parameterizable instructions. Compared with purely prompt based heterogeneity, these systems reduce redundancy and improve reliability, although they also narrow communication flexibility.

A different branch of heterogeneous systems uses persona variation to simulate social diversity Carroll et al. (2019); Chen et al. (2024a); Ahn & Sentis (2021); Liu et al. (2025b); Sun et al. (2025c); Tran et al. (2025b); Li et al. (2024f); Zhang et al. (2023b). In these settings, **who** communicates with **whom** is influenced by social compatibility, trust, or inferred preferences rather than static pipelines. As a result, **what** is communicated often includes beliefs, intentions, or culturally grounded reasoning. These systems improve realism and emergent dynamics but introduce higher variability and reduced predictability compared with role fixed teams.

Strategic heterogeneity appears in debate, negotiation, or opinion shaping contexts Li et al. (2024f); Xu et al. (2023b); Chen et al. (2024a). Agents differ in reasoning style, argumentative strength, or decision heuristics, which creates richer information exchange. Here, the **who talks to whom** pattern becomes fully connected since each agent must respond to all others. Consequently, **what** is communicated tends to include evidence, counterarguments, and justification. Compared with persona based heterogeneity, strategic heterogeneity is better for convergence and critique but less effective at modeling long horizon social dynamics.

Overall, heterogeneous agent settings enable targeted, role dependent communication with clear pathways specifying **who** speaks to **whom**. They also shape **what** is communicated toward specialization, justification, tool use, or persona grounded interaction. Their main advantage is increased coordination power and expressivity. Their main limitation is reduced scalability and greater dependence on accurate role specification.

**Homogeneous agents**   Homogeneous systems employ agents with identical architectures and comparable capabilities, without predefined role assignments or fixed communication hierarchies. In such settings, there is no pre-specified **who talks to whom** structure; instead, symmetry allows any agent to communicate with any other, and interaction patterns emerge from task demands and environmental context rather than from role differentiation. For example, Optima Chen et al. (2024f) demonstrates how homogeneous LLM agents coordinate planning and decision-making through peer-to-peer communication without designated leaders, while scalable graph-based planners Hu et al. (2024c) show that identical agents can dynamically form sparse communication graphs based on task relevance and dependency structure. VillagerAgent Dong et al. (2024) provides a concrete illustration of emergent **who**–**whom** selection: although all agents share the same architecture and reasoning capabilities, communication links arise from spatial proximity, shared observations, and immediate coordination needs in social-deduction environments. In this case, **what** is communicated consists primarily of local observations, suspicion signals, and coordination requests, rather than role-specific commands. Similarly, MedAgents Tang et al. (2023) employs homogeneous medical agents that exchange observations, hypotheses, and intermediate conclusions; despite identical capabilities, agents naturally differentiate their interactions based on case context and informational needs. Across these works, a common pattern emerges: homogeneity simplifies system design and avoids brittle role engineering, but it also constrains **what** is communicated to relatively generic signals—requests, observations, and coordination cues—since no agent is explicitly responsible for specialized planning or supervision. As a result, meaningful interaction structures can still emerge, yet they rely heavily on task structure and environmental signals to induce effective **who talks to whom** patterns.

Homogeneous collaboration remains effective even in embodied environments. Modular embodied agents Zhang et al. (2023b) show that identical agents can jointly perform object manipulation and navigation when communication is grounded in shared perception. This shifts **what** is communicated toward spatial descriptions or action proposals rather than strategy division.

Compared with heterogeneous systems, homogeneous designs scale more easily, simplify message formats, and reduce dependency on accurate role prompting. However, they may lack the expressive power needed for complex task decomposition and often require additional coordination mechanisms to prevent redundancy or conflict.

In summary, the heterogeneous versus homogeneous distinction directly affects specification of **who** interacts with **whom** and the structure of **what** is communicated. Heterogeneous teams excel at specialization and structured pipeline communication, while homogeneous teams excel at scalability, symmetry, and emergent coordination. The appropriate choice depends on the demands of the LLM based multi agent system.

### 7.2.3  Message Design and Communicative Content

Beyond interaction frequency, the structure of communication also determines **what** types of messages agents exchange and how content aligns with addressing assumptions. Structured messages constrain **what** can be communicated to predefined schemas, reducing ambiguity and enabling reliable execution when **who** and **whom** are fixed. Unstructured messages, by contrast, allow agents to convey nuanced reasoning and contextual information, which becomes necessary when addressing patterns are flexible or dynamically determined.

Different content types reflect distinct assumptions about **who** needs access to information and **what** level of abstraction is required. Task execution messages explicitly align responsibilities between agents, as in AutoGen and GPT-in-the-Loop, where **what** is communicated consists of actionable instructions or task states directed to specific executors Wu et al. (2023b); Nascimento et al. (2023a). Knowledge exchange mechanisms emphasize shared understanding and collective reasoning, with agents broadcasting intermediate insights rather than targeting a single recipient, as in CoMM Chen et al. (2024d). Negotiation and strategy-oriented communication emerges when **who** and **whom** have partially misaligned objectives, shifting **what** is communicated toward influence and compromise, exemplified by LLM-Deliberation Abdelnabi et al. (2023).

State updates, feedback, and persona maintenance further illustrate how message content depends on structural assumptions. Systems such as MetaGPT and ReConcile rely on frequent status reporting to maintain global coherence across agents Hong et al. (2023); Chen et al. (2023b), while persona-driven systems require agents to communicate self-consistent reflections to preserve identity and role stability Baltaji et al. (2024); Wang et al.. These designs implicitly assume that **who** receives a message can interpret it within a shared contextual frame, reinforcing the coupling between addressing structure and message semantics.

Taken together, the structure of communication in LLM-based multi-agent systems reveals that decisions about **who** communicates with **whom** directly motivate **what** must be communicated. Environmental context further constrains these choices, as grounded and embodied settings require spatially and temporally localized exchanges, while abstract workflows favor broadcast or pooled communication. Organizing interaction structure and message content jointly is therefore necessary to understand how communication supports coordination, scalability, and robustness in complex multi-agent systems.

### 7.2.4  Framework-Level Design of Communication Structure

This subsection focuses on existing LLM-based multi-agent frameworks because they provide concrete, system-level answers to the fundamental questions of **who** communicates with **whom**, **what** information is exchanged, and ultimately **why** communication is introduced in the first place. Unlike algorithmic studies that isolate communication modules, frameworks operationalize communication choices through explicit design decisions, such as agent roles, orchestration logic, message routing, and memory sharing. As a result, each framework implicitly encodes assumptions about **who** should be allowed to speak, **whom** they should address, and **what** kinds of information are necessary to support coordination, reasoning, or task execution. Examining these frameworks therefore reveals how different application demands motivate distinct answers to the **why communicate** question and how those motivations shape the resulting **who**–**whom**–**what** configurations.

Table 12 summarizes representative LLM multi-agent frameworks through the lens of communication topology, number of agents, and application domain, providing a compact view of how these systems instantiate

communication structure at scale. The topology columns explicitly characterize constraints on **who** can communicate with **whom**, ranging from rigid chains and hierarchies to decentralized and shared-memory designs. The application domains further contextualize **why** these communication patterns are chosen, for example, reliability in software engineering workflows, diversity of reasoning in debate settings, or autonomy in embodied environments. By juxtaposing these dimensions, the table highlights that frameworks supporting similar tasks often converge on similar answers to the **who**–**whom** question, while differing primarily in **what** information is exchanged and how strictly communication is regulated. This overview sets the stage for the detailed analysis that follows, where we compare how different classes of frameworks resolve these tradeoffs in practice.

Table 12: Comparison of LLM Multi-Agent Frameworks

| Framework | Topology | | | | | Num Agents | Application-Domain |
|---|---|---|---|---|---|---|---|
| | Chain | Hier. | Decent. | Cent. | Pool | | |
| LangGraph lan (b) | ✓ | ✓ | ✓ | ✓ | ✓ | Flexible | General |
| CrewAI cre | ✓ | ✓ | | ✓ | | Flexible | General |
| OpenAI Swarm OpenAI | ✓ | | | | | Flexible | General |
| SuperAGI sup | | | | ✓ | | Flexible | General |
| CAMEL Li et al. (2023b) | ✓ | | | | ✓ | 2 | General |
| AutoGen Wu et al. (2023b) | ✓ | ✓ | | ✓ | ✓ | Flexible | Reasoning tasks |
| AgentVerse Chen et al. (2023c) | | ✓ | ✓ | | ✓ | Flexible | General |
| MetaGPT Hong et al. (2023) | ✓ | ✓ | | ✓ | | 5 | Software dev & coding |
| MegaAgent Wang et al. (2024b) | | ✓ | | | | 590+ | Software dev & coding; policy simulation |
| ProAgent Zhang et al. (2024a) | | | ✓ | | | 2 | Embodied - Overcooked |
| AgentCF Zhang et al. (2024b) | | | ✓ | | | Users & Items | Recommender |
| ChatDev Qian et al. (2024a) | ✓ | | | | | 5 | Software dev & coding |
| COLEA Zhang et al. (2023b) | | | ✓ | | | 2 | Embodied - WAH and TDW-MAT |
| MedAgents Tang et al. (2023) | | | ✓ | | ✓ | 3 | Clinical reasoning tasks |
| ReConcile Chen et al. (2023b) | | | ✓ | | ✓ | 3 | Reasoning tasks |
| AutoAgents Chen et al. (2023a) | | ✓ | | ✓ | | Flexible | General task automation |
| DyLAN Liu et al. (2024b) | | | ✓ | | | Flexible | General |
| OpenAgents Xie et al. (2023) | | | | ✓ | | 3 | Data Analysis/API Tools/Web assistant |
| MACRec Wang et al. (2024f) | | | ✓ | | | 5 | Recommendation system |
| Magentic-One Fourney et al. (2024) | | | | ✓ | | 5 | General |
| AgentCoord Pan et al. (2024) | ✓ | ✓ | ✓ | ✓ | ✓ | Flexible | Visualization of coordination strategies |
| AutoML-Agent Trirat et al. (2024) | | | | ✓ | | 5 | ML pipeline (data→deployment) |
| GPT-Swarm Zhuge et al. (2024a) | | | ✓ | | | Flexible | Reasoning tasks |
| MACNet Qian et al. (2024b) | ✓ | ✓ | ✓ | | | Flexible | General |
| Theory-of-Mind Li et al. (2023c) | | ✓ | | | ✓ | 3 | Customized text game (search and rescue) |
| Self-Organized Ishibashi & Nishimura (2024) | | | ✓ | | | Flexible | Software dev & coding |
| SMART-LLM Kannan et al. (2023) | | | | ✓ | | 1 LLM + N robots | Multi-robot task planning |

**Workflow oriented frameworks** Workflow oriented systems such as LangGraph lan (b), CrewAI cre, AutoAgents Chen et al. (2023a), Magentic-One Fourney et al. (2024), AutoML-Agent Trirat et al. (2024), MetaGPT Hong et al. (2023), ChatDev Qian et al. (2024a), MegaAgent Wang et al. (2024b), Self-Organized Ishibashi & Nishimura (2024), OpenAI Swarm OpenAI, and OpenAgents Xie et al. (2023) define a relatively rigid pattern of **who** communicates with **whom**. A central planner or manager agent typically receives the user goal, decomposes it into subtasks, and assigns these subtasks to specialized worker agents, which then report back their outputs. This design constrains **what** is communicated to structured artifacts such as task specifications, intermediate results, bug reports, and tool invocation logs. MetaGPT Hong et al. (2023), MegaAgent Wang et al. (2024b), and ChatDev Qian et al. (2024a) emphasize strong role specialization in software engineering settings, which narrows **what** each role communicates and improves reliability for code production and review. In contrast, LangGraph lan (b), CrewAI cre, OpenAI Swarm OpenAI, and OpenAgents offer more flexible compositions of agents and tools, expanding the range of **what** can be exchanged but increasing coordination overhead. AutoAgents Chen et al. (2023a), Magentic-One Fourney et al. (2024), AutoML-Agent Trirat et al. (2024), and Self-Organized further highlight that central orchestration simplifies debugging and monitoring but can create bottlenecks when many agents depend on a single coordinator. Across these systems, the main tradeoff is between strict control of **who** talks to **whom** for predictable behavior and looser structures that support richer but harder to manage communication.

**Collaborative reasoning and decision making frameworks** Collaborative reasoning frameworks such as CAMEL Li et al. (2023b), ReConcile Chen et al. (2023b), AutoGen Wu et al. (2023b), GPT-Swarm Zhuge

et al. (2024a), MACNet Qian et al. (2024b), and MedAgents Tang et al. (2023) relax some of the workflow constraints to encourage richer dialogue. They typically rely on chain or shared message pool topologies so that each agent can observe and respond to the evolving conversation. This broadens the possible **whom** targets for each agent and shifts **what** is communicated toward rationales, critiques, counterarguments, and confidence assessments. CAMEL Li et al. (2023b) keeps **who** and **whom** simple by pairing two role-playing agents, which facilitates convergence and grounded collaboration. AutoGen Wu et al. (2023b) and MACNet Qian et al. (2024b) extend this to multiple agents with complementary roles, increasing reasoning depth but also introducing redundancy when many agents address the same issue. ReConcile Chen et al. (2023b) and MegaAgent Wang et al. (2024b) use shared message pools to support multi perspective critique in reasoning and clinical settings, respectively, pushing **what** toward evidence based justification and error checking. GPT-Swarm Zhuge et al. (2024a) emphasizes scalability in the number of cooperating agents, showing that more agents increase coverage of the solution space but require mechanisms to filter low-quality contributions. Overall, these frameworks illustrate how enlarging the set of possible **whom** recipients and promoting argumentative **what** content can improve solution quality, at the cost of more complex aggregation and consensus procedures.

**Embodied and interactive environment frameworks**  Embodied and interactive frameworks such as ProAgent Zhang et al. (2024a), COLEA Zhang et al. (2023b), Theory-of-Mind–based agents Li et al. (2023c), and SMART-LLM Kannan et al. (2023) primarily operate in simulated or physical environments where communication is grounded in action and perception. Here, **who** communicates with **whom** is often determined dynamically by spatial relationships, task assignments, or inferred capabilities. In ProAgent Zhang et al. (2024a) and COLEA Zhang et al. (2023b), for example, a small number of agents coordinate in Overcooked or manipulation environments, and **what** is communicated focuses on local state, intended moves, or high-level intent to avoid collisions and deadlocks. Theory-of-Mind systems introduce richer internal models of other agents, so messages can include beliefs, goals, or inferred mental states, changing **what** from purely action-oriented content to explanations of reasoning. SMART-LLM Kannan et al. (2023) uses a central LLM planner with multiple robots, but at the level of the robots, communication still centers on motion plans and task assignments. Compared to workflow and reasoning frameworks, embodied systems are more constrained in **what** they must communicate, yet they are more sensitive to errors in **who** talks to **whom**, since misrouted information can directly degrade physical coordination.

**Personalization frameworks**  Personalization-oriented systems such as AgentCF Zhang et al. (2024b) and MACRec Wang et al. (2024f) define **who** as specialized recommenders that each capture a distinct perspective on users, items, or contexts, while **whom** is the aggregation mechanism that combines these perspectives into final recommendations. Because the goal is to keep the system efficient and interpretable, **what** is communicated is typically lightweight, including scores, embeddings, or short rationales rather than long free form messages. AgentCF Zhang et al. (2024b) allows agents to attend to each other's intermediate outputs, which enriches **what** by incorporating cross-agent critique and adjustment, leading to more diverse recommendations. MACRec Wang et al. (2024f) intentionally restricts communication between agents to control noise and maintain stability, trading some expressiveness in **what** for improved robustness. Comparing these two frameworks highlights how communication density among specialized agents influences both accuracy and interpretability in personalization tasks.

**General purpose frameworks**  General purpose infrastructures such as LangChain lan (a), Super-AGI sup, AutoGen Wu et al. (2023b), AgentVerse Chen et al. (2023c) expose flexible building blocks rather than fixed communication patterns. They leave **who** communicates with **whom** largely under developer control through configurable graphs, routing policies, or orchestration logic. As a result, **what** can be communicated spans tool calls, environment observations, intermediate reasoning traces, and meta level control signals. AgentVerse illustrates how this flexibility supports heterogeneous teams of agents operating under different topologies, while AgentCoord visualizes emergent communication patterns to help diagnose whether information is being routed effectively. LangChain and SuperAGI focus on tool integration and workflow construction, letting developers tune **who** and **whom** for specific applications. AutoGen appears here as well because it provides both ready made interaction patterns and lower level primitives. These frameworks

demonstrate the benefits and risks of high flexibility: they enable rapid experimentation with communication structures but also place the burden of designing stable **who**–**whom**–**what** configurations on practitioners.

In summary, existing LLM based multi-agent frameworks differ systematically in how they organize **who** communicates with **whom** and thus structure **what** information flows through the system. Workflow and centralized frameworks fix these dimensions to ensure reliability and debuggability, collaborative reasoning frameworks broaden them to support critique and consensus, embodied frameworks tie them to environmental and physical constraints, and personalization frameworks optimize them for efficient user modeling. General-purpose platforms span these regimes but expose unresolved challenges in scaling communication without losing coherence. This diversity suggests that designing communication in LLM-based multi-agent systems is fundamentally about controlling the interaction between **who**, **whom**, and **what** rather than only choosing a topology in isolation.

### 7.3 When and Where LLM Agents Communicate

#### 7.3.1 Motivations for Communication Timing

Motivational settings determine not only the alignment of agent objectives but, more fundamentally, **when** agents choose to communicate during interaction. In LLM-based multi-agent systems, communication is often selective and costly, so agents do not exchange messages continuously. Instead, motivational structure governs the timing of communication, such as whether agents speak preemptively, reactively, strategically, or only at key decision points. We categorize motivational settings into three main types: competitive, cooperative, and mixed-motive (Figures 21–23).

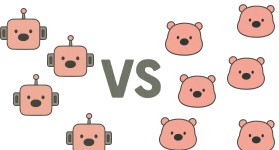
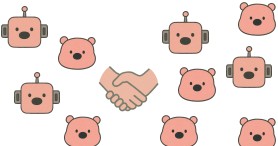
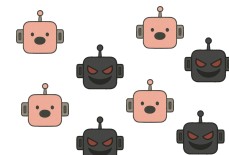

Figure 21: Competitive        Figure 22: Cooperative        Figure 23: Mixed-Motives

**Competitive settings**   This setting (Fig. 21) arises in adversarial or conflicting-goal environments, fundamentally altering the function and timing of communication. Unlike cooperative systems, where communication supports synchronization, in competitive settings **when** to communicate becomes a strategic decision: agents communicate selectively at moments where messages can influence beliefs, commitments, or outcomes. As a result, communication is sparse, high-stakes, and tightly coupled to strategic leverage rather than routine state sharing. **(1) Hidden-role and social-deduction games**, including Werewolf Bailis et al. (2024), Avalon Wang et al. (2023b), and AvalonBench Light et al. (2023), exemplify this regime. In these environments, **when** agents communicate is largely dictated by explicit game phases such as accusation, defense, and voting. Communication is concentrated at points of maximal impact, where revealing, distorting, or withholding information can decisively shift collective beliefs. A recurring trend across these works is that communication prioritizes persuasion and deception over information completeness, making timing more critical than message frequency. **(2) Debate-oriented competitive frameworks**, such as LLM-Debate Du et al. (2023), MAD Liang et al. (2023), and ChatEval Chan et al. (2023), impose externally scheduled communication rounds. Here, **when** communication occurs is fixed by debate protocols—critique, rebuttal, and judgment phases—which improves fairness, comparability, and evaluation consistency across agents. However, this structure limits adaptive timing strategies, preventing agents from opportunistically delaying, withholding, or escalating arguments based on opponent behavior. **(3) Repeated adversarial benchmarks**, including LLM Arena Chen et al. (2024c), embed agents in ongoing competition where **when** to communicate must be decided reactively. Unlike turn-based debates, agents here choose communication timing in response to evolving opponent strategies and performance feedback. These settings expose failure modes such as over-persuasion, delayed response, or ineffective escalation, highlighting the difficulty of timing communication in open-ended competitive interaction. **(4) Strategic dialogue and critique-based systems**, such as negotiation and reconciliation frameworks Abdelnabi et al. (2023); Chen et al. (2023b),

treat conflict detection itself as the trigger for communication. Compared with hidden-role games that reward deception, these systems encourage agents to communicate when disagreements or inconsistencies are identified, shifting the timing objective from belief manipulation toward conflict resolution and outcome stabilization. **(5) Dynamic competitive environments**, including MindAgent Gong et al. (2023) and extended Arena-style settings Chen et al. (2024c), further emphasize temporal adaptivity. In these systems, agents communicate in response to changes in environmental state or opponent strategy, making **when** communication occurs context-dependent rather than phase- or turn-based. This highlights the importance of rapid detection and response over deliberative scheduling. **(6) Learning-based competitive frameworks** Du et al. (2024); Feng et al. (2024) introduce adaptive timing policies learned through self-play or reinforcement learning. Here, agents discover **when** to communicate based on task difficulty, uncertainty, and opponent strength, often outperforming fixed or prompt-driven schedules. This trend suggests that optimal communication timing in competitive settings is itself a learned control problem rather than a static design choice. Overall, competitive settings consistently concentrate communication at strategically critical moments. Compared with cooperative systems, communication is less frequent but more consequential: mistimed messages can directly degrade performance, while well-timed interventions can decisively shift beliefs, votes, or strategic balance.

**Cooperative settings**  This setting (Fig. 22) assumes aligned objectives, shifting the role of communication from persuasion or negotiation toward coordination, synchronization, and error recovery. Importantly, even in fully cooperative systems, agents do not communicate continuously. Instead, **when** communication occurs is typically event-driven, triggered by task decomposition boundaries, dependency resolution, replanning needs, or uncertainty about execution outcomes. **(1) Structured task-driven systems** in software engineering, healthcare, law, and scientific discovery—such as CAMEL Li et al. (2023b), Chat-Dev Qian et al. (2024a), AutoML-style pipelines Du et al. (2024), hospital simulations Li et al. (2024c), legal workflows Sun et al. (2024b), and scientific discovery agents Ghafarollahi & Buehler (2024)—exhibit highly regular communication timing. In these systems, **when** agents communicate is tightly coupled to role hand-off points (e.g., planner to executor, executor to reviewer), which improves reliability and traceability but limits flexibility when task structure or execution order changes. **(2) General cooperative benchmarks and coordination frameworks**, including AgentCF Zhang et al. (2024b), ProAgent Zhang et al. (2024a), AgentCoord Pan et al. (2024), Gentopia Xu et al. (2023a), MARG D'Arcy et al. (2024), and large-scale evaluation studies Agashe et al. (2023); Zheng et al. (2023); Li et al. (2024a), shift communication timing away from fixed pipelines toward coordination boundaries. Here, agents communicate **when** shared plans must be aligned, responsibilities reassigned, or conflicts resolved, reflecting a trend toward adaptive but still structured synchronization rather than continuous exchange. **(3) Human–agent cooperative studies**, such as those on utility alignment Carroll et al. (2019), preference elicitation Feng et al. (2024), reconciliation and critique Chen et al. (2023b), negotiation Yin et al. (2023), reflective reasoning Shinn et al. (2024), and mixed-initiative collaboration Wu et al. (2024a), emphasize a different driver of timing: uncertainty reduction. In these settings, **when** to communicate is dictated less by task stage and more by ambiguity in intent, preferences, or trust, and communication quality is evaluated by its ability to reduce misalignment rather than task completion alone. **(4) Embodied cooperative environments**, including robotic construction Zhang et al. (2023b), scalable multi-robot coordination Chen et al. (2024g), real-world manipulation Mandi et al. (2024), embodied agent teams Guo et al. (2024c), and control-oriented LLM systems Wang et al. (2023c), impose strict real-time constraints. In these works, agents communicate primarily at joint-action boundaries, replanning events, or failure recovery moments, making **when** communication occurs critical for safety, efficiency, and physical feasibility. **(5) Self-organizing and adaptive team frameworks**, such as self-organizing agents Ishibashi & Nishimura (2024), dynamic coordination models Liu et al. (2024b), adaptive navigation systems Rasal & Hauer (2024), long-horizon agent teams Schmidgall et al. (2025), Optima Chen et al. (2024f), AutoAct Qiao et al. (2024), GPT-Swarm Zhuge et al. (2024a), autonomous multi-agent control Liu et al. (2024a), scalable planners Hu et al. (2024c), VillagerAgent Dong et al. (2024), AutoML coordination Trirat et al. (2024), and MedAgents Tang et al. (2023) further relax fixed schedules. Compared to static cooperative pipelines, these systems trigger communication during capability discovery, delegation, and long-horizon state maintenance, highlighting a trend toward adaptive timing at the cost of higher coordination overhead. Finally, **(6) reusable infrastructures** such as AgentVerse Chen et al. (2023c), AutoGen Wu et al. (2023b), MetaGPT Hong et al. (2023), and ProAgent Zhang et al. (2023a)

generalize cooperative communication patterns across tasks and domains. While they broaden coverage of coordination scenarios, they also expose a recurring design tension: without explicit timing regulation via memory, scheduling, or attention mechanisms, cooperative systems risk over-communication that degrades efficiency rather than improving coordination.

**Mixed-motive settings**  This setting (Fig. 23) combines cooperative and competitive incentives, making communication timing inherently context-dependent. Unlike purely cooperative or adversarial regimes, agents must continuously assess whether communication will strengthen alliances, signal commitment, repair trust, or expose strategic vulnerabilities. As a result, **when** to communicate is neither fixed nor sparse, but adaptively modulated by evolving social relationships and incentive structures. **(1) Generative agent societies and social simulations**, such as role-based and persona-driven environments Lan et al. (2023); Hua et al. (2023); Chen et al. (2024a); Abdelnabi et al. (2024), illustrate this tension clearly. In these systems, communication is frequently triggered by social events—e.g., alliance formation, reputation repair, betrayal, or conflict escalation—rather than by task decomposition alone. A recurring pattern is that early communication serves to establish social positioning and intent, while later communication reflects shifts in trust and strategic alignment. **(2) Large-scale self-organizing and open-ended platforms** Cheng et al. (2024); Liu et al. (2025b); Tran et al. (2025b); Sun et al. (2025c); Xie et al. (2023); AL et al. (2024); Hu et al. (2024a) emphasize the role of scale and temporal evolution. In these systems, **when** communication occurs changes over time: early-stage interactions focus on partner discovery, norm formation, and role negotiation, whereas later-stage communication shifts toward maintaining coherence, resolving disputes, and managing long-term dependencies across many agents. This temporal stratification of communication timing is largely absent in smaller or static environments. **(3) Nested, modular, and multi-level architectures** Fourney et al. (2024); Xu et al. (2023c); Gong et al. (2023); Wu et al. (2025); Li et al. (2024f); Ahn & Sentis (2021) further refine communication timing by introducing structural layers. In these systems, agents communicate frequently within local groups for coordination and execution, but communicate selectively across groups when incentives diverge or global alignment is required. This produces layered communication schedules, where **when** to communicate depends jointly on group membership, task phase, and strategic alignment. Overall, mixed-motive settings reveal that communication timing cannot be prescribed by a single principle. Compared with cooperative systems that emphasize coordination events and competitive systems that concentrate communication at leverage points, mixed-motive environments require agents to dynamically adapt **when** they communicate in response to shifting incentives, social structure, and long-horizon strategic considerations.

**Game-Theoretic View of Communication Timing in Mixed-Motive Settings**  Mixed-motive communication settings can be formally interpreted through the lens of *dynamic games with incomplete information*. Let agents $i \in \mathcal{N}$ interact in an extensive-form game characterized by state $s_t$, private information $\theta_i$, actions $a_i^{(t)}$, and messages $m_i^{(t)}$. Communication acts modify agents' information sets and beliefs, $\mu_i^{(t)} = \Pr(\theta_{-i} \mid h_i^{(t)}, m_{-i}^{(t)})$, thereby reshaping strategic incentives and equilibrium behavior. From this perspective, deciding **when** to communicate corresponds to selecting signaling actions at specific nodes of the game tree, balancing information revelation against strategic vulnerability. Equilibrium behavior in mixed-motive multi-agent systems is often *emergent and heterogeneous*. In large or role-structured environments, different subsets of agents may converge to local Nash or Bayesian Nash equilibria, while higher-level coordination arises through aggregation, mediation, or hierarchy. Many MARL and LLM-based systems implicitly approximate Nash or Bayesian Nash equilibria via self-play or reinforcement learning, even when equilibria are not solved explicitly. Recent work, such as Yi et al. (2025), formalizes multi-LLM coordination as an incomplete-information game and explicitly targets the Bayesian Nash equilibrium, demonstrating improved efficiency and regret bounds relative to non-equilibrium coordination. Beyond learning-based approximation, several studies directly examine the ability of LLMs to *compute* or reason about equilibrium. Silva et al. Silva (2024) study LLM performance in mixed-strategy Nash equilibrium games and show that, while LLMs can identify equilibria in canonical settings, especially when equipped with code execution and structured prompts, their performance degrades sharply under slight perturbations of game structure or randomized strategies. These findings highlight both the promise and fragility of LLMs as equilibrium reasoners, underscoring the gap between surface-level strategic competence and robust game-theoretic reasoning. More recent frameworks further extend equilibrium analysis to the *reasoning level*. The LLM-Nash framework Zhu

(2025) defines equilibrium over the prompt or reasoning space, modeling bounded rationality by treating LLM inference itself as part of the strategic process. In this view, actions emerge as behavioral outputs of equilibrium reasoning trajectories, and equilibrium outcomes may diverge from classical Nash predictions due to cognitive constraints and epistemic learning. These perspectives suggest that **communication timing** in mixed-motive settings is best understood as an equilibrium-consistent strategy across multiple levels: agents speak, delay, or withhold messages when doing so improves expected utility under evolving beliefs, whether equilibrium is approximated through learning, computed explicitly, or realized through reasoning-level adaptation. This unifies modern MARL and LLM-based communication with classical notions of signaling, cheap talk, and equilibrium selection, while remaining compatible with data-driven multi-agent learning.

### 7.3.2 Environmental Triggers and Context

The environments in which LLM agents operate play a central role in determining **when** communication is necessary, optional, or costly. Unlike motivational settings, which describe agents' goals, application domains impose concrete constraints on observability, temporal coupling, action dependencies, and feedback latency. These factors directly influence **when to communicate**: agents may need to exchange information before acting, during execution, after observing outcomes, or only when coordination failures arise. We therefore organize application domains as a lens on **when** communication is triggered by environmental structure rather than by incentives alone.

Table 13 summarizes major application domains for LLM-based multi-agent systems and highlights how different environments induce distinct communication timing patterns. Across domains, communication is not uniformly continuous. Instead, **when** agents communicate depends on where uncertainty arises, where dependencies between agents are strongest, and where delayed or missing information would cause coordination breakdowns.

**Coding**  In coding-oriented multi-agent systems, **when** communication occurs is primarily governed by task decomposition, artifact dependencies, and error propagation rather than social incentives or persuasion. Communication is therefore event-driven and tightly coupled to the software development lifecycle, with agents exchanging information when intermediate artifacts become available, inconsistent, or insufficient for progress. **(1) Stage-based and role-driven coding pipelines**, exemplified by early systems such as ChatDev Qian et al. (2024a), adopt predefined workflows that explicitly schedule communication at fixed stages (e.g., design, implementation, testing, and review). In these systems, **when** agents communicate is determined by role handoff points, ensuring clarity and traceability of information flow. This structured timing improves reliability and reduces ambiguity, but it also limits adaptability when task requirements change or unexpected dependencies arise. **(2) Parallel and multi-team coding frameworks**, including systems that relax strict stage ordering Du et al. (2024); Li et al. (2024d), shift communication timing toward periodic synchronization and conflict resolution events. Here, agents or subteams work concurrently and communicate **when** inconsistencies, integration conflicts, or divergent design choices are detected. This trend improves robustness through diversity and redundancy, but introduces additional coordination overhead and requires explicit mechanisms to manage synchronization frequency. **(3) Self-organizing and uncertainty-driven coding agents**, such as those explored in Ishibashi & Nishimura (2024), further decouple communication from predefined schedules. In these systems, **when** to communicate is demand-driven: agents initiate communication only after detecting uncertainty, stalled progress, or insufficient confidence in their local solutions. This adaptive timing reduces unnecessary exchanges but places a greater burden on uncertainty estimation and progress monitoring. **(4) Domain-specific coding environments**, including automated machine learning pipelines Trirat et al. (2024) and competitive programming agents Islam et al. (2024), illustrate how artifact readiness directly governs communication timing. Agents communicate when concrete outputs, such as trained models, feature sets, compiler errors, or failing test cases, become available, making **when** to communicate tightly aligned with artifact maturity rather than abstract task phases. Overall, coding environments consistently favor communication at dependency boundaries rather than continuous dialogue. Across these systems, **when** communication occurs is dictated by artifact availability, integration risk, and uncertainty, reflecting a broader trend toward event-driven and demand-aware communication policies rather than fixed or socially motivated schedules.

| Application Domain | Key Communication Characteristics | Timing, Structure, and Design Implications |
|---|---|---|
| **Coding Systems** | • Role-based decomposition (planner, coder, reviewer): Qian et al. (2024a); Li et al. (2023b);
• Artifact-centric coordination via code, tests, or models: Trirat et al. (2024); Islam et al. (2024);
• Increasing use of self-organizing or adaptive teams: Ishibashi & Nishimura (2024); Li et al. (2024d). | • Communication triggered at dependency boundaries or artifact readiness;
• Fixed pipelines improve reliability but reduce adaptability;
• Adaptive timing improves robustness at the cost of coordination overhead. |
| **Virtual Environments** | • Event-driven social interaction and memory updates: Park et al. (2023); Kaiya et al. (2023);
• Turn-based strategic communication in games: Bailis et al. (2024); Light et al. (2023);
• Open-ended exploration and long-horizon planning: Lan et al. (2023); Hua et al. (2023). | • Communication timing driven by state changes or social events;
• Strict interaction rounds enable evaluation but limit adaptivity;
• Scalability requires selective and delayed communication. |
| **Embodied Environments** | • Grounded language-action coupling: Ahn et al. (2022); Zhang et al. (2023b);
• Coordination under physical and temporal constraints: Mandi et al. (2024); Guo et al. (2024c);
• Mixed centralized and decentralized control strategies: Kannan et al. (2023). | • Communication aligned with perception–action loops;
• Early communication aids planning, late communication aids recovery;
• Over-communication degrades real-time performance. |
| **Human–Agent Workflows** | • Mixed-initiative dialogue and preference elicitation: Feng et al. (2024); Wu et al. (2022);
• Role- and persona-aware interaction: Chen et al. (2024a); Li et al. (2024a);
• Structured debate and critique mechanisms: Xu et al. (2023b); Jin et al. (2024). | • Communication timed to reduce uncertainty or cognitive load;
• Turn-taking improves reasoning but limits responsiveness;
• Effective timing balances informativeness and user trust. |

Table 13: Taxonomy of LLM-based multi-agent systems: linking application domains to communication characteristics and timing-driven design implications.

**Virtual Environments**   Virtual environments are characterized by evolving world states, partial observability, and long-horizon interaction, making **when** communication occurs contingent on environmental events and state transitions rather than fixed task phases. Communication timing in these settings is therefore predominantly event-driven, shaped by changes in social context, spatial proximity, resource availability, or agent beliefs. **(1) Generative agent simulations and social worlds**, such as Smallville-style environments Park et al. (2023); Kaiya et al. (2023), emphasize socially motivated timing. In these systems, agents communicate **when** new information is acquired, social relationships change, or long-term plans are revised. Communication is triggered by events like encounters with other agents, updates to memory or goals, and shifts in social context, reflecting a trend toward communication driven by internal state updates rather than externally imposed schedules. **(2) Social deduction and strategic game environments**, including Avalon and Werewolf benchmarks Lan et al. (2023); Light et al. (2023); Bailis et al. (2024), impose explicit temporal constraints on communication. Here, **when** agents may communicate is tightly regulated by game mechanics, such as discussion rounds or voting phases. This structured timing enables controlled evaluation of reasoning, deception, and belief manipulation, but also restricts adaptive communication strategies that depend on continuous state monitoring. **(3) Competitive and evaluative virtual benchmarks**, such as LLMArena and related platforms Chen et al. (2024c); Agashe et al. (2025), systematically vary interaction schedules to probe agent robustness. These studies reveal that agents often struggle when communication opportunities are sparse, delayed, or strictly time-limited, highlighting sensitivity to **when** information exchange is permitted rather than to message content alone. **(4) Large-scale virtual societies and task-oriented simulations**, including institutional and project-based environments Li et al. (2024c); AL et al. (2024); Wang et al. (2023a); Zhu et al. (2023); Wu et al. (2024b), expose scalability-driven shifts in communication timing. As the number of agents and interactions grows, communication becomes increasingly selective and delayed, typically triggered by resource contention, coordination failures, or significant environmental changes. This trend reflects a move away from frequent interaction toward sparse, high-impact communication events. Overall, virtual environments foreground timing as a central design variable. Across social simulations, strategic games, and large-scale virtual worlds, **when** communication occurs is governed by environmental dynamics and scalability constraints, revealing a fundamental tension between responsiveness and tractability that does not arise in more structured task-driven settings.

**Embodied Environments**   Embodied environments couple communication tightly to real-time perception–action loops, making **when** communication occurs a function of physical dynamics, sensing uncertainty, and safety constraints rather than abstract task stages. In these settings, communication is costly in time and attention, and mistimed messages can directly degrade performance or safety. **(1) Early embodied coordination systems**, including multi-robot and embodied LLM agents Ahn et al. (2022); Agashe et al. (2023); Zhang et al. (2023b), demonstrate that proactive communication before action execution is often beneficial. Agents communicate **when** plans must be aligned spatially (e.g., shared navigation goals or manipulation targets), reducing downstream conflicts and redundant motion. These works establish a baseline insight: in tightly coupled physical tasks, early communication can prevent costly errors. **(2) Centralized versus decentralized planning studies** Mandi et al. (2024); Kannan et al. (2023); Chen et al. (2024g); Dong et al. (2024); Guo et al. (2024c) reveal a systematic divergence in communication timing. Centralized or coordinator-based systems tend to communicate early, aggregating observations and issuing joint plans before execution. In contrast, decentralized systems delay communication until local uncertainty, prediction error, or coordination risk exceeds a threshold. This comparison highlights a key trade-off: early communication improves global consistency, while delayed communication preserves reactivity and scalability. **(3) Cooperative embodied benchmarks**, such as Overcooked-style environments and related evaluations Agashe et al. (2023); Zhang et al. (2024a), emphasize that communication frequency alone is not predictive of success. Instead, performance correlates strongly with **when** communication occurs—particularly at joint-action boundaries, task handoffs, or failure recovery moments. These benchmarks show that sparse but well-timed messages outperform continuous or poorly aligned communication. Overall, embodied environments make **when to communicate** inseparable from perception and action timing. Across centralized planners, decentralized agents, and cooperative benchmarks, the literature converges on a common design principle: effective embodied communication is event-triggered and environment-aware, prioritizing safety, coordination, and responsiveness over message volume Mandi et al. (2024); Agashe et al. (2023); Guo et al. (2024c).

**Human-Agent Workflows and Collaboration**   In human–agent workflows, **when** communication occurs is fundamentally constrained by human attention, trust calibration, and cognitive load, making timing as critical as content. Unlike purely automated systems, excessive or poorly timed communication can directly degrade human performance and trust. **(1) Task-driven human–agent workflows**, such as CAMEL-style collaborative problem solving Li et al. (2023b) and preference-aware assistance systems Feng et al. (2024), show that communication is most effective when triggered at decision-critical moments rather than continuously. These systems demonstrate that delaying communication until uncertainty, ambiguity, or irreversible choices arise reduces cognitive burden while preserving decision quality. **(2) Interactive and mixed-initiative systems**, including human-in-the-loop AI interfaces Wu et al. (2022) and adaptive guidance policies Feng et al. (2024), explicitly model **when** to communicate as a control problem. Here, agents learn to request feedback or intervention only when confidence drops below a threshold or when predicted outcomes diverge, highlighting a trend toward demand-driven rather than schedule-driven communication. **(3) Persona-based and culturally grounded agents** Chen et al. (2024a); Li et al. (2024a) further refine communication timing by adapting to user expectations, roles, and social norms. These works show that **when** communication is perceived as appropriate depends not only on task state but also on social context, with mistimed messages harming engagement even if informationally correct. **(4) Human-in-the-loop multi-agent debates and team-based reasoning environments** Xu et al. (2023b); Stepputtis et al. (2023); Jin et al. (2024); Du & Zhang (2024); Xiong et al. (2023); Chen et al. (2023b) emphasize structured timing mechanisms such as turn-taking, moderation, or intervention windows. These studies consistently find that externally regulated communication timing improves reasoning quality and reduces redundancy, whereas unrestricted communication leads to repetition and cognitive overload. Overall, human–agent collaboration reframes **when to communicate** as a balance between responsiveness and cognitive efficiency. Across task-driven workflows, mixed-initiative systems, persona-aware agents, and debate settings, the literature converges on a shared design principle: effective human–agent communication is sparse, event-triggered, and socially aware, prioritizing human trust and usability over maximal information exchange.

### 7.3.3   Evaluation Settings for Communication

Evaluating LLM-based multi-agent systems requires explicit reasoning about **when** communication occurs and **where** it is situated, because communication is rarely an isolated artifact. Instead, it is embedded within environment dynamics, task structure, and agent interaction schedules. Unlike single-agent evaluation, where outputs can be assessed independently, multi-agent evaluation must consider how information exchange unfolds over time and how the environment exposes or constrains that exchange.

Crucially, the choice of evaluation environment determines **where** communication can happen, such as text channels, embodied actions, or shared state, and this in turn determines **when** communication becomes observable, necessary, or consequential. As a result, benchmarks implicitly encode assumptions about communication timing even when communication itself is not an explicit evaluation target.

**Planning and Collaboration Benchmarks**   Planning-oriented benchmarks such as Collab-Overcooked Sun et al. (2025a) and large-scale coordination tasks Chen et al. (2024g) emphasize joint planning and cooperative execution, making **when** communication occurs tightly coupled to plan formation. In these environments, agents typically communicate before acting to align goals, assign roles, or synchronize plans. Compared to benchmarks that emphasize execution-time adaptation, these settings privilege early, front-loaded communication and implicitly assume that most coordination can be resolved prior to action. The **where** dimension in these benchmarks is usually a structured interaction loop with explicit planning and execution phases. Metrics such as task completion rate and efficiency indirectly reward timely communication but do not distinguish whether communication was minimal, redundant, or delayed. As a result, these benchmarks underrepresent the value of corrective or opportunistic communication that occurs during execution rather than at planning time.

**Embodied Interaction and Navigation Benchmarks**   In contrast to planning-centric benchmarks, embodied benchmarks such as AI2-THOR Kannan et al. (2023) and C-WAH and TDW-MAT Zhang et al. (2023b) shift attention from plan formation to real-time perception–action coupling, where **when** communication occurs is tightly constrained by physical dynamics. Agents must decide whether to communicate

before acting to coordinate intent or after acting to repair mistakes, introducing a continuous tradeoff between proactive and reactive communication. Here, **where** communication is embedded within action sequences rather than isolated dialogue turns. Evaluation metrics such as task success and efficiency implicitly penalize late communication, as mistimed exchanges directly translate into physical inefficiencies. Compared to symbolic planning benchmarks, embodied environments more clearly expose the cost of delayed or insufficient communication, even though they rarely measure communication timing explicitly.

**Social Inference and Strategic Communication Benchmarks** Unlike both planning-oriented and embodied benchmarks, social reasoning benchmarks such as Werewolf Arena Bailis et al. (2024) and Avalon Stepputtis et al. (2023) externalize the decision of **when** to communicate by enforcing discrete, turn-based discussion and voting phases. Communication timing is therefore no longer adaptive but institutionally imposed, shifting the learning problem from deciding *when* to communicate to deciding *what* to communicate at strategically critical moments. The **where** aspect is constrained to shared text-based dialogue channels, enabling controlled evaluation of belief modeling, persuasion, and deception. Metrics such as win rate and role identification accuracy reflect whether communication was effective at specific moments rather than whether agents communicated frequently. Compared to continuous environments, these benchmarks allow clearer attribution between communication timing and outcomes but are limited to stylized social interactions with fixed temporal structure.

**Long-Horizon and Open-Ended Environments** Moving beyond fixed temporal structures, long-horizon environments such as MindCraft Bara et al. (2021) and MineLand Yu et al. (2024) make **when** communication occurs an explicit strategic choice over extended time scales. Agents must decide whether to communicate early to establish shared understanding or delay communication until uncertainty, conflict, or divergence emerges, introducing temporal credit assignment challenges absent from short-horizon or turn-based benchmarks. In these settings, **where** communication is distributed across persistent world states and repeated interactions, which obscures individual communication events. Metrics such as survival rate or resource accumulation capture aggregate performance but fail to reveal how communication timing contributed to coordination. Compared to episodic benchmarks, these environments highlight the need for temporally sensitive evaluation metrics that can attribute long-term outcomes to earlier communication decisions.

**Large-Scale Swarm and Infrastructure Benchmarks** At even larger scales, swarm and infrastructure benchmarks such as SwarmBench Ruan et al. (2025) and Smart Streetlight IoT Nascimento et al. (2023a) introduce scalability as the dominant constraint shaping **when** communication occurs. Unlike small-team or long-horizon environments where communication supports shared understanding, these benchmarks require agents to communicate selectively, often only when local decisions have system-level consequences. The **where** dimension is decentralized and localized, limiting global observability of communication patterns. Metrics such as coordination efficiency and energy usage indirectly reflect communication timing decisions but do not isolate communication events Zhu et al. (2025); Chen et al. (2024c). Compared to small-team benchmarks, these environments foreground tradeoffs between communication frequency, scalability, and robustness, emphasizing that effective systems must learn not only when to communicate, but also when *not* to.

Table 14: Multi-agent environment benchmarks and their characteristics.

| Environment | #Agents | Observation Space | Action Space | Evaluation Method |
|---|---|---|---|---|
| TextStarCraft II Ma et al. (2023) | 2 | partial; text-based | discrete (macro/micro text commands) | win rate; population ratio; resource ratio |
| LLM-Coordination Agashe et al. (2023) | 2 | partial; text-based | discrete (text-based commands) | score; ToM accuracy; planning accuracy |
| C-WAH and TDW-MAT Zhang et al. (2023b) | 2 | partial; RGB-D | discrete (navigate; interact; communicate) | task completion rate; efficiency |
| AI2-THOR Kannan et al. (2023) | 4 | partial; symbolic | discrete (primitive API calls) | success rate; task completion; goal recall |
| GAIA Mialon et al. (2023) | no limit | partial; multi-modal) | discrete (tool use; browse; reason) | answer accuracy |
| BoxNet, Warehouse, BoxLift Chen et al. (2024g) | 32 | full; symbolic; grid-based | discrete (move; pick; place) | success rate; token usage; steps taken |
| CuisineWorld Gong et al. (2023) | 4 | full; text-based | discrete (move; get/put; use tool) | task score; efficiency |

| Environment | #Agents | Observation Space | Action Space | Evaluation Method |
|---|---|---|---|---|
| RoCoBench Mandi et al. (2024) | 3 | partial; symbolic | discrete (move; pick; place) | success rate; coordination |
| GOVSIM Piatti et al. (2024) | 5 | partial; text-based | discrete | survival rate; efficiency; equality |
| VillagerBench Dong et al. (2024) | 3 | partial; text-based | discrete (move; place; craft; manage) | completion rate; efficiency; workload balance |
| Smart Streetlight IoT Nascimento et al. (2023a) | 100 | partial; simulated sensory inputs | discrete (switch; adjust; communicate) | fitness score; latency; energy usage |
| Werewolf Arena Bailis et al. (2024) | 8 | partial; text-based | discrete (vote; act; communicate) | win rate; strategy quality; gameplay metrics |
| GraphInstruct Hu et al. (2024c) | no limit | partial; graph-based (local neighborhood) | discrete (message passing) | task accuracy |
| Agent Hospital Li et al. (2024c) | 100 | partial; text-based | discrete (diagnose; prescribe; examine) | diagnosis accuracy; MedQA score |
| LLMArena Chen et al. (2024c) | 5 | partial; text-based | discrete (game moves; communicate) | TrueSkill score; win rate; skill metrics |
| BOLAA Liu et al. (2023a) | 3 | partial; text-based | discrete (web navigation; query tools) | final reward; intermediate recall |
| Avalon (Role Identification) Stepputtis et al. (2023) | 6 | partial; text-based | discrete (vote; propose; communicate) | win rate; role identification accuracy |
| SwarmBench Ruan et al. (2025) | no limit | partial; grid-based | discrete (move; communicate) | success rate; coordination metrics |
| Collab-Overcooked Sun et al. (2025a) | 2 | partial; grid-based | discrete (move; pick; place; interact) | dishes delivered; efficiency |
| BattleAgentBench Wang et al. (2024d) | 4 | partial; grid-based | discrete (move; shoot) | mission progress; action accuracy; mission score |
| PARTNR Chang et al. (2024) | 2 | partial; 3D | discrete (navigate; manipulate; instruct) | success rate; steps vs human; error recovery |
| MultiAgentBench Zhu et al. (2025) | no limit | partial; text-based | discrete (task-dependent actions; communicate) | task completion; milestone KPIs; collaboration quality |
| MindCraft Bara et al. (2021) | 2 | partial; 3D | discrete (move; build; communicate) | ToM accuracy; task success |
| MineLand Yu et al. (2024) | 48 | partial; 3D | discrete (move; gather; build; communicate) | survival rate; resource collection; social dynamics |
| MeltingPot Agapiou et al. (2022) | 8 | partial; multi-modal | discrete (move; tag; collect) | episode return; generalization to novel partners |

Table 14 provides a cross-sectional view of representative multi-agent benchmarks, revealing how environment design fundamentally shapes what aspects of communication can be evaluated. Across these benchmarks, variation in agent scale, observability, action abstraction, and evaluation metrics leads to markedly different assumptions about *how*, *when*, and *why* agents communicate.

At a high level, the environments in Table 14 span four broad benchmark regimes. **Small-team, task-oriented environments** (e.g., Collab-Overcooked, AI2-THOR, RoCoBench, PARTNR) emphasize tight coordination under partial observability, where communication is closely coupled to action timing and task progress. **Social and strategic games** (e.g., Werewolf Arena, Avalon, LLMArena) impose discrete interaction rounds that externalize communication timing and enable clearer attribution between messages and outcomes. **Long-horizon and open-ended worlds** (e.g., MindCraft, MineLand, MeltingPot) evaluate communication over extended temporal scales, where timing decisions are strategic but difficult to isolate. Finally, **large-scale swarm and infrastructure benchmarks** (e.g., Smart Streetlight IoT, SwarmBench, MultiAgentBench) foreground scalability constraints, forcing communication to be selective and localized.

A key trend evident from the table is that **observation space and action abstraction jointly determine how visible communication timing is to the evaluator**. Benchmarks with symbolic or text-based observations and discrete actions (e.g., TextStarCraft II, VillagerBench, GOVSIM) make communication events explicit but often abstract away physical constraints. In contrast, embodied and 3D environments (e.g., C-WAH, AI2-THOR, MineLand) embed communication within perception–action loops, where delayed or mistimed messages directly manifest as execution failures, yet are rarely measured explicitly as communication errors.

Agent scale further shapes evaluation. Small-team benchmarks allow fine-grained analysis of pairwise or group communication, but may overfit to dense interaction patterns. As the number of agents increases (e.g., Agent Hospital, Smart Streetlight IoT, SwarmBench), communication overhead becomes a dominant concern, and benchmarks increasingly rely on system-level metrics such as efficiency, energy usage, or fairness. While these metrics reflect the consequences of communication decisions, they obscure causal links between specific communication events and outcomes.

Taken together, Table 14 highlights a fundamental evaluation gap: most benchmarks assess *task performance* under communication, rather than communication itself. Discrete, turn-based environments enable clearer attribution but are limited in realism, while continuous and large-scale environments capture realistic constraints but lack metrics that disentangle communication timing, content, and necessity. This observation motivates the need for benchmark designs and evaluation protocols that explicitly expose and measure communication structure across diverse environments.

### 7.4 Why and How LLM Agents Communicate

### 7.4.1 Structure of Communication

The structure of communication in multi-agent systems reflects both **why agents need to communicate** and **how communication is operationalized** to support coordination, reasoning, and task execution. Rather than being a purely architectural choice, interaction structure encodes assumptions about task coupling, uncertainty, error tolerance, and coordination cost. Different structural designs arise from different motivations for communication, and these motivations directly shape how interaction unfolds in practice. **Table 15 provides a consolidated view of these design choices, linking communication motivations to structural patterns and their associated trade-offs.**

Formally, communication in a multi-agent system can be modeled as a time-indexed signaling process. At timestep $t$, each agent $i$ emits a message $m_i^t \in \mathcal{M}$ according to a communication policy $m_i^t \sim C_i(o_i^t, h_i^{t-1})$, where $o_i^t$ is the local observation and $h_i^{t-1}$ summarizes past messages and actions. Actions are then selected as $a_i^t \sim \pi_i(o_i^t, \{m_j^t\}_{j \neq i})$, making communication an explicit information channel that shapes the joint decision process. **Single-turn communication** corresponds to the special case where $C_i$ is evaluated only once (e.g., at $t = 0$) and $h_i^{t-1} = \emptyset$, so messages do not adapt to others' responses. **Multi-turn communication** allows repeated message generation over $t = 1, \ldots, T$, enabling belief updates and iterative alignment through feedback loops. In **grounded and embodied communication**, message generation and action selection are tightly coupled in time, with $m_i^t$ constrained to influence immediate or near-term actions $a_i^t$ under execution and latency constraints.

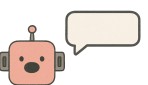

Figure 24: Single-turn

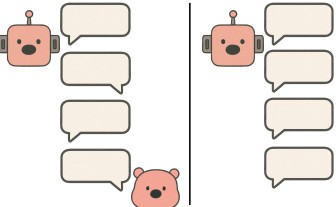

Figure 25: Multi-turn

**Single-Turn Communication: Aggregation-Driven Coordination**  Single-turn communication is adopted when the primary **why to communicate** is to improve robustness, diversity, or solution quality through aggregation rather than explicit coordination. In this setting, agents do not need to adapt their behavior based on others' intermediate states. As a result, **how communication is structured** emphasizes minimal interaction, low latency, and independence across agents. A representative example is *More Agents Is All You Need* Li et al. (2024d), where multiple independently sampled agents generate solutions without inter-agent interaction, and communication occurs only implicitly through a final voting or aggregation step. Compared to interactive approaches, this design reduces coordination overhead and avoids error propagation across rounds, while still improving performance through ensemble effects. However, because communication is non-adaptive, single-turn designs are limited in their ability to resolve ambiguity, handle dependencies, or correct mistakes during execution. **This aggregation-driven design pattern and its limitations are summarized in Table 15.**

**Multi-Turn Communication: Iterative Alignment and Coordination**  Multi-turn communication arises when the **why to communicate** extends beyond one-shot information sharing to resolving uncer-

tainty, managing inter-agent dependencies, or sustaining coordination over time. In these settings, communication is not an auxiliary signal but an iterative control mechanism: agents must repeatedly exchange messages to align beliefs, refine shared artifacts, or adapt plans as new information becomes available. Consequently, the **how to communicate** is shaped by explicit interaction loops that support feedback, revision, and convergence rather than single-pass transmission. **(1) Artifact-centric collaboration systems** represent a first major class of multi-turn communication. In multi-agent software engineering, AutoML pipelines, and workflow-oriented design assistants, such as automated coding and toolchain systems Hu et al. (2024a); Du et al. (2024); Trirat et al. (2024); Chen et al. (2023a); Wang et al. (2024b), project-level orchestration frameworks AL et al. (2024); Fourney et al. (2024); Ahn & Sentis (2021), and scalable task decomposition environments Chen et al. (2024g); Liu et al. (2024b), the primary **why** for communication is iterative refinement of shared artifacts. This motivates a **how** characterized by staged exchanges, versioning, and role-based feedback cycles. Compared to single-turn pipelines, these systems improve solution quality and robustness through repeated correction, but they also incur higher coordination cost and require explicit mechanisms to manage dependency tracking and convergence. **(2) Interactive reasoning and collective sense-making systems** form a second category, where the **why to communicate** is belief alignment or hypothesis refinement rather than artifact production. Multi-round dialogue enables agents to react to intermediate reasoning steps, challenge assumptions, and incorporate diverse perspectives, as seen in collaborative reasoning benchmarks and social inference settings Lan et al. (2023); Xu et al. (2023c); Wu et al. (2023a); Händler (2023); Feng et al. (2024); Li et al. (2024a); Chen et al. (2024c); Ghafarollahi & Buehler (2024). This motivation leads to a **how** that emphasizes turn-taking, explicit reasoning traces, and iterative updates, improving robustness over one-shot aggregation methods. A recurring insight across these works is that allowing agents to revise earlier conclusions through dialogue is often more effective than increasing model capacity or ensemble size alone. **(3) Debate-oriented and adversarial interaction frameworks** further specialize multi-turn communication when the **why to communicate** includes persuasion, disagreement resolution, or strategic influence. Systems based on structured debate, critique, and reconciliation Jin et al. (2024); Li et al. (2024f); Chen et al. (2023b); Du & Zhang (2024); Wang et al. (2024c); Yin et al. (2023); Xiong et al. (2023) motivate a **how** centered on adversarial turn structure, selective disclosure, and explicit challenge–response dynamics. Compared to cooperative dialogue, these designs improve performance under conflicting objectives and reduce correlated errors, but they also introduce higher variance and instability, as communication becomes sensitive to ordering, rhetoric, and strategic withholding of information. Across these categories, a unifying pattern emerges: as the **why to communicate** shifts from information sharing to alignment, correction, or influence, the **how** necessarily evolves from single-turn messages to structured, multi-turn interaction protocols. While multi-turn communication consistently improves robustness and adaptability, the literature also reveals a shared trade-off: greater interaction depth increases coordination overhead and amplifies the need for stopping criteria, memory management, and convergence guarantees. **Table 15 highlights how these motivations map to iterative structures and their coordination trade-offs.**

**Grounded and Embodied Communication: Execution-Oriented Interaction** Grounded and embodied communication arises when the **why to communicate** is driven by real-time execution constraints rather than abstract reasoning or offline planning. In navigation, manipulation, and teamwork tasks, agents must synchronize actions under spatial, temporal, and perceptual uncertainty. As a result, the **how to communicate** prioritizes context-sensitive, temporally aligned exchanges that directly support execution, error recovery, and action coordination. **(1) Execution-driven coordination systems** constitute a first category, where communication is motivated by immediate action alignment. In simulated and embodied environments—such as cooperative task execution and multi-agent control settings Sun et al. (2025c); Piatti et al. (2024); Wu et al. (2025); Cheng et al. (2024)—agents communicate before or during action execution to synchronize movement, allocate subtasks, or resolve local conflicts. This **why** leads to a **how** characterized by frequent, low-latency exchanges tied closely to perception–action loops, improving task success compared to static or single-turn designs. **(2) Socially grounded and interactive environments** extend execution-oriented communication into settings where physical or simulated actions interact with social dynamics. In generative agent societies and interactive simulations Hua et al. (2023); Xu et al. (2023b;a); Wu et al. (2024a), agents communicate not only to coordinate actions but also to negotiate space usage, adapt to others' behavior, or maintain social coherence during execution. Here, the **how** reflects a blend of action-grounded signals

and lightweight social reasoning, enabling agents to adapt to dynamic, multi-agent contexts at the cost of increased communication complexity. **(3) Tool-augmented and hybrid execution frameworks** further enrich grounded communication by integrating external tools or shared representations. Systems combining embodied action with tool use or shared knowledge structures Zhuge et al. (2024a); Li et al. (2023c) motivate a **how** that interleaves execution-level messages with higher-level coordination cues. This hybrid structure allows agents to adapt strategies during execution while preserving responsiveness, though it introduces additional overhead in managing when to switch between action-centric and abstraction-centric communication. Across grounded and embodied settings, a consistent pattern emerges: as the **why to communicate** shifts toward real-time execution and coordination, communication becomes more frequent, localized, and tightly coupled to action timing. Compared to reasoning- or planning-driven domains, these systems trade communication efficiency for adaptability and safety, highlighting a fundamental design tension between minimizing overhead and maintaining robust execution under uncertainty. **This execution-oriented communication regime is contrasted with single- and multi-turn designs in Table 15.**

Table 15: Communication structure in multi-agent systems: motivations, structural choices, and trade-offs.

| Structure | Why Agents Communicate | How Communication Is Structured (and Trade-offs) |
|---|---|---|
| **Single-turn** | • Improve robustness or diversity via aggregation: Li et al. (2024d); 
 • Avoid inter-agent dependency tracking or belief alignment; 
 • Suitable when coordination is unnecessary or harmful. | • Independent generation followed by voting or selection: Li et al. (2024d); 
 • Minimal latency and coordination overhead; 
 • Limited ability to resolve ambiguity or correct errors during execution. |
| **Multi-turn** | • Resolve uncertainty and align beliefs: Lan et al. (2023); Xu et al. (2023c); Feng et al. (2024); 
 • Manage inter-agent dependencies and shared artifacts: Hu et al. (2024a); Du et al. (2024); Trirat et al. (2024); 
 • Support persuasion or disagreement resolution: Jin et al. (2024); Chen et al. (2023b). | • Explicit interaction loops with feedback and revision: AL et al. (2024); Fourney et al. (2024); Chen et al. (2024g); 
 • Turn-taking, staged workflows, or debate protocols: Li et al. (2024f); Du & Zhang (2024); 
 • Improved robustness and adaptability at the cost of coordination overhead and convergence control. |
| **Grounded & Embodied** | • Real-time execution and action synchronization: Sun et al. (2025c); Piatti et al. (2024); 
 • Error recovery under spatial and temporal constraints: Wu et al. (2025); Cheng et al. (2024); 
 • Coordination driven by perception–action coupling: Hua et al. (2023); Xu et al. (2023b). | • Frequent, low-latency, context-sensitive exchanges tied to execution: Sun et al. (2025c); Wu et al. (2025); 
 • Communication aligned with action boundaries and state changes; 
 • Improved safety and adaptability, but higher bandwidth and timing sensitivity: Zhuge et al. (2024a); Li et al. (2023c). |

**Analysis and Implications** Table 15 reveals a consistent organizing principle across multi-agent systems: the structure of communication is not an arbitrary architectural choice, but a direct consequence of the underlying **motivation for communication**. When the goal is robustness through diversity, systems favor single-turn aggregation with minimal coordination overhead. As the motivation shifts toward uncertainty reduction, dependency management, or joint reasoning, multi-turn structures emerge to support feedback, revision, and convergence. In grounded and embodied settings, where communication directly affects physical execution, interaction becomes tightly coupled to perception–action loops, prioritizing responsiveness and safety over efficiency. Importantly, the table highlights a shared trade-off across all regimes. Richer communication structures improve adaptability, error correction, and coordination, but incur higher costs in bandwidth, latency, and convergence control. Conversely, lightweight structures scale well and are easier to stabilize, but struggle with ambiguity and dynamic dependencies. This perspective suggests that future

MA-Comm systems should move beyond selecting a single communication structure, and instead design *adaptive or hybrid mechanisms* that modulate interaction structure based on task phase, uncertainty level, and execution risk.

### 7.4.2 Why to Communicate: Static and Dynamic Communication Protocols

This subsection analyzes communication protocols through the joint lenses of **why agents communicate** and **how communication is operationalized**. These dimensions are inseparable: the motivation for communication determines whether interaction can be predetermined or must remain adaptive, while the protocol structure constrains which motivations can be effectively supported. We group static and dynamic protocols together because rigidity versus adaptability is not an abstract architectural preference, but a direct response to task uncertainty, environmental volatility, and coordination demands.

The **where** dimension plays a mediating role in this discussion. Communication channels, observability, and interaction interfaces determine how motivations translate into concrete protocol behavior. Fixed workflows and well-defined interfaces favor static protocols, whereas open, partially observable, or socially complex environments necessitate dynamic communication to ensure timely and context-aware exchanges.

**Static Communication Protocols**   Static communication protocols arise when the answer to **why communicate** is stable and known in advance. Agents communicate primarily to transmit predefined information required by a fixed workflow, and communication is triggered at predetermined stages rather than by runtime uncertainty. **(1) Sequential and pipeline-based protocols** exemplify this design. Chain-based prompting and sequential reasoning frameworks Wu et al. (2022); Xiao et al. (2023) fix **how** communication unfolds as a unidirectional information flow, where each agent processes an input and passes a structured output downstream. This protocol is motivated by a procedural **why**: decomposing a task into ordered subtasks. Such designs maximize reproducibility and interpretability, but assume early communication is sufficient and cannot correct upstream errors once execution begins. **(2) Fixed-turn debate and roundtable protocols** adopt static structure for a different reason. In structured debate settings Li et al. (2024f), the **why** of communication is to expose diverse reasoning paths rather than to coordinate actions. Consequently, **how** communication is implemented relies on predetermined turn-taking and message formats. Compared to pipelines, these protocols allow richer content, but remain temporally rigid and insensitive to evolving task context. **(3) Workflow-driven applied systems** further illustrate static protocols motivated by procedural coordination. In systems such as ChatDev Qian et al. (2024a), AutoML pipelines Trirat et al. (2024), and large-scale software agent frameworks Schmidgall et al. (2025), agents communicate to exchange specifications, intermediate artifacts, or validation results according to a predefined plan. This improves scalability and debuggability, but limits adaptability when task requirements change mid-execution. **(4) Graph-structured and dependency-driven frameworks** formalize static communication by fixing interaction graphs Qian et al. (2024b); Hu et al. (2024c); Yang et al. (2025). Here, the **why** for communication is localized dependency resolution, and the **how** is constrained by graph topology. While this supports parallelism and scalability, it restricts cross-cutting information flow and makes late-stage adaptation costly.

**Dynamic Communication Protocols**   Dynamic communication protocols emerge when the answer to **why communicate** cannot be determined a priori. Instead, communication is triggered by uncertainty, partial observability, strategic interaction, or evolving goals, requiring the **how** of communication to remain flexible and responsive at runtime. **(1) Social and role-driven interaction systems** represent a first class of dynamic protocols. Generative agent societies and role-playing environments Park et al. (2023); Xu et al. (2023b); Cheng et al. (2024) motivate communication through social context, memory updates, and changing interpersonal states. This **why** leads to a **how** based on open-ended dialogue, memory retrieval, and adaptive response generation rather than fixed pipelines. **(2) Embodied and interactive environments** further amplify the need for dynamic protocols. In embodied task settings Zhang et al. (2023b); Dong et al. (2024); Guo et al. (2024c); Wu et al. (2025), communication is motivated by perception errors, coordination failures, or environmental change. As a result, **how** communication occurs becomes tightly coupled to action timing, enabling plan repair and synchronization. Compared to static workflows, delayed communication in these settings directly degrades performance. **(3) Competitive and mixed-motive**

**environments** impose additional pressure on protocol adaptability. In hidden-role games and adversarial benchmarks Bailis et al. (2024); Wang et al. (2023b); Light et al. (2023); Ma et al. (2023), the **why** of communication includes persuasion, deception, and strategic signaling. Consequently, the **how** must adapt message content, timing, and recipients based on opponent behavior, making static protocols ineffective. **(4) Human–agent collaboration systems** highlight a complementary motivation. In interactive reasoning and feedback-driven settings Feng et al. (2024); Du et al. (2024); Du & Zhang (2024); Chen et al. (2023b), agents communicate to align with human intent or incorporate guidance at critical decision points. This motivates a **how** centered on interruptible reasoning, negotiation, and iterative refinement rather than one-shot exchange. **(5) Large-scale and self-organizing systems** push dynamic protocols toward selective and emergent communication. In scalable coordination frameworks and benchmarks Chen et al. (2024g); Zhuge et al. (2024a); Chen et al. (2024c); Hu et al. (2024a), communication must be sparse to avoid overload, motivating role specialization and partial observability. Fully self-organizing systems Liu et al. (2024b); Ishibashi & Nishimura (2024); Liu et al. (2023a); AL et al. (2024); Jinxin et al. (2023) further relax constraints, allowing communication structures themselves to emerge over time.

Table 16: Static vs. dynamic communication protocols, illustrating the motivations (**why**) and operational mechanisms (**how**) identified in Section 7.4.2.

| Protocol Type | Why Communicate | How Communication is Structured |
|---|---|---|
| **Static protocols** | | |
| | • Predefined task decomposition and ordered subtasks: sequential reasoning pipelines Wu et al. (2022); Xiao et al. (2023)
• Exposure of diverse reasoning paths without runtime adaptation: fixed-turn debates Li et al. (2024f)
• Procedural exchange of artifacts in stable workflows: software, AutoML, and engineering pipelines Qian et al. (2024a); Trirat et al. (2024); Schmidgall et al. (2025)
• Local dependency resolution in known graphs or workflows Qian et al. (2024b); Hu et al. (2024c); Yang et al. (2025) | • Fixed execution order or pipelines with no backtracking Wu et al. (2022); Xiao et al. (2023)
• Predetermined turn-taking and message formats Li et al. (2024f)
• Stage-based handoffs between predefined roles Qian et al. (2024a); Trirat et al. (2024)
• Static interaction graphs limiting cross-cutting information flow Qian et al. (2024b); Hu et al. (2024c) |

| Protocol Type | Why Communicate | How Communication is Structured |
|---|---|---|
| **Dynamic protocols** | | |
| | • Social context evolution and memory updates in open-ended interaction Park et al. (2023); Xu et al. (2023b); Cheng et al. (2024) | • Event-driven, open-ended dialogue with memory and context tracking Park et al. (2023); Xu et al. (2023b) |
| | • Coordination under partial observability and perception uncertainty in embodied settings Zhang et al. (2023b); Dong et al. (2024); Guo et al. (2024c); Wu et al. (2025) | • Action-timed communication tightly coupled to perception–action loops Zhang et al. (2023b); Guo et al. (2024c) |
| | • Strategic persuasion, deception, and opponent modeling in competitive or mixed-motive environments Bailis et al. (2024); Wang et al. (2023b); Light et al. (2023); Ma et al. (2023) | • Adaptive message timing, recipients, and content based on opponent behavior Bailis et al. (2024); Wang et al. (2023b) |
| | • Human–agent alignment, feedback incorporation, and intent clarification Feng et al. (2024); Du et al. (2024); Du & Zhang (2024); Chen et al. (2023b) | • Interruptible reasoning, negotiation, and iterative refinement Feng et al. (2024); Chen et al. (2023b) |
| | • Selective coordination and emergence in large-scale or self-organizing systems Chen et al. (2024g); Zhuge et al. (2024a); Chen et al. (2024c); Liu et al. (2024b); Ishibashi & Nishimura (2024) | • Sparse, selective, and emergent communication structures at scale Liu et al. (2024b); Ishibashi & Nishimura (2024) |

**Comparative Perspective** Across these studies, static protocols prioritize efficiency, predictability, and control by assuming that the **why** of communication is known in advance. Dynamic protocols embrace uncertainty, allowing the **how** of communication to be determined at runtime. This distinction reflects a fundamental trade-off between coordination overhead and adaptability, summarized in Table 16. Crucially, this trade-off is mediated by **where** communication occurs. Fixed workflows and fully observable environments favor static designs, whereas open, embodied, or socially complex environments demand dynamic protocols to ensure timely and context-aware information exchange. Organizing protocol design around why, how, and where thus reveals static versus dynamic communication as an environmental response rather than a purely architectural choice.

**Analysis** Table 16 clarifies that the distinction between static and dynamic communication protocols is driven primarily by **why agents need to communicate**, rather than by architectural or implementation convenience. When task structure is stable and dependencies are known in advance, static protocols align naturally with procedural coordination needs, offering efficiency, interpretability, and ease of scaling. In contrast, as uncertainty, partial observability, social interaction, or strategic behavior increase, dynamic protocols become necessary to support context-sensitive, event-driven communication that cannot be scheduled a priori. A key insight emerging from the table is a recurring design tension across domains: while dynamic protocols consistently improve robustness, adaptability, and error recovery, they introduce higher coordination overhead and amplify the need for filtering, prioritization, and convergence control. This trade-off suggests that future MA-Comm systems should move beyond a binary choice between static and dynamic designs, and instead adopt **hybrid communication protocols** that adapt interaction structure based on task phase, uncertainty level, and coordination demands. Such hybrid designs offer a principled pathway to balance efficiency and flexibility across diverse multi-agent settings.

### 7.5  Discussions

### 7.5.1  Who Communicates with Whom and What Is Shared

The first foundational dimension in LLM-based multi-agent communication concerns **who** participates in communication, **whom** messages are directed toward, and **what** information is exchanged. Across the surveyed systems, these design choices are not incidental but tightly coupled to task structure, coordination requirements, and system scalability. Unlike emergent language settings, where communication channels are typically constrained and learned from scratch, LLM-based systems inherit a rich linguistic prior, shifting the focus from inventing symbols to structuring interaction.

Early LLM multi-agent frameworks primarily adopt small, fixed agent sets with clearly defined roles, resulting in explicit and often static **who**–**whom** relationships. Centralized and hierarchical systems constrain **who** can speak to **whom** through orchestrators or planners, while decentralized and debate-based systems allow all agents to address each other freely. These choices directly shape **what** is communicated. Pipeline-style systems emphasize task instructions, intermediate artifacts, and execution summaries, whereas decentralized or debate-based systems encourage argumentative reasoning, persuasion, and belief updates.

As agent populations scale, several works shift from explicit addressing to implicit aggregation mechanisms, such as shared message pools or memory buffers. In these settings, **whom** a message is intended for becomes ambiguous by design, and **what** is communicated must be general enough to be interpretable by multiple recipients. This transition highlights a core tension in LLM communication: increasing expressiveness and flexibility often comes at the cost of precision and control. Overall, LLM-based communication systems demonstrate that **who**, **whom**, and **what** are jointly determined by architectural constraints rather than learned implicitly, contrasting sharply with emergent language paradigms.

### 7.5.2  When and Where Communication Is Necessary

The second dimension concerns **when** communication is triggered and **where** it occurs within the task or environment. In LLM-based systems, communication is rarely continuous. Instead, it is activated at specific decision points, such as task decomposition, conflict resolution, uncertainty reduction, or coordination checkpoints. This selective invocation reflects both computational cost considerations and the assumption that language is most valuable at moments of ambiguity or coordination failure.

Different application environments impose distinct temporal and spatial pressures on communication. In workflow automation and coding systems, communication typically occurs at stage boundaries, such as requirement analysis, code review, or integration. In embodied or interactive environments, communication is tied to spatial context and temporal urgency, for example during navigation, object handoff, or joint manipulation. Virtual social environments and games further complicate this picture by requiring communication during belief formation, alliance shifts, or adversarial reasoning.

These patterns indicate that **where** agents are situated, whether in symbolic workflows, simulated worlds, or physical environments, directly influences **when** communication becomes necessary. Unlike emergent language benchmarks, where communication frequency is often fixed by experimental design, LLM-based systems increasingly treat communication as a resource to be scheduled. This observation motivates analyzing **when** and **where** jointly, as environmental structure determines not only the availability of communication but also its strategic value.

### 7.5.3  Why Communication Is Used and How It Is Implemented

The final dimension addresses **why** LLM agents communicate and **how** communication mechanisms are realized. In contrast to emergent language, where communication emerges to maximize reward under partial observability, LLM-based communication is typically introduced to compensate for limitations in planning, coordination, or generalization. Communication is used to externalize reasoning, align plans, negotiate roles, or improve robustness in the face of uncertainty.

The **how** of communication reflects these motivations. Static protocols impose fixed interaction patterns that support reliability and reproducibility, particularly in workflows and pipelines. Dynamic protocols, by

contrast, allow agents to adapt communication partners, frequency, and content in response to intermediate outcomes or environmental feedback. Hybrid systems combine both approaches, using structured stages augmented by opportunistic dialogue when conflicts or ambiguities arise.

Importantly, LLM-based systems rarely optimize communication end-to-end. Instead, communication mechanisms are hand-designed and layered on top of pretrained models. This design choice enables rapid deployment but raises questions about efficiency, scalability, and interpretability. Compared to emergent language systems, where communication and policy are jointly learned, LLM-based systems trade optimality for flexibility. Understanding **why** this tradeoff is beneficial and **how** it affects coordination quality remains an open challenge.

Taken together, the **who**–**whom**–**what**, **when**–**where**, and **why**–**how** dimensions reveal that LLM-based multi-agent communication is best understood as a systems design problem rather than a purely learning-driven phenomenon. Communication is introduced deliberately, structured explicitly, and evaluated functionally. While this contrasts with the bottom-up nature of emergent language, it enables practical deployment at scale. Bridging these two paradigms remains a promising direction for future research.

### 7.6 Bridge: LLM-Grounded MARL and Its Limitations

**Purpose of this bridge.** This subsection connects MARL-Comm (Sec. 5) and EL-Comm (Sec. 6) with recent LLM-based multi-agent systems, clarifying *why neither MARL-Comm nor EL-Comm alone fully resolves communication challenges*, and motivating LLM-grounded MARL as a principled intermediate paradigm.

**From MARL-Comm to EL-Comm: Progress and Persistent Gaps** Communication is a fundamental mechanism for collaboration in multi-agent systems, particularly in ad-hoc teamwork settings where agents lack shared protocols or prior agreements Mirsky et al. (2020). **MARL** has shown that agents can learn communication strategies end-to-end by optimizing task rewards, as exemplified by Comm-Net Sukhbaatar et al. (2016) and IC3Net Singh et al. (2018) (reviewed in Sec. 5). However, communication learned purely through task-driven optimization is often highly environment-specific, producing protocols that are difficult for humans or non–co-trained agents to interpret or reuse Kottur et al. (2017); Li et al. (2024b). These limitations motivated **EL** research, which seeks to encourage agents to develop more structured and interpretable communication through interaction. Nevertheless, many **EL-Comm** approaches, particularly referential games trained with symbolic or pixel-level inputs, still exhibit **limited interpretability** and **poor generalization** beyond co-trained agents Lazaridou et al. (2018) (reviewed in Sec. 6). At a deeper level, this challenge stems from how communication is represented and learned: many EL-Comm methods rely on atomic or compositional symbols without explicit semantic grounding, as surveyed in Zhu et al. (2024), making the resulting protocols difficult to interpret or transfer. Although some works explore semantic representations that enable zero-shot EL communication Tucker et al. (2021); Karten et al. (2023), most MARL+EL-based protocols remain task-specific. Moreover, both **MARL-Comm** and **EL-Comm** typically require large amounts of interaction data to overcome joint exploration challenges, reflecting the data-hungry nature of reinforcement learning. Inductive-bias approaches, such as positive signaling and positive listening Eccles et al. (2019), partially alleviate this issue but do not eliminate it. Attempts to further incorporate linguistic structure from human language Tucker et al.; Lazaridou et al. (2020); Lowe et al. (2020); Agarwal et al. (2019a) face fundamental obstacles due to the mismatch between data-intensive RL training and the limited availability of human-annotated interaction data Chaabouni et al. (2019); Lowe et al. (2020). Taken together, these results indicate that learning communication solely through reward-driven interaction, even with emergent structure, leaves a persistent gap between task efficiency, generalization, and human-aligned interpretability.

**Why LLMs Became Attractive for Multi-Agent Communication** These limitations motivated growing interest in **LLMs** as a complementary mechanism for multi-agent communication. LLMs provide access to rich, human-aligned language priors learned from large-scale text corpora, enabling agents to communicate in natural language, express intent, and generalize across tasks without requiring task-specific co-training. Recent instruction-tuned models demonstrate strong performance in cooperative and embodied tasks, producing fluent and context-aware communication Park et al. (2023); Li et al. (2023c). In contrast to MARL-Comm and EL-Comm, LLM-based agents can leverage linguistic structure and world knowledge

that would be infeasible to acquire through interaction alone. However, **LLMs introduce a different class of limitations**. Their language generation is weakly grounded in environment dynamics, which can lead to hallucinated or inconsistent plans that are not executable in real or simulated worlds Mahowald et al. (2024). While attempts have been made to ground LLMs with RL or interactive data collected from environments such as World Models Xiang et al. (2023), Interactive RL Environments Carta et al. (2023), and RL Embodied Environments Tan et al. (2024), these efforts largely focus on single-agent settings and none of them involve teamwork or communication among multiple embodied agents.

**Motivation for LLM–MARL Hybrid Systems** Together, these observations motivate **LLM-grounded MARL** as a bridge between task-optimized communication and language-grounded coordination. Rather than replacing MARL communication with unconstrained language, hybrid systems integrate LLMs as high-level communicative or reasoning components while preserving MARL's grounding in environment dynamics and reward optimization. This hybrid perspective aims to combine the adaptability and interpretability of natural language with the stability and control-oriented guarantees of MARL. At the same time, hybridization exposes unresolved tensions between **linguistic abstraction** (emphasized in EL and LLM-based systems), and **control-oriented grounding** (central to MARL). Understanding how these tensions shape communication behavior is essential for designing reliable multi-agent systems. The remainder of this subsection analyzes how such hybrid systems operationalize communication and what limitations remain.

### 7.6.1 Why and How Communication Emerges in LLM–MARL Hybrid Systems

This part examines **representative LLM–MARL hybrid systems** through the lenses of **why communication is required** and **how communication is operationalized** in our Five Ws Taxonomy. In hybrid systems, communication is not an auxiliary interface but a mechanism shaped directly by environment dynamics, reward structure, and long-horizon optimization constraints. **LLMs** contribute flexible language generation, abstraction, and social reasoning, which explains **why communication is introduced** in many MARL environments: to negotiate, explain intent, reduce coordination uncertainty, or reason under partial observability without exhaustive retraining. **MARL**, in contrast, constrains **how communication is used**: messages must be grounded in policies, influence future actions, or improve long-term reward. As a result, communication in LLM–MARL systems occupies a middle ground between unconstrained natural language interaction and purely latent learned protocols.

Before analyzing individual systems, we situate LLM–MARL hybrids relative to pure MARL and pure LLM-based multi-agent systems along key communication dimensions in Table 17. This comparison clarifies that hybrid systems neither replace learned communication with free-form language nor simply add language on top of MARL; instead, they constrain natural language interaction through policy optimization, shaping both expressiveness and stability.

| Aspect | Pure MARL | Pure LLM-MAS | LLM–MARL Hybrid |
|---|---|---|---|
| Message form | Latent vectors | Natural language | Hybrid (language grounded in policy) |
| Communication driver | Reward optimization | Reasoning and prompting | Reward optimization + reasoning |
| Adaptivity mechanism | Learned end-to-end | Prompt-based adjustment | Policy-constrained adaptation |

Table 17: Comparison of communication mechanisms across pure MARL, pure LLM-based multi-agent systems, and LLM–MARL hybrid architectures.

We now illustrate these design principles through **representative LLM–MARL hybrid systems**, highlighting how different communication motivations lead to distinct structural choices.

**Case Study: Cicero** Cicero (FAIR) exemplifies a setting where the **why to communicate** is driven by strategic alliance formation, persuasion, and long-horizon commitment management in the game of Diplo-

macy. Communication is indispensable because successful play requires agents to coordinate intentions, negotiate agreements, and maintain consistency across many future decision steps under shifting alliances. The **how of communication** in Cicero tightly couples LLM-generated natural language dialogue with MARL-trained strategic policies. The language model generates proposals, explanations, and commitments, while reinforcement learning constrains these utterances through policy optimization to ensure alignment with future actions. As summarized in Table 17, Cicero represents a hybrid communication regime where natural language is explicitly constrained by long-horizon policy optimization. Compared to pure LLM-based negotiators, Cicero stabilizes dialogue by grounding language in reinforcement learning; however, the language and policy modules are trained largely separately and connected through intention estimates, limiting end-to-end alignment between communication learning and control.

**Case Study: LangGround** LangGround Li et al. (2024b) addresses a related but distinct motivation, where the **why to communicate** arises from the need for *human-interpretable and reusable communication* in ad-hoc teamwork scenarios involving unseen teammates and novel task states. Unlike Cicero, the primary objective is not strategic negotiation among agents alone, but alignment between agent communication and human language to support effective human-agent collaboration. The **how of communication** in LangGround differs fundamentally from Cicero by training action and communication *end-to-end* within MARL. Communication policies are jointly optimized using reinforcement learning and supervised grounding losses derived from synthetic interaction data generated by LLM agents. This design aligns the learned communication space with human natural language while preserving reward-driven coordination. As summarized in Table 17, LangGround exemplifies a hybrid architecture where natural language grounding is integrated directly into MARL optimization rather than appended as a separate module. **Compared to Cicero's modular design**, LangGround enables tighter coupling between communication and control, supports zero-shot generalization in ad-hoc teamwork, and facilitates translation between latent embeddings and human-readable language, making communication both task-effective and human-aligned.

**Case Study: Werewolf** *Werewolf* Xu et al. (2023c) represents a contrasting hybrid setting where the **why to communicate** is driven by deception, hidden roles, and belief manipulation rather than explicit coordination or alignment. Communication is essential for influencing other agents' beliefs, obscuring true intentions, and surviving under adversarial and mixed-motive incentives. The **how of communication** in Werewolf reflects this strategic objective through a tight integration of LLM reasoning and reinforcement learning–based decision control. The LLM is used to perform deductive reasoning and generate a diverse set of candidate language actions, helping mitigate intrinsic biases inherited from pretraining. A reinforcement learning policy then selects among these candidates to optimize long-term game outcomes. In this design, language functions as a probabilistic and strategic signal rather than a binding commitment, with RL shaping when agents speak truthfully, deceive, or remain silent across repeated interactions. **Compared to Cicero**, where language is used to form and maintain cooperative commitments aligned with long-horizon planning, communication in Werewolf is adversarial and belief-centric, emphasizing strategic signaling over coordination. **In contrast to LangGround**, which aims to align MARL communication with human-interpretable language for ad-hoc teamwork and generalization to unseen partners, Werewolf does not prioritize interpretability or transferability. Instead, its communication is optimized for in-game strategic advantage under fixed adversarial roles, with reinforcement learning directly regulating the deployment of language to maximize reward. As summarized in Table 17, Werewolf exemplifies a hybrid communication regime where natural language serves as a strategic signaling interface, while reinforcement learning governs action selection and policy consistency under adversarial incentives.

**Case Study: FAMA** FAMA Slumbers et al. (2023) represents a cooperative LLM–MARL hybrid where the **why to communicate** is driven by the need for reliable coordination under partial observability across multiple cooperative tasks. Unlike social or adversarial settings, communication in FAMA is motivated by execution-level coordination: individual agents lack sufficient local information to act optimally without exchanging state summaries, intentions, or intermediate reasoning. The **how of communication** in FAMA reflects this coordination-centric objective through two tightly coupled mechanisms. First, LLM-based agents are aligned with task-specific functional knowledge via a centralized on-policy MARL update rule, ensuring that communication remains reward-aligned and policy-consistent. Second, agents exchange information

using natural language message passing, leveraging LLMs' linguistic strengths to convey task-relevant abstractions in an intuitive and interpretable form. This combination allows FAMA to outperform independent LLM agents and traditional symbolic MARL methods across diverse coordination tasks. **Compared to Cicero**, where communication supports long-horizon negotiation and commitment management, FAMA uses language strictly as a coordination signal rather than a tool for persuasion or alliance formation. **Compared to Werewolf**, where language functions as a strategic and potentially deceptive belief signal, FAMA's communication is cooperative, non-adversarial, and execution-oriented. **Relative to LangGround**, which focuses on aligning MARL communication with human language for ad-hoc teamwork and generalization to unseen partners, FAMA prioritizes functional alignment and task performance within fixed cooperative environments rather than zero-shot human interpretability. As summarized in Table 17, FAMA exemplifies a hybrid communication pattern where natural language serves as an intuitive medium for state and intent sharing, while centralized MARL training constrains communication to support stable, coordinated policy optimization across tasks.

**Case Study: ACC-Collab** ACC-Collab Estornell et al. (2025) studies multi-agent collaboration in complex question-answering and reasoning tasks, where the **why to communicate** is driven by **epistemic uncertainty** and reasoning errors rather than environmental dynamics or partial observability. Individual agents may generate plausible but incorrect reasoning paths, making communication necessary to surface, evaluate, and correct intermediate conclusions through interaction. The **how of communication** in ACC-Collab is realized through an explicit actor–critic dialogue framework inspired by MARL. One agent (the actor) generates candidate reasoning trajectories, while another agent (the critic) evaluates, critiques, and provides feedback, with reinforcement-style updates guiding improvement across iterative dialogue rounds. Unlike **FAMA**, which uses language to support coordination and execution in task environments, ACC-Collab operates in a purely linguistic setting without external state transitions. Unlike **Werewolf**, where language is strategically deceptive and belief-oriented, communication in ACC-Collab is corrective and accuracy-driven. Compared to **Cicero**, which grounds dialogue in long-horizon strategic commitments, ACC-Collab focuses on short-horizon reasoning refinement. In contrast to **LangGround**, which aims to align communication with human-interpretable semantics for ad-hoc teamwork, ACC-Collab emphasizes improving solution quality through structured critique rather than semantic grounding. As summarized in Table 17, ACC-Collab exemplifies a hybrid communication regime in which natural language interaction is structured by MARL-inspired feedback mechanisms, enabling policy-constrained reasoning refinement rather than action coordination or social negotiation.

### 7.6.2 Comparative Pattern Across Hybrid Systems

Table 18 makes clear that LLM–MARL hybrid systems do not share a single communication template; instead, each system *chooses* a communication structure that matches its dominant uncertainty source and incentive landscape. **Cicero** treats language as a **commitment mechanism** for long-horizon coordination: communication must remain **policy-consistent** so that promises map onto executable trajectories. **Werewolf** flips this logic under adversarial incentives: language becomes a **strategic belief signal**, and RL is used to **select** among candidate language actions to counter intrinsic LLM bias and to manage when to reveal, distort, or withhold information. **FAMA** emphasizes cooperative execution under partial observability: communication is primarily a **coordination substrate** for sharing task-relevant state and intent, while centralized MARL updates align the LLM agents toward **functional knowledge** for cooperation. **ACC-Collab** abstracts away the environment entirely and focuses on epistemic uncertainty: communication is structured as **actor–critic critique loops** that improve reasoning quality through iterative feedback, rather than through state sharing or negotiation.

Several design lessons for future LLM–MARL co-design follow directly from these contrasts. **(1) Match the communication role to the objective and uncertainty source.** If the bottleneck is long-horizon coordination, treat language as **commitments** and enforce message-action consistency (Cicero). If the bottleneck is adversarial belief dynamics, treat language as **signals** and let RL learn when and how to deploy them under strategic pressure (Werewolf). If the bottleneck is decentralized execution under partial observability, treat language as **structured state/intent summaries** and couple it to policy learning through coordination-centric updates (FAMA). If the bottleneck is reasoning reliability, treat language as **self-correction infras-**

Table 18: Communication motivations and mechanisms in representative LLM–MARL hybrid systems, highlighting distinct functional roles of language across strategic, adversarial, cooperative, human-aligned, and reasoning-centric settings.

| System | Why Communication Is Needed | How Communication Is Structured |
|---|---|---|
| Cicero (FAIR) | <ul><li>**Strategic commitment**: alliance formation</li><li>Long-horizon coordination</li><li>Managing promises under shifting coalitions</li></ul> | <ul><li>Natural language negotiation</li><li>Policy-consistent dialogue</li><li>RL-constrained commitments aligned with future actions</li></ul> |
| Werewolf Xu et al. (2023c) | <ul><li>**Adversarial belief dynamics**: hidden roles</li><li>Belief manipulation and deception</li><li>Strategic inference under uncertainty</li></ul> | <ul><li>Language as a probabilistic belief signal</li><li>RL-based selection among candidate language actions</li><li>Non-binding, strategically adaptive communication</li></ul> |
| LangGround Li et al. (2024b) | <ul><li>**Human-aligned coordination**: ad-hoc teamwork</li><li>Interoperability with unseen teammates</li><li>Bridging agent protocols and human language</li></ul> | <ul><li>End-to-end MARL training with language grounding</li><li>Alignment between latent embeddings and natural language</li><li>Joint optimization of task reward and linguistic interpretability</li></ul> |
| FAMA Slumbers et al. (2023) | <ul><li>**Cooperative execution**: partial observability</li><li>Task decomposition across agents</li><li>Coordination under execution uncertainty</li></ul> | <ul><li>Centralized on-policy MARL updates</li><li>Natural language message passing</li><li>Functionally aligned, task-relevant abstractions</li></ul> |
| ACC-Collab Estornell et al. (2025) | <ul><li>**Epistemic reliability**: reasoning uncertainty</li><li>Error detection and correction</li></ul> | <ul><li>Actor–critic dialogue roles</li><li>Iterative proposal–critique rounds</li><li>Reward-guided reasoning refinement</li></ul> |

tructure through explicit critique roles and reward-guided refinement (ACC-Collab). **(2) Separate what LLMs are good at from what RL must guarantee.** Across all systems, LLMs provide rich candidate messages and abstractions, while RL is most valuable when it (i) **selects** among candidates to reduce bias (Werewolf), (ii) **constrains** language to remain executable and temporally consistent (Cicero), or (iii) **aligns** communication with cooperative performance objectives (FAMA) and iterative improvement (ACC-Collab). **(3) Prefer structured interfaces when stakes are high.** The more communication must drive control and coordination, the more these systems rely on **structured schedules, roles, or interfaces** (policy-consistency constraints in Cicero, centralized update structure in FAMA, actor–critic roles in ACC-Collab), suggesting that future hybrids should avoid fully free-form chat and instead adopt **protocol-aware natural language**.

Overall, Table 18 supports a unified conclusion: communication in LLM–MARL hybrids is best viewed as a **reward-shaped, protocol-constrained mechanism** rather than a generic language interface. Future LLM–MARL co-design should therefore start from the intended **function of communication** (commitment, signaling, coordination, critique) and then choose the minimal structure—constraints, roles, and update rules—needed to make that function reliable, scalable, and learnable.

### 7.6.3 LLM Limitations in MARL Settings

While LLM-powered agents introduce expressive language, social reasoning, and rapid adaptation, their integration into MARL settings exposes several fundamental limitations that shape both **why** communication is effective and **how** it can be reliably used. These limitations recur across different LLM–MARL systems and evaluation settings, as summarized later in Table 19. **(1) grounding and state alignment remain weak**. In MARL environments, optimal communication is tightly coupled with latent environment states, long-horizon dynamics, and delayed rewards. LLMs, operating primarily on textual context, lack direct access to environment transition models and often rely on imperfect summaries or abstractions provided by other agents. As a result, language-based communication may drift from the true system state, leading to inconsistent coordination or compounding errors over time. This issue is particularly salient when language mediates strategic commitments or hidden information, as in *Cicero* and *Werewolf* (FAIR); Xu et al. (2023c), and has also been observed in broader evaluations of LLM-augmented cooperation Mosquera et al. (2025). **(2) policy consistency and temporal coherence are not guaranteed**. MARL communication protocols are learned jointly with policies, ensuring that messages influence future actions in a predictable and reward-aligned manner. In contrast, LLM-generated messages are typically produced at inference time and may vary across identical states due to stochastic decoding or prompt sensitivity. This variability complicates credit assignment and undermines the stability required for long-horizon coordination. Even when hybrid systems constrain language to align with downstream actions (e.g., *Cicero*), this typically requires substantial scaffolding or task-specific training (FAIR). Recent analyses of multi-agent LLM failures further identify communication inconsistency and robustness as key bottlenecks Cemri et al. (2025a). **(3) scalability and efficiency constraints differ sharply**. MARL communication mechanisms are often explicitly designed to respect bandwidth limits, sparse messaging, and decentralized execution constraints. LLM-based communication, by contrast, incurs high token and computation costs and scales poorly with the number of agents or communication rounds. This limits its applicability in real-time, resource-constrained multi-agent systems and motivates designs that restrict language to high-level negotiation or episodic coordination rather than continuous control signals Slumbers et al. (2023); Mosquera et al. (2025). **(4) theoretical guarantees are largely absent**. Classical MARL communication frameworks can be analyzed through Dec-POMDPs, information bottlenecks, regret bounds, or return-gap guarantees. LLM-based communication currently lacks comparable formal guarantees regarding convergence, optimality, or robustness under partial observability and non-stationarity. This gap is amplified in open-ended language settings where semantics are not optimized directly for control performance Cemri et al. (2025a). **(5) failure modes are difficult to detect and correct**. LLM agents may hallucinate, overgeneralize, or adopt persuasive but incorrect narratives, especially in adversarial or mixed-motive environments. Without explicit coupling to reward signals or environment feedback, such failures can persist undetected and propagate across agents. Mixed-motive dialogue-centric settings such as *Werewolf* make these risks especially clear, since communication can be strategically manipulated Xu et al. (2023c), while broader empirical studies report fragility in cooperation and robustness of LLM-based multi-agent behaviors Mosquera et al. (2025); Cemri et al. (2025a).

Table 19: LLM limitations in MARL settings: manifestations, representative evidence, and practical implications for hybrid LLM–MARL system design.

| Limitation | Manifestation in MARL | Representative Works | Implication for Design |
|---|---|---|---|
| **Grounding & state alignment** | • Language drifts from latent environment state; • Textual summaries omit long-horizon dynamics; • Errors compound over time under partial observability. | • Policy-aligned negotiation with partial grounding (*Cicero*) (FAIR); • Hidden roles and deceptive signaling (*Werewolf*) Xu et al. (2023c); • Empirical evaluation in Melting Pot Mosquera et al. (2025). | • Use structured state interfaces; • Restrict language to high-level intent or episodic coordination. |
| **Temporal coherence & consistency** | • Same state yields different messages; • Unstable influence on downstream actions; • Credit assignment becomes ambiguous. | • Long-horizon commitment alignment requires scaffolding (*Cicero*) (FAIR); • Failure analysis highlights communication inconsistency Cemri et al. (2025a). | • Constrain generation strategies; • Enforce message–action consistency checks; • Incorporate feedback-driven updates. |
| **Scalability & efficiency** | • Token and computation cost scale with agent count; • Communication latency limits real-time control. | • Centralized MARL with language messaging (*FAMA*) Slumbers et al. (2023); • Cooperation fragility at scale Mosquera et al. (2025). | • Prefer sparse, event-triggered language; • Keep low-level control in MARL modules. |
| **Lack of guarantees** | • No convergence guarantees; • Weak robustness under non-stationarity. | • Multi-agent failure analysis emphasizes robustness gaps Cemri et al. (2025a). | • Use MARL for control-critical channels; • Add verification or safety layers. |
| **Hard-to-detect failure modes** | • Hallucination and overgeneralization; • Persuasive but incorrect narratives; • Strategic manipulation propagates across agents. | • Adversarial and mixed-motive signaling (*Werewolf*) Xu et al. (2023c); • Robustness and cooperation failures Mosquera et al. (2025); Cemri et al. (2025a). | • Add grounding and consistency audits; • Use adversarial testing; • Fallback to learned protocols when uncertain. |

Table 19 summarizes these limitations by mapping each failure mode to its typical manifestation in MARL settings, representative systems or empirical studies, and the resulting design implications for hybrid LLM–MARL architectures. Taken together, these limitations indicate that LLMs are best viewed not as replacements for MARL communication, but as complementary components. Hybrid systems benefit most when LLMs are constrained to high-level reasoning, abstraction, or negotiation (e.g., *Cicero*, *ACC-Collab*), while MARL mechanisms retain responsibility for low-level control, temporal consistency, and policy optimization (FAIR); Estornell et al. (2025); Slumbers et al. (2023). From a broader perspective, many of the limitations identified above stem not only from current LLM architectures, but also from a deeper mismatch between **open-ended language generation** and the assumptions underlying **classical decentralized control and communication-constrained decision-making**.

### 7.6.4 LLM Limitations from a Decentralized Control Perspective

Table 20: LLM limitations through a decentralized control lens: each control requirement is paired with representative MARL-Comm mechanisms and empirical evidence on LLM/MAS limitations.

| Decentralized Control Requirement | What It Implies for Communication | Representative MARL-Comm Works (Align With Req.) | LLM/MAS Evidence of Limitation |
|---|---|---|---|
| **Well-defined semantics and grounding** | • Messages encode task-relevant information about latent state;
• Semantics are interpretable under partial observability. | • Dec-POMDP-based modeling Bernstein et al. (2002);
• Learned state sharing via CommNet Sukhbaatar et al. (2016). | • Cooperation may drift from latent state due to weak grounding Mosquera et al. (2025). |
| **Reliability and predictability** | • Signals remain consistent across time;
• Coordination stabilizes under repeated interaction. | • Structured aggregation and attention in TarMAC Das et al. (2018);
• Stable learned coordination in IC3Net Singh et al. (2018). | • Communication inconsistency and brittleness in multi-agent LLM systems Cemri et al. (2025a). |
| **Information efficiency and minimality** | • Messages are compressed and sparse;
• Bandwidth constraints are explicitly respected. | • Information-bottleneck-inspired efficient communication Wang et al. (2020b);
• Selective exchange mechanisms in TarMAC Das et al. (2018). | • Token-heavy communication and scaling fragility Mosquera et al. (2025). |
| **Objective coupling (control optimality)** | • Messages are optimized directly for control performance;
• Communication influences return maximization. | • End-to-end message learning with reward feedback (CommNet, IC3Net) Sukhbaatar et al. (2016); Singh et al. (2018). | • Semantics arise from pretraining rather than control objectives Cemri et al. (2025a). |
| **Robustness under non-stationarity** | • Communication remains effective as policies evolve;
• Coordination resists behavioral drift. | • Decentralized policies with learned communication to mitigate non-stationarity Singh et al. (2018). | • Coordination instability and robustness breakdowns Mosquera et al. (2025); Cemri et al. (2025a). |

From the standpoint of decentralized control and decentralized information theory, communication is not merely an auxiliary interface but a **control-relevant information channel** with well-defined semantics, whose design critically determines system performance Bakule & Papik (2012); Tripakis (2004). Classical team decision theory and communication-constrained control formalize this view by modeling communica-

tion as a structured signaling mechanism subject to partial observability, bandwidth, delay, and noise, and by explicitly characterizing how information flow affects optimal control policies Witsenhausen (1968); Ho et al. (1972); Tatikonda & Mitter (2004). In these formulations, the value of a message is defined by its contribution to state estimation, coordination, and long-horizon control objectives, rather than by linguistic expressiveness.

This perspective is directly reflected in Dec-POMDPs and related decentralized control models, which assume that communicated signals are optimized jointly with control policies and can be analyzed in terms of sufficiency, redundancy, and performance loss (e.g., Dec-POMDP formalizations Bernstein et al. (2002)). Many MARL-Comm mechanisms align naturally with these assumptions: messages are learned end-to-end with policies and shaped by reward feedback, often under explicit efficiency or sparsity constraints. Representative examples include continuous message pooling in CommNet Sukhbaatar et al. (2016), selective attention-based exchange in TarMAC Das et al. (2018), and learned gating or graph-based coordination in IC3Net Singh et al. (2018). Moreover, several MARL-Comm studies explicitly adopt information-theoretic objectives, such as compression, minimality, or information bottlenecks, to control communication cost while preserving task-relevant information Wang et al. (2020b).

In contrast, LLM-based communication departs fundamentally from this control-theoretic paradigm. Language is treated as an **open-ended interface** whose semantics emerge implicitly from large-scale pretraining rather than being optimized for a specific decentralized control objective. As a result, it becomes difficult to formally reason about whether generated messages are sufficient, minimal, or reliable for coordination under partial observability and non-stationarity. Empirical studies of multi-agent LLM systems report failure modes—including inconsistency across time, coordination fragility, and robustness breakdowns—that directly violate predictability and reliability assumptions central to classical decentralized control Mosquera et al. (2025); Cemri et al. (2025a). Furthermore, stochastic decoding and prompt sensitivity introduce unbounded variability, which is incompatible with the stability requirements of long-horizon decentralized decision-making.

These observations suggest a principled division of labor in hybrid systems. LLMs are best integrated as high-level planners, negotiators, or abstraction mechanisms that operate above the control layer, while MARL-based communication remains essential for enforcing decentralized control laws, maintaining temporal coherence, and ensuring robustness under uncertainty. Bridging these paradigms, therefore, requires not only architectural interfaces but also new theoretical tools that reconcile open-ended language with the foundational principles of communication-constrained decentralized control. Table 20 summarizes how key decentralized-control requirements translate into communication desiderata, and how reported LLM multi-agent failure modes violate these desiderata relative to MARL-Comm mechanisms.

# 8 Discussion, Open Problems, and Future Directions

In this section, we provide an in-depth discussion on multi-agent communication (**MARL-Comm**), emergent language in multi-agent systems (**MA-EL**), and multi-agent communication with large language models (**LLM-Comm**). These discussions expand on the main text and outline key challenges, methodologies, and future directions in each area. Our analysis focuses on how communication has been modeled, learned, and optimized within multi-agent systems, providing a comprehensive summary of this paper while offering insights for future research. For clarity, we use the following abbreviations: **MARL-Comm** - Multi-Agent Reinforcement Learning Communication, **MA-EL** - Emergent Language, and **LLM-Comm** - Multi-Agent Communication with Large Language Models.

## 8.1 Discussions on Multi-agent Communications

**A Common Usage of Communication Models in Multi-Agent Systems** Communication models play a crucial role in multi-agent systems by enabling agents to share information, coordinate actions, and enhance decision-making. These models can be broadly categorized based on their structure and learning mechanisms, and they are commonly used in MARL-Comm, MA-EL, LLM-Comm. In MARL, communication models facilitate cooperation by allowing agents to exchange observations, share rewards, or align policies. They can be integrated into centralized learning frameworks, where a global controller aggregates

and distributes information, or into decentralized architectures, where agents autonomously decide whom and when to communicate with. To be specific, the communication process can be formulated as

$$m_{i,j}^{(t)} \sim C\Big(a_i^{(t-1)}, a_j^{(t-1)} \mid s_t\Big), \quad a_i^{(t)} \sim \pi_i\Big(\cdot \mid s_t, \{m_{j,i}^{(t)}\}_{j \neq i}\Big), \quad m_{i,j}^{(t)} \in M,$$

where $m_{i,j}^{(t)}$ represents the message sent from agent $i$ to agent $j$ at time step $t$, $C(\cdot)$ denotes the communication function that generates messages conditioned on the current environment state $s_t$ and the agents' actions from the previous timestep, and $\pi_i(\cdot \mid s_t, \{m_{j,i}^{(t)}\})$ denotes the policy of agent $i$ that selects an action at time $t$ based on its current observation and the set of received messages. This time-indexed formulation makes explicit that communication and action selection are staged across timesteps, thereby avoiding circular dependencies within a single decision epoch. All communication models exhibit unique advantages and trade-offs: (1). **Graph-based communication** encodes agent interactions using predefined or dynamic graphs, allowing for structured message passing but requiring scalability optimizations. (2). **Attention-based communication** uses transformer-based mechanisms to selectively attend to relevant information, improving robustness but increasing computational overhead. (3). **Emergent communication** enables agents to learn their own protocols, enhancing adaptability but often lacking interpretability. (4). **LLM-based communication** leverages pretrained language models to enhance expressiveness but may suffer from grounding issues in task-specific settings. Each of these models provides different capabilities depending on the environment and the desired level of communication efficiency. For instance, graph-based models are well-suited for structured multi-agent interactions, whereas LLM-based approaches excel in human-agent communication and natural language coordination. Future research should focus on integrating adaptive communication mechanisms that dynamically adjust based on task complexity, scalability requirements, and real-time constraints.

**The unique usages of each communication model in multi-agent systems** Each communication model has distinct advantages and is suited for different multi-agent scenarios. (1) **Graph-based communication** structures agent interactions as a communication graph, where nodes represent agents and edges determine message passing. This structured representation is effective for tasks requiring **topology-aware coordination**, such as traffic control and distributed sensor networks. Graph-based models can be either **static**, where communication links are predefined, or **dynamic**, where agents learn whom to communicate with based on relevance and necessity. (2) **Attention-based communication** utilizes mechanisms such as transformers to allow agents to selectively attend to the most relevant messages, improving scalability and robustness. By dynamically weighing information from different sources, attention-based communication enhances **coordination efficiency** in large-scale multi-agent systems while reducing redundancy in message exchange. (3) **Emergent communication** enables agents to develop their own communication protocols through interaction, rather than relying on predefined language or rule-based exchanges. This type of communication is particularly useful in *cooperative tasks* where agents need to establish novel coordination strategies in *zero-shot or few-shot scenarios*. However, ensuring *compositionality and interpretability* remains a challenge. (4) **LLM-based communication** integrates large language models to facilitate natural language-based interactions among agents, making it particularly valuable for *human-agent collaboration* and *generalist AI systems*. LLMs can serve as a *high-level reasoning module*, allowing agents to infer intentions, interpret ambiguous instructions, and align communication with human-like structures. However, challenges include *grounding in task-specific contexts* and *reducing hallucinations in generated messages*. (5) **Adaptive communication policies** leverage reinforcement learning to optimize *when and whom to communicate with*, reducing unnecessary message exchange and improving bandwidth efficiency. These policies are crucial in *scalable multi-agent reinforcement learning (MARL)*, where excessive communication can lead to *overhead and inefficiency*. Future research should explore *hybrid communication architectures* that combine structured message-passing mechanisms with *learned communication strategies*, ensuring scalability, adaptability, and robustness across different multi-agent domains.

**Integrated use of different communication models in multi-agent systems** The unique usages and advantages of each communication model form the basis for integrating different approaches in multi-agent communication, and several works have explored this direction. Here, we highlight some key examples. (1) Graph-based and attention-based communication can be combined to leverage *structured message passing* while dynamically *selecting relevant information*, enhancing scalability in large-scale multi-agent coordination. (2) Emergent communication can be integrated with reinforcement learning to allow agents to *learn*

*communication protocols* while optimizing their policies, enabling more *adaptive and decentralized inter-actions* in complex environments. (3) LLM-based communication can be used as a *high-level reasoning mechanism*, complementing low-level MARL-based message exchange to enable *human-interpretable and goal-oriented* communication. (4) Hybrid models incorporating both explicit and implicit communication strategies allow agents to *switch between direct messaging and behavior-based signaling*, improving efficiency in tasks requiring limited bandwidth or privacy constraints. (5) Adaptive communication policies can be in-corporated into any of these models, allowing agents to *dynamically decide when and whom to communicate with*, balancing message complexity and coordination effectiveness. Future research should further explore the integration of *structured, learned, and language-based communication*, ensuring that multi-agent systems remain robust, interpretable, and scalable across diverse environments.

**A summary of the base communication models in multi-agent systems** The different communica-tion models used in multi-agent systems can be categorized based on their structure and learning mechanisms. (1) In **MARL**, communication strategies can be broadly classified into *explicit* and *implicit* communication. Explicit communication methods include *graph-based* and *attention-based* models, where agents directly ex-change messages to enhance coordination. Implicit communication, on the other hand, relies on agents learning to *infer information from observed behaviors*, reducing bandwidth but requiring stronger inference capabilities. (2) In **emergent communication**, agents develop their own structured protocols through in-teraction, often optimizing for *task-specific coordination* while balancing *generalization and interpretability*. Research has explored different learning objectives, such as maximizing *mutual information*, reinforcement learning rewards, or structured learning constraints to encourage *compositionality and efficiency*. (3) In **large language model-based communication**, pretrained models facilitate agent interactions through *human-interpretable messages*, allowing for seamless integration of *natural language understanding* in multi-agent collaboration. However, ensuring *grounding and alignment* with task-specific requirements remains an open challenge. (4) **Hybrid communication architectures** combine multiple models, such as inte-grating *learned attention mechanisms* with *structured graph-based message passing* or combining *emergent communication* with *LLM reasoning modules*. These approaches aim to balance *interpretability, efficiency, and scalability*. (5) Finally, **adaptive communication policies** allow agents to *dynamically decide when and whom to communicate with*, reducing unnecessary message exchange while maintaining *effective coordi-nation*. Future research should focus on refining the balance between *structured and learned communication approaches*, ensuring that multi-agent systems achieve *robust, interpretable, and scalable communication strategies* across diverse applications.

**Seminal works of communication models in multi-agent systems** The development of communi-cation models for multi-agent systems has followed diverse trajectories, with some models seeing greater adoption in certain domains than others. The impact of a communication model often depends on whether there are seminal works in the category, as many subsequent research efforts extend these foundational studies. Here, we highlight seminal contributions in this area. **Graph-based communication** has been widely studied in reinforcement learning contexts, where agent interactions are modeled as graph structures, and information is propagated through message passing mechanisms. Many works formalize communica-tion as a function over a graph structure: $h_i^{(t+1)} = \phi\left(h_i^{(t)}, \sum_{j \in \mathcal{N}(i)} \psi(h_j^{(t)}, m_{j,i})\right)$, where $h_i^{(t)}$ is the hidden state of agent $i$ at timestep $t$, $m_{j,i}$ is the message received from neighboring agent $j$, and $\phi$ and $\psi$ are learned transformation functions. **Attention-based communication** leverages transformer-based archi-tectures to allow agents to dynamically weigh the importance of messages. This mechanism is particularly effective in large-scale systems where message relevance changes based on task demands. The communi-cation process can be modeled using attention weights: $\alpha_{i,j} = \frac{\exp(f(h_i, h_j))}{\sum_{k \in \mathcal{N}(i)} \exp(f(h_i, h_k))}$, where $\alpha_{i,j}$ represents the attention weight assigned by agent $i$ to agent $j$, and $f(h_i, h_j)$ is a scoring function that determines the relevance of agent $j$'s message. **Emergent communication** models have introduced new paradigms in reinforcement learning, where agents develop their own symbolic language through optimization. Re-search in this area often seeks to maximize the mutual information between the transmitted and received messages: $\max_{\pi_C} I(m; s) = H(m) - H(m|s)$, where $m$ represents the communicated message and $s$ is the observed state. The challenge in emergent communication lies in ensuring compositionality, interpretability, and generalization across different agent populations. **LLM-based communication** introduces pretrained

language models as a foundation for natural communication between agents. Large language models provide agents with rich semantic representations, enabling them to process and generate messages that align more closely with human language. However, challenges such as grounding messages in task-relevant information remain critical: $p(m|s,a) = \frac{\exp(f(s,a,m))}{\sum_{m'} \exp(f(s,a,m'))}$, where $p(m|s,a)$ represents the probability of generating message $m$ given state $s$ and action $a$, and $f(s,a,m)$ is a learned function capturing relevance. Some seminal works have pioneered new paradigms in multi-agent communication. Graph-based communication has been foundational in distributed reinforcement learning, while emergent language research has opened up avenues for self-learned symbolic exchanges. The use of large language models represents a shift toward integrating structured natural language into agent communication. Future research should continue exploring the intersection of these methods, leveraging **hybrid architectures** that integrate structured message-passing with adaptive learning-based communication strategies.

**A summary of the issues and extensions of communication models in multi-agent systems** Communication models in multi-agent systems address various challenges and enable extensions that improve scalability, adaptability, and robustness. Below, we categorize key issues these models aim to solve, along with potential research directions for further improvements. Specifically, six major extensions have been explored: **multi-task (MT)**, **multi-agent (MA)**, **hierarchical (Hier)** learning, **model-based (MB)** communication, *policy safety*, and *policy generalization*. While significant progress has been made, many challenges remain unsolved, requiring further exploration. (1) **Communication efficiency and bandwidth constraints**: Many multi-agent systems operate under limited communication resources, requiring models to minimize message exchange while maintaining effective coordination. Graph-based and attention-based communication mechanisms mitigate unnecessary transmissions by dynamically selecting relevant information. The effectiveness of these approaches is often evaluated through message entropy and communication sparsity: $\min_{\pi_C} H(m|s,a)$ subject to $I(m;s,a) \geq \lambda$, where $H(m|s,a)$ represents the entropy of transmitted messages given state-action pairs, ensuring minimal redundancy while maintaining sufficient information flow through a mutual information constraint $I(m;s,a)$. (2) **Multi-modal and heterogeneous input**: Agents in real-world scenarios must process multiple input modalities, such as vision, language, and structured data. Communication models designed for *multi-modal fusion* use attention-based mechanisms to integrate these diverse signals. A common approach is to embed multiple modalities into a shared latent space before generating communication messages: $z = f_{\text{modality}}(s_1, s_2, \ldots, s_n)$, $m = g(z)$, where $f_{\text{modality}}$ encodes inputs from different modalities $s_1, s_2, \ldots, s_n$, and $g$ generates a structured message $m$. (3) **Generalization to new partners and environments**: Ensuring that communication models generalize beyond their training conditions is an ongoing challenge. Emergent communication systems often struggle with partner adaptation, requiring mechanisms to align agent language representations dynamically. One approach involves optimizing for alignment loss between agents: $\min_{\pi_C} D_{\text{KL}}(p(m_i|s)||p(m_j|s))$, $\forall i,j \in \mathcal{A}$, where $D_{\text{KL}}$ is the Kullback-Leibler divergence ensuring that message distributions between agents $i$ and $j$ remain consistent. (4) **Scalability in large-scale multi-agent communication**: As the number of agents increases, traditional communication protocols face scalability issues due to excessive message overhead. Attention-based models address this by dynamically filtering irrelevant messages. Scalable communication is often formalized as a minimization problem over message complexity while maintaining task performance: $\min_{\pi_C} \sum_{i=1}^{N} |m_i|$, subject to $J(\pi) \geq J_{\text{threshold}}$, where $|m_i|$ represents the message size of agent $i$, and $J(\pi)$ is the expected return of the multi-agent policy, constrained to ensure effective coordination. (5) **Grounding and interpretability in LLM-based communication**: Large language models introduce challenges in aligning generated messages with task-relevant information. Ensuring messages remain *grounded* in the environment requires integrating reinforcement learning with human feedback mechanisms: $\max_{\pi_C} \mathbb{E}[r_{\text{grounded}}(m, s, a)]$, where $r_{\text{grounded}}(m, s, a)$ measures how well the message $m$ aligns with meaningful environmental information. Future research should focus on **hybrid architectures** that combine structured message-passing with adaptive learning, ensuring that multi-agent communication is both *efficient and interpretable* across diverse applications.

**Foundation models, Algorithms, and Data** Among all the communication models in multi-agent systems, transformer-based and LLM-driven communication approaches have demonstrated some of the most promising results. (1) Many algorithms leveraging attention-based communication have achieved success in *complex multi-agent coordination tasks*, including real-world applications such as robotic teamwork, au-

tonomous vehicle swarms, and large-scale multi-agent competitions. (2) The use of *transformers enables agents to process long-range dependencies* in communication, making them highly effective for handling *sequential interactions and multi-modal inputs.* Agents leveraging transformer-based communication can dynamically adjust message relevance, allowing for efficient collaboration in dynamic environments. (3) In particular, large-scale foundation models such as GPT-based architectures have been explored for *multi-agent dialogue and cooperative problem-solving*, where a single model can adapt to multiple domains, from *robotic task execution* to *human-agent interaction.* Recent research has explored the integration of *pretrained language models* with reinforcement learning for communication optimization, demonstrating *cross-domain generalization effects.* The ability to fine-tune large models for *multi-agent reasoning* suggests a paradigm shift towards data-driven communication learning, where pretrained models provide agents with *rich semantic representations* for structured coordination. However, despite these advancements, foundation models introduce new challenges, such as *computational efficiency, real-time adaptability, and grounding in task-specific environments.* The question remains whether future development should be *data-driven* (leveraging large-scale pretraining) or *algorithm-driven* (designing more efficient communication-specific architectures). Regardless, future research must emphasize *scalability and computational efficiency*, ensuring that communication models remain *robust, interpretable, and deployable in real-world multi-agent scenarios.*

## 8.2 Beyond ML: Epistemic Perspectives on Agent Communication

While the preceding sections of this survey focus on algorithmic realizations of communication in MARL, emergent language, and LLM-based multi-agent systems, Communication has long been examined as a form of *belief-aware reasoning* in philosophy Lewis (2008); Hintikka (1962), cognitive science Premack & Woodruff (1978); Tomasello (2009), and general artificial intelligence Russell et al. (1995), where communicative acts are treated as deliberate actions selected based on agents' beliefs about others' knowledge, intentions, and mental states. In these traditions, communication is not merely a channel for information exchange, but a deliberate action chosen based on an agent's beliefs about other agents' knowledge, intentions, and mental states, as formalized in speech act theory and cooperative models of meaning Austin (1975); Searle (1969); Grice (1957).

A central idea in this line of work is that agents communicate in order to influence or coordinate beliefs. Epistemic and doxastic logics formalize this perspective by explicitly modeling agents' beliefs and knowledge Meyer (2003), often using Kripke semantics to reason about nested beliefs (e.g., what one agent believes about another agent's beliefs) Orłowska (1990). Within these frameworks, communication decisions are naturally framed around questions of *when* to communicate, *why* communication is necessary, and *to whom* information should be conveyed—dimensions that closely align with the Five Ws organizing principle adopted in this survey.

From this perspective, many challenges observed in modern multi-agent learning systems can be reinterpreted as instances of implicit belief reasoning Levesque (1984). In MARL-based communication, agents learn message policies that indirectly encode beliefs about other agents' future actions, but these beliefs remain latent and task-specific. Emergent language systems similarly rely on interaction to align internal representations, yet lack explicit mechanisms for modeling or reasoning about others' mental states. LLM-based agents, by contrast, exhibit strong surface-level theory-of-mind capabilities through language, but without formal guarantees that such reasoning is grounded in environment dynamics or aligned with control objectives.

Viewing communication through an epistemic lens helps clarify why issues such as grounding, interpretability, and generalization persist across learning-based approaches. Without explicit representations of belief, intent, or audience, modern systems must rediscover speech-act-like behavior through experience alone. This observation suggests that future progress may benefit from tighter integration between **learning-based communication mechanisms** and **classical models of belief-aware reasoning**, even if such models are used only as conceptual guides rather than explicit symbolic components.

Importantly, the goal of incorporating epistemic perspectives is not to replace data-driven or reinforcement learning approaches, but to situate them within a broader intellectual context. By connecting modern learning-based communication systems to foundational ideas in epistemic reasoning and cognitive science, we highlight that contemporary communication learning revisits long-standing questions about intentional action, belief alignment, and social reasoning under uncertainty. This broader view complements the technical

focus of the survey and provides additional guidance for designing agents that communicate effectively, robustly, and intelligibly in complex multi-agent environments.

## 8.3 Perspectives on Future Directions

Based on the discussions above and the main content in previous sections, we present some perspectives on future research directions in multi-agent communication. We categorize these directions into four key areas: theoretical models, algorithm developments, general benchmarks, and human-centric research. We believe that more open challenges can be identified from the detailed discussions in earlier sections, and we have included specific comments on future work within the relevant sections. Each of these categories represents a crucial aspect of advancing multi-agent communication, ranging from foundational theoretical advancements to practical considerations in real-world deployment.

### 8.3.1 Future Research on Theoretical Guarantees

Most works on multi-agent communication focus on algorithm design and empirical evaluation rather than theoretical analysis, highlighting a need for deeper theoretical research in this field. One promising direction is the formal derivation of **convergence guarantees and performance bounds** for key communication frameworks, such as attention-based message passing and emergent communication protocols. Establishing rigorous mathematical foundations will enable researchers to design communication strategies that are both scalable and provably efficient.

**Theoretical Guarantees for MARL Communication**  Most MARL communication studies focus on learning-based methods that improve coordination, but theoretical results on optimality, convergence, and performance bounds remain scarce. A fundamental question is: *how much does communication improve the return in cooperative MARL*, and *what is the theoretical gap between policies with and without communication?* Given a cooperative MARL setting, where $J^*(\pi)$ is the expected return under an optimal fully centralized policy $\pi^*$, and $J(\pi_C)$ is the expected return under a communication-constrained policy $\pi_C$, the return gap can be defined as:

$$\Delta J = J^*(\pi) - J(\pi_C), \tag{2}$$

where $\Delta J$ quantifies the loss incurred due to limited communication. A major research direction is deriving upper bounds for $\Delta J$, showing under what conditions communication policies can approach optimality. Additionally, scalability and communication efficiency are key concerns in large-scale MARL. Many existing methods use graph-based or attention-based message passing, but the theoretical trade-offs between bandwidth constraints, coordination performance, and sample efficiency remain unclear. A useful framework is minimizing communication costs while ensuring bounded performance loss:

$$\min_{\pi_C} \sum_{t=1}^{T} C_t, \quad \text{subject to} \quad J^*(\pi) - J(\pi_C) \leq \epsilon, \tag{3}$$

where $C_t$ represents communication costs at time $t$ and $\epsilon$ is the maximum allowable performance degradation. Future work should develop formal criteria for when and whom agents should communicate with to optimize efficiency. Finally, robustness in adversarial settings is a crucial but underexplored theoretical problem. Communication channels can be subject to noise, delays, or adversarial tampering. Establishing worst-case bounds on return degradation due to message corruption is essential:

$$J(\pi_C, \eta) = \mathbb{E}\left[\sum_{t=1}^{T} r_t \mid m_t = f(m_t^*, \eta)\right], \tag{4}$$

where $m_t^*$ is the ideal message, $\eta$ represents a perturbation or adversarial noise, and $f(m_t^*, \eta)$ is the received message. Theoretical guarantees on error correction, redundancy mechanisms, and robustness will be necessary for safe deployment of communication-based MARL.

**Theoretical Analysis of Emergent Language (EL) Communication**  In emergent communication, agents develop their own protocols through interaction rather than being explicitly programmed. However, understanding why certain languages emerge, how they generalize, and how to ensure their efficiency remains a theoretical challenge. A major open problem is quantifying the expressiveness and compositionality of emergent languages. Current emergent protocols often lack systematic structure and interpretability, leading to poor generalization across tasks and partners. One research direction is modeling emergent language as an information bottleneck problem:

$$\max_{\pi_C} I(m; s) \quad \text{subject to} \quad H(m|s) \leq \delta, \tag{5}$$

where $I(m; s)$ measures the mutual information between message $m$ and state $s$, while $H(m|s)$ ensures that messages remain concise and interpretable. Establishing conditions under which *compositionality emerges naturally* will be crucial for developing generalizable emergent languages. Another key theoretical question is the *stability and convergence of learned languages*. Many current methods rely on reinforcement learning to evolve communication, but the non-stationarity of language evolution can lead to instability. Understanding the convergence properties of emergent protocols and the conditions under which languages remain stable is a crucial research area. Lastly, *partner adaptation* remains an open challenge in emergent communication. Most current models assume a fixed agent population, but real-world multi-agent systems must interact with new, unseen partners. Theoretically, ensuring *zero-shot generalization* across different agent communities requires defining formal notions of language adaptability:

$$\min_{\pi_C} D_{\text{KL}}(p(m_i|s)||p(m_j|s)), \quad \forall i, j \in \mathcal{A}, \tag{6}$$

where $D_{\text{KL}}$ represents the Kullback-Leibler divergence between the language distributions of different agents. Establishing bounds on language alignment loss will be critical for ensuring communication effectiveness across diverse multi-agent populations.

**Theoretical Challenges in LLM-Based Multi-Agent Communication**  Large language models (LLMs) introduce new possibilities for structured communication among agents, enabling human-like interaction and symbolic reasoning. However, integrating LLMs into multi-agent systems presents new theoretical challenges related to *grounding, consistency, and efficiency*. One major issue is ensuring that LLM-generated messages are *grounded in task-specific knowledge* rather than being syntactically plausible but semantically irrelevant. This can be formalized as a constrained optimization problem:

$$p(m|s, a) = \frac{\exp(f(s, a, m))}{\sum_{m'} \exp(f(s, a, m'))}, \tag{7}$$

where $p(m|s, a)$ represents the probability of generating message $m$ given state $s$ and action $a$, and $f(s, a, m)$ ensures message relevance. Future research should investigate ways to enforce *consistency and alignment* between LLM-based communication and decision-making policies. Another key challenge is *scalability and real-time performance*. LLM-based communication is computationally expensive, and real-time multi-agent interactions require *low-latency message generation*. Developing *bounded complexity guarantees* for LLM inference in multi-agent settings is an important area of theoretical research. Finally, *multi-modal communication* is an emerging field where LLMs integrate language with vision, gestures, and symbolic reasoning. Establishing a *unified theoretical framework* for multi-modal communication will help in designing systems where agents can effectively process and exchange diverse types of information.

### 8.3.2 Future Research on Algorithm Developments

From the discussions in previous sections and the main content in multi-agent communication (MA COMM), we can identify several potential future directions regarding algorithm design. Here, we outline some of these promising directions.

**Under-explored categories in multi-agent communication.**  For existing categories of multi-agent communication algorithms, several areas remain under-explored, such as reinforcement learning-based

communication without explicit message exchange, self-adaptive emergent communication protocols, and transformer-based communication learning that integrates structured memory for improved long-term planning. Additionally, while large-scale multi-agent systems often suffer from communication bottlenecks, more efficient message-passing algorithms that balance communication cost and coordination effectiveness are needed. Specifically, attention-based message filtering and information-theoretic compression techniques could significantly improve the scalability of multi-agent communication.

**Integrated use of various communication paradigms in MARL** Each category of communication strategies—such as explicit message-passing, emergent language, and LLM-based communication—has unique advantages. However, integrating these paradigms into a unified multi-agent communication framework remains an open challenge. Future work could explore hybrid architectures that combine structured communication for task-specific scenarios with emergent protocols for adaptability. For instance, multi-agent reinforcement learning (MARL) systems could benefit from hierarchical communication learning, where low-level agents use emergent communication while high-level agents leverage structured messages generated by LLMs. Another promising research direction is improving multi-agent transformer-based communication, where agents can dynamically decide when to communicate using learned importance scores:

$$\alpha_{i,j} = \frac{\exp(f(h_i, h_j))}{\sum_{k \in \mathcal{N}(i)} \exp(f(h_i, h_k))}, \tag{8}$$

where $\alpha_{i,j}$ represents the attention weight assigned by agent $i$ to the message from agent $j$, and $f(h_i, h_j)$ is a learned relevance function. Developing adaptive attention mechanisms that optimize communication efficiency while maintaining task performance remains a critical challenge.

**Unsolved issues and extensions of multi-agent communication** Several fundamental issues in multi-agent communication remain unsolved, requiring new algorithmic developments: (1). **Scalability in large-scale MARL**: Many existing methods struggle with scalability due to communication overhead. Efficient routing and selective message-passing strategies, such as reinforcement learning-based communication topologies, could be explored. (2). **Handling non-stationarity**: In non-stationary environments where agents continually update their policies, communication protocols must adapt dynamically. One approach could be meta-learning-based strategies that allow agents to learn and adapt their communication strategies over time. (3). **Multi-agent robustness under noisy communication**: Real-world applications introduce message loss, delays, or adversarial tampering. Designing robust multi-agent communication frameworks that employ redundancy mechanisms and adversarial training techniques could improve resilience. Specifically, MARL algorithms for fully competitive or mixed cooperative-competitive tasks, grounded in game-theoretic principles, require more sophisticated communication strategies that balance cooperation and deception.

**Development of generalist multi-agent communication models** One of the most exciting future directions in multi-agent communication is developing a generalist communication model that can adapt to a variety of environments and tasks without requiring task-specific fine-tuning. This could be achieved by leveraging advances in foundation models from NLP and computer vision. To build generalist multi-agent communication systems, several challenges must be addressed: (1). **Scalable architectures for universal communication**: Designing models that can process multi-modal inputs, including textual, visual, and structured data, while maintaining efficient message encoding. (2). **Pretraining and fine-tuning for generalization**: As seen in large-scale foundation models, pretraining communication models on diverse datasets followed by fine-tuning on task-specific interactions could enable better generalization. (3). **Learning from third-person demonstrations**: Future research should explore whether multi-agent communication strategies can be learned from video datasets or human-agent interactions, allowing agents to infer communication protocols without direct reinforcement learning.

**Future works in related areas** Finally, there are several related research areas that could significantly impact the future of multi-agent communication: (1). **Inverse reinforcement learning for communication**: Understanding how communication strategies emerge in expert demonstrations could improve learned policies. (2). **Uncertainty-aware communication in MARL**: Extending multi-agent communication to

handle risk-sensitive decision-making could be critical for high-stakes applications such as autonomous driving. (3). **Incorporating new foundation models**: The emergence of new architectures such as Mamba models and Poisson Flow models provides exciting opportunities to enhance communication strategies for real-time and data-efficient learning.

### 8.3.3 Future Works on Benchmarking

**Development of more realistic, challenging benchmarks**    We observe that most multi-agent communication (MA COMM) algorithms are evaluated on standard MARL benchmarks such as StarCraft Multi-Agent Challenge (SMAC), Multi-Agent Particle Environment (MPE), and Google Research Football (GRF). However, these benchmarks may not fully capture the complexity and challenges of real-world multi-agent communication. To advance the field and properly evaluate new methods, more comprehensive and challenging benchmarks should be developed: (1). **Large-scale datasets for communication evaluation**: The scalability of MA COMM algorithms remains a key challenge. Benchmarks should include large-scale datasets that test how communication methods perform with increasing agent populations, diverse communication constraints, and extensive training data. Understanding the relationship between dataset size and communication efficiency can help assess whether learned communication strategies generalize well. (2). **Multi-modal communication benchmarks**: Real-world multi-agent communication often involves heterogeneous data sources, including language, vision, and symbolic information. Future benchmarks should include multi-modal environments where agents communicate using a mix of textual, visual, and structured messages. For example, agents might receive partial visual observations and complement them with natural language messages to coordinate actions. (3). **More realistic task scenarios**: A major reason for the widespread success of deep learning in computer vision and NLP is that models are trained on large, real-world datasets. In contrast, multi-agent communication is still largely evaluated in simulated environments that lack real-world complexity. Future benchmarks should include real-world datasets of human or agent interactions, such as autonomous vehicle coordination data, multi-robot task execution logs, and human dialogue datasets for cooperative problem-solving. (4). **Beyond return-based evaluation**: Current benchmarks mainly evaluate performance based on cumulative rewards or task completion rates. However, effective communication should also be assessed in terms of **interpretability, efficiency, generalization, and safety**. Future benchmarks should include metrics that could evaluate problems like: **Message efficiency**, How much communication overhead is required for achieving optimal performance? And **Generalization**, such as answering questions like *Can learned communication protocols transfer to unseen environments or new agents?* And **Robustness**: How resilient is the communication strategy to message loss, delays, or adversarial interference? Or for **Emergent communication structure**, an efficient benchmark should help researchers to answer *Do agents develop structured and compositional languages, and how interpretable are they?*

**Comparisons among different multi-agent communication paradigms**    While there exist numerous algorithms for multi-agent communication—ranging from explicit message-passing approaches to emergent language learning and LLM-based multi-agent coordination—there is a lack of standardized comparisons among them. To better understand their relative advantages and limitations, future research should establish benchmarks that allow fair evaluations across different paradigms: (1). **Standardized comparisons across communication strategies**: Future benchmarks should compare traditional MARL communication approaches (e.g., differentiable communication via continuous signals) with emergent language-based communication and LLM-augmented communication frameworks. By evaluating these approaches on the same set of tasks, researchers can better understand when structured communication is preferable over learned protocols. (2). **Scalability of different communication methods**: As the number of agents increases, how do different communication approaches scale? Some methods may perform well in small agent populations but struggle with communication bottlenecks in larger teams. Benchmarks should include scalability evaluations where different communication architectures are tested under increasing numbers of agents and message complexity constraints. (3). **Computational and learning efficiency**: While some multi-agent communication methods achieve high performance, they may require significant computational resources for training. Future benchmarks should track training time, inference speed, and memory usage across different methods, providing insights into their practical feasibility. (4). **Adaptability to new agents and tasks**: Benchmarks should evaluate how well communication strategies generalize to new agents or novel

tasks. For example, if an agent trained in one environment is deployed in another with different teammates, how quickly can it adapt its communication protocol? Evaluating zero-shot and few-shot generalization capabilities would be crucial for developing flexible multi-agent systems.

**Developing a unified benchmark suite for MA COMM**    To accelerate progress in multi-agent communication research, a unified benchmark suite should be developed that provides: (1). **A diverse set of environments**: Benchmarks should cover cooperative, competitive, and mixed-motive multi-agent tasks, ensuring that communication strategies are evaluated across a broad range of scenarios. (2). **Extensive datasets for pretraining and evaluation**: Similar to ImageNet in computer vision and GLUE in NLP, multi-agent communication research would benefit from large-scale datasets that facilitate pretraining and systematic evaluation. (3). **A set of well-defined metrics**: Future benchmarks should move beyond simple reward-based evaluation and incorporate new performance metrics that reflect communication efficiency, interpretability, robustness, and generalization. (4). **Baseline implementations for fair comparisons**: Providing open-source implementations of baseline communication methods would ensure reproducibility and encourage fair comparisons across different approaches.

### 8.3.4   Future Research on Human-Centric Multi-Agent Communication

**How can multi-agent communication systems align with human expectations and interpretability?**    The effectiveness of multi-agent communication (MA COMM) is not solely determined by task performance but also by its interpretability, alignment with human communication norms, and ability to foster trust in human-agent collaboration. One fundamental research direction is to explore how multi-agent communication protocols can be designed to be both machine-efficient and human-comprehensible. (1) Current emergent communication models often develop symbols or representations that are opaque to humans. Ensuring that learned communication protocols exhibit structured, compositional, and generalizable properties similar to natural language is a key challenge. A possible approach is to impose information-theoretic constraints that favor structured communication while maintaining efficiency. Another direction is to explore how learned protocols can be mapped to existing human language structures without compromising performance. (2) Multi-agent communication should be interpretable not only in terms of the message content but also in how communication influences agent behavior. Formal methods for analyzing the causal effect of communication on decision-making remain underdeveloped. Establishing metrics to evaluate whether agents correctly utilize received messages in a way that aligns with human intuition can be beneficial for interpretability and debugging. (3) The degree to which communication can be adapted for diverse user preferences is also an open question. Humans exhibit variation in linguistic styles, levels of detail in communication, and implicit assumptions about shared knowledge. Future research should investigate adaptive communication strategies where agents dynamically adjust their communication style to align with individual user expectations. Such adaptability could be achieved through meta-learning techniques or reinforcement learning policies trained on diverse human interaction datasets.

**How should multi-agent communication systems facilitate collaboration with humans?**    In many real-world applications, agents must not only communicate with each other but also interact with humans as teammates, supervisors, or evaluators. Designing communication protocols that effectively integrate human feedback and decision-making is critical for advancing human-AI collaboration. (1) One challenge is determining the optimal balance between autonomous agent communication and human-in-the-loop intervention. While some tasks require full autonomy, others benefit from human oversight, and communication models should be able to modulate information exchange accordingly. This raises the question of how to design communication-aware policies that incorporate human corrections efficiently without excessive reliance on human intervention. (2) Another open problem is learning communication strategies that generalize across different types of human users. Agents may need to adjust their communication level based on the expertise and role of the human collaborator. For instance, a novice user may require detailed explanations, while an expert user may prefer concise updates. A promising direction is to develop hierarchical communication frameworks where agents can reason about when and how to involve human input. (3) In settings where multiple agents communicate with a human, ensuring that communication remains coherent and non-redundant is crucial. Unlike homogeneous multi-agent settings, human interactions require mechanisms for resolving conflicting information, synthesizing diverse perspectives, and presenting relevant insights in a structured

manner. Future work should investigate how multi-agent communication can be optimized for presenting actionable insights in an efficient and user-friendly manner.

**How can human trust and safety be ensured in multi-agent communication?** Trust in multi-agent communication systems is essential for adoption in high-stakes domains such as healthcare, autonomous driving, and critical infrastructure management. However, ensuring robustness and reliability in communication remains an open challenge. (1) One primary concern is that communication protocols learned purely through reinforcement learning may not always prioritize safety or reliability. Messages exchanged between agents should not only be optimized for task efficiency but also be constrained to avoid potentially harmful or misleading communication. Future research should explore formal safety constraints that ensure multi-agent messages adhere to ethical and regulatory guidelines. (2) Communication robustness under adversarial settings is another key research direction. In environments where agents operate in competitive or adversarial conditions, ensuring that communication remains truthful and resilient to manipulation is essential. Theoretical work on adversarial robustness in communication, such as establishing upper bounds on the probability of message tampering or introducing cryptographic techniques for secure multi-agent communication, is an important avenue of research. (3) Beyond security concerns, communication reliability in dynamic and non-stationary environments is also a challenge. Agents must be able to detect and recover from missing or corrupted messages while ensuring that communication failures do not lead to catastrophic decision errors. Future work should investigate self-correcting communication mechanisms that allow agents to infer missing information or reconstruct degraded messages in real time.

**How can human feedback be effectively incorporated into multi-agent communication learning?** Human-in-the-loop learning is a promising approach for improving multi-agent communication by leveraging human expertise to refine communication strategies. However, incorporating human feedback efficiently and scalably remains an open challenge. (1) One direction is investigating how agents can learn effective communication through a combination of explicit and implicit feedback. While direct human corrections can provide useful supervision, implicit signals such as human engagement, response time, or sentiment analysis from text-based interactions could serve as valuable learning signals for communication adaptation. (2) Future work should explore techniques for efficiently integrating sparse human feedback into multi-agent learning frameworks. Since obtaining human annotations for large-scale multi-agent interactions is costly, techniques such as preference-based reinforcement learning or weak supervision could be useful for extracting meaningful guidance with minimal human effort. (3) Evaluating the effectiveness of human feedback on multi-agent communication learning remains an underexplored research problem. While traditional reinforcement learning relies on reward functions, the impact of human feedback is more complex and may require new evaluation frameworks that capture the long-term benefits of human-informed communication strategies. Establishing benchmarks that explicitly test human-guided communication adaptation would be a valuable contribution to this area.

**How can multi-agent communication be designed to enhance social interaction and fairness?** As multi-agent systems are increasingly deployed in human-facing applications, it is important to consider fairness and social dynamics in communication. (1) One key question is how to ensure equitable information access in multi-agent interactions. In scenarios where multiple agents communicate with each other and with human users, disparities in information access can emerge, leading to biased decision-making. Research should focus on designing fairness-aware communication policies that ensure all participants receive relevant information based on their role and decision-making needs. (2) Another challenge is avoiding emergent behaviors that disadvantage certain agents or human users. For example, in competitive multi-agent environments, agents may develop deceptive communication strategies that provide short-term advantages but undermine trust and cooperation in the long run. Studying game-theoretic mechanisms for enforcing cooperative communication norms could help mitigate such behaviors. (3) Finally, multi-agent communication should support natural and socially appropriate interactions. This includes adapting communication tone, level of detail, and responsiveness based on the social context. Future research should explore computational models of social intelligence that enable agents to adjust their communication style in a human-like manner, improving engagement and user experience.

## 9 Conclusion

This survey presented a comprehensive overview of multi-agent communication through the unifying lens of the **Five Ws of communication**: **who** communicates with **whom**, **what** is communicated, **when** communication occurs, **why** it is beneficial, and **how** it is motivated and operationalized. Using this framework, we organized progress across three major paradigms: **communication in MARL, EL, and LLM–based multi-agent systems**. We traced the evolution of MARL communication from early message-passing mechanisms to learned, end-to-end protocols shaped by reward structure and control objectives, reviewed emergent language research that studies how communication arises through interaction with improved structure and interpretability, and surveyed LLM-powered multi-agent systems where natural language enables reasoning, planning, and collaboration in open-ended environments, including **hybrid LLM–MARL architectures** that balance expressiveness, grounding, and control.

Beyond cataloging methods, this survey emphasized **conceptual integration**. By organizing diverse approaches around shared communication questions, we clarified how communication design reflects underlying assumptions about coordination, uncertainty, incentives, and agency. We further connected modern learning-based communication to foundational perspectives from philosophy, cognitive science, game theory, and classical AI, framing communication as an *action* chosen to influence beliefs, align intentions, and support collective decision-making. The added **bridge** sections explicitly illustrate how newer paradigms emerge in response to the limitations of earlier ones, rather than replacing them outright.

Looking ahead, multi-agent communication research faces persistent challenges in grounding, interpretability, generalization, scalability, and theoretical guarantees, especially as systems move toward open-ended, human-facing, and safety-critical deployments. Progress will require tighter integration of learning, language, and control, informed by both system-level design principles and formal reasoning about belief, knowledge, and interaction. As the first survey to unify MARL-based communication, EL, and LLM-driven multi-agent systems within a single, coherent framework, we hope this work serves as a durable reference and a foundation for developing scalable, robust, and human-aligned communication in future multi-agent systems.

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
