# OpenReview forum: "The Five Ws of Multi-Agent Communication: Who Talks to Whom, When, What, and Why - A Survey from MARL to Emergent Language and LLMs"
_TMLR — Accepted by TMLR_

### Review · Reviewer_ye4q · 2025-10-29

**Summary Of Contributions:**

The authors present a survey in the field of Multi-Agent Reinforcement Learning (MARL) with a particular focus on multi-agent communication (MA-Comm) and emergent language (MA-EL). Within the work the authors consider different aspects of agent communication under varying communication costs, agent motifs, goal alignments and communication constraints.

The first part of the survey is concerned with 'what', 'when' and 'how', comprising  discussion on limitations on the scalability of communication channels, centralized/decentralized communication, as well as the more general aspect of communication towards knowledge acquisition. A key focus of the presented survey is the consideration of emergent language within learning systems. The authors simultaneously focus on emergent language protocols in unregularized settings, as well as the alignment and leverage of communication through human-like natural language. For the former, aspects on the emergence of (symbolic) message sequences under difference scenarios are discussed. For the latter, use-cases of the application of large language models are comprehensively studied. Finally, the authors provide a discussion on open problems in current literature involving theoretical guarantees, benchmarks and challenges in human-computer interactions.



**Strengths**

The individual sections consistently build on top of each other, starting with MARL and transitioning towards emergent- and natural language aspects. The paper is clearly written and the discussed concepts are presented in an easy to understand way. The chosen structure is intuitive and follows a common thread. Individual sections are accompanied by nicely designed figures which help communicating the discussed concepts on an intuitive level and are easy to grasp.

While I do not posses a comprehensive overview of the field, the survey seems to accurately represent common concepts and ideas in MARL and agent communication. The authors seem to comprehensively cite and categorize recent developments in papers and methods. The work furthermore refers to existing surveys contrasting it with the aspects of emergent language and communication discussed in the paper.



**Weaknesses**

**Foundations of Agent Communication.** Given the long-standing line of work on multi-agent planning reaching back to the 1980s, it is striking that most cited works in the survey falls into the time-frame of 2010 onward. While the area undoubtedly gained traction under the past developments in general deep learning and LLMs, giving rise to a new understanding and application areas for agents, the authors ignore the foundations of the field. I would like to recommend adding a small discussion on the foundations of the the field. While not being well versed in the foundations of the field myself, I would kindly refer to, for example, Russel and Norvig "Artificial Intelligence: A Modern Approach", which contains a brief introduction on the matter, citing initial references.



**Alignment with Title.** Although the aspects of 'Who', 'Whom' and 'Why' are implicitly covered within the other sections, the survey lacks an explicit account of those. Given that the particular question of 'Why' fundamentally justifies the need for communication in the first place, I would like to recommend (-also in order to meet expectations of the title. E.g., the word 'Why' only appears 5 times in the paper, 3 of those being the title, abstract and a referenced paper-), to give a more explicit discussion on the missing aspects in separate sections.



**Broader Discussion.** The survey focuses strongly on the technical realizations of agents, but does not touch on the more general philosophical/mathematical considerations of agent communication. Considerations communication might be substantially shaped due to reasoning about an other agents' beliefs and their state of mind. These aspects are commonly considered a key skill for generally 'intelligent' actors, e.g. in cognitive science or general AI literature. Beyond specific realizations of agents, the authors might therefore want to discuss more general frameworks, such as Doxastic logic or more Kripke semantics that allow the formal modeling of such reasoning, and providing reasons for 'When', 'Why' and 'to Whom' to communicate.



**Technical Detail.** Contrary to the previous point, the survey provides rather high-level descriptions on many mentioned concepts. Except for sections 3 and 6, concepts are primarily expressed in natural language, which might somewhat reduce the information that can be drawn directly from the paper. E.g., section 5.2.4 "Structure of Communication" could be complemented by a brief formalization of communication channels. Similarly, work concerned with mixed-motives or competitive settings (sec. 5.2.2 "Motivational Settings") is often concerned with Nash equilibria. While the aim of a survey is not to present excessively detailed formalism, I would at least like to discuss this topic with the authors.



**Minor: Citations.** Given that the type of the presented work is a survey, the authors might want to put a stronger emphasis on providing up-to-date references. Upon checking some of the papers on the first page of references, I found several works cited as their arxiv versions, which where already published at conferences:

* Agarwal, Akshat, et al. "Community Regularization of Visually-Grounded Dialog." *Proceedings of the 18th International Conference on Autonomous Agents and MultiAgent Systems*. 2019.
* Agarwal, Akshat, et al. "Learning Transferable Cooperative Behavior in Multi-Agent Teams." *Proceedings of the 19th International Conference on Autonomous Agents and MultiAgent Systems*. 2020.
* Agashe, Saaket, et al.  "LLM-Coordination: Evaluating and Analyzing Multi-agent Coordination  Abilities in Large Language Models." *Findings of the Association for Computational Linguistics: NAACL 2025*. 2025.
* Andreas, Jacob, et al. "Measuring Compositionality in Representation Learning." *International Conference on Learning Representations*. Vol. 375. Association for Computational Linguistics, 2019.

**Audience:**

Yes

**Audience Explanation:**

Autonomous agent systems are increasingly deployed in practice. Due to the rise in LLM capabilities, the field evolves rapidly as new environments can be targeted by these systems. The authors provide a well curated survey on the state of existing works that might help to navigate and categorize the ever increasing amount of papers.

**Claims And Evidence:**

Yes

**Claims Explanation:**

The paper being a survey paper, no novel technical insights or claims are made. The paper comprehensively cites recent works. The perspective presented in the survey aligns with the common understanding of ml agent systems, covering recent developments in a well curated fashion.

**Requested Changes:**

The requested changes mainly concern the stated weaknesses above.

Generally, I would like to encourage the authors to more strongly consider foundational works in the field. Although, relevance might diminish as restrictions are lifted through the ability of modern systems to handle natural language, they commonly provide the theoretical and formal foundations of modern systems.

Similarly, I would like to encourage the authors to include perspectives beyond 'typical' computer science/ML view on agent systems. While these considerations rarely have an immediate impact on the field, they can provide a more general perspective of the field being studied.

As minor concerns, I would finally like to encourage the authors to, more explicitly, discuss the questions of 'Who', 'Whom' and 'Why' (possibly by implementing the previous changes) and possibly update their references.

---

> ### Author Response · Authors · 2025-12-15
> **Response to Reviewer ye4q**
>
> ## Response to Reviewer ye4q
>
> We thank the reviewer for the thorough and constructive feedback. We appreciate the positive assessment of the paper’s clarity, organization, and coverage of MARL-Comm, emergent language, and natural-language/LLM-based communication. We have revised the manuscript to address the concerns on foundations, alignment with the Five Ws (especially who/whom/why), broader perspectives, technical formalization, and reference updates. **All changes made in response to your review are marked in orange in the revised manuscript.**
>
> ---
>
> ### 1. Foundations of Agent Communication
> **Revisions**:
> - We added a new **Section 2: Foundations of Agent Communication** to explicitly ground the survey in **classical AI, philosophy of language, and early multi-agent planning**.
> - This section cites **Russell and Norvig, *Artificial Intelligence: A Modern Approach*** and foundational works (e.g., speech act theory, formal semantics, early decentralized planning), clarifying how modern MARL-Comm, EL-Comm, and LLM-Comm **extend, not replace the classical views of communication as action**.
> - We further provide a **comparative table** contrasting classical, MARL-based, and LLM-based communication assumptions (semantics, grounding, interpretability, limitations) to make these foundations explicit and accessible.
>
>
> ---
>
> ### 2. Alignment with Title and Explicit Coverage of Who/Whom/Why
> **Revisions**:
> - We revised all major sections to **explicitly align with the Five Ws structure**, organized as **who/whom/what**, **when**, **why**, and **how**.
> - We made **who communicates with whom** and **why communicate** explicit throughout, and clarified how **why motivates how** across MARL-Comm, EL-Comm, and LLM-Comm.
>
> ---
> ### 3. Broader Discussion Beyond a Pure ML View
> **Revisions**:
> - We added **Sec. 8.2: Beyond ML—Epistemic Perspectives on Agent Communication**, grounding the discussion in **belief-aware reasoning** from philosophy, cognitive science, and classical AI.
> - The section links communication to **epistemic/doxastic logic**, clarifying how beliefs about others shape *when*, *why*, and *to whom* agents communicate, and reframes key challenges (grounding, interpretability, generalization) as forms of **implicit belief reasoning**.
>
>
> ---
>
> ### 4. Technical Detail and Formalization of Communication Structure
> **Revisions**:
> - We added **Sec. 7.4.1 (marked in orange)** with a **concise mathematical formalization of communication**, using time-indexed notation to distinguish **single-turn**, **multi-turn**, and **grounded/embodied** structures.
> - We also added a **game-theoretic view of communication timing in mixed-motive settings** (Sec. 7.3.1, marked in orange), explicitly connecting learned communication policies to **Nash/Bayesian Nash equilibrium** concepts.
>
> ---
>
> ### 5. Citation Updates and Versioning
> **Revisions**:
> - We audited and updated references to prioritize **archival conference and journal versions** where available, replacing outdated arXiv citations.
> - We also harmonized citation formatting and terminology for consistency across the survey.

---

### Review · Reviewer_CeNu · 2025-11-21

**Summary Of Contributions:**

This paper presents "The Five Ws of Multi-Agent Communication: Who Talks to Whom, When, What, and Why," a comprehensive survey spanning the evolution from traditional Multi-Agent Reinforcement Learning (MARL) communication to emergent language and LLM-based multi-agent systems. The work is commendable in several respects: it provides extensive coverage of the multi-agent communication landscape, organizes a vast body of literature through the intuitive lens of the "Five Ws" framework, and distinguishes itself from existing surveys by bridging three distinct research paradigms—MARL-Comm, emergent language, and LLM-based multi-agent systems—under a unified conceptual umbrella.
The paper's taxonomies are thorough and well-structured, particularly in Section 5, which provides a detailed categorization of LLM-based multi-agent systems across communication topology, motivational settings, agent configurations, and communication structures. The extensive reference compilation (spanning hundreds of papers) demonstrates the authors' diligent effort to capture the field's breadth.
**However, I have several substantive concerns that affect the paper's coherence and depth:**
1. Disjointed Integration Between MARL and LLM Sections: The transition from traditional MARL communication (Section 3) and emergent language (Section 4) to LLM-based approaches (Section 5) feels abrupt and insufficiently motivated. While the paper acknowledges these as distinct research paradigms, it lacks a compelling narrative arc that explains why and how the field evolved from one to another. The sections read more like parallel surveys concatenated together rather than chapters in a unified story. A stronger conceptual bridge is needed—perhaps discussing what limitations in MARL-Comm motivated emergent language research, and what gaps in both led to the adoption of LLMs.
2. Insufficient Depth in Analysis: While the paper excels at categorization and enumeration, it falls short in providing critical analysis and insights. This is particularly evident in Section 5 on LLM-based multi-agent systems. For example: The communication topology subsection (5.2.1) lists five topologies (chain, decentralized, centralized, nested/hierarchical, and dynamic) with corresponding citations, but provides minimal comparative analysis of their trade-offs, applicability, or empirical performance.
The motivational settings subsection (5.2.2) categorizes systems as cooperative, competitive, or mixed-motive, but doesn't offer deep insights into how these different motivations fundamentally change communication strategies or what design principles emerge.
Throughout Section 5, there is heavy reliance on citation stacking (e.g., listing 20+ papers for a single category) without synthesizing their key findings or identifying trends.

3. Inadequate Treatment of Communication in LLM+RL Integration: Section 5.5, which discusses the integration of LLMs with MARL, is disappointingly brief given its centrality to the paper's thesis of bridging paradigms. More critically, this section largely neglects the communication mechanisms between agents in these hybrid systems. The examples provided (Cicero, Werewolf, FAMA, ACC-Collab) are described primarily in terms of their task domains rather than their communication architectures. Given that the paper's central organizing principle is the "Five Ws of communication," the lack of systematic analysis of how agents communicate in these LLM+RL systems represents a significant gap. Questions like "Do these systems use natural language, learned protocols, or hybrid approaches?" and "How do communication patterns differ from pure MARL or pure LLM systems?" remain largely unaddressed.

**Audience:**

Yes

**Audience Explanation:**

This survey addresses a timely and important topic that intersects multiple active research communities. The paper would be valuable to several audience segments:

1. MARL researchers seeking to understand how communication protocols have evolved and how modern LLM-based approaches compare to traditional methods.

2. LLM researchers interested in multi-agent applications and wanting comprehensive context on how agents coordinate and communicate.

However, the paper's impact would be significantly strengthened by addressing the coherence issue mentioned earlier. Currently, readers from different subfields might find their respective sections useful while struggling to see the connections to other sections. A more integrated narrative would make the survey valuable not just as a reference collection, but as a conceptual contribution that helps researchers understand the field's evolution and identify future research directions at the intersections of these paradigms.

**Broader Impact Concerns:**

No concerns. The paper is a survey that primarily organizes and synthesizes existing research. It does not introduce new methods or systems that would raise ethical concerns.

**Claims And Evidence:**

Yes

**Claims Explanation:**

The paper demonstrates strong scholarly rigor in its evidence and citations:

1. The references are comprehensive and appear accurate, with proper attribution to foundational and recent works across all three research areas.

2. The categorizations are well-grounded in the literature. For instance, the taxonomy of LLM-based multi-agent systems in Section 5.2 effectively captures the diversity of approaches, with each category supported by numerous representative works.

3. The technical background sections (3.1 on MARL-Comm, 4.1 on emergent language, 5.1 on LLMs) provide accurate preliminaries that appropriately contextualize subsequent discussions.

4.The paper's tables (e.g., Table 10 on LLM Agent Communication Taxonomy, Table 11 on evaluation benchmarks) are informative and backed by proper citations.

**Requested Changes:**

I recommend the following revisions to strengthen the paper:
1. Add Bridging Content Between Major Sections.
Insert a subsection at the end of Section 3 (MARL-Comm) that explicitly discusses the limitations that motivated emergent language research. What problems couldn't be solved with traditional MARL communication? Why did researchers turn to learning communication protocols end-to-end?
Similarly, add content at the end of Section 4 explaining why LLMs became attractive for multi-agent systems. What capabilities do LLMs provide that emergent language systems lack?
Consider adding a high-level "Evolution of Multi-Agent Communication" subsection in the introduction that previews this narrative arc.

2. Substantially Expand Section 5.5 on LLM+RL Integration.
This section is crucial for demonstrating how the field is synthesizing insights from both paradigms, yet it currently occupies only ~2 pages. Provide systematic analysis of communication mechanisms in hybrid systems: Do agents use natural language end-to-end? Are there learned message-passing layers beneath language? How do these systems handle the trade-off between language interpretability and communication efficiency? Discuss how the "Five Ws" apply differently in hybrid systems compared to pure MARL or pure LLM approaches. Include empirical comparisons where available (e.g., how does communication in FAMA compare to traditional MARL-Comm approaches in similar domains?).

---

> ### Author Response · Authors · 2025-12-15
> **Response to Reviewer CeNu**
>
> ## Response to Reviewer CeNu
>
> We thank the reviewer for the detailed, thoughtful, and constructive feedback. We appreciate the recognition of the paper’s breadth, the effectiveness of the **Five Ws framework**, and the value of unifying MARL-Comm, emergent language, and LLM-based multi-agent systems. We have substantially revised the manuscript to address the concerns regarding coherence, analytical depth, and integration. **All changes made in response to your review are marked in green in the revised manuscript.**
>
> ---
>
> ### 1. Narrative Coherence and Bridging Across Paradigms
> **Revisions**:
> - We strengthened the **narrative linking MARL-Comm, EL-Comm, and LLM-Comm** by explicitly explaining *why* the field evolved across paradigms.
> - We added **bridging subsections** at the end of each major section (**Secs. 5.7, 6.7, 7.6**) to clarify how limitations in earlier paradigms motivated subsequent ones.
> - We substantially expanded the **Bridge section (Sec. 7.6)** into a structured integration of MARL and LLM systems, with **four subsections (7.6.1–7.6.4)** covering hybrid communication mechanisms, comparative patterns, and LLM limitations (including a decentralized control perspective).
> - We also introduced an **Evolution overview (Sec. 1.1)** to present the high-level narrative upfront.
>
>
> ---
>
> ### 2. Analytical Depth in LLM-Based Multi-Agent Systems
> **Revisions**:
> - We reduced citation stacking and added **explicit comparative analysis** of LLM-based systems.
> - Using the **Five Ws framework**, the revised text highlights **design trade-offs, trends, and limitations** across communication topology, motivation, agent configuration, and structure.
>
> ---
>
> ### 3. Communication-Centric Analysis of LLM+RL Integration
> **Revisions**:
> - We substantially expanded **Sec. 7.6 (Bridge: LLM-Grounded MARL and Its Limitations)** to provide a structured account of **why MARL-Comm and EL-Comm leave gaps**, why LLMs became attractive, and how **LLM-grounded MARL** bridges the paradigms.
> - Sec. 7.6 now gives a **Five Ws–driven analysis** of representative **LLM–MARL hybrid systems** (e.g., Cicero, LangGround, Werewolf, FAMA, ACC-Collab), including **comparative tables** that contrast motivations (*why*) and mechanisms (*how*).
> - We also added dedicated subsections on **LLM limitations in MARL settings** and **limitations from a decentralized control perspective** (with summary tables), clarifying why LLM-based communication is **complementary to** control-grounded MARL communication.
>
> ---
>
> ### 4. Explicit Articulation of “Why Communicate”
> **Revisions**:
> - We revised the paper to **explicitly categorize different motivations for communication** (e.g., partial observability, coordination, efficiency, human–agent interaction).
> - This structure is now applied **consistently across MARL-Comm, EL-Comm, and LLM-Comm**, and we explicitly show how *why* motivates *how* in communication design.

---

### Review · Reviewer_7M5j · 2025-12-03

**Summary Of Contributions:**

Summary:

The paper surveys multi-agent communication across three areas that are usually treated separately: communication in MARL, emergent communication (EL), and communication in LLM-based multi-agent systems. It presents a taxonomy inspired by “Five Ws” (who/whom, when, what, why) and uses it most systematically in the MARL section. It then reviews key methods, provides classification tables, and closes with cross-domain discussion of open challenges and future directions.

Contributions:

The main contribution is a unified survey that spans MARL-Comm, emergent language, and LLM-based agent communication (a combination not covered by existing surveys). The MARL portion offers a clear and well-structured taxonomy, helpful formalization, and comprehensive tables. The EL and LLM-Comm sections give accessible overviews of the literature.

Strengths:
The MARL-Comm taxonomy (“what/whom, when, how”) is well-organized and accurate, with nice comparative tables. The paper is generally clear, detailed, and visually well-supported. It fills a gap, in my view.

Weaknesses:
The “Five Ws” framing is not totally consistently applied in my view, especially “why,” which is not too developed as a true dimension. EL and LLM-Comm sections are not as in-depth as the MARL section and could have more focus on comparing existing works, rather than listing them. Some claims about LLM-based multi-agent systems are perhaps overly optimistic, and one communication formalization is a bit unclear (Section 6). The survey would benefit from explicit acknowledgment of LLM limitations in MARL settings and clearer connection to classical decentralized control literature.

**Audience:**

Yes

**Audience Explanation:**

The paper is relevant. Communication in MARL is a core RL topic, emergent communication has become a standard tool in multi-agent RL research, and LLM-based agents increasingly interact with or guide RL components. A unified survey covering all three areas is timely and useful for researchers in and around the area.

**Broader Impact Concerns:**

Nothing to declare.

**Claims And Evidence:**

Yes

**Claims Explanation:**

The MARL-Comm section is accurate and technically solid; its formalization matches standard Dec-POMDP treatments and its categorization of methods is reliable. The EL and LLM-Comm sections are also factually correct but less analytic and perhaps makes optimistic claims about LLM capabilities. One communication formulation, in Section 6, is conceptually unclear due to circular dependencies (see that the message depends on the action and the action depends on the message). Overall clarity is good, the structure is consistent, and tables/figures are effective, though the “Five Ws” framing is only partially fulfilled and some repetition appears across sections.

**Requested Changes:**

I don't have strict requirements for changes.

Nice-to-haves:
- Make the “Five Ws” framing a bit more explicit and consistent across the three main sections, especially clarifying “why communicate” as its own analytical axis;
- Temper and contextualize claims involving LLM-based multi-agent systems, ensuring that limitations and failure modes are explicitly acknowledged;
- Clarify the generic communication formalization in Section 6 to remove circular action–message dependence or explain its intended abstraction;
- Add explicit links (discussion and references) to classical communication-constrained control and decentralized information theory.

Please also consider the following questions, whose answers could also appear in text:
- What criteria guided which papers were included or excluded across the three domains?
- How many total works were surveyed versus discussed in detail, and how were these proportions determined?
- How do you envision integration between RL-based communication approaches and LLM-based multi-agent systems?
- Can you articulate different categories of “why communicate,” and how each motivation affects communication design?

A few typos, I believe:
- “We begin…” mid-sentence;
- “MARL-COMM” vs “MARL-Comm”;
- Confusing phrase “agents’ numbers settings”;
- “fixed set” instead of “fix set” and "are" instead of "is" in “communication symbols is.”.

---

> ### Author Response · Authors · 2025-12-15
> **Response to Reviewer 7M5j**
>
> ## Response to Reviewer 7M5j
>
> We thank the reviewer for the careful reading and positive assessment of the paper. We appreciate the recognition that this survey fills an important gap by **unifying MARL-Comm, emergent language, and LLM-based multi-agent communication**, and for the constructive suggestions on how to further strengthen the presentation. **All changes made in response to your review are marked in blue in the revised manuscript.**
>
> ---
>
> ### 1. Consistency of the “Five Ws” Framework (Especially *Why Communicate*)
> **Revisions**:
> - We revised all major sections to **explicitly align with the Five Ws structure**, organized as **who/whom/what**, **when**, **why**, and **how**, with *why communicate* serving as a key motivating dimension.
> - We clarified that **different communication motivations (“why”) directly shape communication mechanisms (“how”)**, linking objectives such as partial observability reduction, coordination, efficiency, and human–agent interaction to concrete design choices.
> - Throughout the paper, we made these dependencies explicit so that the discussion consistently reflects how *why* motivates *how* across MARL-Comm, EL-Comm, LLM-Comm.
>
>
> ---
>
> ### 2. Depth and Comparison in EL and LLM-Based Sections
>
> **Revisions**:
> - We now use the **Five Ws framework** to explicitly compare works in the EL-Comm and LLM-Comm sections.
> - We added **explicit cross-work comparisons and summary tables** throughout the EL-Comm and LLM-Comm sections.
> - The revised text emphasizes **contrasts across who/whom/what, when, why, and how**, including differences in assumptions, learning signals, evaluation challenges, and limitations, rather than descriptive listings of papers.
>
>
>
>
>
>
>
> ---
>
> ### 3. Tempering Claims on LLM-Based Multi-Agent Systems
>
> **Revisions**:
> - We integrated discussion of **LLM limitations directly into each Five Ws dimension** in the LLM-Comm section, including lack of formal guarantees, grounding and hallucination issues, and scalability, cost, and real-time constraints.
> - We additionally added a new **Bridge Section (Sec. 7.6)** that **explicitly discusses the limitations of LLM-based multi-agent systems** and clarifies their role as **complementary to, rather than replacements for, classical MARL methods**.
>
> ---
>
> ### 4. Clarifying Communication Formalization (Section 6)
> **Revisions**:
> - We removed the circular dependency by introducing **explicit time indices** for messages and actions.
> - The revised formulation in **Section 8.1 (marked in blue)** stages communication and action selection across timesteps, eliminating same-step circularity.
>
>
> ---
>
> ### 5. Connection to Classical Decentralized Control
>
> **Revisions**:
> - We added **Section 7.6.4 (marked in blue)** to analyze **LLM limitations from a decentralized control perspective**.
> - The section includes a **summary table** contrasting decentralized-control requirements with MARL-Comm mechanisms and reported **LLM multi-agent failure modes**.
>
>
> ### 6. Scope and Paper Selection Criteria
> **Revisions**:
> - We added a new **Methodology and Scope** section (Sec. 4) detailing the **search process, temporal scope, and inclusion/exclusion criteria**.
> - It also clarifies the **coverage scale** (~400 works surveyed, ~100–130 discussed in depth) and the rationale for selecting papers for detailed analysis.
>
> ---
> ### 7. Integration of RL-Based and LLM-Based Multi-Agent Communication
> **Revisions**:
> - We added a new **Section 7.6** that explicitly discusses **integration pathways between RL-based communication and LLM-based multi-agent systems**, clarifying their complementary roles and how they can be combined in hybrid architectures.
>
> ---
> ### 8. Articulation of “Why Communicate”
> **Revisions**:
> - We revised the paper to **explicitly articulate different categories of “why communicate”** and to show how each motivation affects communication design.
> - This structure is now applied **consistently across all three major sections** (MARL-Comm, EL-Comm, and LLM-Comm)
>
> ---
> ### 9. Minor Issues and Typos
> **Revisions**:
> - All reported typos and terminology inconsistencies have been corrected and **marked in blue**, including sentence fragments, naming inconsistencies (e.g., “MARL-COMM” vs. “MARL-Comm”), unclear phrasing, and grammatical errors.

---

### Author Response · Authors · 2025-12-15
**Overall Response to All Reviewers**

# Overall Response to Reviewers

We sincerely thank all reviewers for their careful reading and constructive feedback. Although each reviewer raised distinct points, their comments converged on several common concerns regarding **narrative coherence across paradigms, explicit use of the Five Ws (especially *why communicate*), analytical depth beyond citation enumeration, foundational grounding, technical clarity, and the positioning of LLM-based systems**. We have **systematically addressed all of these concerns** through substantial revisions to both the structure and content of the paper.
All changes are clearly marked in the manuscript (**blue for Reviewer 7M5j, green for Reviewer CeNu, and orange for Reviewer ye4q**).

In addition to the targeted revisions described below, we also made **global consistency updates throughout the manuscript**, including the **abstract, introduction, and opening paragraphs of all major sections**, to ensure that the revised framing, terminology, and narrative structure are reflected consistently across the paper.

In addition to this brief overview, we provide **detailed, point-by-point responses to each reviewer below**, explicitly indicating where and how each concern has been addressed in the revised manuscript.

---

## Summary of Key Revisions Addressing Common Concerns

- **Stronger cross-paradigm coherence**: We added an **Evolution overview (Sec. 1.1)** and **bridging subsections (Secs. 5.7, 6.7, 7.6)** to explain why the field evolved from MARL-Comm to EL-Comm and LLM-Comm.

- **Explicit Five Ws framing**: All major sections were revised to consistently organize discussion around **who/whom, when, why, and how**, with *why* explicitly motivating communication design choices.
- **Deeper comparative analysis**: Citation stacking was revised in favor of **cross-work comparisons, summary tables, and discussion of trade-offs, trends, and limitations**, especially in the LLM-based sections.
- **Foundations and broader perspectives**: We added **Section 2 (Foundations of Agent Communication)** and **Sec. 8.2 (Epistemic Perspectives)** to ground the survey in classical AI, philosophy, and belief-aware reasoning.
- **Targeted formalization**: We introduced **lightweight mathematical formalizations** of communication structure (**Sec. 7.4.1**) and a **game-theoretic view of mixed-motive communication** (**Sec. 7.3.1**).
- **Expanded LLM–MARL hybrid analysis**: We **added Sec. 7.6**, a dedicated Bridge section on **LLM-grounded MARL**, with focused subsections on **why/how communication emerges (7.6.1)** and **comparative hybrid patterns (7.6.2)**, supported by **systematic comparisons and summary tables**.
- **Balanced treatment of LLMs**: Sec. 7.6 also provides a **clear analysis of LLM limitations**, with dedicated discussions of **LLM constraints in MARL settings (7.6.3)** and **from a decentralized control perspective (7.6.4)**, positioning LLMs as **complementary to, not replacements for, MARL**.
- **Clear scope and updated references**: We added a new **Sec. 4: Methodology and Scope of the Survey** section and updated citations to prefer **archival versions**.



We believe these revisions substantially strengthen the paper’s clarity, coherence, and analytical depth. We thank the reviewers again for their insightful feedback and provide detailed responses to each reviewer below.

---

### Decision · Action_Editor_GGaJ · 2026-01-20

**Recommendation:** Accept as is

**Additional Comments:**

The paper surveys multi-agent communication across communication in MARL, emergent communication (EL), and communication in LLM-based multi-agent systems. The paper presents a taxonomy inspired by “Five Ws” (who/whom, when, what, why) and uses it most systematically in the MARL section. The authors reviews key methods, provides classification tables, and closes with cross-domain discussion of open challenges and future directions. All reviewers agree that this is a comprehensive and timely suvery paper, which could be a helpful overview and summary on the field of multi-agent communication to readers in the community.

**Audience:**

Yes

**Audience Explanation:**

Multi-agent systems are increasingly deployed and of great interest in practice. A comprehensive  survey paper that covering key areas of communication across MARL, emergent language, and LLM-based multi-agent systems is timely and useful for readers in the area.

**Claims And Evidence:**

Yes

**Claims Explanation:**

This is a survey paper that comprehensively cites recent works and presents concrete formulation that aligns with the common understanding of multi-agent agent systems.